# Competition between parallel sensorimotor learning systems

Scott T Albert[1,2]*, Jihoon Jang[1,3], Shanaathanan Modchalingam[4], Bernard Marius 't Hart[4], Denise Henriques[4], Gonzalo Lerner[5], Valeria Della-Maggiore[5], Adrian M Haith[6], John W Krakauer[6,7,8], Reza Shadmehr[1]

[1]Department of Biomedical Engineering, Johns Hopkins School of Medicine, Baltimore, United States; [2]Neuroscience Center, University of North Carolina, Chapel Hill, United States; [3]Vanderbilt University School of Medicine, Nashville, United States; [4]Department of Kinesiology and Health Science, York University, Toronto, Canada; [5]IFIBIO Houssay, Deparamento de Fisiología y Biofísia, Facultad de Medicina, Universidad de Buenos Aires, Buenos Aires, Argentina; [6]Department of Neurology, Johns Hopkins School of Medicine, Baltimore, United States; [7]Department of Neuroscience, Johns Hopkins School of Medicine, Baltimore, United States; [8]The Santa Fe Institute, Santa Fe, United States

**Abstract** Sensorimotor learning is supported by at least two parallel systems: a strategic process that benefits from explicit knowledge and an implicit process that adapts subconsciously. How do these systems interact? Does one system's contributions suppress the other, or do they operate independently? Here, we illustrate that during reaching, implicit and explicit systems both learn from visual target errors. This shared error leads to competition such that an increase in the explicit system's response siphons away resources that are needed for implicit adaptation, thus reducing its learning. As a result, steady-state implicit learning can vary across experimental conditions, due to changes in strategy. Furthermore, strategies can mask changes in implicit learning properties, such as its error sensitivity. These ideas, however, become more complex in conditions where subjects adapt using multiple visual landmarks, a situation which introduces learning from sensory prediction errors in addition to target errors. These two types of implicit errors can oppose each other, leading to another type of competition. Thus, during sensorimotor adaptation, implicit and explicit learning systems compete for a common resource: error.

*For correspondence: scottalbert1@gmail.com

## Editor's evaluation

The interaction between implicit and explicit processes is central for motor learning. The present study builds upon diverse and sometimes seemingly conflicting data sets to propose a computational model, delineating a competing relationship between the explicit and implicit learning process during motor adaptation. The model provides a number of conceptual insights about the nature of error-based learning, not just for researchers in sensorimotor learning but also for those studying human learning in general.

## Introduction

When our movements are perturbed, we become aware of our errors, and through our own strategy, or instructions from a coach, engage an explicit learning system to improve our outcome (*Morehead et al., 2017*; *Mazzoni and Krakauer, 2006*). This awareness, is not required to adapt; our brain also uses an implicit learning system that partially corrects behavior without our conscious awareness

(*Morehead et al., 2017*; *Mazzoni and Krakauer, 2006*). How do these two systems interact during sensorimotor adaptation?

Suppose that both systems learn from the same error. In this case, when one system adapts, it will reduce the error that drives learning in the other system; thus, the two parallel systems will compete to 'consume' a common error. Alternatively, suppose the two systems learn from separate errors, and each produces an output to minimize its own error. In this case, when one system adapts to its error, it could change behavior in ways that paradoxically increase the other system's error.

Current models suggest that adaptation is driven by two distinct error sources: a task error (*Leow et al., 2020*; *Körding and Wolpert, 2004*; *Langsdorf et al., 2021*), and a prediction error (*Mazzoni and Krakauer, 2006*; *Tseng et al., 2007*; *Kawato, 1999*). One leading theory suggests that the explicit system acts to decrease errors in task performance, while the implicit system acts to reduce errors in predicting sensory outcomes (*Mazzoni and Krakauer, 2006*; *Taylor and Ivry, 2011*; *Wong and Shelhamer, 2012*). In this model, strategies have no impact on implicit learning. A second theory suggests that task errors can drive learning in both systems (*Leow et al., 2020*; *Kim et al., 2019*; *McDougle et al., 2015*; *Miyamoto et al., 2020*). In this model, implicit and explicit systems will compete with one another.

Suppose implicit and explicit systems share at least one common error source. What will happen when experimental conditions enhance one's explicit strategy? In this case, increases in explicit strategy will siphon away the error that the implicit system needs to adapt, thus reducing total implicit learning without directly changing implicit learning properties (e.g. its memory retention or sensitivity to error). This reduction in implicit learning creates the illusion that the implicit system was directly altered by the experimental manipulation, when in truth, it was only responding to changes in strategy.

Competitive interactions like this highlight the need to distinguish between an adaptive system's learning properties such as its sensitivity to an error, and its learning timecourse, that is the contribution it makes to overall adaptation at any point in time. In a competitive system, an adaptive processes' learning timecourse depends not only on its own learning properties, but also its competitors' learning properties. In cases where implicit and explicit systems share an error source, one system's behavior can be shaped not only by its past experience, but also by changes in the other system. Thus, competition may play an important role in savings (*Haith et al., 2015*; *Coltman et al., 2019*; *Kojima et al., 2004*; *Medina et al., 2001*; *Mawase et al., 2014*) and interference paradigms (*Sing and Smith, 2010*; *Lerner et al., 2020*; *Caithness et al., 2004*) where learning properties change over time. Measuring the interdependence between implicit and explicit learning may help to explain the disconnect between studies that have suggested acceleration in motor learning is subserved solely by explicit strategy (*Haith et al., 2015*; *Huberdeau et al., 2019*; *Morehead et al., 2015*; *Avraham et al., 2020*; *Avraham et al., 2021*), and studies that have pointed to concomitant changes in implicit learning systems (*Leow et al., 2020*; *Yin and Wei, 2020*; *Albert et al., 2021*).

Here, we begin by mathematically (*McDougle et al., 2015*; *Miyamoto et al., 2020*; *Smith et al., 2006*; *Albert and Shadmehr, 2018*; *Thoroughman and Shadmehr, 2000*) considering the extent to which implicit and explicit systems are engaged by task errors and prediction errors. The hypotheses make diverging predictions, which we test in various contexts. Our work suggests that in some contexts (*Mazzoni and Krakauer, 2006*; *Taylor and Ivry, 2011*), prediction errors and task errors both make important contributions to implicit learning (Results Part 3). In other contexts, the data suggest that the implicit system is primarily driven by task errors shared with the explicit system (Results Part 1). In this latter case, the competition theory explains why increases (*Neville and Cressman, 2018*; *Benson et al., 2011*) or decreases (*Fernandez-Ruiz et al., 2011*; *Saijo and Gomi, 2010*) in explicit strategy cause an opposite change in implicit learning. This model explains why in some cases implicit adaptation can saturate as perturbations grow (*Neville and Cressman, 2018*; *Bond and Taylor, 2015*; *Tsay et al., 2021a*), but not others (*Tsay et al., 2021a*; *Salomonczyk et al., 2011*). The model also explains why participants that utilize large explicit strategies can exhibit less implicit (*Miyamoto et al., 2020*) or procedural learning (*Fernandez-Ruiz et al., 2011*), than those who do not. Finally, the theory provides an alternate way to interpret implicit contributions to two learning hallmarks: savings (*Haith et al., 2015*) and interference (*Lerner et al., 2020*) (Results Part 2).

Altogether, our results illustrate that sensorimotor adaptation is shaped by competition between parallel learning systems, both engaged by task errors.

# Results

In visuomotor rotation paradigms, participants move a cursor that travels along a rotated path (*Figure 1A*). This perturbation causes adaptation, resulting in both implicit recalibration (*Figure 1A*, implicit) and explicit (intentional) re-aiming (*Figure 1A*, aim) (*Mazzoni and Krakauer, 2006*; *Taylor and Ivry, 2011*; *Taylor et al., 2014*; *Shadmehr et al., 1998*).

Current models suggest that the rotation $r$ creates two distinct error sources. One error source is the deviation between cursor and target: a target error (*Leow et al., 2020*; *Körding and Wolpert, 2004*; *Langsdorf et al., 2021*). Notably, this target error (*Figure 1A*, target error) is altered by both implicit ($x_i$) and explicit ($x_e$) adaptation:

$$e_{target}^{(n)} = r^{(n)} - \left( x_i^{(n)} + x_e^{(n)} \right)$$

(1)

In addition, a second error is created due to our expectation that the cursor should move toward where we aimed our movement: a sensory prediction error (SPE) (*Mazzoni and Krakauer, 2006*; *Tseng et al., 2007*; *Kawato, 1999*). SPE is the deviation between the aiming direction (the expected cursor motion) and where we observed the cursor's actual motion (*Figure 1A*, sensory prediction error). Critically, because this error is anchored to our aim location, it changes over time in response to implicit adaptation alone:

$$e_{SPE}^{(n)} = r^{(n)} - x_i^{(n)}$$

(2)

How does the implicit learning system respond to these two error sources? State-space models describe implicit adaptation as a process of learning and forgetting (*McDougle et al., 2015*; *Miyamoto et al., 2020*; *Smith et al., 2006*; *Albert and Shadmehr, 2018*; *Thoroughman and Shadmehr, 2000*):

$$x_i^{(n+1)} = a_i x_i^{(n)} + b_i e^{(n)}$$

(3)

Forgetting is controlled by a retention factor ($a_i$) which determines how strongly we retain the adapted state. Learning is controlled by error sensitivity ($b_i$) which determines the amount we adapt in response to an error (e.g. an SPE or a target error).

Here, we will contrast two possibilities: (1) the implicit system responds primarily to target error, or (2) the implicit system responds primarily to SPE. In a target error learning system, explicit strategy will reduce the target error in *Equation 1*. This decrease in target error will lead to a competition between implicit and explicit systems, that is increasing explicit strategy reduces target error, which will then decrease implicit learning. Competition in a target error model will occur over the entire learning timecourse and can lead to unintuitive implicit learning phenotypes (Appendix 1.2). While these implicit behaviors can be observed at any point during adaptation, they are easiest to examine during steady-state adaptation (Appendix 1.1).

Consider how *Equation 3* behaves in the steady-state condition. Like adapted behavior (*Kim et al., 2019*; *Albert et al., 2021*; *Vaswani et al., 2015*; *Kim et al., 2018*), *Equation 3* approaches an asymptote with extended exposure to a rotation. This steady-state (*Figure 1B*, implicit) occurs when learning and forgetting counterbalance each other.

Consider a system where target errors alone drive implicit learning. In this system, total (steady-state) implicit learning is determined by *Equations 1 and 3*:

$$x_i^{ss} = \frac{b_i}{1 - a_i + b_i} \left( r - x_e^{ss} \right)$$

(4)

*Equation 4* demonstrates a competition between implicit and explicit systems; the total amount of implicit adaptation ($x_i^{ss}$) is driven by the difference between the rotation $r$ and total explicit adaptation ($x_e^{ss}$).

Now consider a system where SPEs drive implicit learning. SPEs (*Equation 2*) are unaltered by strategy. In this case, total implicit learning is determined by *Equations 2 and 3*:

$$x_i^{ss} = \frac{b_i}{1 - a_i + b_i} r$$

(5)

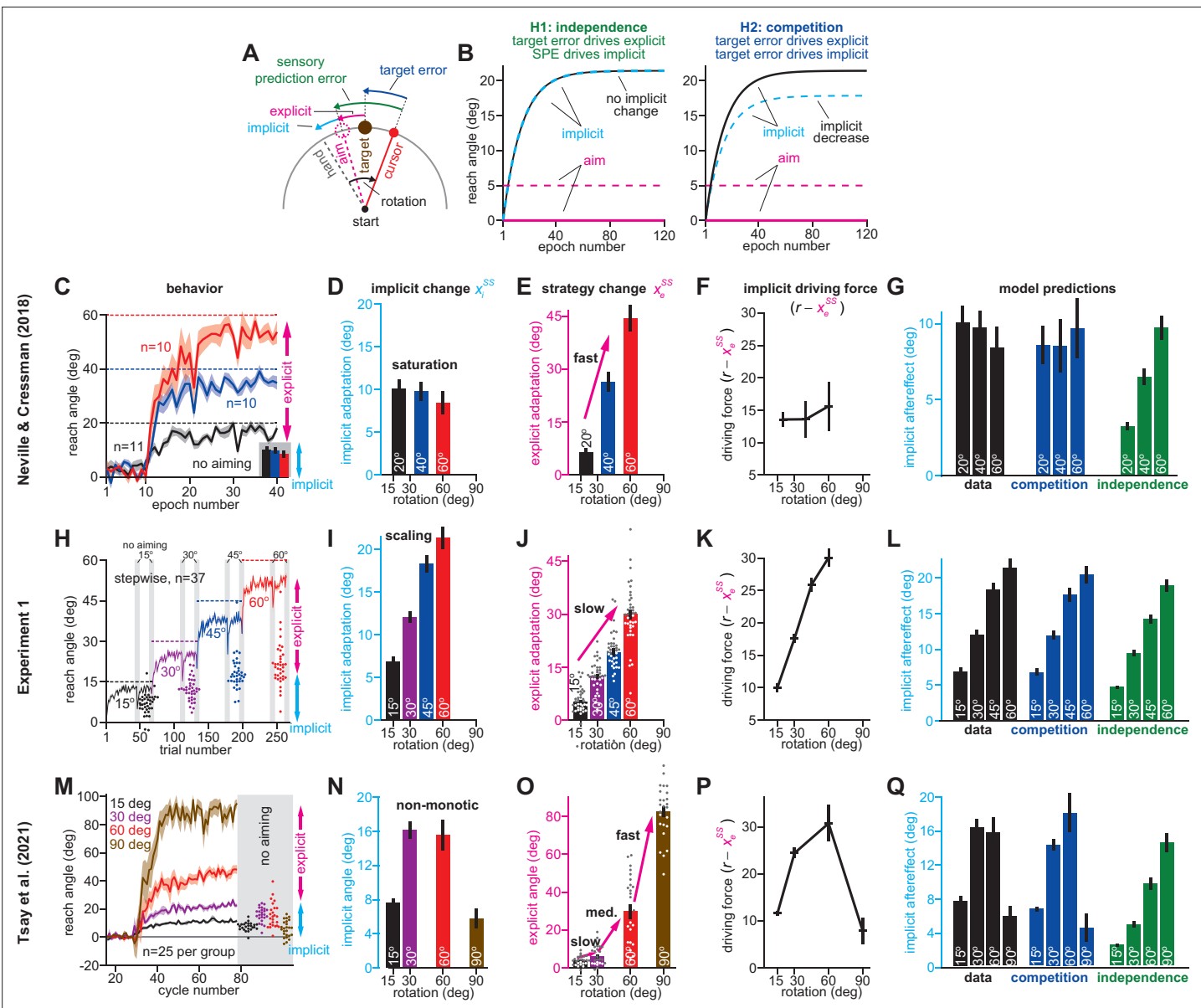

**Figure 1.** Total implicit learning is shaped by competition with explicit strategy. (**A**). Schematic of visuomotor rotation. Participants move from start to target. Hand path is composed of explicit (aim) and implicit corrections. Cursor path is perturbed by rotation. We explored two hypotheses: prediction error (H1, aim vs. cursor) vs. target error (H2, target vs. cursor) drives implicit learning. (**B**) Prediction error hypothesis predicts that enhancing aiming (dashed magenta) will not change implicit learning (black vs. dashed cyan) according to the independence equation. Target error hypothesis predicts that enhancing aiming (dashed magenta) will decrease implicit adaptation (black vs. dashed cyan). (**C**) Data reported by **Neville and Cressman, 2018**. Participants were exposed to either a 20°, 40°, or 60° rotation. Learning curves are shown. The "no aiming" inset shows implicit learning measured via exclusion trials at the end of adaptation. Explicit strategy was calculated as the voluntary reduction in reach angle during the no aiming period. (**D**) Implicit learning measured during no aiming period in Neville and Cressman yielded a 'saturation' phenotype. (**E**) Explicit strategies calculated in Neville & Cressman dataset by subtracting exclusion trial reach angles from the total adapted reach angle. (**F**) The implicit learning driving force in the competition theory: difference between rotation and explicit learning in Neville and Cressman. (**G**) Implicit learning predicted by the competition and independence models in Neville and Cressman. Models were fit assuming that the implicit learning gain was identical across rotation sizes. (**H**) Experiment 1. Subjects in the stepwise group (n = 37) experienced a 60° rotation gradually in four steps: 15°, 30°, 45°, and 60°. Implicit learning was measured via exclusion trials (points) twice in each rotation period (gray 'no aiming'). (**I**) Total implicit learning calculated during each rotation period in the stepwise group yielded a 'scaling' phenotype. (**J**) Explicit strategies were calculated in the stepwise group by subtracting exclusion trial reach angles from the total adapted reach angle. (**K**) The implicit learning driving force in the competition theory: difference between rotation and explicit learning in the stepwise group. (**L**) Implicit learning predicted by the competition and independence models in the stepwise group. Models were fit assuming that implicit learning gain was constant across rotation size. (**M**) Data reported by **Tsay et al., 2021a**. Participants were exposed to either a 15°, 30°, 60°, or 90° rotation. Learning curves are shown. The "no aiming" inset shows implicit learning measured via exclusion trials at the end of adaptation. (**N**) Implicit

*Figure 1 continued on next page*

*Figure 1 continued*

learning measured during no aiming period in Tsay et al. yielded a 'non-monotonic' phenotype. (**O**) Explicit strategies calculated in Tsay et al. dataset by subtracting exclusion trial reach angles from the total adapted reach angle. (**P**) Implicit learning driving force in the competition theory: difference between rotation and explicit learning in Tsay et al. (**Q**) Total implicit learning predicted by the competition and independence models in Tsay et al. Models were fit assuming that the implicit learning gain was identical across rotation sizes. Error bars show mean ± SEM, except in the independence predictions in **G**, **L**, and **Q**; independence predictions show mean and standard deviation across 10,000 bootstrapped samples. Points in **H**, **J**, **M**, and **O** show individual participants.

The online version of this article includes the following source code and figure supplement(s) for figure 1:

**Source code 1.** *Figure 1* data and analysis code.

**Figure supplement 1.** Implicit learning can exhibit various phenotypes in the competition theory.

**Figure supplement 2.** Variations between total learning and implicit learning are consistent with the competition model.

**Figure supplement 2—source code 1.** *Figure 1—figure supplement 2* data and analysis code.

**Figure supplement 3.** Scaling, saturation, and non-monotonic phenotypes across the implicit learning timecourse.

**Figure supplement 3—source code 1.** *Figure 1—figure supplement 3* analysis code.

**Figure supplement 4.** Changes in implicit learning across blocks.

**Figure supplement 4—source code 1.** *Figure 1—figure supplement 4* data and analysis code.

*Equation 5* demonstrates an independence between implicit and explicit systems; the total amount of implicit adaptation depends solely on the rotation's magnitude, not one's explicit strategy.

Here, we explore how implicit learning systems respond to explicit strategy, and whether behavior is more consistent with competition or independence. Competition and independence can be studied at any point during the adaptation timecourse (Appendix 1). We will primarily examine steady-state learning, where the competition equation (*Equation 4*) and independence equation (*Equation 5*) make simple predictions. The critical insight is that in an independent system (SPE learning), increasing the explicit strategy (*Figure 1B*, magenta solid and dashed) does not alter implicit adaptation (*Figure 1B*, independence, compare black and cyan). However, in a competitive system (*Equation 4*), the same increase in strategy will indirectly decrease implicit learning (*Figure 1B*, competition, compare black and cyan).

To analyze these possibilities, we begin by examining how changes in explicit strategy alter implicit learning in response to variations in rotation magnitude, experimental instructions, rotation type, and at the individual participant level (Part 1). Next, we describe how competition between implicit and explicit systems could in principle mask changes in implicit learning (Part 2). Finally, we will examine studies which suggest implicit error sources vary across experimental conditions due to the presence and/or absence of multiple visual stimuli in the experimental workspace (Part 3).

## Part 1: Measuring how implicit learning responds to changes in explicit strategy

Here, we measure how implicit learning and explicit strategy vary across several factors: (1) rotation size, (2) instructions, (3) gradual versus abrupt rotations, and (4) individual subjects. We will ask whether the variations in implicit and explicit learning are consistent with the competition or independence theories.

### Implicit responses to rotation size suggest a competition with explicit strategy

Over extended exposure to a rotation, adaptation appears to saturate (*Morehead et al., 2017*; *Albert et al., 2021*; *Vaswani et al., 2015*; *Kim et al., 2018*). How does implicit learning contribute to steady-state saturation, and what learning model best describes its behavior?

In *Neville and Cressman, 2018*, participants adapted to a 20°, 40°, or 60° rotation (*Figure 1C*). As is common, adaptation reached a steady-state prior to eliminating the target error (*Albert et al., 2021*; *Figure 1C*, solid vs. dashed lines). To measure implicit learning, participants were instructed to reach to the target without aiming (*Figure 1C*, no aiming). The independence model (*Equation 5*) predicts that the implicit response should scale as the rotation increases. On the contrary, total implicit learning was insensitive to rotation size; it reached only 10° and remained constant despite

a threefold increase in rotation magnitude (**Figure 1D**). To estimate explicit strategy, we subtracted the implicit learning measure from the total adapted response. Opposite to implicit learning, explicit strategy increased proportionally with the rotation's size (**Bond and Taylor, 2015**; **Tsay et al., 2021a**; **Figure 1E**).

In the competition model, implicit learning is driven by the difference between the rotation and explicit strategy ($r - x_e^{ss}$ in **Equation 4**). As a result, when an increase in rotation magnitude is matched by an equal increase in explicit strategy (**Figure 1—figure supplement 1A**, same), the implicit learning system's driving force will remain constant (**Figure 1—figure supplement 1B**, same). This constant driving input leads to a phenotype where implicit learning appears to 'saturate' with increases in rotation size (**Figure 1—figure supplement 1C**, same).

To investigate whether this mechanism is consistent with the implicit response, we examined how explicit strategy and the implicit driving force varied with rotation size. As rotation size increased, explicit strategies increased substantially (**Figure 1E**). Under the competition model, these rapid changes in explicit strategy produced an implicit driving force that responded little to rotation magnitude; while the rotation increased by 40°, the driving force changed by less than 2.5° (**Figure 1F**). Thus, the competition Equation (**Figure 1G**, competition) suggested that implicit learning would not vary with rotation size, as we observed in the measured data (**Figure 1G**, data).

In other words, the competition model suggests that the implicit system can exhibit an unintuitive saturation when its driving input remains constant. The key prediction is that by altering explicit strategy, this driving input will change, changing the implicit response to rotation size. One possibility is to weaken the explicit system's response to the rotation (**Figure 1—figure supplement 1A**, slower) which should increase the steady-state of the implicit system (**Figure 1—figure supplement 1C**, slower).

To test this idea, we used a stepwise rotation (**Yin and Wei, 2020**). In Experiment 1, participants (n = 37) adapted to a stepwise perturbation which started at 15° but increased to 60° in 15° increments (**Figure 1H**). Twice toward the end of each rotation block, we assessed implicit adaptation by instructing participants to aim directly to the target (**Figure 1H**, gray regions). Supplemental analysis suggested that the implicit system reached its steady-state during each learning period (Appendix 2), although this is not required to test the competition theory (Appendix 1.2). Critically, the stepwise rotation onset decreased explicit responses relative to the abrupt rotations used by **Neville and Cressman, 2018**; explicit strategies increased with a 94.9% gain (change in strategy divided by change in rotation) across the abrupt groups in **Figure 1E**, but only a 55.5% gain in the stepwise condition shown in **Figure 1J**. In the competition model, this reduction in strategy increased the implicit system's driving input (**Figure 1K**). The increased driving input produced a "scaling" phenotype in the competition model's implicit response (**Figure 1L**, competition) which closely matched the measured implicit data (**Figure 1I**; 1 L, data; rm-ANOVA, $F(3,108)=99.9$, $p < 0.001$, $\eta_p^2=0.735$).

Thus, the implicit system can exhibit both saturation (**Figure 1G**) and scaling (**Figure 1L**), consistent with the competition model. Recent work by **Tsay et al., 2021a** suggests a third steady-state implicit phenotype: non-monotonicity. In their study, the authors examined a wider range in rotation size, 15° to 90° (**Figure 1M**). A no-aiming period revealed total implicit adaptation each group (n = 25/group). Curiously, whereas implicit learning increased between the 15° and 30° rotations, it appeared similar in the 60° rotation group, and then decreased in the 90° rotation group (**Figure 1N**). This non-monotonic behavior was inconsistent with the independence model where implicit learning is proportional to rotation size (**Figure 1Q**, independence).

To determine whether this non-monotonicity could be captured by the competition theory, we considered again how explicit re-aiming increased with rotation size (**Figure 1O**). We observed an intriguing pattern. When the rotation increased from 15° to 30°, explicit strategy responded with a very low gain (4.5%, change in strategy divided by change in rotation). An increase in rotation size to 60° was associated with a medium-sized gain (80.1%). The last increase to 90° caused a marked change in the explicit system: a 53.3° increase in explicit strategy (177.7% gain). Thus, explicit strategy increased more than the rotation had. Critically, this condition produces a decrease in the implicit driving input in the competition theory (**Figure 1—figure supplement 1**, faster). Overall, we estimated that this large variation in explicit learning gain (4.5–80.1% to 177.7%) should yield non-monotonic behavior in the implicit driving input (**Figure 1P**): an increase between 15° and 30°, no change between 30° and 60°,

and a decrease between 60° and 90°. As a result, the competition theory (*Figure 1Q*, competition) exhibited a non-monotonic envelope, which closely tracked the measured data (*Figure 1Q*, data).

Unfortunately, there is a potential problem in our analysis: implicit and explicit learning measures were not independent, because explicit strategy was estimated using implicit reach angles (i.e. explicit learning equals total learning minus implicit learning). Did this bias our analysis towards the competition model? To answer this question, *Equation 4* can be stated as $x_i^{ss} = p_i(r - x_e^{ss})$ where $p_i$ is the learning gain determined by the implicit system's retention and error sensitivity (i.e. $a_i$ and $b_i$). We can replace the explicit strategy ($x_e^{ss}$) appearing in this equation noting that $x_e^{ss} = x_T^{ss} - x_i^{ss}$, where $x_T^{ss}$ equals total steady-state adaptation. With this, the model relates implicit learning to total learning: $x_i^{ss} = p_i(1 - p_i)^{-1}(r - x_T^{ss})$, as opposed to explicit learning, and can be used to test the competition model without correlated learning measures (see Appendix 3). We reexamined all three experiments in *Figure 1*, using total adaptation to predict implicit learning with the competition model (*Figure 1—figure supplement 2*). This alternate method yielded nearly identical predictions (*Figure 1—figure supplement 2*, 'model-2') as *Equation 4* (*Figure 1—figure supplement 2*, 'model-1'). Thus, the qualitative and quantitative correspondence between the competition model and the measured data was not due to how we operationalized implicit and explicit learning (see Appendix 3).

Collectively, these studies demonstrate that the implicit system can exhibit at least three distinct behavioral phenomena: saturation, scaling, or non-monotonicity. The competition model matched all three phenotypes, due to the implicit system's response to explicit strategy. The SPE learning model described by the independence equation, however, could only produce a scaling phenotype (*Figure 1I*). Could the SPE learning model be altered to produce implicit learning phenotypes other than scaling? One possibility is that a saturation phenotype (*Figure 1D*) could be built into the SPE model by adding a restriction, that is an upper bound, on total implicit adaptation, as observed in studies where participants experience invariant error perturbations (*Morehead et al., 2017*; *Kim et al., 2018*). With that said, the 10° implicit responses observed across the three rotations in *Neville and Cressman, 2018*, are much lower than the 20°–25° ceiling suggested by recent error-clamp studies (*Kim et al., 2018*), and the 35–45° implicit responses observed in some standard rotation studies (*Salomonczyk et al., 2011*; *Maresch et al., 2021*). More importantly, a learning model with a rotation-insensitive upper bound on implicit learning would be inconsistent with the scaling (*Figure 1I*) and nonmonotonic (*Figure 1N*; see Appendix 6.6) phenotypes we observed. We will explore other extensions to this SPE model in several analyses in the Control analyses section below.

## Increase in explicit strategy suppresses implicit learning

The competition model predicts that increasing explicit strategy will decrease implicit learning, even when the rotation size is the same. In contrast, the independence theory predicts that implicit learning will be insensitive to differences in explicit strategy (extensions to this model are considered in Control analyses).

To test these ideas, we considered another condition tested by *Neville and Cressman, 2018* where participants were exposed to the same 20°, 40°, or 60° rotation, but received coaching instructions. The coaching sharply improved adaptation over the non-instructed group (*Figure 2A*, compare purple with black). To understand how implicit and explicit learning contributed to these changes, we analyzed the mean implicit and explicit reach angles measured across all three rotation sizes (each individual response is shown in *Figure 2—figure supplement 1*).

Unsurprisingly, explicit adaptation was enhanced in the participants that received coaching instructions. Explicit re-aiming increased by approximately 10° (*Figure 2B*, t(61)=2.29, p = 0.026, d = 0.56). However, while instruction enhanced explicit strategy, it suppressed implicit learning, decreasing total implicit learning by approximately 32% (*Figure 2C*, data, t(61)=2.62, p = 0.011, d = 0.66). To interpret this implicit response, we fit the competition (*Equation 4*) and independence equations (*Equation 5*) to the behavior across all experimental conditions (six groups: 3 rotation magnitudes, 2 instruction conditions), while holding the implicit learning parameters in the model constant (i.e. holding $a_i$ and $b_i$ constant across all conditions).

As in *Figure 1*, implicit learning in the independence model does not respond to explicit strategy, and is not altered by instruction (*Figure 2C*, implicit learning, indep.). On the other hand, the competition model accurately suggested that total implicit learning would decrease by approximately 3° (data showed 2.98° decrease, model produced a 2.92° decrease) in response to increases in explicit

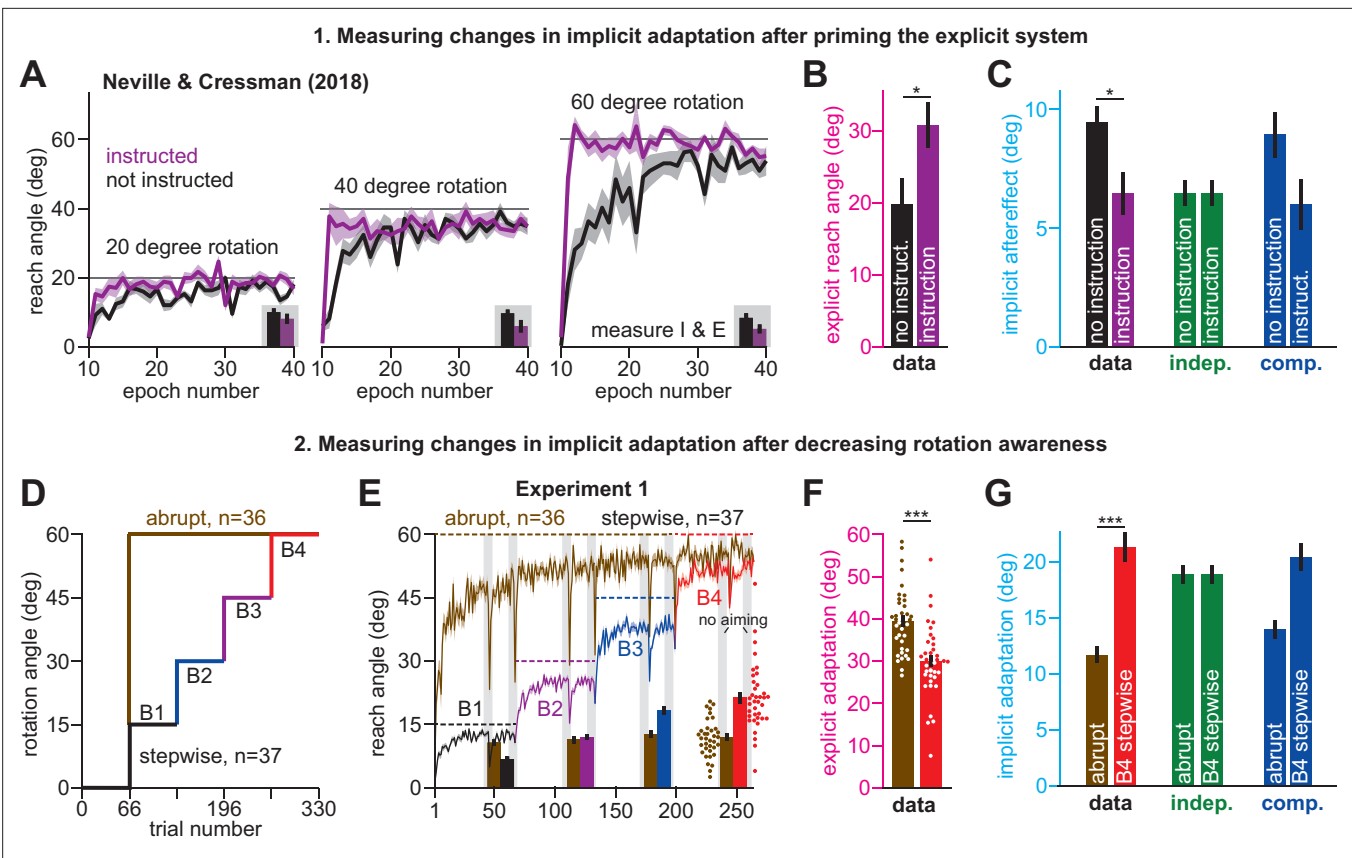

**Figure 2.** Increases or decreases in explicit strategy oppositely impact implicit adaptation. (**A**) *Neville and Cressman, 2018* tested participants in two conditions: an uninstructed condition (black) and an instructed condition (purple) where subjects were briefed about the upcoming rotation and its solution. Instruction increased the adaptation rate across three rotation sizes: 20°, 40°, and 60°. Insets in gray shaded area show implicit adaptation measured via exclusion trials at the end of adaptation. (**B**) Here, we show the average strategy across all rotation sizes in the instructed (black) and uninstructed (purple) conditions. Explicit strategy was calculated by subtracting implicit learning (exclusion trials) from total adaptation. Instruction increased explicit strategy use. (**C**) The data show implicit adaptation averaged across all three rotation sizes. The independent (SPE learning) and competition (target error learning) models were fit to these data assuming that implicit error sensitivity and retention were identical across rotation sizes and instruction conditions (i.e. identical $a_i$ and $b_i$ across all six groups). Error bars for model predictions refer to mean and standard deviation across 10,000 bootstrapped samples. (**D**) In Experiment 1 we tested participants in either an abrupt condition or a stepwise (gradual) condition. Here, we show the rotation schedule. (**E**) Here, we show learning curves in the abrupt and stepwise conditions in Experiment 1. Bars show implicit adaptation measured during each rotation period (four blocks total) via exclusion trials. Individual learning measures are shown in the terminal 60° learning period for both groups (points at bottom-right). (**F**) We calculated explicit strategies during the terminal 60° learning period by subtracting implicit learning measures from total adaptation (mean over last 20 trials). Gradual onset reduced explicit strategy use. (**G**) The data show total implicit learning measured in the 60° rotation period. The competition (blue) and independence (green) models were fit to the data assuming that the implicit learning parameters were the same across the abrupt and stepwise groups. Error bars for the model show the mean and standard deviation across 1,000 bootstrapped samples. Statistics in **B**, **F**, and **G** denote a two-sample t-test: *p < 0.05, ***p < 0.001. Error bars in **A**, **B**, **C** (data), **E**, **F**, and **G** (data) denote mean ± SEM. Points in **E** and **F** show individual participants.

The online version of this article includes the following source code and figure supplement(s) for figure 2:

**Source code 1.** *Figure 2* data and analysis code.

**Figure supplement 1.** Changes in implicit adaptation in response to awareness and rotation size.

**Figure supplement 1—source code 1.** *Figure 2—figure supplement 1* data and analysis code.

**Figure supplement 2.** Total implicit adaptation varies slowly with changes in implicit error sensitivity.

**Figure supplement 2—source code 1.** *Figure 2—figure supplement 2* analysis code.

**Figure supplement 3.** Suppressing explicit strategy increases total implicit adaptation.

**Figure supplement 3—source code 1.** *Figure 2—figure supplement 3* data and analysis code.

**Figure supplement 4.** The competition model is compatible with various explicit strategy levels in *Saijo and Gomi, 2010*.

**Figure supplement 4—source code 1.** *Figure 2—figure supplement 4* data and analysis code.

strategy (*Figure 2C*, implicit learning, competition, t(61)=2.05, p = 0.045, d = 0.52). Altogether, the competition theory parsimoniously captured how the implicit system responded to explicit instruction (*Figure 2C*) as well as changes in rotation size (*Figure 1G*) with the same model parameter set (same $a_i$ and $b_i$ in the competition equation).

## Decrease in explicit strategy enhances implicit learning

Next, we examined how implicit learning responds to decreases in explicit strategy. *Yin and Wei, 2020* recently demonstrated that explicit strategies can be suppressed using gradual rotations. The competition theory predicts that decreasing explicit strategy will lead to greater implicit adaptation. We tested this prediction in Exp. 1. Participants were exposed to a 60° rotation, either abruptly (n = 36), or in a stepwise manner (n = 37) where perturbation magnitude increased by 15° across four distinct learning blocks (*Figure 2D*). We measured implicit and explicit learning during each block, as in *Figure 1*. To compare gradual and abrupt learning, we analyzed reach angles during the 4th learning block, where both groups experienced the 60° rotation size (*Figure 2E*).

As in *Yin and Wei, 2020*, participants in the stepwise condition exhibited a 10° reduction in explicit re-aiming (*Figure 2F*, two-sample t-test, t(71)=4.97, p < 0.001, d = 1.16). Reductions in strategy led to a decrease in total adaptation in the stepwise group by approximately 4°, relative to the abrupt group (*Figure 2E*, right-most gray region (last 20 trials); two-sample t-test, t(71)=3.33, p = 0.001, d = 0.78), but an increase in implicit learning by approximately 80% (*Figure 2G*, data, two-sample t-test, t(71)=6.4, p < 0.001, d = 1.5). Thus, the data presented a curious pattern; greater total adaptation in the abrupt condition was paradoxically associated with reduced implicit adaptation. As expected, these surprising patterns did not match the independence model (*Figure 2G*, indep.), in which implicit learning does not respond to changes in explicit strategy.

To test whether implicit learning patterns matched the competition model we fit *Equation 4* to implicit and explicit reach angles measured in Blocks 1–4, across the stepwise and abrupt conditions, while holding the model's implicit learning parameters ($a_i$ and $b_i$) constant. The competition model correctly predicted that the decrease in strategy in the gradual condition should produce an increase in implicit learning (*Figure 2G*, comp., two-sample t-test, t(71)=4.97, p < 0.001, d = 1.16). In addition, the competition model predicted a decrease in total learning, consistent again with the data (the model yielded 53.47° total adaptation in abrupt, and 50.42° in gradual: values not provided in *Figure 2*). The model's negative correlation between implicit learning and total adaptation occurred in two steps: (1) greater abrupt strategies increased overall adaptation, but (2) siphoned away target errors, reducing implicit adaptation.

We analyzed another hypothesis: changes in implicit adaptation were caused by variation in error sensitivity (e.g. greater implicit error sensitivity in the stepwise condition), rather than competition. Note, however, that the implicit learning gain, $p_i$, is given by $p_i = b_i(1 - a_i + b_i)^{-1}$. Because the $b_i$ term appears in both numerator and denominator, total implicit learning varies slowly with changes in $b_i$ (Appendix 4). Accordingly, supplemental analyses (Appendix 4, *Figure 2—figure supplement 2*) showed that no change in $b_i$ could yield the 80% increase in stepwise implicit learning in *Figure 2G*, let alone the 46% increase in implicit learning in the no-instruction group in *Figure 2C*. Thus, while variation in implicit error sensitivity might contribute to changes in steady-learning learning, its role is minor compared to error competition.

In summary, we observed that explicit strategies could be suppressed by increasing the rotation gradually. Reductions in explicit strategy were associated with increased implicit adaptation (*Figure 2G*) as predicted by the competition theory. Furthermore, the same competition theory parameter set (i.e. same $a_i$ and $b_i$, see Materials and methods) accurately matched the extent to which implicit learning responded to decreases in explicit strategy (*Figure 2G*) as well as increases in rotation size (*Figure 1L*). It is interesting to note that these implicit patterns are broadly consistent with the observation that gradual rotations improve procedural learning (*Saijo and Gomi, 2010*; *Kagerer et al., 1997*), although these earlier studies did not properly tease apart implicit and explicit adaptation (see the Saijo and Gomi analysis described in Appendix 5).

## Implicit adaptation responds to between-subject differences in explicit adaptation

Use of explicit strategy is highly variable between individuals (*Miyamoto et al., 2020*; *Fernandez-Ruiz et al., 2011*; *Bromberg et al., 2019*). According to the competition theory (*Equation 4*), implicit and explicit learning will negatively co-vary according to a line whose slope and bias are determined by the properties of the implicit learning system ($a_i$ and $b_i$). In Experiment 2, we tested this prediction. In one group, we limited preparation time to inhibit time-consuming explicit strategies (*Fernandez-Ruiz et al., 2011*; *McDougle and Taylor, 2019*; *Figure 3D–F*, Limit PT). In the other group, we imposed no preparation time constraints (*Figure 3A–C*, No PT Limit). We measured $a_i$ and $b_i$ in the Limit PT group and used these values to predict the implicit-explicit relationship across No PT Limit participants.

As expected, Limit PT participants dramatically reduced their reach latencies throughout the adaptation period (*Figure 3F*), whereas the No PT Limit participants exhibited a sharp increase in movement preparation time after perturbation onset (*Figure 3C*), indicating explicit re-aiming (*Langsdorf et al., 2021*; *Haith et al., 2015*; *Albert et al., 2021*; *Fernandez-Ruiz et al., 2011*; *McDougle and Taylor, 2019*). Consistent with explicit strategy suppression, learning proceeded more slowly and was less complete under the preparation time limit (compare *Figure 3B&E*; two-sample t-test on last 10 adaptation epochs: t(20)=3.27, p = 0.004, d = 1.42).

Next, we measured the retention factor $a_i$ during a terminal no feedback period (*Figure 3E*, dark gray, no feedback) and error sensitivity $b_i$ during the steady-state adaptation period. Steady-state implicit error sensitivity (note errors are small at steady-state creating high $b_i$) was consistent with recent literature (*Figure 3—figure supplement 1A-C*). Together, this retention factor ($a_i$ = 0.943) and error sensitivity ($b_i$ = 0.35), produced a specific form of *Equation 4*, $x_i = 0.86 (30 - x_e)$. We used this result to predict how implicit and explicit learning should vary across participants in the No PT Limit group (*Figure 3G*, blue line).

To measure implicit and explicit learning in the No PT Limit group, we instructed participants to move their hand through the target without any re-aiming at the end of the rotation period (*Figure 3B*, no aiming). The precipitous change in reaching angle revealed implicit and explicit components of adaptation (post-instruction reveals implicit; voluntary decrease in reach angle reveals explicit). We observed a striking correspondence between the No PT Limit implicit-explicit relationship (*Figure 3G*, black dot for each participant; $\rho = -0.95$) and that predicted by the competition equation (*Figure 3G*, blue). The slope and bias predicted by *Equation 4* (–0.86 and 25.74°, respectively) differed from the measured linear regression by less than 5% (*Figure 3G*, black line, $R^2 = 0.91$; slope is –0.9 with 95% CI [–1.16,–0.65] and intercept is 25.46° with 95% CI [22.54°, 28.38°]).

In addition, we also asked participants to verbally report their aiming angles prior to concluding the experiment. These responses were variable, with 25% reported in the incorrect direction. Because strategies are susceptible to sign-flipped errors (*McDougle and Taylor, 2019*), we assumed these misreported strategies represented the correct magnitude, but the incorrect sign, and thus took their absolute value. While reported explicit strategies were on average greater than our probe-based measure, and report-based implicit learning was on average smaller than our probe-based measure (*Figure 3—figure supplement 2A*&B; paired t-test, t(8)=2.59, p = 0.032, d = 0.7), the two report-based measures exhibited a strong correlation which aligned with the competition theory's prediction (*Figure 3—figure supplement 2C*; $R^2 = 0.95$; slope is –0.93 with 95% CI [–1.11,–0.75] and intercept is 25.51° with 95% CI [22.69°, 28.34°]).

In summary, individual participants exhibited an inverse relationship between implicit and explicit learning; participants who used large explicit strategies inadvertently suppressed their implicit learning, a pattern consistent with error-based competition.

## Limiting reaction time strongly suppresses explicit strategy and increases implicit learning

Our analysis in Experiment 2 had two important limitations. First, the competition theory used implicit learning parameters measured under limited preparation time conditions (*Leow et al., 2020*; *Fernandez-Ruiz et al., 2011*; *Leow et al., 2017*): how effectively does this condition suppress explicit learning? Second, our individual-level implicit and explicit learning measures were intrinsically correlated because they both depended on probe-based reach angles (i.e. implicit is no aiming probe, and explicit is total learning minus no aiming probe).

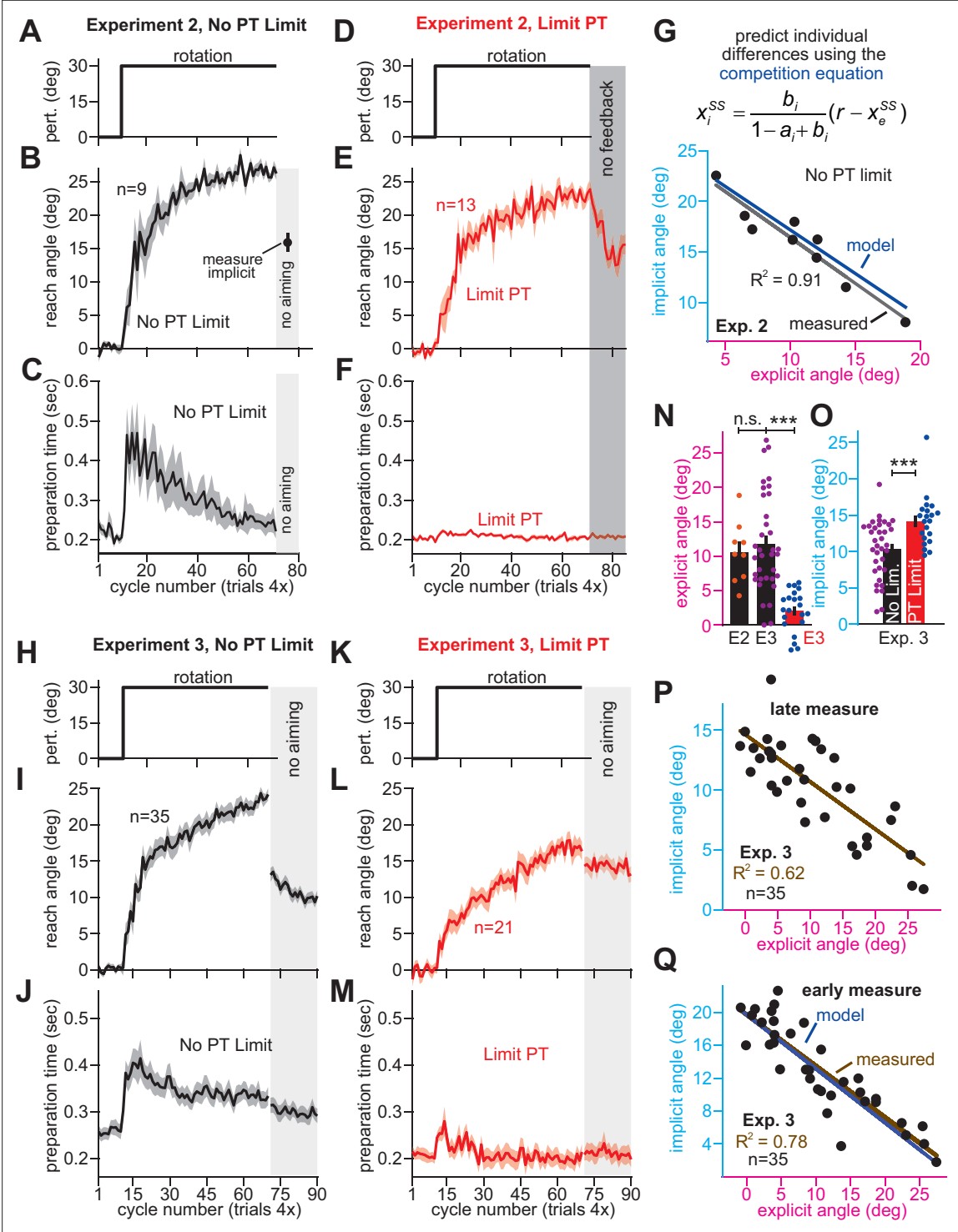

**Figure 3.** Strategy suppresses implicit learning across individual participants. (**A–C**) In Experiment 2, participants in the No PT Limit (no preparation time limit) group adapted to a 30° rotation. The paradigm is shown in **A**. The learning curve is shown in **B**. Implicit learning was measured via exclusion trials (no aiming). Preparation time is shown in **C** (movement start minus target onset). (**D–F**) Same as in **A–C**, but in a limited preparation time condition (Limit PT). Participants in the Limit PT group had to execute movements with restricted preparation time (**F**). The task ended with a prolonged no visual feedback period where memory retention was measured (**E**, gray region). (**G**) Total implicit and explicit adaptation in each participant in the No PT Limit condition (points). Implicit learning measured during the terminal no aiming probe. Explicit learning represents difference between total adaptation (last 10 rotation cycles) and implicit probe. The black line shows a linear regression. The blue line shows the theoretical relationship predicted by the competition equation which assumes implicit system adapts to target error. The parameters for this model prediction (implicit error sensitivity and

*Figure 3 continued on next page*

*Figure 3 continued*

retention) were measured in the Limit PT group. (**H–J**) In Experiment 3, participants adapted to a 30° rotation using a personal computer in the No PT Limit condition. The paradigm is shown in **H**. The learning curve is shown in **I**. Implicit learning was measured at the end of adaptation over a 20-cycle period where participants were instructed to reach straight to the target without aiming and without feedback (no aiming seen in **I**). We measured explicit adaptation as difference between total adaptation and reach angle on first no aiming cycle. We measured 'early' implicit aftereffect as reach angle on first no aiming cycle. We measured 'late' implicit aftereffect as mean reach angle over last 15 no aiming cycles. (**K–M**) Same as in **H–J**, but for a Limit PT condition. (**N**) Explicit adaptation measured in the No PT Limit condition in Experiment 2 (E2), No PT Limit condition in Experiment (E3, black), and Limit PT condition in Experiment 3 (E3, red). (**O**) Late implicit learning in the Experiment 3 No PT Limit group (No Lim.) and Experiment 3 Limit PT group (PT Limit). (**P**) Correspondence between late implicit learning and explicit strategy in the Experiment 3 No PT Limit group. (**Q**) Same as in **G** but where model parameters are obtained from the Limit PT group in Experiment 3, and points represent subjects in the No PT Limit group in Experiment 3. Early implicit learning is used. Throughout all insets, error bars indicate mean ± SEM across participants. Statistics in **N** and **O** are two-sample t-tests: n.s. means p > 0.05, ***p < 0.001.

The online version of this article includes the following source code and figure supplement(s) for figure 3:

**Source code 1.** *Figure 3* data and analysis code.

**Figure supplement 1.** Implicit error sensitivity varies with error.

**Figure supplement 1—source code 1.** *Figure 3—figure supplement 1* data and analysis code.

**Figure supplement 2.** Comparing implicit and explicit adaptation via reported strategies.

**Figure supplement 2—source code 1.** *Figure 3—figure supplement 2* data and analysis code.

**Figure supplement 3.** Movement paths in Experiment 3 were straight and brisk.

**Figure supplement 3—source code 1.** *Figure 3—figure supplement 3* data and analysis code.

To address these limitations, we conducted a laptop-based control experiment (Experiment 3). Participants (n = 35) adapted to a 30° rotation (***Figure 3I***), but this time, we measured implicit adaptation using the no-aiming instruction over an extended 20-cycle period (***Figure 3I***, no aiming). We calculated early (the first no-aiming cycle; ***Figure 3Q***) and late (last 15 no-aiming cycles; ***Figure 3P***) implicit learning measures. Explicit strategy was estimated by subtracting the first no-aiming cycle from total adaptation. Thus, our explicit strategy measure was not calculated using late implicit learning trials; these two measures were no longer spuriously correlated. Regardless, we still observed a strong relationship between explicit strategy and late implicit learning; greater strategy use was associated with reduced late implicit adaptation (***Figure 3P***, $\rho = -0.78$, p< 0.001).

Next, we repeated this experiment, but under limited preparation time conditions in a separate participant cohort (***Figure 3L***, Experiment 3, Limit PT, n = 21). As for the Limit PT group in Exp. 2, we imposed a strict bound on reaction time to suppress movement preparation time (compare ***Figure 3J&M***). Once the rotation period ended, participants were told to stop re-aiming. The decrease in reach angle revealed each participant's explicit strategy (***Figure 3N***). When no reaction time limit was imposed (No PT Limit), re-aiming totaled 11.86° (***Figure 3N***, black). In addition, we did not detect a statistically significant difference in re-aiming across Exps. 2 and 3 (t(42)=0.50, p = 0.621). As in earlier reports (***Leow et al., 2020***; ***Albert et al., 2021***; ***Fernandez-Ruiz et al., 2011***; ***Leow et al., 2017***), limiting reaction time dramatically suppressed explicit strategy, yielding only 2.09° of re-aiming (***Figure 3N***, red). Thus, these data showed that our limited reaction time technique was highly effective at suppressing explicit strategy.

Consistent with the competition theory, suppressing explicit strategy increased implicit learning by approximately 40% (***Figure 3O***, No PT Limit vs. Limit PT, two-sample t-test, t(54)=3.56, p < 0.001, d = 0.98). We again used the Limit PT group's behavior to estimate implicit learning parameters ($a_i$ and $b_i$) as we did in Exp. 2 (***Figure 3G***). Using these parameters, the competition theory (***Equation 4***) predicted that implicit and explicit adaptation should be related by the line: $x_i = 0.658(30 - x_e)$. As in Exp. 2, we observed a striking correspondence between this model (***Figure 3Q***, bottom, model) and the actual implicit-explicit relationship measured in participants in the No PT Limit group (***Figure 3Q***, bottom, points). The slope and bias predicted by ***Equation 4*** (–0.665 and 19.95°, respectively) differed from the measured linear regression by less than 5% (***Figure 3Q***, bottom brown line, $R^2 = 0.78$; slope is –0.63 with 95% CI [-0.74,–0.51] and intercept is 19.7° with 95% CI [18.2°, 21.3°]).

In summary, Exp. 3 provided additional evidence that implicit and explicit systems compete with one another at the individual-participant level. Participants who relied more on strategy exhibited reductions in implicit learning, as predicted by the competition theory. Moreover, by limiting

preparation time on each trial, explicit strategies were strongly suppressed, allowing us to estimate the time course of the implicit system's adaptation.

## Control analyses

Implicit learning exhibits generalization: a decay in adaptation measured when subjects move to positions across the workspace (*Hwang and Shadmehr, 2005*; *Krakauer et al., 2000*; *Fernandes et al., 2012*). Implicit generalization is centered where participants aim (*Day et al., 2016*; *McDougle et al., 2017*). For this reason, implicit learning measured when aiming towards the target, can underapproximate total implicit learning. Subjects that aim more (larger strategy) can exhibit a larger reduction in measured implicit learning. Might this contribute to the negative implicit-explicit correlations in Exps. 1–3?

To test this idea, we compared our data to generalization curves measured in past studies (*Krakauer et al., 2000*; *Day et al., 2016*; *McDougle et al., 2017*; *Figure 4*). Absolute implicit responses are shown in *Figure 4B*, and normalized measures are shown in *Figure 4A* (see Appendix 6.1). Implicit learning in Exps. 2&3 declined 300% more rapidly than predicted by past generalization studies (*Figure 4A&B*). Moreover, this comparison in *Figure 4A–C* is not appropriate under the generalization hypothesis. In Exps. 2&3, explicit strategies are estimated as total learning minus implicit learning. If implicit learning measured at the target underapproximates total implicit learning measured at the aim location, then the explicit strategies we calculate will overapproximate the actual strategy used by each participant. We need to correct these strategies prior to comparing to past generalization curves (Appendix 6.2). The corrected generalization curves (*Figure 4C*, E2 and E3 lines) that produce the patterns in *Figure 4A&B* exhibited an unphysiological narrowing: their standard deviation (width) was 85% smaller than that reported in recent studies (*Krakauer et al., 2000*; *Day et al., 2016*; *McDougle et al., 2017*) ($\sigma$ is about 5.5° versus 37.76° in McDougle et al., see Appendix 6.1). These same issues occurred in the group-level phenomena that we analyzed in *Figures 1 and 2*: no plausible generalization curve could explain the implicit response to instruction, rotation onset (abrupt/gradual), and rotation size (Appendices 6.4 and 6.5). As an example, the variations in implicit learning across abrupt and stepwise groups in Exp. 1 would require a generalization curve that is 90% narrower than recent estimates (*McDougle et al., 2017*) (see Appendix 6.4 and *Figure 4—figure supplement 1*; $\sigma$ = 3.87° versus 37.76° in *McDougle et al., 2017*).

We extended the independence model with implicit generalization and compared its behavior to the competition theory. The competition model is given by $x_i^{ss} = p_i(r - x_e^{ss})$, where $p_i$ is an implicit learning gain. The SPE generalization model is $x_i^{measured} = p_i r g(x_e^{ss})$, where $g(x_e^{ss})$ encodes generalization (derivation in Appendix 6.2). We specified $g(x_e^{ss})$ with *McDougle et al., 2017*. We considered models where $g(x_e^{ss})$ was linear (*Figure 4D–F*, SPE gen. linear) and $g(x_e^{ss})$ was normal (SPE gen. normal). Then we fit each model's $p_i$ to match implicit learning during the 60° stepwise rotation in Exp. 1. We used this gain to predict the implicit-explicit relationship across the three earlier learning periods (B1-B3 in *Figure 4D*). The generalization models yielded poor matches to the held-out data (model RMSE in *Figure 4E*, rm-ANOVA, F(2,72)=13.7, p < 0.001, $\eta_p^2$ = 0.276). Further, a model comparison showed that competition best described individual subject data, minimizing AIC in 84% of stepwise participants (*Figure 4G*, Appendix 6.3). Poor SPE generalization model performance was not due to misestimating generalization curve properties; we conducted a sensitivity analysis in which we varied the generalization curve's width. The competition model was superior across the entire range (*Figure 4H*, Appendix 6.3).

To understand why the competition theory alone generalized across rotation sizes, we fit linear regressions to the data in each rotation period. The regression slopes and 95% CIs are shown in *Figure 4F* (data). Remarkably, the measured implicit-explicit slope appeared to be constant across all rotation sizes. This invariance was directly consistent with the competition theory (*Figure 4F*, competition) which possesses an implicit gain $p_i$ that remains constant across rotations (like the data). But in generalization models (*Figure 4F*, generalization), the gain relating implicit and explicit learning is not constant; it changes as the rotation gets larger (see Appendix 6.3). In sum, data in Exps. 1–3 were poorly explained by an SPE model extended with generalization.

We considered one last control analysis. The competition equation predicts that implicit-explicit correlations are caused by the implicit system's response to variations in strategy. An SPE learning model could create correlations the opposite way: individuals who possess less implicit learning

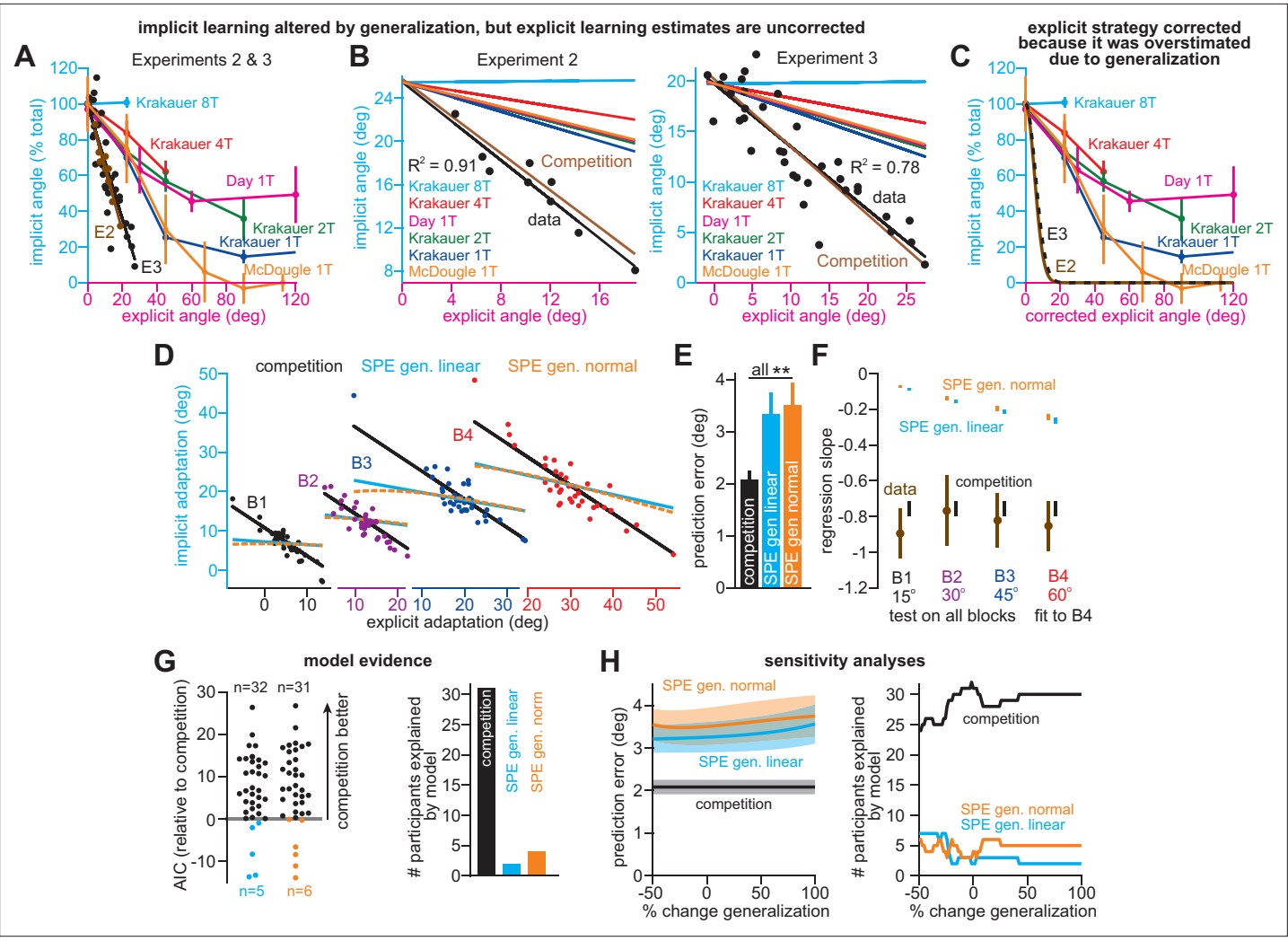

**Figure 4.** Correlations between implicit and explicit learning are consistent with competition, not SPE generalization. (**A**) Aim-centered generalization could create the illusion that implicit and explicit systems compete. To evaluate this possibility, we compared the implicit-explicit relationship in Exps. 2 and 3 to generalization curves reported in *Krakauer et al., 2000*, *Day et al., 2016*, and *McDougle et al., 2017*. The 1T, 2T, 4T, and 8T labels correspond to the number of adaptation targets in Krakauer et al. The gold McDougle et al. curve is particularly relevant because the authors controlled aiming direction on generalization trials and counterbalanced CW and CCW rotations. Data in Exps. 2 and 3 are shown overlaid in the inset. Implicit learning declined about 300% more rapidly with increases in re-aiming than that observed by Day et al. The solid black and brown lines show the competition theory predictions. Implicit learning in Experiments 2 and 3 was normalized to its theoretical maximum, reached when re-aiming is equal to zero. The value used to normalize was determined via linear regression (25.5° in Exp. 2, 19.7° in Exp. 3). (**B**) Same as in **A**, but without normalizing implicit learning. Generalization curves were converted to degrees by multiplying the curves in **A** by the max. implicit learning value in Exp. 2 (25.5°) or Exp. 3 (19.7°). (**C**) The comparisons in **A** and **B** are not correct. Under the generalization hypothesis, each data point's explicit strategy needs to be corrected according to generalization. This inset shows the true implicit-explicit generalization curve that would be required to produce the data in **A** and **B**. The E2 and E3 lines show the Exp. 2 and Exp. 3 curves. (**D**) Points show implicit and explicit learning measured in the stepwise individual participants studied in Exp. 1 (B1 is 15° period, B2 is 30° period, B3 is 45° period, and B4 is 60° period). Three models were fit to participant data in the 60° period. Competition model fit is shown in black. A linear generalization (SPE gen. linear) with slope set by McDougle et al. is shown in cyan. A Gaussian generalization (SPE gen. normal) with width set by McDougle et al. is shown in gold. Since models were fit to B4 data, the B1, B2, and B3 lines represent predicted behavior. (**E**) The prediction error (RMSE) in each model's implicit learning curve across the held-out 15°, 30°, and 45° periods in D. (**F**) Linear regressions fit to each rotation block in (**D**). Brown points and lines (data) show the regression slope and 95% CI. The black (competition), cyan (SPE gen. linear), and gold (SPE gen. normal) are model predictions where lines are 95% CIs estimated via bootstrapping. (**G**) All three models in **D**–**F** were fit to individual participant behavior in the stepwise group. At left, the AIC for each model is compared to that of the competition model. At right, the total number of subjects best captured by each model is shown. (**H**) Same as **E** and **G** but where the generalization width was varied in a sensitivity analysis. We tested values between one-half the McDougle et al. generalization curve (–50%) and twice the McDougle et al. generalization curve (+100%). Error bars in **E** show mean ± SEM. Statistics in **E** are post-hoc tests following one-way rm-ANOVA: **p < 0.01.

The online version of this article includes the following source code and figure supplement(s) for figure 4:

*Figure 4 continued on next page*

*Figure 4 continued*

**Source code 1.** *Figure 4* data and analysis code.

**Figure supplement 1.** Implicit variations are inconsistent with generalization.

**Figure supplement 1—source code 1.** *Figure 4—figure supplement 1* data and analysis code.

**Figure supplement 2.** Differences in generalization across visuomotor rotation tasks.

**Figure supplement 2—source code 1.** *Figure 4—figure supplement 2* data and analysis code.

compensate by increasing their explicit strategy. This scenario can be described by $x_e^{ss} = p_e(r − x_i^{ss})$ where $p_e$ is the explicit response gain. This model has three properties (Appendix 7.2). First, implicit and explicit learning will show a negative relationship (*Figure 5A*). Second, increases in implicit learning will tend to increase total adaptation (*Figure 5C*). Finally, increasing implicit learning leaves smaller errors to drive explicit strategy, resulting in a negative correlation between strategy and total adaptation (*Figure 5B*). While the competition model also predicts negative implicit-explicit correlations (*Figure 5D*), the other pairwise correlations differ (Appendix 7.1). Increases in explicit strategy lead to greater total learning (*Figure 5E*), but reduce the error which drives implicit learning, leading to a negative correlation between implicit learning and total adaptation (*Figure 5F*).

We analyzed these predictions in the No PT Limit group in Exp. 3 (Appendix 7.4). Our observations matched the competition theory; greater explicit strategy was associated with greater total adaptation (*Figure 5G*, $\rho = 0.84$, p < 0.001), whereas greater implicit learning was associated with lower total adaptation (*Figure 5H*, $\rho = −0.70$, p < 0.001). We repeated these analyses in other datasets (Appendix 7.4) that measured implicit learning with no-aiming probe trials: (1) 60° rotation groups (combined across gradual and abrupt groups) in Experiment 1, (2) 60° groups reported by *Maresch et al., 2021* (combined across the CR, IR-E, and IR-EI groups), and (3) 60° rotation group in *Tsay et al., 2021a* These data matched the competition theory: negative implicit-explicit correlations (*Figure 5—figure supplement 1G-I*), positive explicit-total correlations (*Figure 5—figure supplement 1D-F*), and negative implicit-total correlations (*Figure 5—figure supplement 1A-C*).

In summary, while an SPE learning model could exhibit negative correlations between implicit and explicit adaptation, it does not predict a negative correlation between steady-state implicit learning and total adaptation (nor a positive relationship between steady-state explicit strategy and total adaptation), as we observed in the data. The data were consistent with the competition theory, where the implicit system responds to variations in explicit strategy. However, there is a critical caveat. The predictions outlined above assumed that implicit learning properties (contained within $p_i$) are the same across every participant. This is unlikely to be true, and variation in $p_i$ across subjects (e.g. changes in error sensitivity) will undermine some correlations in *Figure 5*, particularly the relationship between implicit learning and total adaptation. This phenomenon and past studies where it appears to occur are treated in Appendix 8.

## Part 2: Competition with explicit learning can mask changes in the implicit learning system

Here, we show that in the competition model, implicit learning may undergo savings, without changing its learning timecourse. Next, we limit preparation time to detect increases and decreases in implicit learning.

### Two ways to interpret the implicit response in a savings paradigm

When participants are exposed to the same perturbation twice, they adapt more quickly the second time. This phenomenon is known as savings and is a hallmark of sensorimotor adaptation (*Smith et al., 2006*; *Herzfeld et al., 2014*; *Zarahn et al., 2008*). Multiple studies have attributed this process solely to changes in explicit strategy (*Haith et al., 2015*; *Huberdeau et al., 2019*; *Morehead et al., 2015*; *Avraham et al., 2021*; *Huberdeau et al., 2015*).

For example, in an earlier work (*Haith et al., 2015*), we trained participants (n = 14) to reach to one of two targets, coincident with an audio tone (*Figure 6A*). By shifting the displayed target approximately 300ms prior to tone onset on a minority of trials (20%), we forced participants to execute movements with limited preparation time (Low preparation time; *Figure 6A*, middle). On all other trials (80%) the target did not switch resulting in high preparation time movements (*Figure 6A*, left).

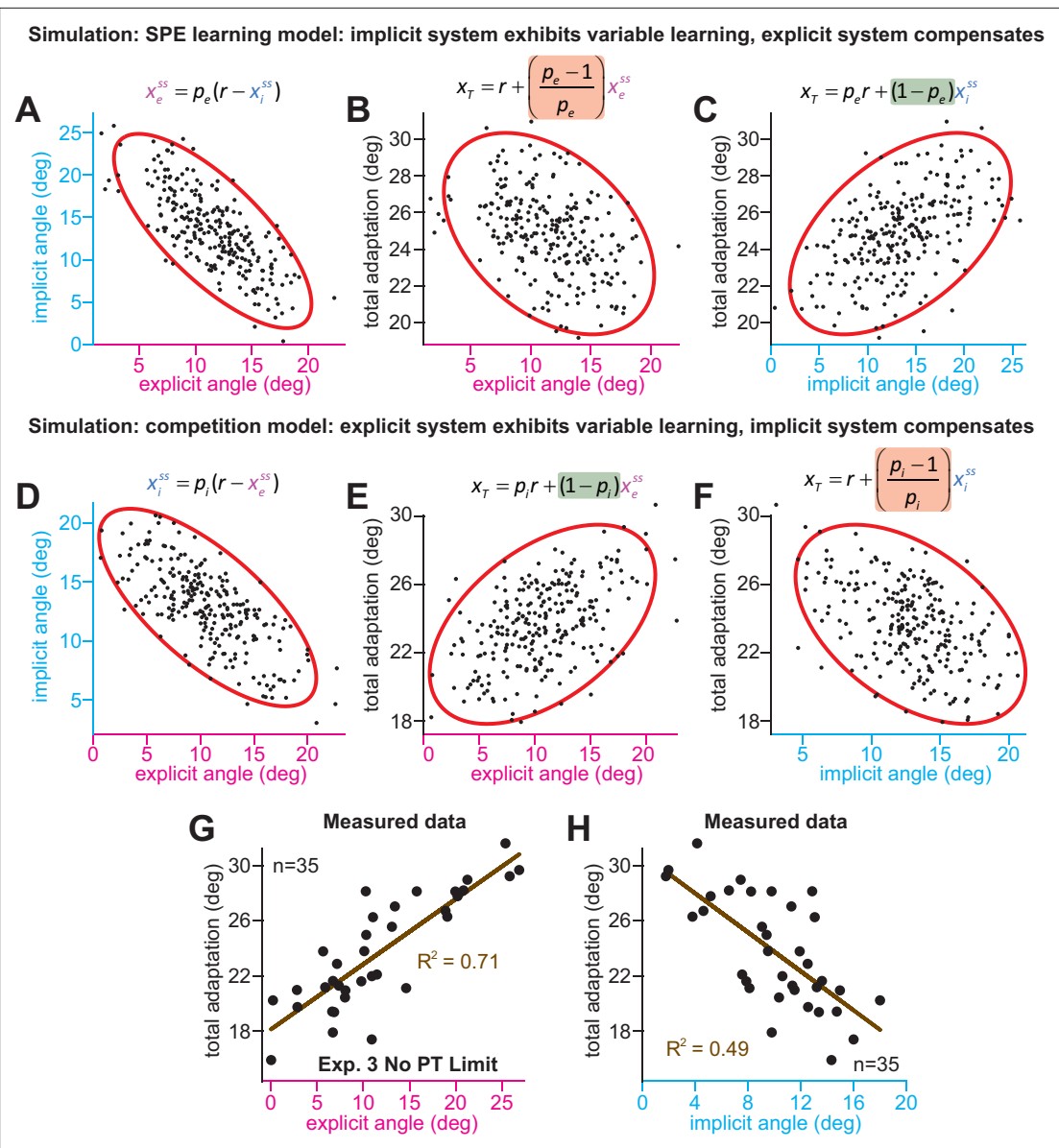

**Figure 5.** Implicit-explicit correlations with total adaptation match the competition theory. The competition equation states that $x_i^{ss} = p_i(r - x_e^{ss})$, where $p_i$ is a scalar learning gain depending on $a_i$ and $b_i$. The competition between steady-state implicit ($x_i^{ss}$) and explicit ($x_e^{ss}$) adaptation predicted by this model is simulated in **D** across 250 hypothetical participants. The model $p_i$ is fit to data in Experiment 3. Total learning is given by $x_T^{ss} = x_i^{ss} + x_e^{ss}$. These two equations can be used to derive expressions relating total learning ($x_T^{ss}$) to steady-state implicit ($x_i^{ss}$) and explicit ($x_e^{ss}$) learning. In **E**, we show that the competition theory predicts a positive relationship between explicit learning and total adaptation (equation at top derived in Appendix 7, green denotes a positive gain). In **F**, we show that the competition theory predicts a negative relationship between implicit learning and total adaptation (equation at top derived in Appendix 7, red shading denotes negative gain). In (**A–C**), we consider an alternative model. Suppose that implicit learning is immune to explicit strategy and varies independently across participants. This is equivalent to the SPE learning model. But in this case, the explicit system could respond to variability in implicit learning via another competition equation: $x_e^{ss} = p_e(r - x_i^{ss})$. Here, $p_e$ is an explicit learning gain (must be less than one to yield a stable system). In **A**, we show the negative relationship between implicit and explicit adaptation predicted by this alternate SPE learning model. In **B**, we show that when the explicit system responds to implicit variability (SPE learning) there is a negative relationship between total adaptation and explicit strategy. The equation at top is derived in Appendix 7. In **C**, we show that the SPE learning model will yield a positive relationship between implicit learning and total adaptation. Equation at top derived in Appendix 7. (**G**) We measured the relationship between explicit strategy and total adaptation in Exp. 3 (No PT Limit group). Total learning exhibits a positive correlation with explicit strategy. (**H**) Same concept as in **G**, but here we show the relationship between total learning and implicit adaptation. The patterns in **G** and **H** are consistent with the competition theory (compare with **E** and **F**).

The online version of this article includes the following source code and figure supplement(s) for figure 5:

*Figure 5 continued on next page*

*Figure 5 continued*

**Source code 1.** *Figure 5* data and analysis code.

**Figure supplement 1.** Relationships between implicit, explicit, and total learning indicate competition.

**Figure supplement 1—source code 1.** *Figure 5—figure supplement 1* data and analysis code.

**Figure supplement 2.** Factors that weaken the correlation between implicit learning and total adaptation.

**Figure supplement 2—source code 1.** *Figure 5—figure supplement 2* data and analysis code.

**Figure supplement 3.** Correlations between explicit learning and total adaptation are more robust to between-subject implicit variability.

**Figure supplement 3—source code 1.** *Figure 5—figure supplement 3* data and analysis code.

**Figure supplement 4.** Variance in implicit learning properties weakens the relationship between implicit learning and total adaptation in the competition theory.

**Figure supplement 4—source code 1.** *Figure 5—figure supplement 4* analysis code.

We measured adaptation to a 30° rotation during high preparation time (*Figure 6B*, left) and low preparation time trials (*Figure 6B*, middle) across two separate exposures (Day 1 and Day 2).

To detect savings, we calculated the learning rate on low and high preparation time trials. Savings appeared to require high preparation time; learning rate increased during the second exposure on high preparation time trials, but not low preparation time trials (*Figure 6B*, right; two-way rm-ANOVA, preparation time by exposure number interaction, F(1,13)=5.29, p = 0.039; significant interaction followed by one-way rm-ANOVA across Days 1 and 2: high prep. time with F(1,13)=6.53, p = 0.024, $\eta_p^2$=0.335; low preparation time with F(1,13)=1.11, p = 0.312, $\eta_p^2$=0.079). To corroborate this rate analysis, we also measured savings via early changes in reach angle (first 5 rotation cycles) across Days 1 and 2 (*Figure 6C*, left and middle). Only high preparation time trials exhibited a statistically significant increase in reach angle, consistent with savings (*Figure 6C*, right; two-way rm-ANOVA, prep. time by exposure interaction, F(1,13)=13.79, p = 0.003; significant interaction followed by one-way rm-ANOVA across days: high prep. time with F(1,13)=11.84, p = 0.004, $\eta_p^2$=0.477; low prep. time with F(1,13)=0.029, p = 0.867, $\eta_p^2$=0.002).

Because explicit strategies can be suppressed by limiting movement preparation time under some conditions (*Huberdeau et al., 2019*; *Fernandez-Ruiz et al., 2011*; *McDougle and Taylor, 2019*), in our initial study we interpreted these data to mean that savings relied solely on time-consuming explicit strategies. Multiple studies have reached similar conclusions (*Haith et al., 2015*; *Huberdeau et al., 2019*; *Morehead et al., 2015*; *Avraham et al., 2021*; *Huberdeau et al., 2015*), suggesting that the implicit learning system is not improved by multiple exposures to a rotation.

However, the competition theory provides an alternate possibility: changes in the implicit learning system may occur but are hidden because of competition with explicit learning. To show this unintuitive phenomenon, we fit the competition model to individual participant behavior under the assumption that low preparation time trials relied solely on implicit adaptation, but high preparation time trials relied on both implicit and explicit adaptation. The model generated implicit (*Figure 6D*, blue) and explicit (*Figure 6D*, magenta) states that tracked the behavior well on high preparation time trials (*Figure 6D*, solid black line) and also low preparation time trials (*Figure 6D*, dashed black line).

Next, we considered the implicit and explicit error sensitivities estimated by the model, which are commonly linked to changes in learning rate (*Coltman et al., 2019*; *Mawase et al., 2014*; *Lerner et al., 2020*; *Albert et al., 2021*; *Herzfeld et al., 2014*). The model unmasked a surprising possibility: even though savings was observed only on high preparation time trials, but not low preparation time trials (*Figure 6B&C*), the model suggested that both the implicit and explicit systems exhibited a statistically significant increase in error sensitivity (*Figure 6D*, right; two-way rm-ANOVA, within-subject effect of exposure number, F(1,13)=10.14, p = 0.007, $\eta_p^2$=0.438; within-subject effect of learning process, F(1,13)=0.051, p = 0.824, $\eta_p^2$=0.004; exposure by learning process interaction, F(1,13)=1.24, p = 0.285).

In contrast, a model where the implicit system adapted to SPEs as opposed to target errors (the independence model) suggested that only the explicit system exhibited a statistically significant increase in error sensitivity (*Figure 6E*; two-way rm-ANOVA, learning process (i.e. implicit vs explicit) by exposure interaction, F(1,13)=7.016, p = 0.02; significant interaction followed by one-way rm-ANOVA across exposures: explicit system, F(1,13)=9.518, p = 0.009, $\eta_p^2$=0.423; implicit system, F(1,13)=2.328, p = 0.151, $\eta_p^2$=0.152).

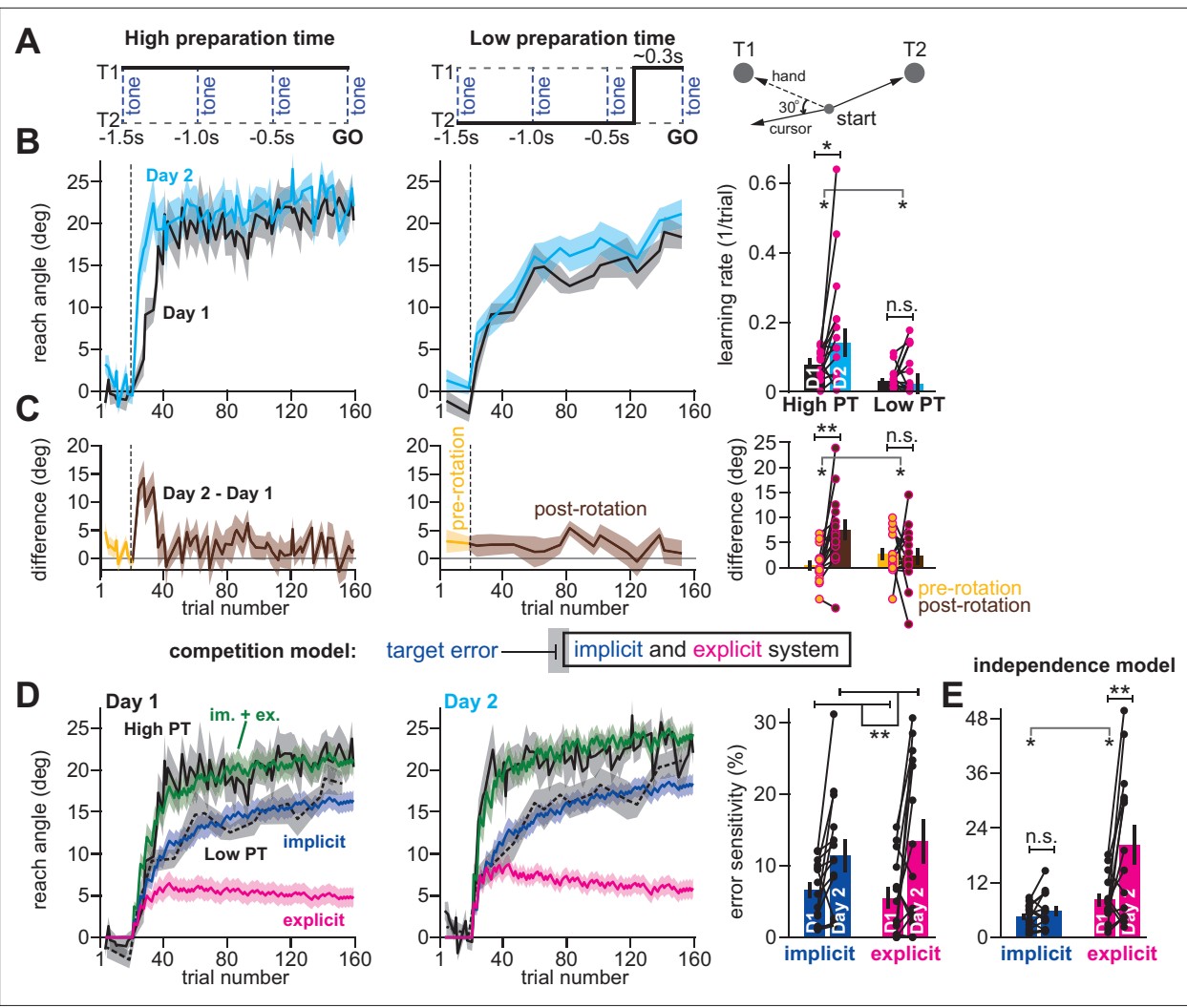

**Figure 6.** Competition predicts changes in implicit error sensitivity without changes in implicit learning rate. (**A**) *Haith et al., 2015* instructed participants to reach to Targets T1 and T2 (right). Participants were exposed to a 30° visuomotor rotation at Target T1 only. Participants reached to the target coincident with a tone. Four tones were played with a 500ms inter-tone-interval. On most trials (80%) the same target was displayed during all four tones (left, High preparation time or High PT). On some trials (20%) the target switched approximately 300ms prior to the fourth tone (middle, Low preparation time or Low PT). (**B**) On Day 1, participants adapted to a 30° visuomotor rotation (Day 1, black) followed by a washout period. On Day 2, participants again experienced a 30° rotation (Day 2, blue). At left, we show the reach angle expressed on High PT trials during Days 1 and 2. Dashed vertical line shows perturbation onset. At middle, we show the same but for Low PT trials. At right, we show learning rate on High and Low PT trials, during each block. (**C**) As an alternative to the rate measure shown at right in **B**, we calculated the difference between reach angle on Days 1 and 2. At left and middle, we show the learning curve differences for High and Low PT trials, respectively. At right, we show difference in learning curves before and after the rotation. 'Pre-rotation' shows the average of Day 2 – Day 1 prior to rotation onset. 'Post-rotation' shows the average of Day 2 – Day 1 after rotation onset. (**D**) We fit a state-space model to the learning curves in Days 1 and 2 assuming that target errors drove implicit adaptation. Low PT trials captured the implicit system (blue). High PT trials captured the sum of implicit and explicit systems (green). Explicit trace (magenta) is the difference between the High and Low PT predictions. At right, we show error sensitivities predicted by the model. (**E**) Same as in **D**, but for a state-space model where implicit learning is driven by SPE, not target error. Model-predicted error sensitivities are shown. Error bars across all insets show mean ± SEM, except for the learning rate in **B** which displays the median. Two-way repeated-measures ANOVA were used in **B**, **C**, **D**, and **E**. For **B** and **C**, exposure number and preparation time condition were main effects. For **D** and **E** exposure number and learning system (implicit vs explicit) were main effects. Significant interactions in **B**, **C**, and **E** prompted follow-up one-way repeated-measures ANOVA (to test simple main effects). Statistical bars where two sets of asterisks appear (at left and right) indicate interactions. Statistical bars with one centered set show main effects or simple main effects. Statistics: n.s. means p > 0.05, *p < 0.05, **p < 0.01.

The online version of this article includes the following source code for figure 6:

**Source code 1.** *Figure 6* data and analysis code.

In summary, when we reanalyzed our earlier data, the competition and independence theories suggested that our data could be explained by two contrasting hypothetical outcomes. If we assumed that implicit and explicit systems were independent, then only explicit learning contributed to savings, as we concluded in our original report. However, if we assumed that the implicit and explicit systems learned from the same error (competition model), then both implicit and explicit systems contributed to savings. Which interpretation is more parsimonious with measured behavior?

## Competition with explicit strategy can alter measurement of implicit learning

The idea that implicit error sensitivity can increase without any change in implicit learning rate (*Figure 6*) is not intuitive. What the competition model suggests is that when the explicit system increases its learning rate as in *Figure 6D*, it leaves a smaller target error to drive implicit learning. However, despite this decrease in target error, low preparation time learning was similar on Days 1 and 2 (*Figure 6B*). Because we assumed that low preparation time learning relied on the implicit system, the competition theory required that the implicit system must have experienced an increase in error sensitivity to counterbalance the reduction in target error magnitude. In other words, though increase in implicit error sensitivity did not increase total implicit learning, it still contributed to savings. That is, had implicit error sensitivity remained the *same*, low preparation time learning would *decrease* on Day 2, and *less* overall savings would occur.

To understand how our ability to detect changes in implicit adaptation can be altered by explicit strategy we constructed a competition map (*Figure 7A*). Imagine that we want to compare behavior across two timepoints or conditions. *Figure 7A* shows how changes in implicit error sensitivity (x-axis) and explicit error sensitivity (y-axis) both contribute to measured implicit aftereffects (denoted by map colors), based on the competition equation (note that the origin denotes a 0% change in error sensitivity relative to Day 1 adaptation in *Haith et al., 2015*). The left region of the map (cooler colors) denotes combinations of implicit and explicit changes that decrease implicit adaptation. The right region of the map (hotter colors) denotes combinations that increase implicit adaptation. The middle black region represents combinations that manifest as a perceived invariance in implicit adaptation ( < 5% absolute change in implicit adaptation).

This map defines several distinct areas (*Figure 7B*). Region A denotes a 'matching' decrease between implicit adaptation and error sensitivity; total implicit learning will decline across two separate learning periods due to a reduction in implicit error sensitivity. Region D is similar. Here, total implicit learning will increase across two separate learning periods due to an increase in implicit error sensitivity.

The other regions show less intuitive cases. In Region B, there is a 'mismatching' change in total implicit learning and implicit error sensitivity; here total implicit learning decreases even though implicit error sensitivity has increased or stayed the same. Likewise, in Region E, total implicit learning will increase across two separate learning periods, though implicit error sensitivity has decreased or stayed the same.

Indeed, we have already described these cases in *Figure 2*. For example, by enhancing the explicit system via coaching (*Figure 2A–C*), implicit learning decreased. This scenario is equivalent to moving up the y-axis of the map (*Figure 7C*, top). The same implicit system will decrease its output (*Figure 7C*, bottom) when normal levels of explicit strategy are increased (*Figure 7C*, middle). On the other hand, suppressing explicit strategy by gradually increasing the rotation (*Figure 2D–G*), or limiting reaction time (*Figure 3N&O*), increased implicit learning without changing any implicit learning properties. This scenario is equivalent to moving down the y-axis of the competition map (*Figure 7D*, top). The same implicit system will increase its output (*Figure 7D*, bottom) when normal levels of explicit strategy are then suppressed (*Figure 7D*, middle).

Now, let us consider the savings experiment in *Figure 6*. The competition theory predicted (*Figure 6D*) that explicit error sensitivity increased by approximately 70.6% during the second exposure, whereas the implicit system's error sensitivity increased by approximately 41.5% (*Figure 7E*, middle). These changes in implicit and explicit adaptation describe a single point in the competition map, denoted by the gray circle in *Figure 7E* (top). This experiment occupies Region C, which indicates that despite the 41.5% increase in implicit error sensitivity, the total implicit learning will increase by less than 5% (*Figure 7E*, bottom). In other words, the competition model suggests the possibility

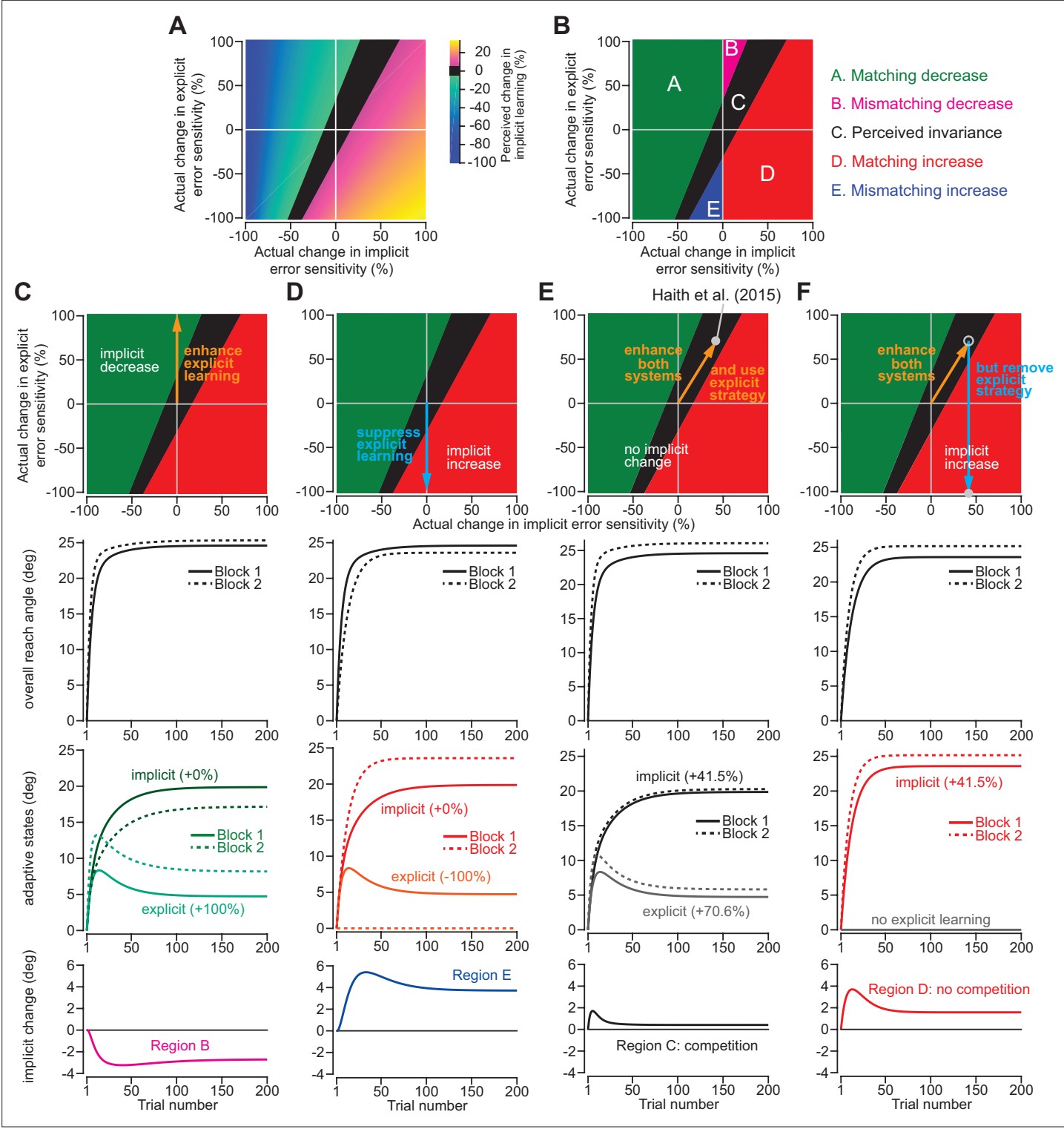

**Figure 7.** Changes in implicit adaptation depend on both implicit and explicit error sensitivity. (**A**) Here we depict the competition map. The x-axis shows change in implicit error sensitivity between reference and test conditions. The y-axis shows change in explicit error sensitivity. Colors indicate the percent change in implicit adaptation (measured at steady-state) from the reference to test conditions. Black region denotes an absolute change less than 5%. The map was constructed with *Equation 8*. (**B**) The map can be described in terms of five different regions. In Region A (matching increase), implicit error sensitivity and total implicit adaption both increase in test condition. Region D is same, but for decreases in error sensitivity and total adaptation. In Region B (mismatching decrease), implicit learning decreases though its error sensitivity is higher or same. In Region E (mismatching increase), implicit learning increases though its error sensitivity is lower or same. Region C shows a perceived invariance where implicit

*Figure 7 continued on next page*

*Figure 7 continued*

adaptation changes less than 5%. (**C**) Row 1: effect of enhancing explicit learning. Row 2: total learning increases. Row 3: implicit and explicit learning shown in Blocks 1 and 2, where only difference is 100% increase in explicit error sensitivity. Row 4: change in implicit learning (Block 2–1). (**D**) Row 1: effect of suppressing explicit learning. Row 2: total learning decreases. Row 3: implicit and explicit learning shown in Blocks 1 and 2, where explicit error sensitivity decreases 100%. Row 4: implicit learning change (Block 2–1). (**E**) Row 1: model simulation for *Haith et al., 2015*. Row 2: Total learning increases. Row 3: implicit and explicit learning during Blocks 1 and 2 where implicit error sensitivity increases by 41.5% and explicit error sensitivity increases by 70.6%. Row 4: negligible change in implicit learning (Block 2–1). (**F**) Same as in **E** except here explicit strategy is suppressed during Blocks 1 and 2.

The online version of this article includes the following source code for figure 7:

**Source code 1.** *Figure 7* analysis code.

that implicit learning improved between Exposures 1 and 2, but this change was hidden by a dramatic increase in explicit strategy (which suppressed implicit learning during Exposure 2).

To test this prediction, we can suppress explicit adaptation, thus eliminating competition (*Figure 7F*, middle). Such an intervention would move our experiment from Region C to Region D (*Figure 7F*, top) where we will observe greater change in the implicit process (*Figure 7D*, bottom). We examined this possibility in a new experiment.

## Savings in implicit learning is unmasked by suppression of explicit strategy

In Exp. 4 (*Figure 8*), participants experienced two 30° rotations, separated by washout trials with veridical feedback (mean reach angle over last three washout cycles was 0.55 ± 0.47°, one-sample t-test against zero, $t(9)=1.16$, $p = 0.28$; not shown in *Figure 8*). To suppress explicit strategy, we restricted reaction time on every trial, which in Exp. 3, greatly reduced explicit learning (*Figure 3N*; re-aiming decreases from 12° to about 2°). Under these reaction time constraints, participants exhibited reach latencies around 200ms (*Figure 8B*, top).

While occasionally limiting preparation time prevented savings in *Haith et al., 2015* (*Figure 8A*, low preparation time on 20% of trials), inhibiting strategy use on every trial in Experiment 4 yielded the opposite outcome (*Figure 8B*). Low preparation time learning rates increased by more than 80% in Experiment 4 (*Figure 8C* top; mixed-ANOVA exposure number by experiment type interaction, $F(1,22)=5.993$, $p = 0.023$; significant interaction followed by one-way rm-ANOVA across exposures: Haith et al. with $F(1,13)=1.109$, $p = 0.312$, $\eta_p^2=0.079$; Experiment 4 with $F(1,9)=5.442$, $p = 0.045$, $\eta_p^2=0.377$). Statistically significant increases in reach angle were detected immediately following rotation onset in Experiment 4 (*Figure 8B*, bottom), but not our earlier data (*Figure 8C*, bottom; mixed-ANOVA exposure number by experiment interaction, $F(1,22)=4.411$, $p = 0.047$; significant interaction followed by one-way rm-ANOVA across exposures: Haith et al. with $F(1,13)=0.029$, $p = 0.867$, $\eta_p^2=0.002$; Experiment 4 with $F(1,9)=11.275$, $p = 0.008$, $\eta_p^2=0.556$).

In sum, when explicit learning was inhibited on every trial, low preparation time behavior showed savings (*Figure 8B*). But when explicit learning was inhibited less frequently, low preparation time behavior did not exhibit a statistically significant increase in learning rate (*Figure 8A*). The competition theory provided a possible explanation; that an implicit system expressible at low preparation time exhibits savings, but these changes in implicit error sensitivity can be masked by competition with explicit strategy.

However, the savings we measured at limited preparation time may not be solely due to changes in implicit learning, but also cached explicit strategies (*Huberdeau et al., 2019*; *McDougle and Taylor, 2019*). Indeed, when we limited preparation time in Exp. 3, participants still exhibited a small decrease (2.09°) in reach angle when we instructed them to stop aiming (*Figure 3L*, no aiming; *Figure 3N and E*, red). These small residual strategies could have contributed to the 8° reach angle measured early during the second rotation in Exp. 4 (*Figure 8C*, implicit difference, no comp.).

What that said, the 'aiming angle' we measured in the Limit PT group in Exp. 3, may overestimate the extent to which participants can use explicit strategy in our limited preparation time paradigm. That is, the decrease in reach angle we observed when participants were told to stop aiming (*Figure 3L*, no aiming) may be due to time-based decay in implicit learning (*Neville and Cressman, 2018*; *Maresch et al., 2021*) over the 30 s instruction period, as opposed to a voluntary reduction in strategy.

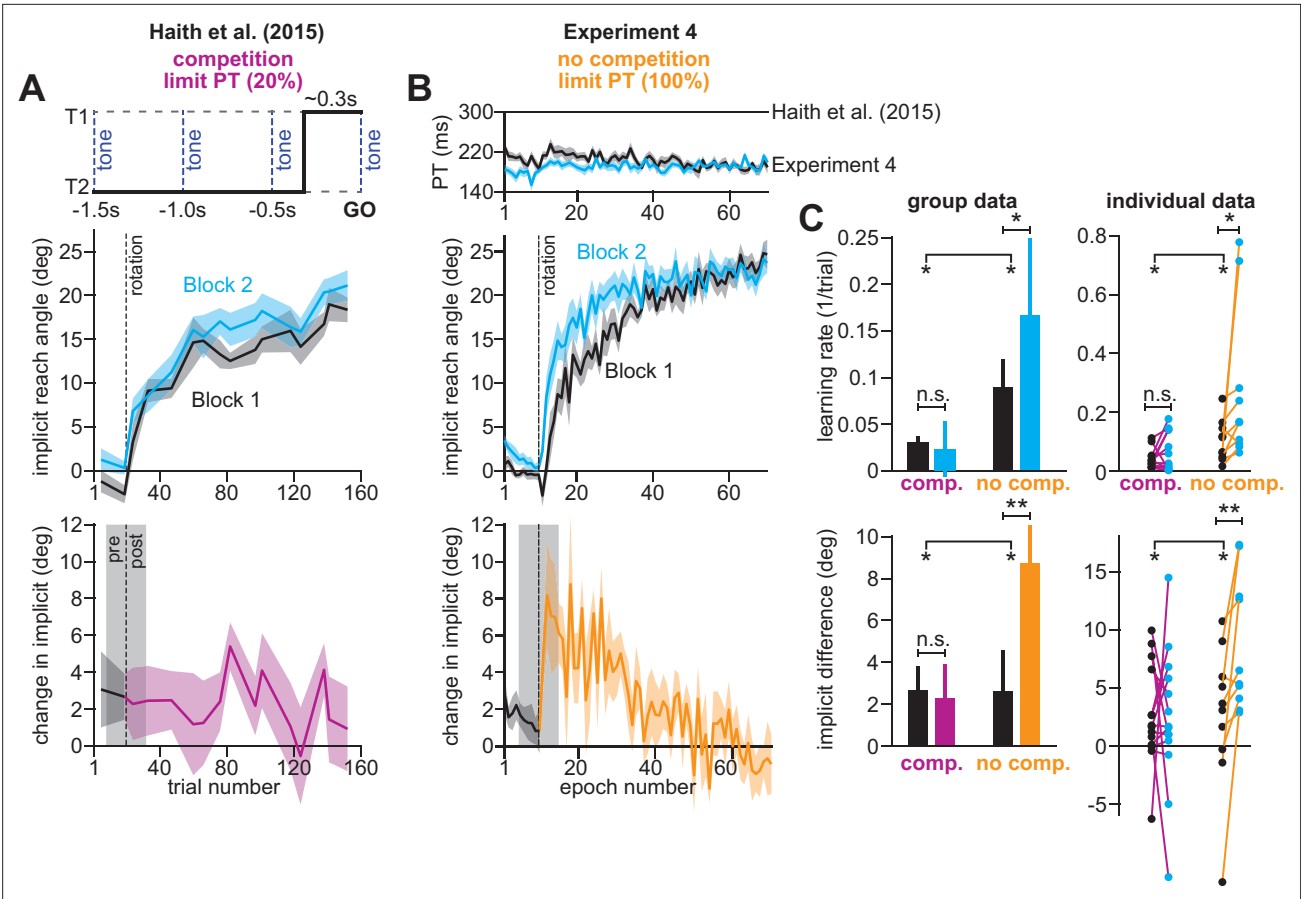

**Figure 8.** Removing explicit strategy reveals savings in implicit adaptation. (**A**) Top: Low preparation time (Low PT) trials in *Haith et al., 2015* used to isolate implicit learning. Middle: learning during Low PT in Blocks 1 and 2. Bottom: difference in Low PT learning between Blocks 1 and 2. (**B**) Similar to **A**, but here (Experiment 4) explicit learning was suppressed on every trial, as opposed to only 20% of trials. To suppress explicit strategy, we restricted reaction time on every trial. The reaction time during Blocks 1 and 2 is shown at top. At middle, we show how participants adapted to the rotation under constrained reaction time. At bottom, we show the difference between the learning curves in Blocks 1 and 2. These two periods were separated by washout cycles with veridical feedback (not shown). (**C**) Here, we measured savings in Haith et al. (20% of trials had reaction time limit) and Experiment 3 (100% of trials had reaction time limit). Top row: we quantify savings by fitting an exponential curve to each learning curve. Data are the rate parameter associated with the exponential. Left column shows group-level data (median). Right column shows individual participants. Bottom row: we quantify savings by comparing how Blocks 1 and 2 differed before perturbation onset (black), and after perturbation onset (purple and yellow). At left, error bars show mean ± SEM. At right, individual participants are shown. Error bars in **A** and **B** indicate mean ± SEM. Statistics in **C** show mixed-ANOVA (exposure number is within-subject factor, experiment type is between-subject factor). Significant interactions were observed both in rate (top) and angular (bottom) savings measure. Follow-up simple main effects were assessed via one-way repeated-measures ANOVA. Statistical bars where two sets of asterisks appear (at left and right) indicate interactions. Statistical bars with a centered set show simple main effects. Statistics: n.s. means p > 0.05, *p < 0.05, **p < 0.01.

The online version of this article includes the following source code and figure supplement(s) for figure 8:

**Source code 1.** *Figure 8* data and analysis code.

**Figure supplement 1.** Limiting preparation time eliminates explicit strategy use.

**Figure supplement 1—source code 1.** *Figure 8—figure supplement 1* data and analysis code.

To test this alternate interpretation, we collected another limited preparation group (n = 12, *Figure 8—figure supplement 1A*, decay-only, black). But this time, participants were instructed that the experiment's disturbance was still on, and that they should continue to move the 'imagined' cursor through the target during the terminal no feedback period. Despite this instruction, reach angles decreased by approximately 2.1° (*Figure 8—figure supplement 1B*, black). Indeed, we detected no statistically significant difference between the change in reach angle in this decay-only group, and the Limit PT group in Experiment 3 (*Figure 8—figure supplement 1B*; two-sample t-test, t(31)=0.016, p = 0.987).

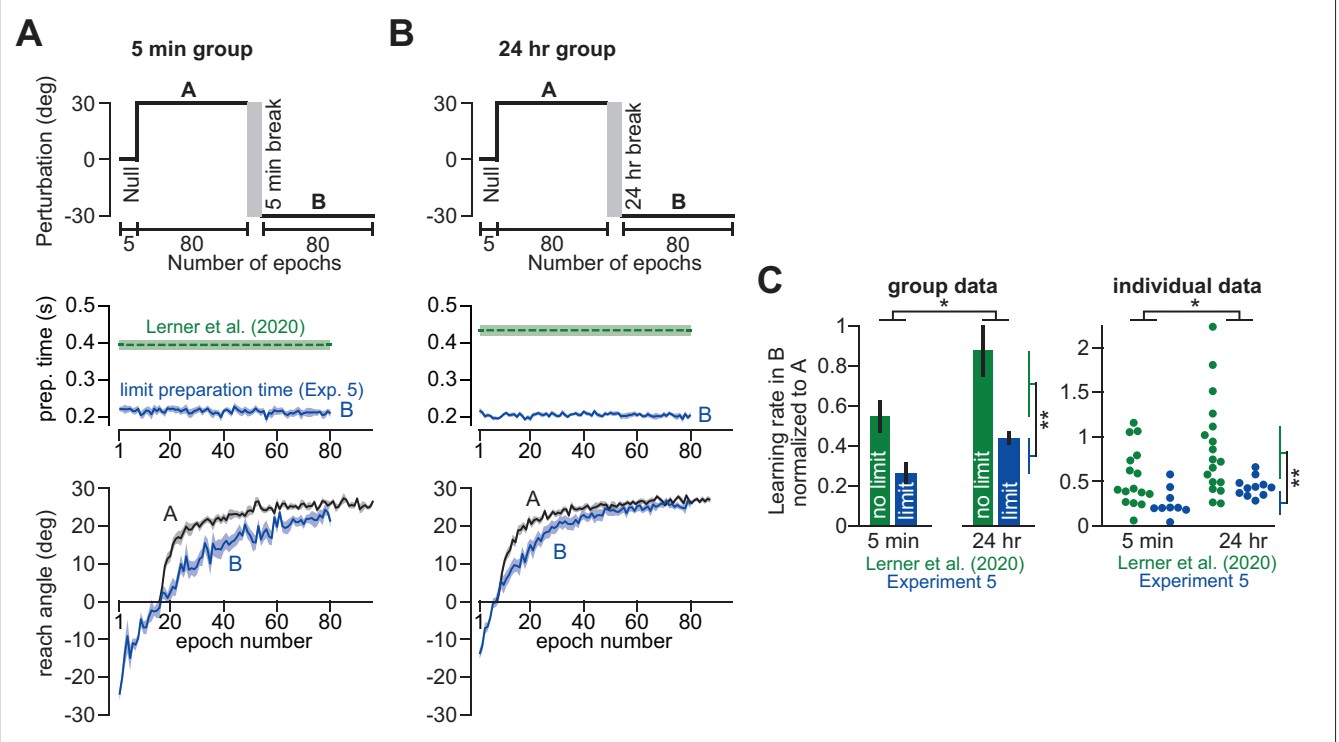

**Figure 9.** Removing explicit strategy reveals anterograde interference in implicit adaptation. (**A**) Top: participants were adapted to a 30° rotation (A). Following a 5-min break, participants were then exposed to a –30° rotation (B). This A-B paradigm was similar to that of *Lerner et al., 2020* Middle: to isolate implicit adaptation, we imposed strict reaction time constraints on every trial. Under these constraints, reaction time (blue) was reduced by approximately 50% over that observed in the self-paced condition (green) studied by *Lerner et al., 2020*. Bottom: learning curves during A and B in Experiment 5; under reaction time constraints, the interference paradigm produced a strong impairment in the rate of implicit adaptation. To compare learning during A and B, B period learning was reflected across y-axis. Furthermore, the curves were temporally aligned such that an exponential fit to the A period and exponential fit to the B period intersected when the reach angle crossed 0°. This alignment visually highlights differences in the learning rate during the A and B periods. (**B**) Here, we show the same analysis as in **A** but when exposures A and B were separated by 24 hr. (**C**) To measure the amount of anterograde interference on the implicit learning system, we fit an exponential to the A and B period behavior. Here, we show the B period exponential rate parameter divided by the A period rate parameter (values less than one indicate a slowing of adaptation). At left, group-level statistics are shown. At right, individual participants are shown. Data in the Limit PT (limited preparation time) condition in Experiment 5 are shown in blue. Data from Lerner & Albert et al. (no preparation time limit) are shown in green. A two-way ANOVA was used to test for differences in interference (preparation time condition (i.e. experiment type) was one between-subject factor, time-elapsed between exposures (5 min vs 24 hr) was the other between-subject factor). Statistical bars indicate each main effect. Statistics: *p < 0.05, **p < 0.01. Error bars in each inset show mean ± SEM.

The online version of this article includes the following source code for figure 9:

**Source code 1.** *Figure 9* data and analysis code.

This control experiment suggested that 'explicit strategies' we measured in the Limit PT condition were more likely caused by time-dependent decay in implicit learning. Indeed, our Limit PT protocol may eliminate explicit strategy. This additional analysis lends further credence to the hypothesis that savings in Experiment 4 was primarily due to changes in the implicit system rather than cached explicit strategies.

### Impairments in implicit learning contribute to anterograde interference

Exp. 4 suggested that the implicit system can exhibit savings. We next wondered whether these changes are bidirectional: can the implicit learning rate decrease? When subjects learn two opposing perturbations in sequence, their adaptation slows due to another hallmark of adaptation, anterograde interference.

In Experiment 5, we exposed two groups of participants to opposing visuomotor rotations of 30° and –30° in sequence (Experiment 5). In one group, the perturbations were separated by a 5-min break (*Figure 9A*). In a second group, the break was 24 hr in duration (*Figure 9B*). We inhibited explicit strategies by strictly limiting reaction time. Under these constraints, participants executed

movements at latencies near 200ms (*Figure 9A&B*, middle, blue). These reaction times were approximately 50% lower than those observed when no reaction time constraints were imposed on participants, as in our earlier work (*Lerner et al., 2020*; *Figure 9A&B*, middle, green).

To assess changes in low preparation time learning, we measured the adaptation rate during each rotation period. In addition, we re-analyzed the adaptation rates obtained in our earlier work (*Lerner et al., 2020*) where participants were tested in a similar paradigm but without any reaction time constraints. While both low preparation time and high preparation time trials exhibited decreases in learning rate which improved with the passage of time (*Figure 9C*; two-way ANOVA, main effect of time delay, $F_{(1,50)}=5.643$, $p = 0.021$, $\eta_p^2=0.101$), these impairments were greatly exacerbated by limiting preparation time (*Figure 9C*; two-way ANOVA, main effect of preparation time, $F_{(1,50)}=11.747$, $p = 0.001$, $\eta_p^2=0.19$). This result was unrelated to initial differences in error across rotation exposures; we obtained analogous results (see Materials and methods) when learning rate was calculated after the 'zero-crossing' in reach angle (two-way ANOVA, main effect of time delay, $F_{(1,50)}=4.23$, $p = 0.045$, $\eta_p^2=0.067$; main effect of prep. time, $F_{(1,50)}=8.303$, $p = 0.006$, $\eta_p^2=0.132$).

Thus, inhibiting explicit strategy via preparation time constraints revealed a strong and sustained anterograde deficit in implicit learning. Under normal reaction time conditions, adaptation rates were less impaired, suggesting that explicit strategies may have partially compensated and masked lingering deficits in the implicit system's sensitivity to error.

## Part 3: Limitations of the competition theory

The competition theory assumes that learning in the implicit system is driven by only one error. Here we show that this single error hypothesis is unlikely to be true in every condition. To demonstrate the theory's limitations, we examine two earlier studies and speculate how the theory might be extended to account for these more sophisticated behaviors.

### The implicit system may adapt to multiple target errors at the same time

In *Mazzoni and Krakauer, 2006*, we tested two sets of participants. In a no-strategy group, participants adapted to a standard 45° rotation (*Figure 10A*, blue, no-strategy, adaptation) followed by washout (*Figure 10A*, blue, no-strategy, washout). In a second group, participants made two initial movements with the rotation (*Figure 10A*, red, strategy, 2 movements no instruction). Then we coached subjects to aim toward a neighboring target (45° away) which entirely compensated for the rotation. Participants adopted the aiming strategy, bringing the primary target error to zero (*Figure 10A*, red, strategy, instruction). Curiously, even though the primary target error had now been eliminated, reaching movements gradually drifted beyond the primary target, overcompensating for the rotation. These involuntary changes implicated an implicit process.

When we compared the rate of learning with and without strategy in *Mazzoni and Krakauer, 2006*, we found that it was not different during the initial exposure to the perturbation (*Figure 10B*, gray, mean adaptation over rotation trials 1–24, Wilcoxon rank sum, $p = 0.223$). This statistical test led us to conclude in Mazzoni and Krakauer, that implicit adaptation was driven by a sensory prediction error that did not depend on the primary target and was not altered by explicit strategy.

However, there remained an unsolved puzzle. While the initial rates of adaptation were the same irrespective of strategy, adaptation diverged later in learning (*Figure 10B*, compare strategy and no-strategy curves after initial gray region; two-sample t-test, $p < 0.005$), with the no-strategy group exhibiting a larger aftereffect (see aftereffect in *Figure 10C*; two-sample t-test, $p < 0.005$). Might these late differences have been caused by participants in the strategy group abandoning their explicit strategy as it led to larger and larger errors? This possibility seemed unlikely. When we asked participants to stop using their aiming strategy and to move instead toward the primary target (*Figure 10A*, do not aim rotation on) their movement angle changed by 47.8° (difference between three movements before and three movements after instruction), indicating that they had continued to maintain the instructed explicit re-aiming strategy near 45°.

We wondered if interactions between implicit and explicit learning could help solve this puzzle. First, we considered the competition model that best described the experiments in *Figures 1–7*. In this model, the implicit system is driven exclusively by error with respect to the primary target (*Equation 1*) (*Figure 10D*, top, $e_1$). While this model predicted learning in the standard no-strategy condition, it failed to account for the drift observed when participants were given an explicit strategy (*Figure 10D*,

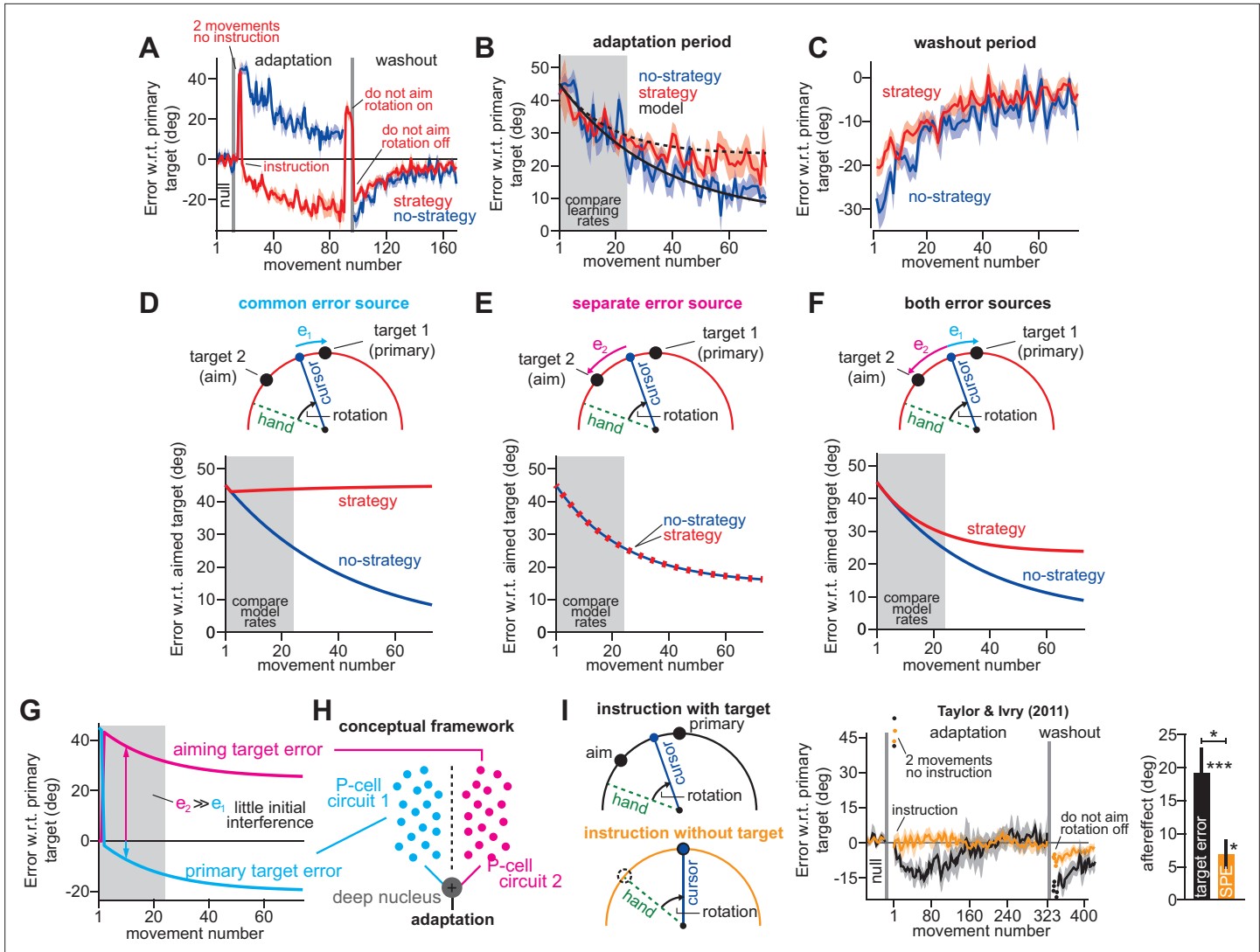

**Figure 10.** Two visual targets create two implicit error sources. (**A**) Data reported in *Mazzoni and Krakauer, 2006*. Blue shows error between primary target and cursor during adaptation and washout. Red shows the same, but in a strategy group that was instructed to aim to a neighboring target (instruction) to eliminate target errors, once participants experienced two large errors (two cycles no instruction). (**B**) The error between the cursor and the aimed target during the adaptation period. These curves are the same as in **A** except we use the aimed target rather than primary target, so as to better compare learning curves across groups. (**C**) The washout period reported in **A**. Here, error is relative to primary target, though in this case aimed and primary targets are the same. (**D**) We modeled behavior when implicit learning adapts to primary target errors $e_1$. Note that the no-strategy learning group resembles data. However, strategy learning exhibits no drift because the implicit system has zero error. Note here that the primary target error of 0° is a 45° aimed target error in the strategy group. (**E**) Similar to **D**, except here the implicit system adapts to errors between the cursor and aimed target, termed $e_2$. (**F**) In this model, the strategy group adapts to both the primary target error and the aimed target error ($e_1$ and $e_2$ at top). The no-strategy group adapts only to the primary target error. Learning parameters are identical across groups. (**G**) We show how aiming target and primary target errors evolve in the strategy group in **F**. (**H**) A potential neural substrate for implicit learning. The primary target error and aiming target error engage two different sub-populations of Purkinje cells in the cerebellar cortex. These two implicit learning modules combine at the deep nucleus. (**I**) Data reported in *Taylor and Ivry, 2011*. Before adaptation, subjects were taught to re-aim their reach angles. In the 'nstruction with target' group, participants re-aimed during adaptation with the aid of neighboring aiming targets (top-left). In the 'instruction without target' group, participants re-aimed during adaptation without any aiming targets, solely based on the remembered instruction from the baseline period. The middle shows learning curves. In both groups, the first two movements were uninstructed, resulting in large errors (two movements no instruction). Note in the 'instruction with target' group, there is an implicit drift as in **A**, but participants eventually reverse this by changing explicit strategy. There is no drift in the 'instruction without target' group. At right, we show the implicit aftereffect measured by telling participants not to aim (first no feedback, no aiming cycle post-adaptation). Greater implicit adaptation resulted from physical target. Error bars show mean ± SEM. Statistics: *p < 0.05, ***p < 0.001.

The online version of this article includes the following source code for figure 10:

**Source code 1.** *Figure 10* data and analysis code.

no learning in strategy group). This was not surprising. If implicit learning is driven by the primary target's error, it will not adapt in the strategy group because participants explicitly reduce target error to zero at the start of adaptation (note that 45° in *Figure 10D* means a 0° primary target error).

We next considered the possibility that implicit learning was driven exclusively by an error with respect to the aimed target (target 2, *Figure 10E*, top, $e_2$), as we concluded in our original study (*Mazzoni and Krakauer, 2006*). While this model correctly predicted non-zero implicit learning in the no-strategy and strategy groups, it could not account for any differences in learning that emerged later during the adaptation period (*Figure 10E*, bottom).

Finally, we noted that participants in the strategy group were given two contrasting goals. One goal was to aim for the neighboring target, whereas the other goal was to move the cursor through the primary target (both targets were always visible). Therefore, we wondered if participants in the strategy group learned from two distinct target errors: cursor with respect to target 1, and cursor with respect to target 2 (*Figure 10F*, top). In contrast, participants in the no-strategy group attended solely to the primary target, and thus learned only from the error between the cursor and target 1. Thus, we imagined that implicit learning in the strategy group was driven by two target errors: $e_1$ was cursor with respect to target 1, and $e_2$ was cursor with respect to target 2:

$$x_{i,1}^{(n+1)} = a_i x_{i,1}^{(n)} + b_i e_1^{(n)}$$
$$x_{i,2}^{(n+1)} = a_i x_{i,2}^{(n)} + b_i e_2^{(n)}$$
(6)

These two modules then combined to determine the total amount of implicit learning (i.e. $x_i = x_{i,1} + x_{i,2}$).

Interestingly, when we applied the dual target error model (*Equation 6*) to the strategy group, and the single target error model (*Equations (1) and (3)*) to the no-strategy group, the same implicit learning parameters ($a_i$ and $b_i$) closely tracked the observed group behaviors (black model in *Figure 10B*). These models correctly predicted that initial learning would be similar across the strategy and no-strategy conditions but would diverge later during adaptation (*Figure 10F*). How was this possible?

In *Figure 10G*, we show how the primary target error and aiming target error evolved over time in the instructed strategy group. Initially, strategy reduces primary target error to zero (*Figure 10G*, primary target error). Thus, early in learning, the implicit system is driven predominantly by aiming target error. For this reason, initial learning will appear similar to the no-strategy group which also adapts to only one error. However, as the error with respect to the aimed target decreases, error with respect to the primary target increases but in the opposite direction (*Figure 10G*; see schematic in *Figure 10F*). Therefore, the primary target error opposes adaptation to the aiming target error. This counteracting force causes implicit adaptation to saturate prematurely. Hence, participants in the no-strategy group, who do not experience this error conflict, adapt more.

It is important, however, to note a limitation in these analyses. Our earlier study did not employ the standard conditions used to measure implicit aftereffects: that is instructing participants to aim directly at the target, and also removing any visual feedback. Thus, the proposed model relies on the assumption that differences in washout were primarily related to the implicit system. These assumptions need to be tested more completely in future experiments.

In summary, the conditions tested by Mazzoni and Krakauer show that the simplistic idea that adaptation is driven by only one target error, or only one SPE, cannot be true in general (*Tsay et al., 2021b*). We propose a new hypothesis that when people move a cursor to one visual target, while aiming at another visual target, cursor error with respect to each target contributes to implicit learning. When one target error conflicts with the other target error, the implicit learning system may exhibit an attenuation in total adaptation.

This experiment alone does not reveal the nature of aiming target error. That is, in the strategy group, the error between the aim direction and the cursor is both an SPE, but also a target error (because participants are aiming at a neighboring target). We explore this distinction in the next section.

### The persistence of sensory prediction error, in the absence of target error

Our analysis in *Figure 10A–G* suggested that when participants see two targets, one to aim toward with their hand and one to move the cursor to, the landmarks can act as two different target errors. To what extent do these errors depend on the target's physical presence in the workspace? *Taylor and Ivry, 2011* tested this idea, repeating the instruction paradigm used by Mazzoni and Krakauer, though with nearly four times the number of adaptation trials (*Figure 10I*, instruction with target, black). Interestingly, while the reach angle exhibited the same implicit drift described by Mazzoni and Krakauer, with many more trials participants eventually counteracted this drift by modifying their explicit strategies, bringing their target error back to zero (*Figure 10I*, black). At the end of adaptation, participants exhibited large implicit aftereffects when instructed to stop aiming (*Figure 10I*, right, aftereffect; t(9)=5.16, p < 0.001, Cohen's d = 1.63).

In a second experiment, participants were taught how to re-aim their reach angles during an initial baseline period, but during adaptation itself, they were not provided with physical aiming targets (*Figure 10I*, instruction without target). In this case, only SPEs (not a target error) could drive implicit learning towards the aimed location. Even without physical aiming landmarks, participants immediately eliminated error at the primary target after being instructed to re-aim (*Figure 10I*, middle, yellow). Curiously, without the physical aiming target, these participants did not exhibit an implicit drift in reach angle at any point during the adaptation period and exhibited only a small implicit aftereffect during the washout period (*Figure 10I*, right, t(9)=3.11, p = 0.012, Cohen's d = 0.985). In fact, the aftereffect was approximately three times larger when participants aimed towards a physical target during adaptation than when this target was absent (*Figure 10I*, right, aftereffect; two-sample t-test, t(18)=2.85, p = 0.012, Cohen's d = 0.935).

A target error (competition) model is consistent with some of these results, but not all. The model correctly predicts that when only a single target is present, performance during adaptation will not exhibit a drift, even though people are aiming. However, it does not explain why this condition still leads to the small aftereffect. Further, with two targets, it correctly predicts that adaptation will drift, as in Mazzoni and Krakauer, but it does not explain how this is eliminated late during adaptation; this reversal in drift would seem to indicate a compensatory and gradual reduction in explicit strategy (*Taylor and Ivry, 2011*; *McDougle et al., 2015*; *Taylor et al., 2014*).

Together, the data suggested a remarkable depth to the implicit system's response to error. While implicit learning was greatest in response to target error, removing the physical target still permitted SPE-driven learning, albeit to a smaller degree. Whether this aiming-related error is both a target error and an SPE occurring together, or solely an SPE enhanced by a salient visual stimulus, remains unknown.

## Discussion

Sensorimotor adaptation relies on an explicit process shaped by intention (*Taylor et al., 2014*; *Hwang et al., 2006*), and an implicit process driven by unconscious correction (*Morehead et al., 2017*; *Mazzoni and Krakauer, 2006*; *Kim et al., 2018*). Here, we examined the possibility that these two parallel systems can become entangled when they respond to a common error source: target (i.e. task) error (*Leow et al., 2020*; *Kim et al., 2019*). The data suggested that this coupling resembles a competition by which enhancing the explicit system's response rapidly depletes error, decreasing the driving force for implicit adaptation. Thus, providing instructions on how to reduce errors enhances the explicit system, but comes at the cost of robbing the implicit system of what it needs to adapt.

This simple rule explained why the implicit system can operate in three modes, one that appears insensitive to perturbation magnitude, another that scales with the perturbation's size, and a third that exhibits non-monotonic behavior (*Figure 1*). It also predicted that priming or suppressing explicit awareness can inversely change implicit adaptation (*Figure 2*). As a result, subjects that utilize strategies inadvertently suppress their implicit learning (*Figures 3–5*). This inhibition can continue to the extent that improvements in implicit learning (e.g. savings) are masked by dramatic upregulation in strategic learning (*Figures 6–8*).

The task-error driven implicit system likely exists in parallel with other implicit processes (*Leow et al., 2020*; *Kim et al., 2019*; *Morehead and Orban de Xivry, 2021*). For example, in cases where primary target errors are eliminated, small amounts of implicit adaptation persist (*Figure 10*). These

residual changes are likely due to sensory prediction errors (*Mazzoni and Krakauer, 2006*; *Leow et al., 2020*; *Taylor and Ivry, 2011*; *Kim et al., 2019*) as well as other target errors that remain in the workspace (*Figure 10I*). When these error sources oppose one another, competition between parallel implicit learning modules may inhibit the overall implicit response (*Figure 10A–C*).

In a broader sense, these competitive interactions extend beyond implicit and explicit processes, to other parallel neural circuits that respond to a common error. Changes in one neural circuit's response to error may be indirectly driven, or hidden, by a parallel circuit. Thus, competition may lead to long-range interactions between neuroanatomical regions that subserve separate neural processes. For example, strategic learning systems housed within the cortex (*Shadmehr et al., 1998*; *Milner, 1962*; *Gabrieli et al., 1993*), may exert indirect changes on a subcortical structure like the cerebellum, which is widely implicated in subconscious adaptation (*Tseng et al., 2007*; *Donchin et al., 2012*; *Smith and Shadmehr, 2005*; *Izawa et al., 2012*; *Wong et al., 2019*; *Becker and Person, 2019*; *Morton and Bastian, 2006*).

## Flexibility in the implicit response to error and its contribution to savings

When two similar perturbations are experienced in sequence, the rate of relearning is enhanced during the second exposure (*Haith et al., 2015*; *Coltman et al., 2019*; *Mawase et al., 2014*; *Zarahn et al., 2008*; *Kording et al., 2007*). This hallmark of memory (*MacLeod, 1988*; *Ebbinghaus, 1885*) is referred to as savings, which is often quantified based on differences in the learning curves for each exposure (*Haith et al., 2015*; *Morehead et al., 2015*), or the rate of adaptation (*Kitago et al., 2013*). These conventions are based on an underlying assumption: when a learning system is enhanced, its total adaptation will also change. Here, we showed that this intuition is incorrect.

The state space model (*Smith et al., 2006*; *Albert and Shadmehr, 2018*; *Thoroughman and Shadmehr, 2000*) quantified behavior using two processes: learning and forgetting. This model described savings as a change in sensitivity to error (*Coltman et al., 2019*; *Mawase et al., 2014*; *Herzfeld et al., 2014*). When similar errors are experienced on consecutive trials, the brain becomes more sensitive to their occurrence and responds more strongly on subsequent trials (*Albert et al., 2021*; *Herzfeld et al., 2014*; *Leow et al., 2016*). Generally, as error sensitivity increases, so too does the rate at which we adapt to the perturbation (e.g. High PT trials in *Figure 6*). However, under certain circumstances, changes in one's implicit sensitivity to error may not lead to differences in measured behavior (e.g. Low PT trials in *Figure 6*).

The reason is competition. When strategy is enhanced, it reduces the error available for implicit learning. Therefore, although the implicit system may become more sensitive to error, this increase in sensitivity is canceled out by the decrease in error size.

For example, recent lines of work have suggested that increases in learning rate depend solely on the explicit recall of past actions. Implicit adaptation does not seem to contribute to faster re-learning, whether implicit learning is estimated via reported strategies (*Morehead et al., 2015*), or by intermittently restricting movement preparation time (*Haith et al., 2015*; *Huberdeau et al., 2019*; *Figure 6*). These results suggested that implicit processes do not show savings. Our data suggest a different possibility. When we limited reaction time on all trials in Experiment 4, thus suppressing explicit contributions to behavior, we found that the implicit system exhibited savings (*Figure 8*). The disconnect between studies that have detected changes in both implicit and explicit learning rates (*Leow et al., 2020*; *Yin and Wei, 2020*; *Albert et al., 2021*), versus studies that have only observed changes in explicit learing (*Haith et al., 2015*; *Huberdeau et al., 2019*; *Morehead et al., 2015*; *Avraham et al., 2020*; *Avraham et al., 2021*), can be resolved by the competition equation (*Equation 4*).

The competition equation links steady-state implicit learning to both implicit and explicit learning properties (*Figure 7*). When both implicit and explicit systems become more sensitive to error, the explicit response can hide changes in the implicit response (*Figure 7B*, Region C). Moreover, dramatic enhancement in explicit adaptation could even lead to a decrease in implicit learning, even when implicit error sensitivity has increased (*Figure 7B*, Region B). Indeed, this prediction can explain cases whereby re-exposure to a rotation increases explicit strategies, but can attenuate implicit learning (*Huberdeau et al., 2019*; *Avraham et al., 2021*; *Wilterson and Taylor, 2021*). For example, in a recent study by *Huberdeau et al., 2019*, seven exposures to a rotation dramatically enhanced the

strategic learning system, but simultaneously attenuated implicit learning. Prolonged multi-day exposure to a rotation appears to have a similar outcome (*Wilterson and Taylor, 2021*).

It is critical to distinguish between cases where implicit learning is indirectly reduced by increases in explicit strategy, versus contexts that lead to direct impairments in the implicit system's sensitivity to error. For example, when two opposing perturbations are experienced sequentially, the response to the second exposure is impaired by anterograde interference (*Sing and Smith, 2010*; *Caithness et al., 2004*; *Smith et al., 2006*; *Miall et al., 2004*). Recently, we linked these impairments in learning rate to a transient reduction in error sensitivity which recovers over time (*Lerner et al., 2020*). Here, we limited reaction time to try and isolate the implicit contributions to this impairment. Impairments in learning at low preparation time were long-lasting, persisting even 24 hr, and exceeded those measured at normal movement preparation times (*Figure 9C*). These results suggested that less-inhibited explicit strategies may sometimes compensate, at least in part, for lingering deficits in implicit adaptation (*Leow et al., 2020*; *Huberdeau et al., 2019*). Our analysis in *Figure 9*, however, compares Exp. 5 to our earlier work in *Lerner et al., 2020* where we did not tease apart implicit and explicit learning. Thus, future work needs to test these ideas more carefully.

There is a possible limitation in this interpretation. Recent studies have demonstrated that with multiple exposures to a rotation, explicit responses can be expressed at lower reaction times: a process termed caching (*Huberdeau et al., 2019*; *McDougle and Taylor, 2019*). Thus, changes in low preparation time adaptation commonly ascribed to the implicit system, may be contaminated by cached explicit strategies. This possibility seems unlikely to have altered our results. First, it is not clear why caching would occur in Experiment 4, but not our earlier study in *Haith et al., 2015*; *Figure 8*; these earlier data implied that caching remains limited with only two exposures to a rotation (at least during the initial exposure to the second rotation over which savings was assessed). Nevertheless, to test the caching hypothesis, we measured explicit re-aiming under limited preparation time conditions in Experiment 3. We found that our method restricted explicit re-aiming to only 2°, compared to about 12° in the standard condition (*Figure 3N*). Moreover, this 2° decrement in reach angle was more likely due to forgetting in implicit learning (*Neville and Cressman, 2018*; *Hadjiosif and Smith, 2015*; *Alhussein et al., 2019*; *Hosseini et al., 2017*; *Joiner et al., 2017*; *Zhou et al., 2017*). That is, we a similar 2° decrease in reach angle occurred over the 30 s instruction period, even when participants were not told to stop aiming (*Figure 8—figure supplement 1*). Thus, while it appears that caching played little role in our results, our results should be taken cautiously. It is critical that future studies investigate how caching varies across experimental methodologies, and how cached strategies interact with implicit learning. In addition, such experiments should dissociate these cached explicit responses from associative implicit memories that may be rapidly instantiated in the appropriate context.

## Competition-driven enhancement and suppression of implicit adaptation

The competition theory cautions that increases or decreases in implicit learning do not necessarily imply that the implicit system has altered its response to error. That is, changes in implicit learning may occur indirectly through competition with explicit strategies.

For example, when participants are coached about a visuomotor rotation prior to its onset, their explicit strategies are greatly enhanced (*Neville and Cressman, 2018*; *Benson et al., 2011*). These increases in explicit strategy are coupled to decreases in implicit adaptation (*Figure 2*). A similar phenomenon is observed in other experiments where participants report their strategy using visual landmarks. In such paradigms, increased reporting frequency leads to increased explicit strategy, but decreased implicit learning (*Maresch et al., 2021*; *Bromberg et al., 2019*; *de Brouwer et al., 2018*). Subjects themselves exhibit substantial variations in strategic learning, leading to negative individual-level correlations between implicit and explicit learning (*Neville and Cressman, 2018*; *Benson et al., 2011*; *Fernandez-Ruiz et al., 2011*; *Figure 3*).

The competition theory helps to reveal the input that drives implicit learning. This competitive relationship (*Equation 4*) naturally arises when implicit systems are driven by errors in task outcome (*Equation 1*). We can observe these negative interactions not solely when enhancing explicit strategy, but also when suppressing re-aiming. For example, in cases where perturbations are introduced gradually, thus reducing conscious awareness, implicit "procedural" adaptation appears to increase (*Yin*

and Wei, 2020; Saijo and Gomi, 2010; Kagerer et al., 1997; Figure 2, Figure 2—figure supplement 3, and Appendix 5). Similarly, when participants are required to move with minimal preparation time, thus suppressing time-consuming explicit re-aiming (Haith et al., 2015; Fernandez-Ruiz et al., 2011; McDougle and Taylor, 2019), the total extent of implicit adaptation also appears to increase (Figure 3O; Albert et al., 2021; Fernandez-Ruiz et al., 2011).

Although the implicit system varies with experimental conditions, a common phenomenon is its invariant response to changes in rotation size (Morehead et al., 2017; Neville and Cressman, 2018; Bond and Taylor, 2015; Tsay et al., 2021a; Kim et al., 2018). For example, in the (Neville and Cressman, 2018) data examined in Figure 1, total implicit learning remained constant despite tripling the rotation's magnitude. While this saturation in implicit learning is sometimes due to a restriction in implicit adaptability (Morehead et al., 2017; Kim et al., 2018), in other cases this rotation-insensitivity may have another cause entirely: competition. That is, when rotations increase in magnitude, rapid scaling in the explicit response may prevent increases in total implicit adaptation. In the competition theory, implicit learning is driven not by the rotation, but by the residual error that remains between the rotation and explicit strategy. Thus, when we used gradual rotations to reduce explicit adaptation (Experiment 1), prior invariance in the implicit response was lifted: as the rotation increased, so too did implicit learning (Salomonczyk et al., 2011; Figure 1I). The competition theory readily described these two implicit learning phenotypes: saturation and scaling (Figure 1G&L). Furthermore, it also provided insight as to why implicit learning can even exhibit a non-monotonic response, as in Tsay et al., 2021a.

With that said, changes in implicit learning occur not solely due to error-based competition, but also variations in implicit learning properties such as error sensitivity. For example, Morehead et al., 2017 show that total implicit learning paradoxically decreases when rotations exceed about 90°. A possible cause is error sensitivity, which declines as errors become larger (Kim et al., 2018; Marko et al., 2012; Wei and Körding, 2009). Because no aiming was permitted in their study, steady-state errors were >80°, which would dramatically reduce error sensitivity. Reductions in error sensitivity could contribute to the non-monotonic phenotype we described in Tsay et al. (2021). On the other hand, Tsay et al. permitted aiming, so steady-state errors were only about 5° in the 90° rotation group. These residual errors would not be associated with dramatic reduction in error sensitivity, so error-based competition seems a more likely mechanism (Figure 1Q). In addition, note that total implicit learning varies strongly with error but not error sensitivity; the implicit learning gain $p_i = b_i(1 - a_i + b_i)^{-1}$, responds weakly to changes in $b_i$ (see Appendix 4). Thus, large changes in total implicit learning are much more likely driven by a competition for error, than by changes in implicit error sensitivity (Appendix 6.6 provides additional comparisons between Morehead et al. and Tsay et al.).

In addition, there may be other ways to cast the adaptation model, that also produce competition between implicit learning and explicit strategy. Here, implicit and explicit systems are treated as parallel states that adapt to the same error. A recent inference-based model of motor adaptation (Heald et al., 2021) suggests the possibility that implicit and explicit systems participate in a credit assignment problem: with the explicit state estimating the external perturbation, and implicit state estimating the mismatch between vision and proprioception. This inference model will also produce a competition because both states attempt to sum to total state feedback. When more credit is assigned to the external perturbation, explicit adaptation will increase, and implicit adaptation will decrease. All in all, this model will produce similar phenotypes to the competition equation, given that they both describe a competitive learning process.

## Variations in individual learning unveil competition between implicit and explicit processes

Individuals exhibit substantial variation in how they adapt to rotations (Miyamoto et al., 2020; Fernandez-Ruiz et al., 2011; Tsay et al., 2021d). For example, in Experiments 1–3, we observed that individuals who relied more on explicit strategy inadvertently suppressed their own implicit learning. In one prime example, Miyamoto et al., 2020 exposed participants to sum-of-sines rotations. Curiously, participants with more vigorous explicit responses to the perturbation exhibited less vigorous implicit learning. In a second case, Fernandez-Ruiz et al., 2011 observed that increases in movement preparation time helped participants adapt more rapidly, but led to reductions in aftereffects. As a third example, when Bromberg et al., 2019 measured eye movements during adaptation,

participants who tended to look toward their re-aiming locations not only exhibited greater explicit strategies, but less implicit adaptation.

These results suggest that a subject's strategy suppresses their implicit learning (*Fernandez-Ruiz et al., 2011*). To explain these individual-level correlations, *Miyamoto et al., 2020* suggested that there may be an intrinsic relationship between implicit and explicit sensitivity to error: when an individual's explicit error sensitivity is high, their implicit error sensitivity is low. Here, our results describe another way to account for a similar observation (*Figure 3*). In Exps. 2 and 3, we used the competition equation (*Equation 4*) to predict an individual's implicit adaptation from their measured explicit strategy, assuming each participant had the same sensitivity to error. This equation could accurately predict the negative relationship between implicit and explicit learning. Thus, negative individual-level correlations between implicit and explicit adaptation can arise from variation in strategy, even under an extreme scenario where implicit error sensitivity is constant across participants.

There are alternate ways that such negative correlations between implicit and explicit learning might arise. For example, here we described an implicit-centered competition equation where explicit strategies suppress implicit learning. The opposite is also possible; implicit learning might be immune to explicit strategy, but strategies respond to variation in implicit learning. These contrasting possibilities both predict negative relationships between implicit learning and explicit strategy but diverge in how total adaptation should vary with implicit and explicit states (*Figure 5*). When we tested these ideas in Experiment 3, our data were highly consistent with the competition model: increases in total learning were associated with greater strategy, but less implicit learning (*Figure 5G&H*). We observed similar phenomena across three additional studies (*Figure 5—figure supplement 1*). Thus, in cases where implicit learning is dominated by target errors, greater total adaptation may be supported by less implicit learning. Note, however, that negative correlations at the individual-level are more nuanced. Variation in implicit learning properties will weaken the relationship between implicit learning and total adaptation (Appendices 7 and 8). Further, in conditions with enhanced SPE learning (e.g. multiple visual landmarks), these correlations can easily be invalidated.

These results imply that implicit learning responds to variations in explicit strategy, but strategies are immune to implicit learning. A similar phenomenon was noted by *Miyamoto et al., 2020*, using structural equation modeling. This unidirectional causality, however, is not true in general. For example, early during learning, it is common that explicit strategies increase, peak, and then decline. That is, when errors are initially large, strategies increase rapidly. But as implicit learning builds, the explicit system's response can decline in a compensatory manner (*Taylor and Ivry, 2011*; *McDougle et al., 2015*; *Taylor et al., 2014*). This dynamic phenomenon can also occur in the competition theory, where both implicit and explicit systems respond to target error (*Figure 6D*). But in many cases, a second error source may drive this behavioral phenotype. That is, in cases with aiming landmarks (*Taylor and Ivry, 2011*; *McDougle et al., 2015*; *Taylor et al., 2014*), errors between the cursor and primary target can be eliminated, but implicit learning persists. This implicit learning is likely driven by SPEs and target errors that remain between the cursor and aiming landmark (*Taylor and Ivry, 2011*). Persistent implicit learning is counteracted by decreasing explicit strategy to avoid overcompensation. In sum, competition between implicit learning and explicit strategy is complex. Both systems can respond to one another in ways that change with experimental conditions.

## Comparisons to invariant error-clamp experiments

The competition and independence models described here apply solely to standard visuomotor rotations where target errors decrease throughout the adaptation process. Another popular visuomotor paradigm is an invariant error-clamp: experiments where the target error is fixed to a constant value, noncontingent on the participant's movement. In this paradigm, implicit adaptation reaches a ceiling whose value varies somewhere between 15 degrees (*Morehead et al., 2017*) and 25 degrees (*Kim et al., 2018*) and does not change with rotation size. It is important not to conflate this rotation-invariant saturation, with the implicit saturation phenotype we explored with the competition model in our *Neville and Cressman, 2018* analysis (*Figure 1G*). The ceiling in the invariant error-clamp paradigm appears to be due to an upper bound on implicit corrections (*Kim et al., 2018*). The saturation phenotype in *Figure 1G* is due to implicit competition with explicit strategy.

In invariant error-clamp studies, there is no explicit strategy. In such a case, the competition and independence models are equivalent. However, the models encoded in *Equations (4) and (5)* only

describe the standard rotation learning conditions considered in our Results. When there is no explicit strategy, these models predict implicit learning via: $x_i^{ss} = b_i(1-a_i+ b_i)^{-1}$ r. In an error-clamp study, however, the correct model would be $x_i^{ss} = b_i(1-a_i)^{-1}$ r. These equations differ in their implicit learning gains: $b_i(1-a_i)^{-1}$ for constant error-clamp and $b_i(1-a_i+ b_i)^{-1}$ for standard rotations. This has critical implications. For example, in an error-clamp condition, for $a_i = 0.98$ and $b_i = 0.3$, the state-space model predicts an implicit steady-state of 15 times the imposed rotation, $r$. In other words, implicit adaptation would need to exceed the rotation size by at least an order of magnitude to reach its steady-state; a 5° error-clamp would require 75° of implicit learning to reach a dynamic steady-state and a 30° rotation would require 450°. In sum, error-clamp rotations require implicit learning that cannot reach the dynamic steady-state described by the state-space model. For these reasons, the steady-states reached in error-clamp studies are likely caused by another mechanism: the ceiling effect shown in *Morehead et al., 2017* and *Kim et al., 2018*. However, in a standard rotation, implicit learning must be less than the rotation size (proportional to difference between rotation size and explicit strategy: proportionality constant between 0.6 and 0.8 in the data sets we consider here). Under these conditions, the dynamic steady-state described by the competition model is attainable.

Now, a separate but related question, is what causes the implicit system's upper limit and does it vary across experimental settings. We suspect it does. For example, in *Morehead et al., 2017* the implicit system was limited to 10–15° learning, but in *Kim et al., 2018* this limit increased to 20–25°. It may be that these limits relate to a reliance on proprioceptive error signals (*Tsay et al., 2021c*): implicit learning may be 'halted' by some unknown mechanism when the hand deviates too far from the target. This would make sense, as participants are told to move their hand straight to the target and ignore the cursor in this paradigm. In standard rotation paradigms, however, visual errors between the cursor and target may dominate this proprioceptive signal, extending the implicit system's capacity. This might explain why some studies have observed implicit learning levels (e.g. about 35° in *Salomonczyk et al., 2011*, and even 45° in *Maresch et al., 2021*) which greatly exceed the error-clamp limits observed in Morehead et al. and Kim et al.

A critical puzzle that remains, however, is savings. The savings in implicit adaptation observed in Exp. 4 (*Figure 8*) contrasts with error-clamp behavior (*Avraham et al., 2021*), where implicit learning decreases during the second exposure. We can only speculate why these phenotypes differ. The discrepancy may relate to a divergence in goals. In error-clamp studies, the overall objective is to move straight to the target: to not change one's reach angle. In standard rotation studies, the objective is to move the cursor to the target: to change one's reach angle. This goal could play a role in enhancing or suppressing the implicit system's response; some utility associated with adapting more rapidly may be necessary to obtain savings. On the other hand, responses to visual errors may be suppressed over time during error-clamp, as they are irrelevant to the arm's motion. Interestingly, interacting with the visual target in error-clamp does appear to attenuate the implicit response to the rotation (*Kim et al., 2019*).

A second idea relates to the reward system. *Sedaghat-Nejad and Shadmehr, 2021* has shown that saccade adaptation is accelerated when learning improves task success. Learning speeds up when this leads to an increase in reward probability. Adaptation rates are not improved when learning does not impact reward probability. In other words, a higher level 'desire' to obtain reward may be needed to increase learning rate. Again, such motivation is clear in standard rotation experiments where adaptation will improve task success and reward probability. There is no motivation to adapt more rapidly in error-clamp paradigms; participants are never rewarded. Moreover, as noted above, hitting the target in invariant error-clamp paradigms appears to attenuate the implicit response (*Kim et al., 2019*). Interestingly, a link between reward and savings may be present in the cerebellum. Several studies (*Medina, 2019*) have shown that both granule cell layers (*Wagner et al., 2017*) and climbing fiber inputs (*Heffley et al., 2018*; *Kostadinov et al., 2019*) carry reward-related signals to the cerebellum. Thus, it may be that the cerebellum, a potential locus for implicit adaptation (*Tseng et al., 2007*; *Donchin et al., 2012*; *Smith and Shadmehr, 2005*; *Izawa et al., 2012*; *Wong et al., 2019*; *Becker and Person, 2019*; *Morton and Bastian, 2006*) responds to errors differently when rewards are not attainable (*Morehead et al., 2017*; *Avraham et al., 2021*; *Kim et al., 2018*) such as error-clamp paradigms, versus conventional rotations where more rapid learning promotes reward acquisition. These ideas are speculative and remain to be tested.

Overall, our data suggest that some implicit learning properties may vary across standard rotation and error-clamp paradigms. Considerable future work is needed to better compare these paradigms and test the suppositions outlined above.

## The relationship between competition and implicit generalization

One potential limitation in our analyses relates to implicit generalization. Earlier studies have shown that implicit learning generalizes around the reported aiming direction (*Day et al., 2016*; *McDougle et al., 2017*). Thus, participants who aim further away from the target may show smaller implicit adaptation when asked to 'move straight to the target'. While generalization could have contributed to the negative implicit-explicit correlation, its role would be small relative to competition. In earlier studies (*Figure 4B&C*), implicit learning decayed only 5° or so with 22.5°–30° changes in aiming (*Figure 4—figure supplement 2B* shows 22.5° re-aiming, *McDougle et al., 2017*; *Figure 4—figure supplement 2A* shows 30° re-aiming, *Day et al., 2016*). However, in Exps. 1–3, we observed between 15°–20° changes in implicit learning (see *Figure 4B&C*) over similar ranges in explicit strategy. Thus, generalization-based decay in implicit learning would need to occur over 300% more rapidly than earlier reports to match our data.

Critically, in Exps. 1–3 explicit strategy was estimated as total adaptation minus implicit learning. Had generalization reduced the implicit measures, it would falsely inflate our explicit measures. While it is tempting to compare our data in *Figure 2* or *Figure 4* with past generalization curves, this should not be done without correcting the explicit strategy measures. These corrections revealed that implicit generalization would need to exhibit an implausible narrowing to explain our group-level (e.g., response to stepwise rotation, or instruction) and individual-level results (Appendices 6.1–6.5, *Figure 4C*, and *Figure 4—figure supplement 1*). Altogether, generalization is not a viable alternative to the competition theory.

Generalization may have played a smaller role in the studies we analyzed, because participants trained with 2 (*Tsay et al., 2021a*), 3 (Exp. 1, *Neville and Cressman, 2018*), 4 (Exps. 2–4), 8 (Exp. 5, *Maresch et al., 2021*), or 12 (*Saijo and Gomi, 2010*) targets. Past studies that measured plan-based generalization, only used one training target (*Day et al., 2016*; *McDougle et al., 2017*; *Figure 4—figure supplement 2A and B*). Thus, decreases in implicit learning would likely be smaller in our studies, because the generalization curve widens with additional training targets (*Krakauer et al., 2000*; *Tanaka et al., 2009*). For, example, in *Neville and Cressman, 2018*, subjects trained with three targets. Given the targets' geometries, 2 had coincided with the neighboring target's aim direction, but one did not. A narrow generalization curve would predict a larger aftereffect for the targets that coincided with aim directions, yet no variations in implicit learning were detected across targets (see their supplementary analyses).

Note that unlike past generalization studies (*Day et al., 2016*; *McDougle et al., 2017*), we did not use aiming reports to measure explicit strategy (*Day et al., 2016*; *McDougle et al., 2017*). We speculate this may play a role in generalization, given that aiming landmarks themselves drive implicit learning (*Taylor and Ivry, 2011*; *Figure 10*). For example, past generalization studies observed a discrepancy between exclusion-based implicit learning and report-based implicit learning: the exclusion measures were smaller due to plan-based generalization (*Figure 4—figure supplement 2C*). But in Exp. 2, the opposite occurred. Exclusion-based implicit learning was larger than implicit learning estimated with reporting (*Figure 4—figure supplement 2E*). The same phenomenon was noted by Maresch and colleagues (*Maresch et al., 2021*) in a condition where reporting was used sparsely during adaptation (*Figure 4—figure supplement 2D*).

Exp. 1 provided a direct way to test how our data may have been impacted by generalization. In Exp. 1, a 60° rotation resulted in 22° of implicit learning, whereas a 15° rotation caused about 7° (*Figure 1I*). Suppose that implicit learning exhibits about 20% generalization-based decay with a 15° change in aiming direction as in *McDougle et al., 2017*. This decay causes a (0.2)(22°) = 4.4° decrease in implicit learning in the 60° rotation, but only a (0.2)(7) = 1.4° in the 15° rotation (i.e., 7° implicit learning). Thus, the absolute change in implicit learning driven by generalization depends on total implicit learning achieved at steady-state, or in Exp. 1, the rotation's size. This is not true in the competition theory: *Equation 4* predicts that the gain relating implicit and explicit adaptation does not depend on rotation size. We tested these diverging predictions in *Figure 4D–F*. Critically, behavior matched the competition theory (*Figure 4F*). AIC indicated that the competition model

better described participant behavior than SPE learning models extended with plan-based generalization (*Figure 4G&H*, Appendix 6.3).

With that said, while the generalization hypothesis did not match important patterns in our data, it remains a very important phenomena that may alter implicit learning measurements. It is imperative that implicit generalization is more thoroughly examined to determine how it varies across experimental methodologies. These data will be needed to accurately evaluate the competitive relationship between implicit and explicit learning.

## Error sources that drive implicit adaptation

*Mazzoni and Krakauer, 2006* exposed participants to a visuomotor rotation, but also provided instructions for how to re-aim their hand to achieve success. While participants immediately used this strategy to move the cursor through the target, the elimination of task error failed to stop implicit adaptation. These data suggested that the implicit system responded to errors in the predicted sensory consequence of their actions (*Tseng et al., 2007*; *Shadmehr et al., 2010*), rather than errors in hitting the target.

However, such a model, where implicit systems learn solely based on the angle between aiming direction and the cursor (*Equation 2*), could not account for the implicit-explicit interactions we observed in our data (*Figures 1–5*). These interactions could only be described by an implicit error source that is altered by explicit strategy, such as the angle between the cursor and the target (*Equation 1*). For example, in Experiments 2 and 3, participants did not aim straight to the target, but rather adjusted their aiming angle by 5–20° (*Figure 3*). These changes in re-aiming appeared to alter implicit adaptation via errors between the cursor and the target. This target-cursor error source (*Equation 1*) appeared to provide an accurate account of short-term visuomotor adaptation across a number of studies (*McDougle et al., 2015*; *Miyamoto et al., 2020*; *Albert et al., 2021*; *Neville and Cressman, 2018*; *Benson et al., 2011*; *Fernandez-Ruiz et al., 2011*; *Saijo and Gomi, 2010*).

We do not mean to suggest, however, that implicit adaptation is solely driven by a single target error. In fact, there are many cases where this idea fails (*Leow et al., 2020*; *Taylor and Ivry, 2011*; *Taylor et al., 2014*). We speculate that one feature which alters implicit learning is the simultaneous presence of multiple visual targets. In *Figures 1–9*, there was only one visual target on the screen at a time. However, in Mazzoni and Krakauer (*Figure 10*), there were two important visual targets: the adjacent target towards which participants explicitly aimed their hand, and the original target toward which the cursor should move. In theory, the brain could calculate errors with respect to both targets. When we considered the idea that the implicit system adapted to both errors at the same time, we could more completely account for these earlier data (*Figure 10F*).

The idea that both kinds of visual error (cursor with respect to the primary target, and cursor with respect to the aimed target) drive implicit learning, could account for other surprising observations. For example, in cases where landmarks are provided to report explicit aiming (*McDougle et al., 2015*; *Taylor et al., 2014*; *Day et al., 2016*), target-cursor error is often rapidly eliminated, but implicit adaptation persists. A dual-error model (*Equation 6*) would explain this continued adaptation based on persistent aim-cursor error. In other words, aiming landmarks may continue to drive adaptation even when primary target errors have been eliminated.

However, the nature of aim-cursor errors remains uncertain. For example, while this error source generates strong adaptation when the aim location coincides with a physical target (*Figure 10I*, instruction with target), implicit learning is observed even in the absence of a physical aiming landmark (*Taylor and Ivry, 2011*; *Figure 10I*, instruction without target), albeit to a smaller degree. This latter condition may implicate SPE learning that does not require an aiming target. Thus, it may be that the aim-cursor error in Mazzoni and Krakauer is actually an SPE that is enhanced by the presence of a physical target. In this view, implicit learning is driven by a target error module and an SPE module that is enhanced by a visual target error (*Leow et al., 2020*; *Kim et al., 2019*; *Leow et al., 2018*).

These various implicit learning modules are likely strongly dependent on experimental contexts, in ways we do not yet understand. For example, *Taylor and Ivry, 2011* would suggest that all experiments produce some implicit SPE learning, but less so in paradigms with no aiming targets. Yet, the competition equation accurately matched single-target behavior in *Figures 1–9* without an SPE learning module. It is not clear why SPE learning would be absent in these experiments. One idea may be that the aftereffect observed by *Taylor and Ivry, 2011* in the absence of an aiming target,

was a lingering associative motor memory that was reinforced by successfully hitting the target during the rotation period. Indeed, such a model-free learning mechanism (*Huang et al., 2011*) should be included in a more complete implicit learning model. It is currently overlooked in error-based systems such as the competition and independence equations.

Another idea is that some SPE learning did occur in the no aiming target experiments we analyzed in *Figures 1–9* but was overshadowed by the implicit system's response to target error. A third possibility is that the SPE learning observed by *Taylor and Ivry, 2011* was contextually enhanced by participants implicitly recalling the aiming landmark locations provided during the baseline period. This possibility would suggest SPEs vary along a complex spectrum: (1) never providing an aiming target causes little or no SPE learning (as in our experiments), (2) providing an aiming target during past training allows implicit recall that leads to small SPE learning, (3) providing an aiming target that disappears during the movement promotes better recall and leads to medium-sized SPE learning (i.e. the disappearing target condition in Taylor and Ivry), and (4) an aiming target that always remains visible leads to the largest SPE learning levels. This context-dependent SPE hypothesis may be related to recent work suggesting that target errors and SPEs drive implicit learning, but SPEs are altered by distraction (*Tsay et al., 2021b*).

We speculate that the cerebellum might play an important role in supporting multiple implicit learning modules (*Smith and Shadmehr, 2005*; *Wong et al., 2019*; *Hanajima et al., 2015*; *Kojima and Soetedjo, 2018*; *Bastian et al., 1996*). Current models propose that complex spikes in Purkinje cells (P-cells) in the cerebellar cortex cause LTD (Marr-Albus-Ito hypothesis). These complex spikes are reliably evoked by olivary input in response to a sensory error (*Kojima and Soetedjo, 2018*; *Herzfeld et al., 2018*; *Herzfeld et al., 2015*). However, different P-cells are activated by different error directions, thus organizing P-cells into error-specific subpopulations (*Herzfeld et al., 2018*; *Herzfeld et al., 2015*). Therefore, our model suggests that two different sources of error might simultaneously transduce learning in two different P-cell subpopulations, which then combine their adapted states into a total implicit correction at the level of the deep nuclei. Thus, errors based on the original target, and the aiming target, might simultaneously activate two implicit learning modules in the cerebellum (*Figure 10H*).

Alternatively, it is equally possible that these aim-cursor errors and target-cursor errors engage separate brain regions both inside and outside the cerebellum. In this view, an interesting possibility is that patients with cerebellar disorders (*Tseng et al., 2007*; *Gabrieli et al., 1993*; *Izawa et al., 2012*; *Maschke et al., 2004*; *Martin et al., 1996*) may have learning deficits specific to one error but not the other, as recent results suggest (*Wong et al., 2019*). These possibilities remain to be fully tested.

## Materials and methods

Our work involves reevaluation of earlier literature; this includes data from *Haith et al., 2015* in *Figures 6 and 8*, data from *Lerner et al., 2020* in *Figure 9*, data from *Neville and Cressman, 2018* in *Figures 1 and 2*, data from *Saijo and Gomi, 2010* in *Figure 2—figure supplement 3*, data from *Mazzoni and Krakauer, 2006* in *Figure 10*, data from *Taylor and Ivry, 2011* in *Figure 10*, data from *McDougle et al., 2017* in *Figure 4*, data from *Day et al., 2016* in *Figure 4*, data from *Tsay et al., 2021a* in *Figure 1*, data from *Maresch et al., 2021* in *Figure 5—figure supplement 1*, and data from *Krakauer et al., 2000* in *Figure 4*. Relevant details for all studies are summarized in the sections below alongside the new data collected for this work (Exps. 1–5). Note that some methods are described in Appendices 1–8.

### Participants

Here we report the sample sizes used in past studies analyzed here: Haith and colleagues (*Haith et al., 2015*) (n = 14), *Lerner et al., 2020* (n = 16 for 5 min group, n = 18 for 24 hr group), *Neville and Cressman, 2018* (no strategy: n = 11 for 20°, n = 10 for 40°, n = 10 for 60°; strategy: n = 10 for 20°, n = 11 for 40°, n = 10 for 60°), *Mazzoni and Krakauer, 2006* (n = 18), *Saijo and Gomi, 2010* (n = 9 for abrupt, n = 9 for gradual), *Maresch et al., 2021* (n = 40 across the CR, IR-E, and IR-EI groups), *Tsay et al., 2021a* (n = 25/rotation size), McDougle et al. (n = 15), and *Taylor and Ivry, 2011* (n = 10 for instruction with visual target, n = 10 for instruction without visual target).

All volunteers (ages 18–62) in Experiments 1–5 were neurologically healthy and right-handed. Experiment 1 included n = 36 participants in the abrupt group (12 Male, 24 Female), n = 37 participants in the stepwise group (6 Male, 30 Female, 1 opted to not report). Experiment 2 included n = 9 participants (5 Male, 4 Female) in the No PT Limit group and included n = 13 participants (6 Male, 7 Female) in the Limit PT group. Experiment 3 included n = 35 participants in the No PT Limit group (7 Male, 14 Female), n = 21 participants in the Limit PT group (20 Male, 15 Female), and n = 12 (5 Male, 7 Female) participants in the decay-only group. Experiment 4 included n = 10 participants (6 Male, 4 Female). Experiment 5 included n = 20 participants (10 Male, 10 Female) with n = 9 in the 5 min group and n = 11 in the 24 group. Experiment 1 was approved by the York Human Participants Review Subcommittee. Experiments 2–5 were approved by the Institutional Review Board at the Johns Hopkins School of Medicine.

## Data extraction

When acquiring data from published figures we first attempted to open it in Adobe Illustrator. Depending on how these figures were saved and embedded, occasionally the figure could be decomposed into its layers. This allowed us to extract the *x* and *y* pixel values for each data point (which appeared as an object) to interpolate the necessary data from the figure. However, in some cases, objects and layers could not be obtained in Illustrator. In these cases, we used the utility GRABIT in MATLAB to extract the necessary data. We clearly indicate which approach was used when discussing each dataset below. Note that the authors provided source data for our *Maresch et al., 2021* and *Tsay et al., 2021a* analyses.

## Apparatus

In Experiments 1, 2, 4, and 5 participants held the handle of a robotic arm and made reaching movements to different target locations in the horizontal plane. The forearm was obscured from view by an opaque screen. An overhead projector displayed a small white cursor (diameter = 3 mm) on the screen that tracked the hand's motion. We recorded the position of the handle at submillimeter precision with a differential encoder. Data were recorded at 200 Hz. Protocol details were similar for *Haith et al., 2015*, *Neville and Cressman, 2018*, *Saijo and Gomi, 2010*, and *Maresch et al., 2021* in that participants gripped a two-link robotic manipulandum, were prevented from viewing their arm, and received visual feedback of their hand position in the form of a visual cursor. In *Lerner et al., 2020*, participants performed pointing movements with their thumb and index finger while gripping a joystick with their right hand. In *Mazzoni and Krakauer, 2006*, participants rotated their hand to displace an infrared marker placed on the index finger. In *Taylor and Ivry, 2011*, hand position was tracked via a sensor attached to the index finger while participants made horizontal reaching movements along the surface of a table. In *Day et al., 2016*, *Krakauer et al., 2000*, and *McDougle et al., 2017*, participants moved a stylus over a digitizing tablet. In Experiment 3, participants were tested remotely on a personal computer. They moved a cursor on the screen by sliding their index finger along the track pad. These conditions were similar in *Tsay et al., 2021a*.

## Visuomotor rotation

Experiments 1–5 followed a similar protocol. At the start of each trial, the participant brought their hand to a center starting position (circle with 1 cm diameter). After maintaining the hand within the start circle, a target circle (1 cm diameter) appeared in 1 of 4 positions (0°, 90°, 180°, and 270°) at a displacement of 8 cm (Experiments 2, 4, and 5). In Experiment 5, eight targets were used, spaced in increments of 45°. In Experiment 1, three targets were used positioned in a triangular wedge (45°, 90°, and 135°). Participants made a brisk movement that terminated on (Exp. 1) or moved through (Exps. 2–5) the target. Each experiment consisted of epochs of four trials (three trials for Experiment 1, 8 trials for Experiment 5) where each target was visited once in a pseudorandom order.

Participants were provided audiovisual feedback about their movement speed and accuracy. If a movement was too fast (duration <75ms) or too slow (duration >325ms) the target turned red or blue, respectively. If the movement was the correct speed, but the cursor missed the target, the target turned white. Successful movements were rewarded with a point (total score displayed on-screen), an on-screen animation, and a pleasing tone (1000 Hz). If the movement was unsuccessful, no point was awarded, and a negative tone was played (200 Hz). Participants were instructed to obtain as many

points as possible throughout the experimental session. Experiment 1 was similar but used 10 cm reach displacements and had no upper bound on movement duration.

Once the hand reached the target, visual cursor feedback was removed, and a yellow marker was frozen on-screen to indicate the final hand position. At this point, participants were instructed to move their hand back to the starting position (in Exp. 1, this return movement was aided by a circle centered on the start position, whose radius matched the hand's displacement). The cursor remained hidden until the hand was moved within 2 cm of the starting circle (1 cm in Exp. 1).

Movements were performed in one of three conditions: null trials, rotation trials, and no feedback trials. On null trials, veridical feedback of hand position was provided. On rotation trials, the on-screen cursor was rotated relative to the start position. On no feedback trials, the subject cursor was hidden during the entire trial. No feedback was given regarding movement endpoint, accuracy, or timing.

As a measure of adaptation, we analyzed the reach angle on each trial. The reach angle was measured as the angle between the hand and the target (relative to the start position), at the moment where the hand exceeded 95% of the target displacement. In Experiment 1, reach angles were measured at the hand's maximum velocity.

Experiments in *Haith et al., 2015*, *Lerner et al., 2020*, *McDougle et al., 2017*, *Taylor and Ivry, 2011*, *Neville and Cressman, 2018*, *Saijo and Gomi, 2010*, *Day et al., 2016*, *McDougle et al., 2017*, *Krakauer et al., 2000*, *Maresch et al., 2021*, *Tsay et al., 2021a*, and *Mazzoni and Krakauer, 2006* were collected using similar, but separate protocols. Important differences between these studies and the rotation protocol mentioned above are briefly described in the sections below.

## Statistics

Parametric t-tests were performed in MATLAB R2018a. For these tests, we report the t-statistic, p-value, and Cohen's d as a measure of effect size. A repeated measures ANOVA (rm-ANOVA) was used to measure differences in prediction error in *Figure 4E*. Two-way repeated measures ANOVAs were used in *Figure 6B–D* to measure how preparation time (low vs high) and exposure number (Day 1 vs. Day 2) altered learning rate, reach angle, and model-based error sensitivity measurements, respectively. Mixed-ANOVAs were used in *Figure 8C* to examine how learning (both rate and mean over initial trials) was altered by preparation time conditions (between-subject factor: *Haith et al., 2015* vs Experiment 4) and exposure number (within-subjects factor, exposure 1 vs exposure 2). A two-way ANOVA was used in *Figure 9C* to determine how interference patterns changed with movement preparation time (no limit vs limit) and time passage (5 min vs 24 hr). For all two-way and mixed-ANOVAs, we initially determined whether there was a statistically significant interaction effect between each factor. In cases where this interaction effect was statistically significant, we next measured simple main effects via one-way ANOVA.

## Competition map

In *Figure 7*, we created a competition map to describe the interactions between explicit strategy and implicit learning predicted by the competition theory. To generate this map, we used a state-space model (*Equations 1-3*) where implicit learning and explicit learning were both driven by target errors:

$$x_i^{(n+1)} = a_i x_i^{(n)} + b_i e^{(n)}$$
$$x_e^{(n+1)} = a_e x_e^{(n)} + b_e e^{(n)}$$

(7)

The terms $a_i$ and $a_e$ represent implicit retention and explicit retention. The terms $b_i$ and $b_e$ represent implicit error sensitivity and explicit error sensitivity.

Because implicit and explicit systems share a common error source in this target error model, their responses will exhibit competition. That is, increases in explicit adaptation will necessarily be coupled to decreases in implicit adaptation. To summarize this interaction, we created a competition map. The competition map describes common scenarios in which the goal is to compare two different learning curves. For example, one might want to compare the response to a 30° visuomotor rotation under two different experimental conditions. Another example would be savings, where we compare adaptation to the same perturbation at two different timepoints. In these cases, it is common to measure the amount of implicit and explicit adaptation, and then compare these across conditions or timepoints.

The critical point is that changes in the amount of implicit adaptation reflect the modulation of both implicit and explicit responses to error. This competition will occur at all points during the adaptation timecourse (Appendix 1), but is easiest to mathematically validate at steady-state. As described in the main text, the steady-state level of implicit adaptation can be derived from *Equations 1-3*. This derivation resulted in the competition equation shown in *Equation 4*. Note that *Equation 4* predicts the steady-state level of implicit learning from the implicit retention factor, implicit error sensitivity, mean of the perturbation, and critically, the steady-state explicit strategy. If the explicit system is also described using a state-space model as in *Equation 7*, it can be shown that *Equation 4* can be equivalently expressed in terms of the implicit and explicit learning parameters according to *Equation 8*:

$$x_i^{ss} = \frac{b_i(1-a_e)}{(1-a_i+b_i)(1-a_e+b_e)-b_ib_e}r \tag{8}$$

*Equation 8* provides the total amount of implicit adaptation as a function of the retention factors, $a_i$ and $a_e$, as well as the error sensitivities, $b_i$ and $b_e$. We used *Equation 8* to construct the competition map in *Figure 7A*, by comparing the total amount of implicit learning across a reference condition and a test condition.

For our reference condition, we fit our state space model to the mean behavior in *Haith et al., 2015* (*Figure 6B*, Day 1, left). This model best described adaptation during the first perturbation exposure using the parameter set: $a_i = 0.9829$, $a_e = 0.9278$, $b_i = 0.0629$, $b_e = 0.0632$. Next, we imagined that implicit error sensitivity and explicit error sensitivity differed across the reference and test conditions. On the x-axis of the map, we show a percent change in $b_i$ from the reference condition to the test condition. On the y-axis of the map, we show a percent change in $b_e$ from the reference condition to the test condition. The retention factors were held constant across conditions. Then for each condition we calculated the total amount of implicit learning using *Equation 8*. The color at each point in the map represents the percent change in the total amount of implicit learning from the reference condition to the test condition.

As described in the main text, the competition map (*Figure 7A*) is composed of several important regions (*Figure 7B*). In Region A, there is a decrease in implicit error sensitivity (from reference to test) as well as a decrease in the total amount of implicit adaptation predicted by *Equation 8*. In Region B, *Equation 8* predicts a decrease in implicit adaptation, despite an increase in implicit error sensitivity. In Region D, there is an increase both in implicit error sensitivity as well as steady-state implicit learning. In Region E, there is an increase in implicit adaptation, despite a decrease in implicit error sensitivity. Finally, Region C shows cases where there are changes in implicit error sensitivity, but the total absolute change in implicit adaptation (*Equation 8*) is less than 5%. To localize this region, we solved for the linear bounds that describe a 5% increase or a 5% decrease in the output of *Equation 8*.

## Neville and Cressman, 2018

To understand how enhancing explicit strategy might alter implicit learning, we considered data collected by *Neville and Cressman, 2018*. Here, the authors tested how awareness of a visuomotor rotation altered the adaptation process. To do this, participants (n = 63) were divided into several groups. In the instructed groups (*Figure 2A*, purple), participants were instructed about the rotation and a compensatory strategy prior to perturbation onset. In other groups, no instruction was provided (*Figure 1C*; *Figure 2A*, black). During rotation periods, participants reached to three potential targets. Implicit contributions to behavior were measured at four different periods using 'exclusion' trials. During exclusion trials, the authors instructed participants to reach (without visual feedback) as they did during the baseline period prior to perturbation onset (without using any knowledge of the perturbation gained thus far). Exclusion trial reach angles served as our implicit learning measure. The difference between total adaptation and exclusion trial reach angles served as our explicit learning measure.

At the start of the experiment, all participants performed a baseline period without a rotation for 30 trials. Baseline implicit and explicit reach angles were then assayed. At this point, participants in the strategy group were briefed about the perturbation with an image that depicted how feedback would be rotated, and how they could compensate for it. Then all groups were exposed to the first block of a visuomotor rotation for 30 trials. Some participants experienced a 20° rotation, others a 40° rotation,

and others a 60° rotation. After this first block, implicit and explicit learning were assayed. This block structure was repeated two more times.

Here, we focused on implicit and explicit adaptation measures obtained at the end of the final block. To obtain these data, we extracted the mean participant response and the associated standard error of the mean, directly from the primary figures reported by *Neville and Cressman, 2018* using Adobe Illustrator CS6. The implicit and explicit responses in all six groups are shown in *Figure 2—figure supplement 1*. The marginal effect of instruction (average over rotation sizes) is shown in *Figure 2B and C*.

Finally, we tested whether the competition equation (*Equation 4*) or independence equation (*Equation 5*) could account for the levels of implicit learning observed across rotation magnitude and awareness conditions. To do this, we used a bootstrapping approach. Using the mean and standard deviation obtained from the primary figures, we sampled hypothetical explicit and implicit aftereffects for 10 participants. We then calculated the mean across these 10 simulated participants. After this, we used *fmincon* in MATLAB R2018a to find an implicit error sensitivity that minimized the following cost function:

$$\theta_{fit} = \underset{\theta}{argmin} \sum_{n=1}^{6} \left(x_{i_n}^{ss} - \hat{x}_{i_n}^{ss}\right)^2 \tag{9}$$

This cost function represents the difference between the simulated level of implicit adaptation, and the amount of implicit learning that would be predicted for a given perturbation size and simulated explicit adaptation, according to our competition framework (*Equation 4*) or independence framework (*Equation 5*). For this process, we set the implicit retention factor to 0.9565 (see *Measuring properties of implicit learning*). Therefore, only the implicit error sensitivity remained as a free parameter. In sum, we aimed to determine if a single implicit error sensitivity could account for the amount of adaptation across the no instruction group, instruction group, and each of the three perturbation magnitudes (20, 40, and 60°). The combination of instruction and perturbation magnitude yielded six groups, hence the upper limit on the sum in *Equation 9*. We repeated this process for a total of 10,000 simulated groups.

In *Figure 2B&C*, we show the marginal effect of instruction on the implicit aftereffect. This was obtained by averaging across each of the three rotation magnitudes shown in *Figure 2—figure supplement 1*, for each model. In *Figure 1*, we show the implicit learning levels predicted by the model across all rotation sizes in the no-instruction group. Model predictions across all rotations sizes in the instruction group are shown in *Figure 2—figure supplement 1*. Again, all model predictions were made using the same underlying implicit learning parameter set.

## Experiment 1

To examine how changes in rotation onset and magnitude altered implicit learning, we recruited two participant groups. In the abrupt group, subjects (n = 36) experienced a 60° visuomotor rotation abruptly. In the stepwise group, subjects (n = 37) experienced four separate rotation magnitudes in sequence: 15°, 30°, 45°, and 60°. Thus, the experiment had four learning periods, one for each rotation size. Each period lasted 66 trials, over which three targets (45°, 90°, 135°) were visited 22 times. This same structure was used in the abrupt group, though the rotation magnitude remained constant over each learning block. Twice during each block (about 75% into each block and again at the end), exclusion trials were used to measure implicit adaptation. On these trials, subjects were told to stop using explicit strategies and to reach as they had during the baseline period. The average exclusion trial reach angle (across both probe periods in each block) served as our implicit learning measure. The difference between total adaptation and the average exclusion trial reach angle served as our explicit learning measure. Total adaptation was calculated as the average reach angle on the last 20 trials in each learning period.

Here, we focused on implicit and explicit adaptation measures obtained during each block. These measures are shown in *Figures 1I and 2J*. In *Figures 1 and 2*, we tested how well these measures were predicted by the competition and independence equations. The same model parameters were used in *Figures 1 and 2*, although *Figure 1* only shows data in the stepwise condition. Note that the competition equation can be written as $x_i^{ss} = p_i(r - x_e^{ss})$ and the independence equation can be written

as $x_i^{ss} = p_i r$, where $p_i$ is a scalar gain determined by $a_i$ and $b_i$. Thus, the gain $p_i$ is the only unknown model parameter.

Our goal was to identify one gain (one for each model) that could parsimoniously explain behavior across the stepwise and abrupt groups. Thus, we identified the optimal gain that minimized the squared error between the model predictions and implicit adaptation across five measures: 15° step-wise learning, 30° stepwise learning, 45° stepwise learning, 60° stepwise learning, and 60° abrupt learning. For the abrupt condition, we did not observe a statistically significant difference in implicit aftereffect across the four learning periods (rm-ANOVA, F(3,105)=2.21, p = 0.091, $\eta_p^2$=0.059); thus, we averaged across learning periods to obtain a single implicit measure. We then identified the $p_i$ parameter that minimized squared error according to *Equation 9*, with all five terms described above appearing in the sum.

To construct the model predictions shown in *Figures 1 and 2*, we used a bootstrapping approach. Participants in the stepwise and abrupt group were resampled with replacement 1000 times. Each time the average implicit learning measure was calculated across the five conditions described above. Each model was then fit to these average data. Thus, *Figures 1 and 2* show the mean implicit learning predicted by each model across all 1000 iterations, as well as the associated standard deviation.

In the main text, we also report a statistical comparison between implicit learning predicted by the competition theory in the 60° stepwise and 60° abrupt conditions. This statistic was obtained using a different procedure. Here the optimal $p_i$ was determined again using *Equation 9*, but without boot-strapping. Average across-subject implicit adaptation in the 15° stepwise period, 30° stepwise period, 45° stepwise period, 60° stepwise period, and 60° abrupt period appeared within the sum in *Equation 9*. Then implicit learning was predicted using *Equation 4* assuming that each participant had the same $p_i$ learning gain. We then conducted a paired t-test between 60° stepwise and 60° abrupt implicit learning predicted by the model.

Exp. 1 was used extensively to compare the competition theory with an SPE generalization model. All details concerning this analysis are provided in Appendix 6. Results are depicted in *Figure 4*.

Finally, we analyzed subject-to-subject pairwise relationships between implicit learning, explicit strategy, and total adaptation (average over last 40 rotation trials) in *Figure 5—figure supplement 1B,E&H*. For these analyses, we combined subjects across the 60° rotation period in the abrupt and stepwise groups. Note we excluded three outlying participants whose reach angles differed by more than three median absolute deviations from the total population on at least 33% of all trials. This yielded a total dataset of n = 70. To analyze each pairwise relationship, we used linear regressions. In addition, we analyzed the same relationships during the 30° rotation period in the stepwise group (*Figure 5—figure supplements 2A and 3B*).

### Tsay et al., 2021a

To evaluate the competition and independence models, we analyzed how implicit and explicit systems responded to rotation sizes between 15° and 90° in experiments conducted by *Tsay et al., 2021a*. Data in these experiments was collected remotely via a laptop-based experiment. Participants moved to targets at 45° and 135°, which alternated across trials. Participants were exposed to a 15°, 30°, 60°, or 90° rotation (n = 25/rotation size). The reach angles during an initial baseline period, rotation period, and terminal no aiming period are shown in *Figure 1M*. During the no aiming period, participants reached to each target 10 times (20 trials total). To calculate implicit learning (*Figure 1M*, no aiming; *Figure 1N&Q*, data) we averaged the reach angle across the 20 no aiming trials. To calculate total adaptation, we measured the average reach angle over the last 40 reaching trials (*Figure 5—figure supplement 1C,1F,2A&3A*). To calculate explicit strategy, we computed the difference between total adaptation and implicit learning (*Figure 1O*). We also calculated the implicit driving input in the competition theory (rotation minus explicit strategy) in *Figure 1P*. We also reported an explicit gain in the main text. This gain was calculated by dividing the difference between explicit strategies by the difference in rotation sizes corresponding to each strategy (and then multiplying by 100 to obtain a percentage).

To investigate the non-monotonic relationship between implicit learning and rotation size (*Figure 1N*), we used the competition and independence models. In *Figure 1Q*, we fit each model to the measured data. To do this, we estimated the implicit retention factor using the reach angle decay rate during the terminal no aiming period (see *Measuring properties of implicit learning*, estimate =

0.974). Next, we used a least-squares approach to determine the optimal implicit error sensitivity ($b_i$) that best matched the implicit reach angles measured across all four rotation sizes. Note that since $a_i$ and $b_i$ appear together in the implicit learning gain, $p_i$, fitting the gain directly would produce the same results.

For the competition theory, we averaged the implicit and explicit responses within each rotation group, and then identified the $b_i$ value that best predicted implicit learning across rotation sizes according to the competition equation (*Equation 4*). To do this, we used the *fminbnd* utility in MATLAB R2018a. This yielded $b_i$ = 0.0319. We then used the same $a_i$ and $b_i$ parameter values to predict total implicit learning across all four rotation sizes via *Equation 4*, assuming all participants had the same implicit learning parameters (*Figure 1Q*, competition). Again, this is equivalent to directly fitting the implicit learning gain $p_i$.

We used a bootstrapping procedure to identify the optimal $b_i$ parameter in the independence model. To do this, we sampled participants in each rotation group with replacement 10,000 times. Each time, we calculated the average implicit response, and then minimized the squared error (*fminbnd* in MATLAB R2018a) between this implicit response and that predicted by the independence model (*Equation 5*), across all four rotations sizes. We used the $a_i$ and $b_i$ estimated in the bootstrapping procedure to predict total implicit learning according to the independence model (*Figure 1Q*, independence).

Finally, we analyzed subject-to-subject pairwise relationships between implicit learning, explicit strategy, and total adaptation in *Figure 5—figure supplement 1C, F and I*. For this, we considered participants in the 60° rotation group. To analyze each relationship, we used linear regressions. We also analyzed these same relationships during the 30° rotation period (*Figure 5—figure supplements 2A and 3B*).

Note that Tsay et al. also tested participants in an invariant error-clamp experiment. We did not analyze these data here for two reasons. First, no strategy is used in invariant error-clamp paradigms. This means that SPE and target errors are the same, meaning that the competition model and independence model cannot be distinguished (they make the same predictions). Second, as described in our Discussion (see the section on invariant error-clamp learning), the competition and independence models derived in *Equations (4) and (5)* only apply to standard rotation learning. The implicit learning gain in the invariant error-clamp paradigm is not the same and predicts implicit learning levels that cannot be physically achieved (see Discussion).

## Experiment 2

To test whether changes in explicit strategy altered implicit learning at the individual-level, we tested two adaptation conditions. In the first experiment, participants adapted to a visuomotor rotation without any limits applied to preparation time (No PT Limit), thus allowing participants to use explicit strategy. In a second experiment, we strictly limited preparation time in order to suppress explicit strategy (Limit PT).

Participants in the No PT Limit condition began with 10 epochs of null trials (one epoch = 4 trials), followed by a rotation period of 60 epochs. Other details concerning the experiment paradigm are described in *Visuomotor rotation*. At the end of the perturbation period, we measured the amount of implicit and explicit learning. To do this, participants were instructed to forget about the cursor and instead move their hand through the target without applying any strategy to compensate for the perturbation. Furthermore, visual feedback was completely removed during these trials. All four targets were tested in a randomized sequence. To quantify the total amount of implicit learning, we averaged the reach angle across all targets (*Figure 3B&G*). To calculate the amount of explicit adaptation, we subtracted this measure of implicit learning from the mean reach angle measured over the last 10 epochs of the perturbation prior to the verbal instruction (results did not change whether we used 5, 10, 15 or 20 epochs to calculate total learning). Explicit measures are shown in *Figure 3G&N* (E2).

In the Limit PT group, we suppressed explicit adaptation for the duration of the experiment by limiting the time participants had to prepare their movements. To enforce this, we limited the amount of time available for the participants to start their movement after the target location was shown. This upper bound on reaction time was set to 225ms (we corrected reaction times by the average screen delay, 55ms). If the reaction time of the participant exceeded the desired upper bound, the

participant was punished with a screen timeout after providing feedback of the movement endpoint. In addition, a low unpleasant tone (200 Hz) was played. This condition was effective in limiting reaction time (*Figure 3F*). This experiment started with 10 epochs (one epoch = 4 trials) of null trials. After this, the visuomotor rotation was introduced for 60 epochs. At the end of the perturbation period, we measured retention of the visuomotor memory in a series of 15 epochs of no feedback trials (*Figure 3E*, no feedback).

Our goal was to test whether the putative implicit learning properties measured in the Limit PT group could be used to predict the subject-to-subject relationship between implicit and explicit adaptation in the No PT Limit group (according to *Equation 4*). To do this, we measured each participant's implicit retention factor and error sensitivity in the Limit PT condition (see *Measuring properties of implicit learning* below). We then averaged each parameter across participants. Next, we inserted these mean parameters into *Equation 4*. With these variables specified, *Equation 4* predicted a specific linear relationship between implicit and explicit learning (*Figure 3G*, model). We overlaid this prediction on the actual amounts of implicit and explicit adaptation measured in each No PT Limit participant (*Figure 3G*, black dots). We performed a linear regression across these measured data (*Figure 3G*, black line, measured). We report the slope and intercept of this regression as well as the corresponding 95% confidence intervals.

Lastly, we also asked participants to verbally report their explicit strategy. After the implicit probe trials, we showed each target once again, with a ring of small white landmarks placed at an equal radial distance around the screen (*McDougle et al., 2015*). A total of 108 landmarks was used to uniformly cover the circle. Each landmark was labeled with an alphanumeric string. Subjects were asked to report the nearest landmark that they were aiming towards at the end of the experiment in order to move the cursor through the target when the rotation was on. The mean angle reported across all four targets was calculated to provide an additional assay of explicit adaptation. However, several (25% across all participants and trials) reports appeared inaccurate in that they had the incorrect sign (participants reported aiming with, not opposite to, the rotation). Noting that explicit re-aiming is prone to erroneous sign errors (*McDougle and Taylor, 2019*) (errors of same magnitude, opposite sign), we took each report's absolute value when calculating explicit recalibration.

Next, we calculated a report-based implicit measure by subtracting report-based explicit strategy from total adaptation. While report-based implicit learning was smaller than reach-based implicit learning (*Figure 3—figure supplement 2B*), and report-based explicit strategy was larger than reach-based strategy (*Figure 3—figure supplement 2A*), the two exhibited close correspondence with *Equation 4* (*Figure 3—figure supplement 2C*).

Lastly, we also analyzed our reach-based implicit and explicit learning measures in a generalization analysis (*Figure 5A&B*). This analysis is described in Appendix 6.

## Experiment 3

We remotely tested three participant groups (No PT Limit, Limit PT, and decay-only). Participants controlled a cursor by moving their index finger across the track pad of their personal computer. The experiment was coded in Java. To familiarize themselves with the task, participants watched a 3 minute instructional video. In this video, the trial structure, point system, and feedback structure were described. After this video, there was a practice period. During the practice period, the software tracked the participant's reach angle on each trial. If the participant achieved success on fewer than 65% of trials (measured based on an angular target-cursor discrepancy ≤30°, reaction time ≤1 sec, and movement duration ≤0.6 sec), they had to re-watch the instructional video and re-do the practice period. Movements were brisk and straight, as in standard in-person rotation studies (two example participants are shown in the No PT Limit and Limit PT groups in *Figure 3—figure supplement 3*).

After the practice period ended, the testing period began. This testing period was similar to the No PT Limit condition in Experiment 2. On each trial, participants reached to 1 of 4 targets (up, down, left, and right). Each target was visited once pseudorandomly in a cycle of 4 targets. After an initial 10-cycle null period, a 30° visuomotor rotation was imposed that lasted for 60 epochs. At the end of the rotation period, we measured implicit and explicit adaptation. The experiment briefly paused, and an audiovisual recording was played that instructed participants to not use any strategy and to move their hand straight through the target. After the experiment resumed, feedback was removed, and

participants performed 20 cycles of no-feedback probe trials. In the No PT-Limit group, participants were told to stop aiming on these no-feedback trials, and to move their hand straight to the target.

We measured subject-to-subject correlations between implicit and explicit adaptation in the No PT Limit group. For this, we calculated two implicit learning measures. The early implicit aftereffect was simply the aftereffect observed on the first no-aiming, no-feedback probe cycle (*Figure 3Q*). The late implicit aftereffect was the average aftereffect observed on the last 15 cycles of this no-aiming, no-feedback period (*Figure 3P*). To measure explicit learning, we calculated the difference between the total amount of adaptation (mean reach angle over last 10 cycles of the rotation period) and the first cycle of the no-aiming, no-feedback period. We investigated the relationship between explicit adaptation and the early and late implicit aftereffects via linear regression in *Figure 3P&Q*, respectively. For the early implicit aftereffect, we measured the 95% CI for the slope and intercept. Note that explicit learning measures are also reported in *Figure 3N* (E3, black) and late implicit learning measures are reported in *Figure 3O* (No Lim.).

In addition, we also analyzed the relationship between total adaptation and implicit and explicit adaptation in the No PT Limit group. As described in the main text, the competition theory predicted that total adaptation and explicit strategy should have a positive relationship, whereas total adaptation and implicit learning should have a negative relationship (see Appendix 7). In *Figure 5G*, we show the relationship between total adaptation and the explicit learning measure. In *Figure 5H*, we show the relationship between total adaptation and the late implicit learning measure. The brown lines denote a linear regression across individual participants.

Finally, we also considered the No PT Limit data in our generalization analyses in *Figure 4A&C*. This process was the same as for Experiment 2 as shown in *Figure 4A&B*. See Appendix 6.

Next, we also tested a Limit PT group in Exp. 3. Here, we attempted to suppress explicit strategies by limiting movement preparation time. To determine the limiting preparation time, we used an adaptive algorithm during the baseline period to decrease or increase the preparation time limit in response to a correct or incorrect reach responses (i.e. reaches to the correct or incorrect target). This limit was capped at 350ms, but this upper bound did not include screen delay. We used audiovisual feedback throughout the experiment to enforce the preparation time limit. If the reaction time of the participant exceeded the desired upper bound, the participant was played a low-pitched tone during which the screen briefly timed out and shown a message to "react faster". This condition produced the preparation times shown in *Figure 3M*. Apart from this, the experiment protocol was the same as the No PT Limit group.

To test whether limiting preparation time was successful in inhibiting explicit strategy, we calculated explicit strategy as in the No PT Limit. Explicit strategies were dramatically inhibited by limiting preparation time (*Figure 3E&O*, red). Second, we wanted to measure implicit learning properties in the Limit PT condition and use these to predict the implicit-explicit relationship in the No PT Limit group, with the competition theory. For the latter, we used the same method described above for Experiment 2 (also see *Measuring properties of implicit learning*). Using the Limit PT data, the competition theory predicted the line shown in blue in *Figure 3Q*. The black data points show the implicit and explicit learning measures in the No PT Limit group. Also note that consistent with the competition theory, limiting preparation time led to an increase in implicit learning (*Figure 3O*, PT Limit).

As stated above, in the No PT Limit and Limit PT groups, participants were instructed to stop re-aiming during the no feedback period, and to move their hand straight to the target (*Figure 3I&L*, no aiming). We used the voluntary change in reach angle to estimate explicit strategy. However, the instruction period lasted about 30 s, which may have caused decay in temporally labile implicit learning (*Neville and Cressman, 2018*; *Maresch et al., 2021*; *Hadjiosif and Smith, 2015*). To measure how much implicit learning had decayed over this time delay, we varied the instruction condition in a decay-only group (n = 12). The decay-only group adapted using the same restricted reaction time paradigm as the Limit PT group. However, prior to the no feedback period, participants were told that the disturbance between the cursor and their movement would still be present when they returned to the experiment, but they would no longer be able to see the cursor. Still, they were told to imagine this disturbance and to try and move the imagined cursor to the target. Changes in reach angle in this group, would be due solely to decay in implicit learning (*Figure 8—figure supplement 1*). We compared the behavior in the decay-only group to the Limit PT group in *Figure 8—figure supplement 1*.

Finally, we used a separate procedure to estimate screen delay. To do this, participants were told to tap a circle that flashed on and off in a regular, repeating cycle. Participants were told to predict the appearance of the circle, and to tap exactly as the circle appeared. Because the stimulus was predictable, the difference between the appearance time, and the participant's button press, revealed the system's visual delay. The average visual delay we measured was 154ms. This average value was subtracted out in the preparation times reported in *Figure 3J&M*, as well as *Figure 8—figure supplement 1*.

## Day et al., 2016

A recent study by Day et al. measured implicit generalization. Participants were exposed to a 45° rotation at a single target. On each trial, they reported their aiming direction, using a ring of visual landmarks. This study measured implicit generalization by instructing participants to aim towards untrained targets. We reproduce this curve in *Figure 4A* (Day 1T). We only show the curve starting at the average aiming direction (0° on the x-axis), towards the training target direction (i.e. in the direction participants will change their aim when instructed to aim to the primary target). Note in *Figure 4B and C*, only the initial two points along the curve are shown.

Last, we also compared implicit learning measured across two groups reported in their Figure 2. In the 'target' group in *Figure 4—figure supplement 2A*, implicit aftereffects were periodically probed at the trained target location, by asking subjects to reach to the target without aiming. In the 'aim' group, implicit aftereffects were probed at a target location 30° away from the trained target, consistent with the direction of the most frequently reported aim. In *Figure 4—figure supplement 2A*, we show the implicit aftereffect measured on the first aftereffect trial at the end of the experiment. In *Figure 4—figure supplement 2C* we again show the implicit aftereffect measured at the trained target location in the 'probe' condition. The 'report' condition shows the amount of implicit learning estimated by subtracting the reported explicit strategy from the reported reach angle on the last cycle of the rotation. Note that all data were extracted using the primary source's figures with MATLAB's GRABIT utility.

## Krakauer et al., 2000

*Figure 4A* reproduces generalization curves measured by Krakauer et al. We extracted curves shown in Figure 7B in *Krakauer et al., 2000* using GRABIT in MATLAB R2018a. To demonstrate how generalization curves are altered by the number of adaptation targets, we show the one target (1T), 2 target (2T), four target (4T), and eight target (8T) curves reported in *Krakauer et al., 2000*. In this study, participants moved a stylus across a digitized tablet and adapted to a 30° rotation.

## McDougle et al., 2017

In *Figure 4—figure supplement 2B*, we show data collected by *McDougle et al., 2017*, reported in *Figure 3A* of the original manuscript. Here, participants were exposed to a 45° rotation while reaching to a single target. At the end of the experiment, participants were exposed to an aftereffect block where they reached 3 times to 16 different targets spaced in varying increments around the unit circle. In this aftereffect block feedback was removed and participants were told to move straight to the target without re-aiming. This aftereffect block was used to construct a generalization curve. In *Figure 4—figure supplement 2B* we show data only from two relevant locations on this curve. The 'target' condition represents aftereffects probed at the training target. The 'aim' condition shows the aftereffect measured at 22.5° away from the primary target, which was the target closest to the mean reported explicit re-aiming strategy of 26.2°.

We also use the study's implicit generalization curve (their Figure 3A) in our SPE generalization model analysis. This curve is reproduced in *Figure 4*. We extracted only one side: the one pointing along the vector which connected the aiming direction and the adaptation target. We also normalized the curve by dividing by the maximum implicit learning they measured along the aiming direction. These data were extensively used in our generalization analysis in Appendix 6. All relevant details are provided there. We selected this study because implicit and explicit learning were dissociated and because CW and CCW were counterbalanced across participants (alleviating potential position-based biases). Note that all data were extracted using the primary source's figures with MATLAB's GRABIT utility.

## Maresch et al., 2021

To evaluate the competition and independence models, we analyzed how implicit and explicit learning varied across individual participants in a study conducted by Maresch and colleagues (*Maresch et al., 2021*). In this analysis, we collapsed across participants in the CR, IR-E, and IR-EI groups (n = 40 total). Note that we did not include participants in the IR-I group, because implicit learning was only measured at one timepoint, unlike the three other groups. In this task, participants reached to eight targets (45° between each target) while holding a robotic manipulandum. Participants were exposed to a 60° rotation. Implicit learning and explicit strategy were probed in various ways throughout the experiment. Here, we used the authors' exclusion-based implicit and explicit learning measures. In other words, implicit learning was measured by telling subjects to stop aiming. Explicit strategy was estimated as the voluntary decrease in reach angle that occurred when participants were told not to aim (the difference between total adaptation and implicit learning). To calculate total adaption, we averaged the reach angle over the 40 terminal rotation trials. We analyzed subject-to-subject pairwise relationships between implicit learning, explicit strategy, and total adaptation in *Figure 5—figure supplement 1A,D&G*. To analyze each pairwise relationship, we used linear regressions.

Lastly, in *Figure 5—figure supplement 3* we show data collected by *Maresch et al., 2021*, reported in *Figure 4b* of the original manuscript. This study calculated implicit learning directly with exclusion trials and indirectly with aim reports. In *Figure 5—figure supplement 3D* we show data from the IR-E group. This group was comparable to our data because aim was reported intermittently (4 times every 80 trials), meaning that on most trials, aiming targets would not cause adaptation (only the primary target). In addition, there were eight adaptation targets, which will widen implicit generalization. The probe condition in *Figure 5—figure supplement 3D* corresponds to the total implicit learning measured at the end of adaptation by telling participants to reach without re-aiming. The 'report' condition corresponds to total implicit learning estimated at the end of adaptation by subtracting the reported aim direction from the measured reach angle.

## Haith et al., 2015

To investigate savings, *Haith et al., 2015* used a forced preparation time task. Briefly, participants (n = 14) performed reaching movements to two targets, T1 and T2, under a controlled preparation time scenario. To control movement preparation time, four audio tones were played (at 500ms intervals) and participants were instructed to reach coincident with the 4th tone. On high preparation time trials (High PT), target T1 was shown during the entire tone sequence. On low preparation time trials (Low PT), T2 was initially shown, but was then switched to target T1 approximately 300ms prior to the 4th tone. High PT trials were more probable (80%) than Low PT trials (20%).

After a baseline period (100 trials for each target), a 30° visuomotor rotation was introduced for target T1 only. After 100 rotations trials (Exposure 1), the rotation was turned off for 20 trials. After a 24 hr break, participants then returned to the lab. On Day 2, participants performed 10 additional reaching movements without a rotation, followed by a second 30° rotation (Target T1 only) of 100 trials (Exposure 2). The experiment then ended with a washout period of 100 trials for each target.

We quantified the amount of savings expressed upon re-exposure to the perturbation, on High PT and Low PT trials. We measured savings using two metrics. First, we measured the rate of learning during each exposure to the perturbation using an exponential fit. We fit a two-parameter exponential function to both Low PT and High PT trials during the first and second exposure (we constrained the third parameter to enforce that the exponential begin at each participant's measured baseline reach angle). We compared the exponential learning rate across high PT trials, low PT trials, and Exposures 1 and 2 with a two-way repeated-measures ANOVA (two within-subject factors: PT and exposure number), followed by one-way repeated-measures ANOVA to test simple main effects (*Figure 6B*, right).

We also quantified savings in a manner similar to that reported by *Haith et al., 2015*; we calculated the difference between the reach angles before and after the introduction of the perturbation, during each exposure (*Figure 6C*, 1st and 2nd columns). For High PT trials, we then computed the mean reach difference over the three trials preceding, and three trials following perturbation onset. Given their reduced frequency, for Low PT trials, we focused solely on the trial before and trial after perturbation onset. We used the same statistical testing procedure (two-way rm-ANOVA with follow-up

simple main effects) to test for savings in the pre-perturbation and post-perturbation differences (*Figure 6C*, right).

Finally, we also used a state-space model of learning to measure properties of implicit and explicit learning during each exposure. We modeled implicit learning according to *Equation 3* and explicit learning according to *Equation 7*. In our competition theory, we used target error as the error in both the implicit and explicit state-space equations. In our SPE model, we used target error as the explicit system's error, and SPE as the implicit system's error.

The total reach angle was set equal to the sum of implicit and explicit learning. Each system possessed a retention factor and error sensitivity. Here, we asked how implicit and explicit error sensitivity might have changed from Exposure 1 to Exposure 2, noting that savings is related to changes in error sensitivity (*Coltman et al., 2019*; *Mawase et al., 2014*; *Lerner et al., 2020*; *Albert et al., 2021*; *Herzfeld et al., 2014*). Therefore, we assumed that the implicit and explicit retention factors were constant across perturbations but allowed a separate implicit and explicit error sensitivity during Exposures 1 and 2. Therefore, our modeling approach included six free parameters. We fit this model to the measured behavior by minimizing the following cost function using *fmincon* in MATLAB R2018a:

$$\theta_{fit} = \underset{\theta}{argmin} \sum_{n=1}^{N} \left( y_1^{(n)} - \hat{y}_1^{(n)} \right)^2 + \left( y_2^{(n)} - \hat{y}_2^{(n)} \right)^2 \tag{10}$$

Here $y_1$ and $y_2$ represent the reach angles during the 1st and 2nd rotation. These reach angles are composed of High PT and Low PT trials. On Low PT trials, the reach angle is equal to the implicit process. On High PT trials, the reach angle is equal to the sum of the implicit adaptive process and the explicit adaptive process.

We fit this model to individual participant behavior, in the case where implicit learning was driven by target errors (*Equation 1*), and also in the alternate case where it was driven by SPEs (*Equation 2*). The implicit and explicit model simulations in *Figure 6D* (columns 1 and 2) represent the competition theory (target error learning). For the SPE model, these states are not shown, but model parameters are reported in *Figure 6E*.

We used a two-way repeated-measures ANOVA to test whether error sensitivity differed across implicit and explicit learning (within-subject factor) and across exposures (within-subject factor). We used follow-up one-way repeated measures ANOVA to test for differences across exposures (separately for implicit and explicit learning) for the SPE model, after detecting a statistically significant interaction effect.

Finally, we also fit the target-error (*Equation 1*) model to the mean behavior across all participants in Exposure 1 and Exposure 2. We obtained the parameter set: $a_i = 0.9829$, $a_e = 0.9278$, $b_{i,1} = 0.0629$, $b_{i,2} = 0.089$, $b_{e,1} = 0.0632$, $b_{e,2} = 0.1078$. Note that the subscripts 1 and 2 denote error sensitivity during Exposure 1 and 2, respectively. These parameters were used for our simulations in *Figure 7* (see *Competition Map*).

## Experiment 4

The competition theory (*Figure 7*) predicted that more consistently suppressing explicit strategy, relative to the conditions used by *Haith et al., 2015*, should reveal savings in the implicit system. That is, *Haith et al., 2015* inhibited strategy only on 20% of all trials. Strategies were able to compete with the implicit system on the remaining 80% of trials. To test this prediction, we inhibited strategy on every trial in Exp. 4. To inhibit strategies, we limited reaction time using the procedure described above for Experiments 2 and 3. In Exp. 3, we observed that limiting movement preparation time drastically suppressed explicit re-aiming (*Figure 3N*). Limiting preparation time in Exp. 4 was effective in reducing reaction times (*Figure 8B*, top row), even lower than the 300ms threshold used by *Haith et al., 2015*.

Experiment 4 used the 4-target protocol reported in *Visuomotor rotation*. Apart from that, its trial structure was similar to that of *Haith et al., 2015*. After a familiarization period, subjects completed a baseline period of 10 epochs (one epoch = 4 trials for each target). At that point, we imposed a 30° visuomotor rotation for 60 epochs (Exposure 1). At the end of this first exposure, participants completed a washout period with no perturbation that lasted for 70 epochs. At the end of the washout period, subjects were once again exposed to a 30° visuomotor rotation for 60 epochs (Exposure 2).

We quantified savings in a manner consistent with *Haith et al., 2015*. First, we fit a two-parameter exponential function to the reach angle during Exposures 1 and 2 (third parameter was used to constrain the fit so that the exponential curve started at the reach angle measured prior to perturbation onset). Second, we also tested for differences in the initial response to the perturbation across each exposure. To do this, we calculated the difference between reach angle during Exposures 1 and 2 (*Figure 8A&B*, bottom row). We then calculated the difference in reach angle between the five epochs preceding and five epochs following rotation onset. Differences between these two savings indicators (rate and early learning) were tested with a mixed-ANOVA, to determine how adaptation differed across each perturbation exposure (within-subject) in Exp. 4 and Haith et al. (between-subject factor). Statistically significant interaction effects were followed by one-way repeated-measures ANOVA (testing simple main effect of exposure number). Results are shown in *Figure 8C*.

## Experiment 5

*Lerner et al., 2020* demonstrated that anterograde interference slows the rate of learning after 5 min (also 1 hr), but dissipates over time and is nearly gone after 24 hr. Here, we wondered if this reduction in learning rate could at least be in part driven by impairments in implicit learning. Because *Lerner et al., 2020* did not constrain preparation time, one would expect that participants used both implicit and explicit learning processes. In Experiments 2–4, we isolated the implicit component of adaptation by limiting reaction time. We used the same technique to limit reaction time in Experiment 5. The experiment paradigm is described in *Visuomotor rotation* above. With that said, we used eight adaptation targets as opposed to four targets, to match the protocol used by *Lerner et al., 2020*.

The perturbation schedule is shown in *Figure 9A&B* at top. We recruited two groups of participants, a 5 min group (n = 9), and a 24 hr group (n = 11). After familiarization, all participants were exposed to a baseline period of null trials lasting five epochs (one epoch = 8 trials). Next participants were exposed to a 30° visuomotor rotation for 80 cycles (Exposure A). At this point, the experiment ended. After a break, participants returned to the task. For the 5 min group, the second session occurred on the same day. For the 24 hr group, participants returned the following day for the second session. At the start of the second session, participants were exposed to a 30° visuomotor rotation (Exposure B) whose orientation was opposite to that of Exposure A. This rotation lasted for 80 epochs.

We analyzed the rate of learning by fitting a two-parameter exponential function to the learning curve during Exposures A and B (the third parameter was used to constrain the exponential curve to start from the behavior on the first epoch of the rotation). For each participant, we computed an interference metric by dividing the exponential rate of learning during Exposure B, by that measured during Exposure A (*Figure 9C*, blue). We tested how interference was impacted by passage of time between Exposures A and B (5 min or 24 hr) as well as by the preparation time condition (no limit in Lerner and Albert et al., limit in Exp. 5) using a two-way ANOVA. In addition, we calculated each exponential's x-intercept (i.e. zero-crossing), which we used in the control analysis described below.

One potential issue with this technique, is that it does not consider differences in the initial errors experienced during re-exposure to the rotation (*Figure 9A&B*, bottom row), which could alter sensitivity to error (*Albert et al., 2021*; *Kim et al., 2018*; *Marko et al., 2012*). To examine this, we recalculated learning rate during the second rotation exposure only after the zero-crossing in reach angle (i.e. the point at which the error reached 30°, as in the initial exposure). To estimate this zero-crossing point, we used the exponential model's x-intercept as described above. Then we used a two-way ANOVA (same as above) to test how this alternate interference metric was altered by time passage (between exposures) and preparation time.

## *Lerner et al., 2020*

Recently, *Lerner et al., 2020* demonstrated that slowing of learning in anterograde interference paradigms is caused by reductions in sensitivity to error. Here, we re-analyze some of these data.

*Lerner et al., 2020* studied how learning one visuomotor rotation altered adaptation to an opposing rotation when these exposures were separated by time periods ranging from 5 min to 24 hr. Here, we focused solely on the 5 min group (n = 16) and the 24 hr group (n = 18). A full methodological description of this experiment is provided in the earlier manuscript. Briefly, participants gripped a joystick with the thumb and index finger which controlled an on-screen cursor. Their arm was obscured from view using a screen. Targets were presented in eight different positions equally spaced at 45°

intervals around a computer monitor. Each of these eight targets was visited once (random order) in epochs of eight trials. On each trial, participants were instructed to shoot the cursor through the target.

All experiment groups started with a null period of 11 epochs (one epoch = 8 trials). This was followed by a 30° visuomotor rotation for 66 epochs (Exposure A). At this point, the experiment ended. After a break, participants returned to the task. For the 5 min group, the second session occurred on the same day. For the 24 hr group, participants returned the following day for the second session. At the start of the second session, participants were immediately exposed to a 30° visuomotor rotation (Exposure B) whose orientation was opposite to that of Exposure A. This rotation lasted for 66 epochs. Short set breaks were taken every 11 epochs during Exposures A and B.

Here, as in the earlier work (*Lerner et al., 2020*), we analyzed the rate of learning by fitting a two-parameter exponential function to the learning curve during Exposures A and B (the third parameter was used to constrain the exponential curve to start from the behavior on the first epoch of the rotation). For each participant we computed an interference metric by dividing the exponential rate of learning during Exposure B, by that measured during Exposure A (*Figure 9C*, green). In addition, we also analyzed the reaction time of the participants during Exposure B. The mean reaction time over the first perturbation block is shown in *Figure 9A&B* (middle, green traces).

### Mazzoni and Krakauer, 2006

In this study, subjects sat in a chair with their arm supported on a tripod. An infrared marker was attached to a ring placed on the participant's index finger. The hand was held closed with surgical tape. Participants moved an on-screen cursor by rotating their hand around their wrist. These rotations were tracked with the infrared marker. On each trial, participants were instructed to make straight out-and-back movements of a cursor through 1 of 8 targets, spaced evenly in 45° intervals. A 2.2 cm marker translation was required to reach each target. Note that all eight targets remained visible throughout the task.

Two groups of participants were tested with a 45° visuomotor rotation. In the no-strategy group, participants adapted as per usual, without any instructions. After an initial null period, the rotation was turned on (*Figure 10A*, blue, adaptation). After about 60 trials of adaptation, the rotation was turned off and participants performed another 60 washout trials (*Figure 10A*, blue, washout). The break between the adaptation and washout periods in *Figure 10A*, no-strategy, is simply for alignment purposes.

The strategy group followed a different protocol. After the null period, participants reached for two movements under the rotation (*Figure 10A*, 2 cycles no instruction, red). At this point, the subjects were told that they made two errors, and that they could counter the error by reaching to the neighboring clockwise target (all targets always remained onscreen). After the instruction, participants immediately reduced their error to zero (point labeled instruction in red, *Figure 10A*). They continued to aim to the neighboring target throughout the adaptation period. Note that the directional errors became negative. This convention indicates overcompensation for the rotation, that is, participants are altering their hand angle by more than their strategic aim of 45°. Toward the end of the adaptation period, participants were told to stop re-aiming, and direct their movement back to the original target (*Figure 10A*, do not aim, rotation on). Then after several movements, the rotation was turned off as participants continued to aim for the original target during the washout period.

In *Figure 10A* we show the error between the primary target (target 1) and cursor during the entire experiment. In *Figure 10B*, we show the error between the aimed target (target 2) and cursor during the adaptation period. Note that the aimed and primary targets are related by 45° when the strategy group is re-aiming. We observed that initial adaptation rates (over first 24 movements, gray area in *Figure 10B*) were similar, but the no-strategy group ultimately achieved greater implicit adaptation. These data were all obtained by using the GRABIT routine in MATLAB 2018a to extract the mean (and standard error of the mean) performance in each group from the figures shown in the primary article.

We fit 1 of 3 models to the direction error during the adaptation period shown in *Figure 10B*. In all cases, we modeled explicit re-aiming in the strategy group as an aim sequence that started at zero during the initial two movements, and then 45° for the rest of the adaptation period (i.e. after the instruction to re-aim). In the no-strategy group, we modeled explicit learning as an aim sequence that remained at zero throughout the adaptation period.

In *Figure 10D*, we modeled implicit learning based on the state-space model in *Equation 3* and target error term defined in *Equation 1*. This target error was defined as the difference between the primary target (i.e. the target associated with task outcome) and the cursor. In *Figure 10E*, we modeled implicit learning based on the state-space model in *Equation 3* and the aim-cursor error defined in *Equation 2*. This aim-cursor error was defined as the difference between the aimed target (either 0° or 45°) and the cursor. *Figure 10F*, shows our third and final model. In this model, implicit learning in the strategy group was modeled using the dual-error system shown in *Equation 6*. That is, there were two implicit modules, one which responded to the target errors as in *Figure 10D*, and the other which responded to aim-cursor errors as in *Figure 10E*. The evolution of these errors is shown in *Figure 10G*. In the no-strategy group, we modeled implicit learning based on the primary target error and cursor alone.

Each model in *Figure 10D–F* was fit in an identical manner. We fit the implicit retention factor and implicit error sensitivity to minimize squared error according to:

$$\theta_{fit} = \underset{\theta}{argmin} \sum_{n=1}^{N} \left( y_{strategy}^{(n)} - \hat{y}_{strategy}^{(n)} \right)^2 + \left( y_{no\text{-}strategy}^{(n)} - \hat{y}_{no\text{-}strategy}^{(n)} \right)^2 \tag{11}$$

In other words, we minimized the sum of squared error between our model fit and the observed behavior across the strategy and no-strategy groups in *Figure 10B*. Therefore, we constrained each group to have the same implicit learning parameters. In the case of our dual-error model in *Figure 10F*, we assumed that each implicit module also possessed the same retention and error sensitivity. In sum, all model fits had two free parameters (error sensitivity and retention) which were assumed to be identical independent of instruction. This fit was performed using *fmincon* in MATLAB R2018a. The predicted behavior is shown in *Figure 10D–F* at bottom. For our best model (*Figure 10F*), the model behavior is also overlaid in *Figure 10B*.

### Taylor and Ivry, 2011

In *Figure 10I*, we show data collected and originally reported by *Taylor and Ivry, 2011*. In this experiment, participants moved their arm at least 10 cm toward 1 of 8 targets, that were pseudorandomly arranged in cycles of eight trials. Only endpoint feedback of the cursor position was provided. The hand was slid along the surface of a table while the position of the index finger was tracked with a sensor. After an initial familiarization block (five cycles), participants were trained how to explicitly rotate their reach angle clockwise by 45°. That is, on each trial they were shown veridical feedback of their hand position, but were told to reach to a neighboring target, that was 45° away from the primary illuminated target. After this training and another null period, the adaptation period started where the cursor position was rotated by 45° in the counterclockwise direction for 40 cycles. The first two movements in the rotation exhibited large errors (*Figure 10I*, 2 movements no instruction). As in *Mazzoni and Krakauer, 2006*, the participants were then instructed that they could minimize their error by adopting the aiming strategy they learned at the start of the experiment. Using this strategy, participants immediately reduced their direction error to zero.

Here, we report data from two critical groups in this experiment. In the 'instruction with target' group (*Figure 10I*, black, n = 10) participants were shown the neighboring targets during the adaptation period to assist their re-aiming. However, in the 'instruction without target' group (*Figure 10I*, yellow, n = 10) participants were only shown the primary target; the neighboring targets did not appear on the screen to help guide re-aiming. Only participants in the 'instruction with target' group exhibited the drift reported by *Mazzoni and Krakauer, 2006*. However, both groups exhibited an implicit aftereffect (*Figure 10I*, aftereffect; first cycle of washout period as reported in *Figure 4C* of the original manuscript *Taylor and Ivry, 2011*).

Data were extracted from the primary figures in *Taylor and Ivry, 2011* using Adobe Illustrator CS6. We used the means and standard deviations for our statistical tests on the implicit aftereffect in *Figure 10I*.

### Measuring properties of implicit learning

Many of our model's predictions depended on estimates of implicit retention factor and error sensitivity. We obtained these using the Limit PT groups in Experiments 2 and 3. To calculate the retention factor for each participant, we focused on the no feedback period at the end of Experiment 2

(*Figure 3E*, no feedback) and the no aiming period at the end of Experiment 3 (*Figure 3L*, no aiming). During these error-free periods trial errors were hidden, thus causing decay of the learned behavior. The rate of this decay is governed by the implicit retention factor according to:

$$y^{(n)} = a_i^n y_{ss} \qquad (12)$$

Here, $y^{(n)}$ refers to the reach angle on feedback trial $n$, and $y_{ss}$ corresponds to the asymptotic behavior prior to the no feedback period. We used *fmincon* in MATLAB R2018a to identify the retention factor which minimized the difference between the decay predicted by *Equation 12* and that measured during the no feedback period. For Experiment 2, we obtained an epoch-by-epoch retention factor of 0.943 ± 0.011 (mean ± SEM). Note that an epoch consisted of four trials, so this corresponded to a trial-by-trial retention factor of 0.985. When modeling *Neville and Cressman, 2018* (*Figure 1*), we cubed this trial-by-trial term because each cycle consisted of 3 different targets (final retention factor of 0.9565). For Experiment 3, we obtained an epoch-by-epoch retention factor of 0.899 (trial-by-trial: 0.9738).

Next, we measured implicit error sensitivity in the Limit PT group during rotation period trials. To measure implicit error sensitivity on each trial, we used its empirical definition:

$$b^{(n_1)} = \frac{y^{(n_2)} - a^{n_2 - n_1} y^{(n_1)}}{e^{(n_1)}} \qquad (13)$$

*Equation 13* determines the sensitivity to an error experienced on trial $n_1$ when the participant visited a particular target T. This error sensitivity is equal to the change in behavior between two consecutive visits to target T, on trials $n_1$ and $n_2$ divided by the error that had been experienced on trial $n_1$. In the numerator, we account for decay in behavior by multiplying the behavior on trial $n_1$ by a decay factor that accounted for the number of intervening trials between trials $n_1$ and $n_2$. For each target, we used the retention factor estimated for that target with *Equation 12*.

Using this procedure, we calculated implicit error sensitivity as a function of trial in Experiment 2. To remove any potential outliers, we identified error sensitivity estimates that deviated from the population median by over three median absolute deviations within windows of 10 epochs. As reported by *Albert et al., 2021*, implicit error sensitivity increased over trials. *Equations (4) and (5)* require the steady-state implicit error sensitivity observed during asymptotic performance. To estimate this value, we averaged our trial-by-trial error sensitivity measurements over the last 5 epochs of the perturbation. This yielded an implicit error sensitivity of 0.346 ± 0.071 (mean ± SEM).

To corroborate this value, we compared our estimate to data reported in *Kim et al., 2018*. There, error sensitivity is reported as a function of error size across various experiments in *Figure 3a*. These data are reproduced in *Figure 3—figure supplement 1C*. Note that error sensitivity increases as errors get smaller. For our analyses, we required steady-state error sensitivity, which is the error sensitivity reached at the end of the training period. *Figure 3—figure supplement 1B* shows how error in the PT-Limit group changed with adaptation. The terminal error (horizontal black line) corresponding to the steady-state condition was equal to about 7.6° (*Figure 3—figure supplement 1B*). For this error, error sensitivity fell somewhere between 0.25 and 0.35 (see *Figure 3—figure supplement 1C*) according to Experiments 1 and 2 reported by *Kim et al., 2018*. Thus, our value 0.346 appeared in agreement with these data.

Finally, we conducted a similar analysis in Experiment 3. However, trial-by-trial behavior was more variable and overall adaptation was lower in this laptop-based experiment. Thus, to obtain a more stable steady-state implicit error sensitivity estimate, we averaged error sensitivity over the asymptotic period apparent in *Figure 3—figure supplement 1D* (cycles 37–60). The average error sensitivity was approximately 0.193 (*Figure 3—figure supplement 1D*). To corroborate this value, we calculated the terminal error in the Limit PT group. This value was approximately 13.1° (*Figure 3—figure supplement 1E*). This error corresponded to an error sensitivity between about 0.13 and 0.22 (*Figure 3—figure supplement 1F*) according to *Kim et al., 2018*. Thus, our Limit PT error sensitivity estimate 0.193 was within this range.

## Acknowledgements

This work was supported by grants from the National Institutes of Health (R01NS078311, F32NS095706), and the National Science Foundation (CNS-1714623).

## Additional information

### Funding

| Funder | Grant reference number | Author |
|---|---|---|
| National Institute of Neurological Disorders and Stroke | F32NS095706 | Scott T Albert |
| National Science Foundation | CNS-1714623 | Reza Shadmehr |
| National Institute of Neurological Disorders and Stroke | R01NS078311 | Reza Shadmehr |

The funders had no role in study design, data collection and interpretation, or the decision to submit the work for publication.

### Author contributions

Scott T Albert, Conceptualization, Data curation, Formal analysis, Investigation, Methodology, Project administration, Software, Writing – original draft, Writing – review and editing; Jihoon Jang, Conceptualization, Data curation, Investigation, Methodology, Software, Writing – review and editing; Shanaathanan Modchalingam, Conceptualization, Data curation, Investigation, Methodology, Resources, Writing – review and editing; Bernard Marius 't Hart, Conceptualization, Data curation, Investigation, Methodology, Resources; Denise Henriques, Conceptualization, Data curation, Investigation, Methodology, Project administration, Resources; Gonzalo Lerner, Conceptualization, Data curation, Formal analysis, Investigation, Methodology, Resources, Writing – review and editing; Valeria Della-Maggiore, Conceptualization, Data curation, Formal analysis, Investigation, Methodology, Resources, Supervision, Writing – review and editing; Adrian M Haith, Conceptualization, Data curation, Investigation, Methodology, Resources, Supervision, Writing – review and editing; John W Krakauer, Conceptualization, Data curation, Funding acquisition, Investigation, Methodology, Resources, Supervision, Writing – review and editing; Reza Shadmehr, Conceptualization, Funding acquisition, Investigation, Methodology, Project administration, Supervision, Writing – original draft, Writing – review and editing

### Author ORCIDs

Scott T Albert http://orcid.org/0000-0001-9140-1077
Gonzalo Lerner http://orcid.org/0000-0002-7791-9408
Adrian M Haith http://orcid.org/0000-0002-5658-8654
John W Krakauer http://orcid.org/0000-0002-4316-1846
Reza Shadmehr http://orcid.org/0000-0002-7686-2569

### Ethics

Human subjects: Informed consent was obtained from all study participants. All human subjects work was approved by the Johns Hopkins School of Medicine Institutional Review Board (protocol number NA_00037510) or the York Human Participants Review Sub-committee.

### Decision letter and Author response

Decision letter https://doi.org/10.7554/eLife.65361.sa1
Author response https://doi.org/10.7554/eLife.65361.sa2

## Additional files

### Supplementary files

• Transparent reporting form

## Data availability

Source data files generated or analyzed during this study, as well as the associated analysis code, are included as supplements to Figures 1-10, as well as their associated Figure Supplements, and have also been deposited in OSF under accession code MZS6A.

The following dataset was generated:

| Author(s) | Year | Dataset title | Dataset URL | Database and Identifier |
|---|---|---|---|---|
| Albert ST, Jang J, Modchalingam S, Hart M, Henriques D, Lerner G, Della-Maggiore V, Haith AM, Krakauer JW, Shadmehr R | 2022 | Competition between parallel sensorimotor learning systems | https://osf.io/mzs6a/ | Open Science Framework, 10.17605/OSF.IO/MZS6A |

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

# Appendix 1

In the main text, the competition and independence models predict steady-state implicit learning, $x_i^{ss}$. But each theory applies to the entire timecourse, not solely asymptotic learning. Here, we will derive general expressions that apply on each trial $k$. We will then show that the scaling, saturation, and non-monotonic phenotypes in *Figure 1*, occur throughout the entire implicit learning timecourse (in the competition model).

## 1.1 General model derivation

Consider the state-space model where the implicit system adapts to target error driven by the rotation $r$, as in the competition model (this is the competition model simulated in *Figures 6 and 7*):

$$x_i^{(n)} = a_i x_i^{(n-1)} + b_i(r - x_i^{(n-1)} - x_e^{(n-1)}) \tag{A1.1}$$

Recall that $a_i$ and $b_i$ are implicit retention and error sensitivity. The $x_i^{(n)}$ and $x_e^{(n)}$ terms represent implicit and explicit learning on a given trial (numerator). This equation can be rewritten recursively to represent $x_i^{(n)}$ with respect to all prior trials:

$$x_i^{(n)} = (a_i - b_i)^{n-1} x_i^{(1)} + b_i \sum_{k=1}^{n-1} (a_i - b_i)^{n-k-1}(r - x_e^{(k)}) \tag{A1.2}$$

In the case where implicit learning starts at zero (naïve learner), this equation simplifies to:

$$x_i^{(n)} = b_i \sum_{k=1}^{n-1} (a_i - b_i)^{n-k-1}(r - x_e^{(k)}) \tag{A1.3}$$

*Equation A1.3* shows how the implicit system on trial $n$ is driven by the explicit system on all prior trials, the rotation, and the implicit error sensitivity and retention factor. An excellent approximation to this equation can be obtained by replacing $x_e^{(k)}$ as the average $x_e$ across all prior trials this approximation's accuracy can be seen in the red and cyan lines in *Figure 1—figure supplement 3A*; cyan shows true implicit learning in *Equation A1.3*; red shows the approximation in A1.4 below. This approximation yields:

$$x_i^{(n)} \approx b_i \frac{1-(a_i-b_i)^{n-1}}{1-(a_i-b_i)}(r - x_e^{(avg)}) \tag{A1.4}$$

*Equation A1.4* is analogous to the competition model in *Equation 4* in the main text. It states that implicit learning on a given trial $n$ is approximately proportional to $(r - x_e^{avg})$, the difference between the rotation and the average explicit strategy used by the participant. Note that in the limit as $n$ goes to infinity (i.e., steady-state), we obtain the competition model in *Equation 4*. The implication of *Equation A1.4* is that a linear competition between implicit learning and explicit learning can be observed throughout the adaptation process, not only in the asymptotic learning phase. Thus, the scaling, saturation, and non-monotonic phenotypes we describe in *Figure 1*, can be observed throughout the adaptation process.

Finally, note that the derivations above apply for the independence (SPE) model in *Equation 5*. For this model, simply set all $x_e$ terms in A1.1-A1.4 to zero.

## 1.2 Scaling, saturation, and non-monotonic phenotypes across the entire implicit learning timecourse

Here we illustrate the scaling, saturation, and non-monotonic implicit learning phenotypes produced by the competition model. In *Figure 1—figure supplement 3A*, we simulate the implicit and explicit response to a 30° rotation using A1.1 and its explicit learning analogue. This means that the implicit and explicit systems are driven by target error. Note that the red line shows the implicit approximation in A1.4 above, the blue line shows exact implicit learning in A1.3 above, and the magenta line shows explicit strategy. In this simulation, we used the $a_i$, $b_i$, and $a_e$ parameters identified in our model fit to *Haith et al., 2015*, as in the competition map shown in *Figure 7*.

The scaling, saturation, and nonmonotonic phenotypes in *Figure 1* are due to how the explicit system responds to changes in rotation size. When strategy increases slower than the rotation, implicit learning will increase in the scaling phenotype. When it increases at the same rate as the rotation, implicit learning will stay the same, leading to the saturation phenotype. When it increases more

rapidly than the rotation, implicit learning will decreases as the rotation increases: the nonmonotonic phenotype. All these scenarios are depicted in *Figure 1—figure supplement 1*.

Thus, to produce these phenotypes, we must change the way explicit strategy 'responds' to error: least vigorously in the scaling phenotype, and most vigorously in the nonmonotonic phenotype. To do this we simulated implicit and explicit responses to 30° and 45° rotations and tuned explicit error sensitivity: $b_e$. For all 30° simulations, $b_e$ remained 0.15. To obtain the scaling implicit phenotype, explicit error sensitivity remained 0.15 in the 45° simulation. To obtain the saturation implicit phenotype, we increased $b_e$ to 0.435 in the 45° condition (i.e., the explicit system became more reactive to the higher rotation). Lastly, to obtain the nonmonotonic phenotype, we increased the explicit error sensitivity dramatically, to 0.93. In *Figure 1—figure supplement 3B and C*, we calculate implicit and explicit learning at various points across these simulations: from left to right, 5, 10, 20, 40, and 150 rotation cycles. The explicit responses are shown in inset B. Implicit responses are shown in inset A.

There are many critical things to note. Firstly, at all time points, changes in explicit error sensitivity produce three distinct levels: less explicit learning when $b_e$ = 0.15, medium explicit learning when $b_e$ = 0.435, and high explicit learning when $b_e$ = 0.93. These changes have dramatic effects on the implicit system. For the low explicit strategy level, the implicit system scales when the rotation increases from 30° to 45°. For the medium explicit strategy level, the implicit system remains the same when the rotation increases from 30° to 45°. For the high explicit strategy level, implicit learning decreases when the rotation increases from 30° to 45°. Most critically, all three phenotypes occur at all phases in the implicit learning time course: as early as cycle 5, and as late as cycle 150 (compare each set of bars in *Figure 1—figure supplement 3B*). The only thing that changes is the total difference (i.e. effect size) between 30° and 45° implicit learning, which is smallest and hardest to detect early in learning (e.g. cycle 5), and largest (easiest to detect) when implicit learning reaches its steady-state (e.g. cycle 150).

What this means is that the competition model does not solely pertain to asymptotic learning. The same phenomena that occur due to implicit-explicit competition, appear throughout all phases in the learning process. To simplify matters we have chosen in our main text to focus on steady-state learning, where the mathematical relationship between steady-state implicit and explicit learning converges and is easy to test (i.e. the competition equation).

## Appendix 2

In *Figure 1H–L*, we applied the competition and independence theories to the stepwise group in Exp. 1. While these models do not solely apply to asymptotic learning (see Appendix 1), here we analyze whether the exposures during B1 (15°), B2 (30°), B3 (45°), and B4 (60°), lasted long enough to achieve implicit steady-state adaptation. That is, the scaling phenotype we noted in the implicit response could be influenced by two factors: (1) rotation size, and (2) exposure time.

In the stepwise condition in Exp. 1 we observed that implicit adaptation increased across the 15°, 30°, 45°, and 60° rotation blocks (*Figure 1—figure supplement 4A*; rm-ANOVA, F(3,38)=99.9, p < 0.001, $\eta_p^2$=0.735). Were these increase due to changing the rotation's magnitude, or additional accumulation in the implicit response over time which had not yet saturated? Comparison with other comparable data sets, suggests that implicit learning likely saturated during each block in the Exp. 1 stepwise group. Here, we will describe each data set in turn.

Most importantly, the abrupt rotation group provides a way to verify the timecourse of implicit learning in Exp. 1. Total implicit learning was assayed four times in the abrupt group. Each probe overlapped with a stepwise rotation period. For example, the initial probe in the abrupt condition occurred during B1, when implicit learning was measured in the 15° stepwise rotation. Similarly, the B2, B3, and B4 implicit measurements in the abrupt group overlapped with the 30°, 45°, and 60° rotation periods in the stepwise group. Implicit learning measures across all four abrupt periods are shown in *Figure 1—figure supplement 4B*. We tested whether total implicit learning varied across the four blocks in the abrupt condition. We did not detect a statistically significant effect of block number on implicit learning (rm-ANOVA, F(3,105)=2.21, p = 0.091, $\eta_p^2$=0.059). The same was true when we compared solely the first and last blocks (paired t-test, t(35)=1.53, p = 0.134). This indicated that in the 60° rotation condition there was little to no change in implicit learning following B1. This suggests that a single rotation block was sufficient to achieve steady-state adaptation even in the largest rotation condition tested: 60°.

Does this generalize to smaller rotation sizes? While we did not measure implicit learning across multiple blocks in the 15°, 30°, and 45° stepwise conditions, we used a very similar experimental protocol in *Salomonczyk et al., 2011*: here three learning targets were also used, separated by 45° between targets, in an upper triangular wedge (the exact same conditions as Exp. 1). Note that these data provided another example where the implicit response scales with rotation size during a stepwise rotation sequence (*Figure 1—figure supplement 4C*). That is, when we increased the rotation in a stepwise manner across three blocks: 30° then 50° and lastly 70°, we observed strong increases in asymptotic implicit learning (p < 0.001). In this study we exposed another participant group to a 30° rotation and measured implicit learning across three consecutive blocks (*Figure 1—figure supplement 4D*). Critically, we did not detect any change in implicit learning across the three learning periods (all contrasts, p > 0.05). This matched our 60° rotation analysis in the abrupt group in Exp. 1. Thus, for both 30° and 60° rotations, a single block provides enough time to reach steady-state learning.

Lastly, we did not measure extended exposures to a 15° or 45° rotation, but *Neville and Cressman, 2018* tested prolonged exposure to a 20° and 40° rotation in a similar paradigm (three targets, separated by 45° in an upper triangular wedge). No-instruction group implicit learning is shown in *Figure 1—figure supplement 4E and F*. We do not have access to the raw data, and so cannot run a repeated-measures ANOVA to test whether these groups exhibited a block-by-block change in implicit learning. However, comparing the first and third rotation blocks, we can calculate a maximum 0.7°–1.4° c hange in implicit learning. This represents a change between 7.6–16.5%, which are daunted by the 313% increase in implicit learning exhibited from the 15° to the 60° blocks in the Exp. 1 stepwise group.

In sum, the 60° implicit learning timecourse we measured in the abrupt group, the 30° implicit learning timecourse measures in *Salomonczyk et al., 2011*, and the 20° and 40° groups in *Neville and Cressman, 2018.*, suggest that the implicit learning system exhibits little to no increase beyond the initial learning block. Each study was extremely comparable, testing participants with three learning targets, located in an upper triangular wedge spanning 45–135° in the workspace. We estimate that any changes in implicit learning due to exposure time alone were limited to about 1.4°, which is an order magnitude smaller than that observed across the stepwise blocks in Exp. 1 (14.5°). Therefore, our Exp. 1 analyses in *Figures 1, 2 and 4*, likely reflects steady-state implicit learning, or very nearly so.

## Appendix 3

In Section 1.1, we note a potential concern in our *Figure 1* analysis, where our implicit and explicit learning measures were not independent; explicit strategy was estimated as total adaptation minus implicit learning: $x_e^{ss} = x_T^{ss} - x_i^{ss}$. Thus, when total adaptation is constant, implicit and explicit learning will exhibit a negative correlation simply due to the dependencies in our implicit and explicit learning measures. At a glance, this negative correlation may appear similar to the negative correlation between implicit and explicit learning embedded within the competition equation: $x_i^{ss} = p_i(r - x_e^{ss})$. Thus, it is reasonable to question whether the correspondence between our empirical data and the competition model in *Figure 1*, is unfairly biased by the intrinsic implicit-explicit dependencies in our empirical measures.

In short, suppose that two conditions, *A* and *B*, exhibit an increase in implicit learning. Using our explicit learning measures, we would expect a decrease in explicit strategy. This negative correspondence between implicit learning and explicit strategy may appear trivially predicted by the competition model. This, however, is not the case. For this trivial relationship to occur, total adaptation must be the same in *A* and *B*. In *Figure 1*, we calculate implicit and explicit learning measures across changes in rotation size, over which total adaptation varies in proportion to rotation magnitude. Thus, our empirical implicit and explicit learning measures will not necessarily match the competition model.

To explain this, let us consider a hypothetical scenario similar to the data recorded by Neville and Cressman. Suppose total adaptation and implicit learning are measured over three rotations sizes (20°, 40°, and 60°): total learning (19°, 36°, 53°), implicit learning (10°, 10°, 10°). As expected, total adaptation increases with rotation size. Implicit learning remained constant, as in the Neville and Cressman data set. Next, suppose we estimate explicit strategy, by subtracting total adaptation and implicit learning. This will yield the explicit strategy estimates: (9°, 26°, 43°). Thus, in this example, our implicit and explicit learning measures are dependent. How well does the competition model match these data?

To answer this question, we can calculate the implicit learning gain, $p_i$, for each rotation size. This can be calculated as $p_i = x_i^{ss}/(r - x_e^{ss})$. For the example above, the implicit learning gain would be equal to the set: (0.909, 0.714, 0.588). What this means, is that as the rotation increases, the implicit learning gain decreases by approximately 35.3%. These variations in $p_i$ will not produce a good match between the data and model. To show this, we can estimate the optimal $p_i$ that would minimize the squared error between the measured implicit learning, and model-predicted implicit learning. In this example, the optimal $p_i$ value is 0.693. Using this gain, we can now use the model to predict how much implicit learning should occur in each rotation size condition. This would yield: (7.62°, 9.70°, 11.78°) in predicted learning. We can see even though implicit learning remained constant in the 'real' data, the competition equation predicts a 54.5% increase in implicit learning across the three rotation sizes.

What does this mean? This hypothetical example demonstrates that calculating explicit learning via a subtraction between total adaptation and implicit learning, will not automatically yield a good match between the empirical data and the competition model. Rather, the only way that the competition model will match data, is in the case where the conditions tested in an experiment abide by the principle that $p_i$ is constant (or at least very similar) across various experimental conditions (e.g. rotation sizes). The three data sets we analyze in *Figure 1*, must obey this property, and thus, are intrinsically compatible with *Equation 4*.

Nevertheless, there is another way to corroborate the relationship between the model and data in *Figure 1* that does not involve using implicit and explicit learning measures at the same time. By noting that $x_e^{ss} = x_T^{ss} - x_i^{ss}$, we can substitute this expression into *Equation 4* to obtain a relationship between implicit learning and total adaptation: $x_i^{ss} = p_i(1 - p_i)^{-1}(r - x_T^{ss})$. This equation represents an alternate way to express the competition model which can be compared with the empirical data. Fortunately, here we compare $x_i^{ss}$ and $x_T^{ss}$ which were directly measured on separate trials (they are not dependent, in a statistical sense).

A simple thought experiment shows that when data do not agree with the competition model, $x_i^{ss}$ and $x_e^{ss}$ can show a correlation, but $x_i^{ss}$ and $x_T^{ss}$ will not (in general). Suppose 3 participants have 10°, 15°, and 5° implicit learning, respectively, and all have 20° total learning. Subtracting total learning and implicit learning will yield an estimated 10°, 5°, and 15° explicit strategy, respectively. The

competition model will predict that implicit-explicit and implicit-total will show negative correlations. For these participants, the implicit-explicit correlation is –1, but the implicit-total correlation is 0, inconsistent with competition. That is, implicit-explicit show a correlation spuriously, but implicit-total were sampled independently, yielding a robust way to test the competition theory.

Thus, in the main text we repeated our competition model analysis in *Figure 1* using the alternate equation. We used total adaptation measured in Exp. 1, Neville and Cressman, and *Tsay et al., 2021a* to predict implicit learning. To do this, we identified a $p_i$ value in each data set that minimized the squared error between measured implicit learning and model-predicted implicit learning (in Neville and Cressman, the $p_i$ value was calculated across six conditions (three rotations, 2 instruction types), in Exp. 1, the $p_i$ value was calculated across five conditions (all four rotation sizes in the stepwise group, as well as the 60° abrupt condition), in Tsay et al., the $p_i$ value was calculated across all four rotation sizes).

Our results are shown in *Figure 1—figure supplement 2*. The competition model predicted nearly identical implicit learning patterns when total adaptation was used as the independent variable ('model-2') and when explicit adaptation was used as the independent variable ('model-1'). This control analyses shows that the close correspondence between measured behavior and model-predicted behavior in *Figure 1*, is not due to the way we empirically operationalized our learning measures. Rather, the properties embedded in the competition model (with a constant implicit learning gain) were intrinsically present within the data.

## Appendix 4

At various points in our work, we analyze how well the competition model, or independence model, can predict changes in implicit learning across various conditions. These models have one free parameter, the implicit learning gain: $p_i = b_i(1 - a_i + b_i)^{-1}$. In each case, we assumed that the implicit learning gain is the same across experimental groups, that is only one gain is selected and applied to all experimental conditions. We made this choice to minimize the number of free variables; allowing $p_i$ to vary across groups, would allow arbitrarily precise matches to the data. However, holding $p_i$ constant across conditions, shows that the same equation applies across conditions, limiting overfitting and increasing model confidence.

This assumption, however, may seem inappropriate given that the implicit learning gain depends on error sensitivity, which varies with conditions such as error size (*Albert et al., 2021*; *Kim et al., 2018*; *Marko et al., 2012*). However, it is important to note that the implicit learning gain responds weakly to changes in error sensitivity. This insensitivity is due to the appearance of error sensitivity ($b_i$) in both the gain's numerator and denominator: $p_i = b_i(1 - a_i + b_i)^{-1}$. For example, let us suppose that participants in Condition 1 have $b_i = 0.2$, but in Condition 2 $b_i = 0.3$, a 50% increase. For an implicit retention factor of 0.9565 (see in Materials and methods: *Measuring properties of implicit learning*), the implicit learning gain in Condition 1 would be $p_i = 0.821$ versus $p_i = 0.873$ in Condition 2. Thus, even though implicit error sensitivity was 50% larger in Condition 2, the implicit learning gain would change only 6.3%. For a more extreme case where implicit error sensitivity was double (0.4) in Condition 2, there would still only be a 9.8% change in implicit learning gain.

For these reasons, there are no physiologic changes in $b_i$ that could create the 46.2% increase in implicit learning in *Figure 2C* (ratio of no instruction to instruction groups), and 82.3% increase in learning in *Figure 2G* (ratio of stepwise to abrupt). To show this, we conducted sensitivity analyses. Imagine that across two conditions, a reference condition, and a test condition, implicit sensitivity varies. We set the reference implicit error sensitivity to 0.1, 0.1625, 0.225, 0.2875, 0.35. We chose these values, because steady-state implicit error sensitivity determines steady-state implicit learning, which varied between 0.19 and 0.35 in Exps. 2 and 3; see *Figure 3—figure supplement 1*. Then we calculated how much $p_i$ (total learning) will change, when this reference error sensitivity increases to a test error sensitivity: test error sensitivity was capped at the physiologic, yet exceedingly unlikely, upper bound of 1. Because $p_i$ also depends on implicit retention, we also tested several possible values between 0.95–0.99. The results are depicted in *Figure 2—figure supplement 2*. Each column denotes the same analysis, but with a different retention factor. Each line denotes a different reference error sensitivity. The x-axis continuously varies the test error sensitivity.

For example, for the point highlighted by the black arrow in the second column: this point shows that total implicit learning will increase by about 20% (y-axis) in a scenario where implicit retention = 0.96, and implicit error sensitivity increases from 0.1625 (i.e., it is on the red line) in the reference condition to 0.8 (i.e., the x-axis value) in the test condition. This example is illustrative. Here $b_i$ increase from 0.1625 to 0.8, a 392% increase, but total implicit learning only increases 20%. In sum, extreme variations in implicit error sensitivity, still produce small changes in total implicit learning.

In *Figure 2—figure supplement 2*, we indicated the 46.2% and 82.3% changes in *Figure 2C&G* with dashed horizontal lines. Note how no curve crosses either level, even though these represent several hundred % changes in implicit error sensitivity. Thus, it is not possible that variations in implicit error sensitivity could generate the changes in total implicit learning we examined in *Figure 2*.

On the other hand, consider the competition model, $x_i^{ss} = p_i(r - x_e^{ss})$. Distributing the $p_i$ term yields: $x_i^{ss} = p_i r - p_i x_e^{ss}$. Note that $p_i$ varied between about 0.6–0.8 in our studies, meaning that implicit learning is very sensitive to competition. The implicit system will decrease by approximately 60–80% each unit increase in explicit strategy. For these reasons, the competition model more readily produces large fluctuations in implicit learning, such as those observed in *Figure 2*.

## Appendix 5

In Experiment 1 in the main text, we analyze how abrupt and gradual perturbation onset alters the steady-state distribution for implicit and explicit learning. In this appendix, we conduct a similar investigation for data collected by *Saijo and Gomi, 2010*. In the first section, we describe the results of our analysis. In the second section, we describe the experimental paradigm and detail the methods used in our analysis.

### 5.1 Suppressing explicit strategy enhances procedural learning

In the main text (*Figure 2D–G*), we observed that gradual perturbations reduce explicit re-aiming relative to abrupt rotations. The competition theory predicts that these reductions in aiming will lead to increases in implicit adaptation. That is, suppose participants adapt with an explicit strategy (*Figure 2—figure supplement 3B*, aim, solid magenta line), but this strategy is then suppressed (*Figure 2—figure supplement 3B*, aim, dashed magenta line). Reductions in explicit strategy will increase the residual steady-state target error that drives implicit adaptation. Thus, *Equation 4* predicts that suppressing explicit aiming will increase implicit learning (*Figure 2—figure supplement 3B*, H2, right, compare dashed blue and solid black implicit lines). However, a model where the implicit system responds to SPEs (*Equation 5*) does not predict any change in implicit learning.

We corroborated these predictions in Experiment 1 (*Figure 2D–G*). Here, we describe similar data collected by *Saijo and Gomi, 2010*. Participants were exposed to an abrupt (*Figure 2—figure supplement 3A*, abrupt) or gradual (*Figure 2—figure supplement 3A*, gradual) rotation. The abrupt perturbation was immediately set to 60°, but the gradual perturbation reached this magnitude over time, in six 10° steps. Each rotation size persisted for 36 trials (three cycles, 12 adaptation targets). Participants in the abrupt condition adapted rapidly to the perturbation, greatly decreasing their target error to about 5° over about 10 perturbation cycles (*Figure 2—figure supplement 3C*, abrupt). In the gradual group, target errors remained small throughout training. Curiously, total adaptation was smaller in the gradual condition; participants exhibited a terminal error nearly three times greater than the abrupt condition (*Figure 2—figure supplement 3C*, gradual).

At this point, the perturbation was abruptly removed, revealing large aftereffects in each group. Paradoxically, even though participants in the gradual group had adapted less completely to the rotation, they exhibited larger aftereffects (*Figure 2—figure supplement 3F*, data), which remained elevated throughout the entire washout period (*Figure 2—figure supplement 3C*, aftereffect).

Here, we demonstrate that these patterns are consistent with the competition theory, provided we make several assumptions. First, we assume that the initial washout cycle is driven by implicit learning. While this is likely not entirely true, we should note that when *Morehead et al., 2015* measured aim reports during washout with a 45° rotation, participants appeared to immediately "turn off" their explicit strategy. Nevertheless, this assumption likely introduces some error in our modeling approach. Second, we assume that participants in the abrupt group used larger strategies than participants in the gradual group. This assumption seems reasonable, considering that we also measured larger strategies in the abrupt group in Experiment 1, as did *Yin and Wei, 2020*. Furthermore, as shown in *Figure 2* in the primary manuscript (*Saijo and Gomi, 2010*), subjects in the abrupt group exhibited a sharp increase in reaction time upon rotation onset, consistent with explicit strategy use (*Fernandez-Ruiz et al., 2011*; *McDougle and Taylor, 2019*).

With these assumptions, we investigated how well the competition and independence models predicted the observed data. To simulate these models, we estimated the explicit strategies in each group. (*Neville and Cressman, 2018*) measured the explicit response to a 60° rotation, demonstrating that participants re-aimed their hand approximately 35° consistently over the adaptation period (see yellow points in *Figure 2—figure supplement 3D and E*, explicit aim). This estimate agreed well with the data; participants in the abrupt condition adapted 55° and exhibited an aftereffect of approximately 20° (*Figure 2—figure supplement 3F*, data, abrupt), suggesting about 35° of re-aiming. In the gradual group, we assumed that little to no re-aiming occurred. This also seemed consistent with the data; participants in the gradual group adapted approximately 40° and exhibited an aftereffect of approximately 38° (*Figure 2—figure supplement 3F*, data, gradual) suggesting <5° of re-aiming. Using these estimates, we constructed hypothetical explicit learning timecourses, as shown in *Figure 2—figure supplement 3D and E*, explicit aim.

We next used the state-space model to simulate the implicit learning timecourse, in cases where the implicit system learned solely due to SPE (*Figure 2—figure supplement 3D*, implicit angle) or solely due to target error (*Figure 2—figure supplement 3E*, implicit angle), under the assumption that participants in both the abrupt and gradual groups had the same implicit error sensitivity ($b_i$) and retention factor ($a_i$). The parameter sets that yielded the closest match to the measured behavior (*Figure 2—figure supplement 3C*) are shown in *Figure 2—figure supplement 3D and E*. In both cases, the models predicted that abrupt learning would be more complete than gradual learning (i.e. steady-state error is smaller in the abrupt condition).

However, the implicit states predicted by SPE learning and target error learning possessed a critical difference. According to *Equation 4*, the target error model predicted that the total extent of implicit learning would be suppressed by explicit strategy in the abrupt condition, yielding a smaller aftereffect (*Figure 2—figure supplement 2E*, implicit angle). However, according to *Equation 5*, the SPE model predicted that implicit learning should reach the same level, yielding identical aftereffects (*Figure 2—figure supplement 3D*, implicit angle).

In summary, the differences in aftereffects across the abrupt and gradual conditions (*Figure 2—figure supplement 3F*, data) were accurately predicted by the competition theory (*Figure 2—figure supplement 3F*, competition), but not the independence equation (*Figure 2—figure supplement 3F*, indep.). Suppressing explicit strategy revealed competition between implicit and explicit systems which suggested that the implicit system predominantly responded to target error. Furthermore, it is interesting to note that these data were consistent with our observation that steady-state adaptation is greater when explicit learning is large and implicit adaptation is small (*Figure 5G&H*). These trends are consistent with competition (*Figure 5D–F*).

## 5.2 Methods used to analyze *Saijo and Gomi, 2010*

To understand how suppressing explicit strategy might alter implicit learning, we considered data collected by *Saijo and Gomi, 2010*. In one of their experiments, the authors tested how perturbation onset altered the adaptation process. Subjects were divided into either an abrupt (n = 9) or gradual group (n = 9), and reached to 1 of 12 targets, which were ordered pseudorandomly in each cycle of 12 trials. After a baseline period of 8 cycles, a visuomotor rotation was introduced. The perturbation period lasted 32 cycles. After this, the perturbation was removed for 6 cycles of a washout condition. Participants were exposed to either an abrupt rotation where the perturbation magnitude suddenly changed from 0° to 60°, or a gradual condition where the perturbation magnitude increased over smaller increments (10° increments that lasted three cycles each, *Figure 2—figure supplement 3A*).

Here, we considered why participants in the abrupt perturbation condition achieved greater adaptation during the rotation period (smaller error in *Figure 2—figure supplement 3C*) but exhibited a smaller aftereffect when the perturbation was removed. Our theory suggested that this may be due to competition. If the gradual condition suppressed explicit awareness of the rotation (*Yin and Wei, 2020*), then *Equation 4* would predict increases in implicit learning which were observed in the aftereffects measured during the washout period (where explicit strategies were disengaged). However, the SPE model (*Equation 5*) would predict the same amount of implicit adaptation: the same aftereffect in each condition.

To test these hypotheses, we simulated implicit adaptation using the state-space model in *Equation 3*. In *Figure 2—figure supplement 3D*, we used an SPE for the error term in *Equation 3*. In *Figure 2—figure supplement 3E*, we used the target error for the error term in *Equation 3*. We imagined that the total reach angle was determined based on the sum of implicit and explicit learning. However, these authors did not directly measure explicit strategies. Fortunately, *Neville and Cressman, 2018* measured explicit strategies using inclusion and exclusion trials during a 60° abrupt rotation (yellow points, explicit aim in *Figure 2—figure supplement 3D and E*).

We used these measurements in our abrupt simulations. Neville and Cressman observed that explicit strategies rapidly reached 35.5° and remained stable during adaptation. To approximate these data, we simulated abrupt explicit strategy using the exponential curve: $x_e = 35.5 - 10e^{-2t}$ (*Figure 2—figure supplement 3D and E*, explicit aim, black line). Note that the nature of this exponential curve is entirely inconsequential to our analysis, apart from its saturation level. Outside of the rotation period, we assumed explicit strategy was zero. This is consistent with data from *Morehead et al., 2015* that showed almost immediate disengagement in aiming strategy during washout (though this assumption likely introduced error into our modeling approach). For the gradual

condition, we assumed explicit strategy was zero throughout the entire experiment (*Figure 2—figure supplement 3D*, explicit aim, gradual), as the participants remained largely unaware of the rotation. This seemed consistent with the data; gradual participants adapted approximately 40°, and exhibited an aftereffect of about 38°, indicating a re-aiming angle less than even 5°. Note, our primary results (*Figure 2—figure supplement 3F*) were unchanged in a sensitivity test where we assumed 10° of re-aiming in the gradual group (*Figure 2—figure supplement 4*).

Thus, our simulations included two free parameters: error sensitivity ($b_i$) and retention factor ($a_i$) for the implicit system. In each simulation, we assumed that these parameters were identical across the gradual and abrupt groups. To fit these parameters, we minimized the following cost function:

$$\theta_{fit} = \underset{\theta}{argmin}\sum_{n} \left( e^{(n)}_{abrupt} - \hat{e}^{(n)}_{abrupt} \right)^2 + \sum \left( e^{(n)}_{gradual} - \hat{e}^{(n)}_{gradual} \right)^2 \tag{A5.1}$$

*Equation A5.1* is the sum of squared errors between the directional errors predicted by the model (*Figure 2—figure supplement 3D and E*, directional error) and observed in the data (*Figure 2—figure supplement 3C*) across all trials in the abrupt and gradual conditions. Note that each simulation incorporated variability. We simulated noisy directional errors using the standard errors shown in the data in *Figure 2—figure supplement 3C*. In the explicit state, we added variability to each trial using the standard error in explicit strategy reported by *Neville and Cressman, 2018*. For the implicit state, we used 20% of the explicit variability, given that aiming strategies are more variable than implicit corrections (*Miyamoto et al., 2020*). We repeated these simulations 20,000 times, each time resampling our noise sources and then fitting our parameter set ($a_i$ and $b_i$) by minimizing *Equation A5.1* with *fmincon* in MATLAB R2018a. The mean implicit curves for the SPE learning model and target error learning models are shown in *Figure 2—figure supplement 3D and E* respectively (implicit angle; mean ± SD). Critically, in each simulation we measured the aftereffect that occurred on the first cycle of the washout period (*Figure 2—figure supplement 3D and E*, aftereffect). The mean and standard deviation in these aftereffects is reported in *Figure 2—figure supplement 3F*.

Finally, note that we obtained the directional errors in *Figure 2—figure supplement 3C* directly from the primary figure in the original manuscript (using the GRABIT routine in MATLAB R2018a). Please also note in the actual experiment, on some trials (7.1% of all trials), the perturbation was introduced midway during the reach to test feedback corrections at only one target location (the 0° target). These trials were not relevant for our current analysis. Otherwise, the visuomotor rotation was applied during the entire movement. Also note that because the authors analyzed feedback responses, participants made 15 cm movements, with a 0.6 second movement duration at baseline. Here, we only wanted to consider the feedforward adaptive component. Fortunately, the authors reported initial movement errors 100ms following movement onset that could not have been altered by feedback. Therefore, we used these early measures of adaptation in the current study.

## Appendix 6

Adapted movement patterns exhibit generalization: a decay in adaptation measured when participants reach towards new areas in the workspace (*Hwang and Shadmehr, 2005*; *Krakauer et al., 2000*; *Fernandes et al., 2012*). Recent studies have observed that this generalization is centered where participants aim their movement, as opposed to the visual target (*Day et al., 2016*; *McDougle et al., 2017*). In Exps. 1–3, we measured implicit adaptation by instructing participants to aim directly to the target without an explicit strategy. Had implicit learning truly been 'centered' at the aiming location and not the target, these no aiming trials may underapproximate total implicit learning. This discrepancy will increase with re-aiming. Thus, it may appear that individuals who aim more, possess less implicit learning. These generalization properties could contribute to the negative subject-to-subject relationships we observed in *Figure 3*. In other words, could an SPE model with plan-based generalization produce the implicit-explicit correlations that we observed in *Figure 3*? Our main text explores this possibility in *Figure 4*. Here, we expand on these analyses to provide additional intuition, derivations, and technical background.

### 6.1 Comparing our data to past generalization curves

We compared data in Exps. 2 and 3 (*Figure 4A–C*), with generalization curves measured by *Krakauer et al., 2000*, *Day et al., 2016*, and *McDougle et al., 2017*. This last study is most important because aiming was controlled during generalization measurements and CW and CCW rotations were counterbalanced. In *Figure 4A* we normalized our data so that true implicit learning equals measured implicit learning (100% generalization) when the re-aiming angle is 0°. To estimate total implicit learning (the value we normalized to), we used the y-intercepts corresponding to the regressions in *Figure 3* (25.5° and 19.7° in Exps. 2 & 3) which extrapolates total implicit learning when explicit strategy is zero. Note that dimensioned data (in degrees) are shown in *Figure 4B&C*. As described in our Results, these comparison showed that implicit learning in Exps. 2 and 3 (*Figure 4A*) declined nearly 300% faster than the generalization curves predicted. For example, given the explicit strategies used in the No PT Limit groups in Exps. 2 and 3, McDougle et al. would predict about a 5° reduction in implicit learning, whereas the actual data varied by about 15–20°. SPE generalization was too small in magnitude to match our data.

This analysis, however, has a critical limitation. Assuming that our implicit learning measures are altered by generalization, the explicit strategy we estimated will be affected too. That is, in Experiments 1–3 explicit strategy was estimated as total adaptation minus implicit learning. Had generalization reduced the implicit measures, it would falsely inflate our explicit measures. While it is tempting to compare our data in *Figure 2* or *Figure 4A* with past generalization curves, this should not be done without correcting the explicit strategy measures. Such corrections create a substantial narrowing in the generalization curve. In *Figure 4C* we show corrected implicit-explicit generalization curves that best match the data in Exps. 2 and 3. This process is described in Appendix 6.2. These curves had σ = 5.16° and 5.76° in Exps. 2 and 3, respectively, which is about 80–85% narrower than that observed in *McDougle et al., 2017* (σ = 37.76°). Altogether, to obtain the implicit-explicit relationships we report in Exps. 2 and 3 would require unphysiological generalization curves that are an order of magnitude narrower than any study has reported to date.

### 6.2 Deriving SPE generalization models

The competition model (*Equation 4*) can be represented as $x_i^{ss} = p_i(r - x_e^{ss})$, where $p_i$ is an implicit learning gain (between 0 and 1). Multiplying out $p_i$ yields, $x_i^{ss} = p_i r - p_i x_e^{ss}$. Assuming $p_i$ is roughly constant (see Appendix 4), this expression predicts implicit learning with vary with explicit strategy according to a line with slope $-p_i$ and bias $p_i r$. In other terms, while the bias in implicit learning will increase with rotation size, the slope that relates implicit and explicit learning should not be altered by the rotation's magnitude.

Let us now derive an SPE generalization model. In this model implicit learning is driven by SPEs, and also exhibits plan-based generalization when it is measured. To derive this model, we begin with *Equation 5*, the independence model: $x_i^{ss} = p_i r$. This encodes SPE learning. Next, we add generalization; measured implicit learning (what we obtained when participants are instructed to stop aiming) is related to steady-state implicit learning by the generalization curve, $g$, which varies with steady-state explicit strategy, $x_e^{ss}$. Thus, we have: $x_i^{measured} = x_i^{ss} g(x_e^{ss})$. Normally, this generalization curve would be modeled via a nonlinear cosine, or normal, tuning function. However, our data suggested that implicit and explicit varied linearly. Thus, we considered two model classes: (1) $g(x_e^{ss})$

is linear and (2) $g(x_e^{ss})$ is Gaussian. For the linear model, $g(x_e^{ss}) = 1 + mx_e^{ss}$ and for the normal model, $g(x_e^{ss}) = \exp(-(x_e^{ss}/\sigma)^2)$. By combining together all equations, we get: $x_i^{measured} = p_i r(1 + mx_e^{ss})$ for the linear model and $x_i^{measured} = p_i r \exp(-(x_e^{ss}/\sigma)^2)$ for the normal model. Here $m$ is the generalization line's slope (negative), and $\sigma$ is the normal distribution's standard deviation.

For the linear model, this indicates that measured implicit learning and explicit strategy will vary according to a line with slope $p_i rm$ and bias $p_i r$. Thus, the rotation size will alter the slope relating implicit and explicit learning. In the Gaussian generalization model, nonlinearity also contributes to this variation. In the competition model, the slope does not vary with rotation size (see equation above). Thus a key way to compare competition with generalization is to examine whether the implicit-explicit slope changes with rotation size. This is discussed in more detail in Appendix 6.3.

Critically, $x_e^{ss}$ in the SPE generalization model is equal to the total steady-state explicit strategy. In Exps. 1–3, our Tsay et al. analysis, and our Neville and Cressman analysis, we did not measure total explicit strategy. Rather our explicit strategies were estimated by subtracting total adaptation and implicit learning. Thus, we cannot fit the SPE generalization model to our data without correcting our explicit strategy measures as our explicit strategies will overapproximate true explicit strategy. To say this another way, suppose that the implicit learning we measured, $x_i^{measured}$ is less than total implicit learning $x_i^{ss}$. The explicit strategy we calculated was total adaptation, $x_T^{ss}$, minus $x_i^{measured}$. It should instead be $x_T^{ss} - x_i^{ss}$. We need to correct the explicit measures in the SPE generalization model.

To do this, start with a Gaussian generalization model:

$$x_i^{measured} = x_i^{ss} exp(-0.5(x_e^{ss}/\sigma)^2). \tag{A6.1}$$

Generalization causes a discrepancy between measured implicit learning and total implicit learning. The total amount that implicit learning is underapproximated, $x_i^{ss} - x_i^{measured}$, is equal and opposite to the total amount that explicit learning is overapproximated. Thus, we have:

$$x_e^{measured} = x_e^{ss} + x_i^{ss} - x_i^{measured}. \tag{A6.2}$$

*Equation A6.2* can be rearranged:

$$x_e^{ss} = x_e^{measured} - x_i^{ss} + x_i^{measured}. \tag{A6.3}$$

Combining *Equations A6.1 and A6.3* yields the expression:

$$x_e^{ss} = x_e^{measured} - x_i^{ss} + x_i^{ss} exp(-0.5(x_e^{ss}/\sigma)^2). \tag{A6.4}$$

*Equations A6.1 and A6.4* together express and constrain the relationship between (1) total implicit learning, (2) total explicit learning, (3) measured implicit learning, and (4) measured explicit learning. The same process can be used to correct the linear SPE generalization model. Here, we begin with the linear implicit generalization equation:

$$x_i^{measured} = x_i^{ss}(1 + mx_e^{ss}). \tag{A6.5}$$

As above, the discrepancy $x_i^{ss} - x_i^{measured}$ will be equal and opposite to the discrepancy between $x_e^{ss}$ and $x_e^{measured}$. Thus,

$$x_e^{measured} = x_e^{ss} + x_i^{ss} - x_i^{measured}. \tag{A6.6}$$

Combining these equations together yields

$$x_i^{measured} = x_i^{ss}(1 + mx_e^{measured}/(1 - mx_i^{ss})). \tag{A6.7}$$

We fit the SPE generalization models to the data noting that $x_i^{ss} = p_i r$ in the generalization model. These models have two unknown parameters: $p_i$ and $m$ in the linear model, and $p_i$ and $\sigma$ in the normal model. In some cases (e.g. *Figure 4D–F*), $m$ and $\sigma$ were set equal to the implicit generalization properties in *McDougle et al., 2017* (more on this below). When these parameters were set, we used *fmincon* to identify the optimal $p_i$ that minimized the squared error between measured implicit learning, and the implicit learning predicted by *Equations A6.1* (Gaussian) and *Equation A6.5*

(linear) above. Note in the Gaussian model, we also needed to constrain that the model's solutions satisfy *Equation A6.4*.

In other cases, we did not constrain *m* or σ. Instead, we used *fmincon* (MATLAB R2021a) to identify the optimal $p_i$ and *m* (linear), or $p_i$ and σ (Gaussian), that minimized the squared error as described above.

### 6.3 Comparing the competition model to SPE generalization models

In *Figure 4D–F*, we compare the competition model to the SPE generalization models above. We did this in multiple ways. Note that the competition model possesses one unknown parameter, $p_i$. The SPE models possess two, *m* and $p_i$ in the linear model, and *m* and σ in the normal model. In one analysis, we used the data collected by *McDougle et al., 2017*, to estimate the generalization parameters *m* and σ. To estimate *m*, we used the initial two points on the generalization curve in *Figure 4A*. This yielded *m* = –0.011. For the σ parameter, we used the value calculated by McDougle et al. for their data: σ = 37.76°. These parameters were used in *Figure 4D–F*, in the SPE gen. linear and SPE gen. normal models.

In *Figure 4D–F*, we fit all three models to the stepwise group's implicit and explicit learning measures in B4 in Exp. 1 (the 60° rotation period). The fitting procedure is described in Appendix 6.2 above. This revealed the optimal $p_i$ that best matched the implicit learning measures. Next, we used these parameters to predict the implicit-explicit relationship across the held-out rotation sizes (i.e., the B1, B2, and B3 periods). Each model's curve is shown in *Figure 4D*.

To determine how well each model generalized to the held-out 15°, 30°, and 45° rotations, we calculated the RMSE between the model and measured data. We compared this error using an rm-ANOVA. Interestingly, the Gaussian model had slightly worse predictive power than the linear model (*Figure 4E*, p < 0.01). This was likely because the underlying data did not appear to be normally distributed.

Issues with the linear and normal SPE generalization functions were due to an intrinsic property possessed by the SPE generalization model; the relationship between implicit and explicit learning should vary as implicit learning increases. To understand this, suppose two participants have an explicit strategy of 20°, which hypothetically yields 80% generalization. If the first participant has 15° implicit learning, they will exhibit a 15° (0.2) = 3° decrease in implicit learning. If the second participant has 7.5° implicit learning, they will exhibit only a 1.5° decrease. What this means is that as total implicit learning changes, the gain that relates implicit learning and explicit learning will also vary. Because implicit learning increased with rotation size in Exp. 1 (stepwise), then the generalization curve will differ in slope across each period. For the Gaussian generalization model, there is an additional factor which alters the gain: sampling across the nonlinear distribution. As the rotation gets larger, explicit strategies increase, which results in systematic changes in where the normal distribution is sampled, hence yielding variable implicit-explicit relationships when assessed via a linear regression within the B1, B2, B3, and B4 periods.

These variations in slope/gain did not match the measured data (*Figure 4F*). Here, we fit a separate linear regression to each learning period in the stepwise group and calculated the regression slope as well as the 95% CI. Remarkably, the measured implicit-explicit slope appeared to be constant across all rotation sizes. This invariance was consistent with the competition theory (*Figure 4F*, competition) which possesses an implicit gain $p_i$ that remains constant across rotations (like the data). It was not consistent with each generalization model, where slope varied across rotation sizes. Note, the error bars on model predictions in *Figure 4F* were estimated with bootstrapping; we sampled participants with replacement, fit the models to collapsed participant behavior, and calculated their slope. For the Gaussian models, there is no one slope (nonlinear) so we calculated the slope in the region bounded by the explicit strategies seen in the measured data over each rotation period. Also note that any negative explicit strategies (only three points in the 15° period and 1 in the 45° period) were ignored during this calculation.

We also compared the models using AIC. For this analysis, we used the stepwise participants in Exp. 1. This is the only experiment where the model can be fit to individual participants, because implicit and explicit learning was assayed across the four rotation sizes. Thus, we fit all the models in *Figure 4D–F* to individual participants. We used the same process described above, where *m* and σ were set to McDougle et al. values in the SPE generalization models. The results are shown in *Figure 4G*. As expected, AIC strongly favored the competition model. At left, we show the

generalization model's AIC values relative to that of the competition model (positive values mean competition is more likely to explain the data). At right, we show how many participants are best described by each of the three models.

A potential issue is that generalization properties differed between McDougle et al. and our data. Perhaps the SPE generalization model would have exhibited better performance with another $m$ or $\sigma$ value. To assess this possibility, we conducted a sensitivity analysis. We repeated the entire analysis in *Figure 4D–G* described above, but varied $m$ and $\sigma$ across a wide range. For the range's lower bound we reduced the McDougle et al. generalization parameters by 50%. For the upper bound we doubled the values. The sensitivity analysis results are shown in *Figure 4H*. The left inset shows the prediction error at each generalization width, similar to *Figure 4E*. The right inset counts the total participants best explained by each model, according to AIC as in *Figure 4G*. Across the entire range, the competition model had smaller error and better explained the data.

In sum, data in Exps. 1–3 were poorly explained by an SPE model extended with generalization. While plan-based generalization may promote negative implicit-explicit correlations, its contribution is small relative to the competition theory.

## 6.4 Abrupt and gradual rotations in Exp. 1

In Exp. 1 we observed that perturbing individuals in a stepwise manner led to an increase in implicit learning and a reduction in explicit strategy. These observations qualitatively and quantitatively matched the competition model (*Figure 2D–G*). But there is an alternate possibility. Suppose that the implicit system is driven by SPE as in the independence model but exhibits plan-based generalization. In Appendix 6.2 we derive this SPE generalization model. This model could predict a reduction in implicit learning via two steps: (1) both abrupt and stepwise groups have equal implicit learning, but the abrupt rotation leads to greater re-aiming. (2) More aiming in the abrupt rotation results in a decrease in the implicit learning measured at the target due to plan-based generalization. This hypothesis could be summarized with an SPE generalization model in which measured implicit learning at the target is equal to total implicit learning via: $x_i^{\text{measured}} = x_i^{\text{ss}}\exp(-0.5(x_e^{\text{ss}}/\sigma)^2)$. Could this model lead to the observed data?

Initially, let us assume that $\sigma = 37.76°$ as measured by *McDougle et al., 2017*. To estimate the change in implicit learning between abrupt and gradual implicit learning, we can calculate the reduction in implicit aftereffect expected given a normal distribution with $\sigma = 37.76°$, for the 39.5° and 29.9° explicit strategies measured in the abrupt and stepwise groups. Aiming straight to the target in the abrupt group would yield 57.86% remaining implicit aftereffect. In the stepwise group, it would yield 73.09% remaining aftereffect. Altogether, the model predicts that stepwise implicit learning should increase by 100(73.09/57.86–1), or 26.3%, over the abrupt group. On the contrary, the abrupt and stepwise implicit aftereffects were 11.72° and 21.36°, respectively. This is an 82.3% increase in implicit learning.

In sum, similar to our analysis of Exps. 2 and 3 in *Figure 5*, while generalization will produce a negative implicit-explicit relationship, the implicit learning variations we observed were much larger in Exp. 1 than predicted by generalization alone. In this case, the 82.3% increase in implicit learning is more than threefold larger than the 26.3% increase predicted by the implicit generalization properties measured by *McDougle et al., 2017*. Suppose $\sigma = 37.76°$ does not accurately represent our data. To match the measured data, $\sigma$ would need to be smaller, to narrow the generalization curve. This is unlikely to be the case given that in Exp. 1 used three targets, whereas McDougle et al. used 1. Additional targets do not narrow the generalization curve, they would widen it (*Krakauer et al., 2000*). Still, let us proceed. Rather than assume that $\sigma = 37.76°$, we can fit a normal distribution to the measured data. In the abrupt group, implicit and explicit learning were 11.72° and 39.5°, respectively. In the stepwise group, implicit and explicit learning were 21.36° and 29.9°, respectively. Fitting a normal curve to these data, would yield the curve shown in *Figure 4—figure supplement 1A*. The optimal $\sigma$ is 23.6° and total implicit learning would need to be 47.8°.

While implicit learning equal to 47.8°, or 90% the total adapted responses, appears high, there is a more important issue. These values create an unphysical scenario. For abrupt learning an $x_i^{\text{ss}}$ of 47.8° and $x_e^{\text{ss}}$ of 39.5° would indicate that total adaptation should equal 47.8° + 39.5° = 87.3° (*Figure 4—figure supplement 1B*). This is larger than the rotation's size and thus, is unphysical. In the stepwise group as well, total predicted learning would be about 77.7°, which is also larger than the rotation size.

There is a deep issue here, as described in Appendix 6.2. The problem is that as the generalization curve narrows (e.g., σ = 23.6° vs. 37.76°), not only does implicit learning measured at the target drastically underapproximate total implicit learning at the aim location, but the explicit strategy we estimated via $x_e^{ss} = x_T^{ss} - x_i^{ss}$ will substantially overestimate true explicit strategy, leading to unphysical systems. To understand this, suppose $x_i^{ss}$ is larger than $x_i^{measured}$. Explicit strategy in Exp. 1, $x_e^{measured} = x_T^{ss} - x_i^{measured}$. When this measured strategy is taken as actual explicit strategy in the generalization curve, this is equivalent to saying that total learning will be equal to $x_e^{measured}$, plus total implicit learning, $x_i^{ss}$. Total learning would be strategy ($x_T^{ss} - x_i^{measured}$) plus total implicit learning $x_i^{ss}$. This would be $x_T^{ss} - x_i^{measured} + x_i^{ss}$. Herein lies the contradiction. Because $x_i^{ss}$ is larger than $x_i^{measured}$, by using the estimated explicit strategy as the actual explicit strategy in the model, total learning in the model will automatically be larger than actual total learning. As we described above, this problem can progress so far as to predict that total learning is larger than the rotation.

The key idea is that both implicit and explicit learning need to be corrected by the generalization curve in our data. This correction is outlined in Appendix 6.2. Using *Equations A6.1 and A6.4*, we identified the σ and $x_i^{ss}$ that minimized the squared error between $x_i^{measured}$ predicted by an SPE generalization model, and the measured stepwise and abrupt implicit values. The model revealed that the optimal σ and $x_i^{ss}$ were 3.87° and 45.69°, which produced the curve shown in *Figure 4—figure supplement 1C* (corrected model). This curve shows how measured implicit learning and explicit learning will interact. This is not the true implicit-explicit generalization curve. That curve is in *Figure 4—figure supplement 1D* (corrected model). The generalization curve required by the model, was implausible: it had a width of σ = 3.87°, compared to 37.76° as measured by McDougle et al. This is why the model's distribution in *Figure 4—figure supplement 1D* is so narrow.

The relationship between *Figure 4—figure supplement 1C and D* may not be intuitive. To explain how these curves are entwined, consider the stepwise learning point in inset C. This point lies roughly at 20° implicit learning and 30° explicit strategy. This explicit strategy is the estimated strategy calculated in Exp. 1: total adaptation – measured implicit learning. Note that total implicit learning is about 45°. Thus, measured implicit learning is about 45°–20° = 25° smaller than total implicit learning. This means that our estimated explicit strategy at 30°, is about 25° too large. Thus, the actual strategy is much smaller: 30°–25° = 5°. These corrections reveal the mapping between insets C and D. The point, $x_i = 20°$ and $x_e = 30°$, will approximately lie at $x_i = 20/45 \times 100 = 44.4\%$, and $x_e = 5°$.

In sum, we conclude that our abrupt vs. stepwise analysis in *Figure 2D–G* does not match implicit generalization. While incorrect, we began with the assumption that the implicit learning measures in Exp. 1 represent generalized learning, but explicit strategies represent total re-aiming. But in this model, the change in implicit learning in the actual data, was three times larger than that predicted by generalization as measured in McDougle et al. Moreover, this analysis is flawed, in that only correcting implicit measures with generalization, but not explicit measures, produced a situation where total adaptation would have exceeded the rotation's magnitude. This is because explicit strategies are estimated using total adaptation minus implicit learning. When we corrected the SPE generalization model so that both the implicit and explicit learning we measured were corrected by a generalization curve, the model required that plan-based generalization resemble a Gaussian with σ = 3.87°, an unphysiological scenario. The generalization model is not a viable alternate to the competition theory.

## 6.5 Instructions and variations in rotation size

In *Neville and Cressman, 2018*, implicit learning did not vary across 20°, 40°, and 60° rotations. Saturated responses like this resemble implicit learning properties exhibited in invariant error-clamp (*Morehead et al., 2017*; *Kim et al., 2018*) paradigms. In such experiments, the implicit system appears to reach a ceiling that does not depend on the rotation's magnitude (at least when rotations are less than 90°). Does this same phenotype cause the saturation we explored using the competition model in *Neville and Cressman, 2018*?

In isolation this might appear plausible, but there is one issue: the response to instruction. The authors also tested how implicit learning and explicit strategies responded to instructions. Coaching participants increased explicit strategy but decreased implicit learning. This variation in implicit learning would not be explained by invariant error-clamp implicit learning properties, which would predict that implicit learning should always saturate at the same level. One idea that could potentially

rescue this alternate hypothesis is generalization; perhaps implicit learning truly is the same across the instruction group and no instruction group but only appears variable because instructed participants used larger strategies. This idea, however, would directly contradict the implicit response to rotation size. Supposing all groups had the same implicit learning, greater explicit strategy in the 40° and 60° rotations, should produce a reduction in the measured implicit learning due to plan-based generalization.

In sum, an SPE learning model with a ceiling on implicit learning, would require complete (100%) generalization, to show the saturation phenotype in *Figure 1D*. However, there would be no way to capture the reduction in implicit learning seen in the instruction group with complete implicit generalization – a contradiction. Conversely, the reduction in implicit learning in the instruction group would need implicit generalization. But variations in explicit strategy across the 20°, 40°, and 60° rotations would alter implicit learning, violating the saturated implicit learning phenotype in the data. Thus, there is no way that these data can be described by an upper ceiling on implicit learning as in invariant error-clamp studies (*Morehead et al., 2017*; *Kim et al., 2018*).

As discussed in our Results, this is not true in the competition model. The exact same competition model parameters (i.e. implicit learning gain $p_i$) parsimoniously explained implicit responses to rotation size in *Figure 1G* and instruction in *Figure 2C*.

There is one last possibility to consider. Perhaps plan-based generalization alone could cause the decrease in implicit learning due to instruction, and the saturation in implicit learning across rotation size. In an SPE generalization model, the instruction and no-instruction groups could reach the same implicit learning level but show differences in implicit learning measured at the target due to variations in explicit strategy. In addition, implicit learning should scale according to $p_i$ as the rotation increases. Perhaps true implicit learning does vary across the 20°, 40°, and 60° rotation periods, but appears saturated because as rotation size increases so do strategies, reducing the implicit learning measured at the target due to generalization. We evaluated both these possibilities.

Let us begin with the response to rotation size. In *Figure 4—figure supplement 1E and F*, we fit a Gaussian SPE generalization model to the implicit and explicit responses measured in the no-instruction group. As described in Appendix 6.2, inset E shows uncorrected explicit strategy estimates: total adaptation minus implicit learning. Inset F shows the true implicit-explicit plan-based generalization curve that produces the data in inset E. These curves were produced by an implicit learning gain $p_i = 0.56$, and σ = 11.2°. This shows that a generalization curve could yield a saturation phenotype, as shown in inset E. Here, the same implicit curve is shown ($p_i = 0.56$ and σ = 11.2°) but is scaled by the rotation size $r$ as predicted by the SPE model. Increases in implicit learning due to the rotation size are counterbalanced by increases in explicit strategy which generalize less at the target. However, while such a model produces a saturation phenotype, the generalization curve's width as shown in inset F, is not physiological. Rather, to produce the measured responses, the curve's width (11.2°) would need to be 70% narrower than the generalization properties measured by McDougle et al. (σ = 37.76°). This extreme narrowing has not been observed in past studies. Moreover, the notion that generalization in Neville and Cressman would be narrower than that in McDougle et al., is inconsistent with known implicit generalization properties (*Krakauer et al., 2000*). As shown in the Krakauer et al. generalization curves in *Figure 4A*, increasing the number of training targets in Neville and Cressman (three targets) would widen the generalization curve relative to McDougle et al., which only used one training target.

Next, we repeated the analyses described above, on the implicit-explicit responses to instruction in *Figure 4—figure supplement 1G, H*. The best generalization model ($p_i = 0.5$ and σ = 9.8°) could produce changes in generalized implicit learning that were consistent with the data, as shown in inset G. However, as in the response to rotation size described above, the required generalization properties were not physiologically consistent with past measurements as shown in inset H. The generalization curve would need to be about 74% narrower than that measured by McDougle et al. Thus, again, while in principle generalization could produce changes in implicit learning, it would require implausible implicit learning properties.

## 6.6 Nonmonotonic implicit learning in *Tsay et al., 2021a*

In *Tsay et al., 2021a*, participants exhibited a non-monotonic implicit response to 15°, 30°, 60°, and 90° rotations as shown in *Figure 1N*. In the main text, we explain how this phenotype could be explained by the competition model. Namely, variations in strategy, could lead to changes in the

residual target error that drives implicit learning in the competition model. Could another model also produce these data?

We considered that the decrease in implicit learning observed in the 90° rotation group resembles a pattern shown by the implicit process in invariant error-clamp paradigms (*Morehead et al., 2017*; *Kim et al., 2018*); rotations larger than 90° cause a drop in implicit learning. *Morehead et al., 2017* suggested that a similar drop occurs in response to standard rotations, at least when participants are told to ignore the cursor and aim to the target (i.e. they did not use explicit strategies). Thus, might it be that reductions in implicit learning in response to large rotations are an intrinsic property of the implicit system, rather than a phenomenon caused by error competition? The experiments conducted by Morehead et al., have another similarity to the Tsay et al. observations. In particular, in standard rotation conditions (plus a no aiming instruction), Morehead et al. observed that a 7.5° rotation caused a reduction in implicit learning relative to larger rotation sizes tested. This decrease in implicit learning cannot be due to error competition, because participants did not aim in this study. In sum, could it be that both the increase in implicit learning between 15° and 30° in Tsay et al. as well as the drop in implicit learning between 60° and 90° are caused by the implicit system's intrinsic learning properties, versus a competition with explicit strategy?

This is unlikely. First, in Morehead et al., subjects in the 7.5° standard rotation condition achieved complete adaptation: a total reach angle of 7.5°. This 7.5° level was smaller than that achieved in a 7.5° invariant error-clamp. To explain these results, Morehead et al., suggested that implicit learning stopped in the standard rotation condition, because the error was completely canceled (i.e. both the rotation and total implicit learning achieved were 7.5°, creating a 0° error). In the error-clamp condition, error never decreased and continued to drive implicit learning to its saturation point.

The data in Tsay et al., however, cannot be explained by the error cancellation mechanism. In the 15° rotation in Tsay et al., implicit learning reached only 7.6°, and thus did not completely cancel the error. Morehead et al. would have predicted that implicit learning should continue until 15° to cancel the error. Thus, unlike Morehead et al., the increase in implicit learning between the 15° and 30° rotations in Tsay et al. cannot be explained by an error cancellation mechanism. This increase clearly violates invariant error clamp learning properties, where implicit learning reaches the same saturation point across error sizes less than 95°, unless error is canceled. This same argument can be made in the scaling phenotype in Exp. 1, as well as the scaling phenotype observed earlier by *Salomonczyk et al., 2011*.

Next, consider the decrement in implicit learning shown in *Figure 1N* with the 90° rotation. It remains possible that this decrease is due to a rotation-insensitivity that is intrinsic to the implicit process (rather than error competition). However, it is error that drives learning, not rotations. While the large rotations used by Morehead et al. resemble the 90° group in Tsay et al., target errors were totally mismatched in these two studies. In Morehead et al., participants in both the error-clamp and standard rotation groups were told not to aim and to ignore the cursor. Because there was no strategy, the implicit learning curve reached approximately 10°, leaving an 85° target error. Past studies have shown that error sensitivity will be exceedingly small in response to such extreme errors (*Kim et al., 2018*; *Marko et al., 2012*; *Wei and Körding, 2009*). In our view, this insensitivity to extremely large errors likely led to the attenuation in implicit learning observed in Morehead et al. Instructions to 'ignore the cursor' may further exacerbate reductions in sensitivity to these large errors.

However, in Tsay et al., subjects were allowed to aim. Total learning reached about 85°, leaving a 5° target error: an error much more inclined to drive implicit learning. Comparing steady-state adaptation to this 5° residual error with the 85° residual error in Morehead et al., is not reasonable in our view.

In sum, the increase in implicit learning in the 15° and 30° groups could only be described by error competition, not an error cancellation mechanism as in Morehead et al. Second, the residual target errors experienced in Morehead et al. were about 80° larger in their 95° rotation group, than the 90° rotation group in Tsay et al. For these reasons, attenuation in implicit learning in these two studies was likely caused by differing mechanisms: a drastic reduction in target errors (the competition hypothesis) in Tsay et al., and an unresponsiveness to extreme target error in Morehead et al. (which could have been exacerbated by telling participants to ignore the cursor). For the learning patterns in Tsay et al., the competition model seems the most parsimonious choice, not

only given its quantitative match to the data (*Figure 1Q* and *Figure 1—figure supplement 2*), but also because it alone (not the error-clamp learning properties in Morehead et al.) can explain implicit responses across the many other cases described in *Figures 1 and 2*: abrupt and stepwise responses in Exp. 1 (as well as Salomonczyk et al.), rotation responses between 15 and 60° in Tsay et al., as well as implicit behavior in Neville and Cressman. This is not to mention that the implicit learning properties in Morehead et al. provide no clear way to interpret the pairwise relationships between implicit learning, explicit strategy, and total learning detailed at length in *Figures 3–5* at the individual-participant level.

# Appendix 7

In our Results, we considered how an SPE model might also predict negative correlations between implicit learning and explicit strategy. Suppose that implicit learning is driven by SPEs and is not altered by explicit strategy. However, a subject with a better implicit learning system (e.g. a higher implicit error sensitivity) will require less explicit re-aiming to reach a desired adaptation level. In other words, individuals with large SPE implicit learning may use less explicit strategy relative to those with less SPE implicit learning. Like the competition theory, this scenario would also yield a negative relationship between implicit and explicit learning, due to the way explicit strategies respond to variation in the implicit system. We will now show the diverging predictions this model makes, relative to the competition theory.

## 7.1 Competition model predictions: implicit learning responds to variations in explicit strategy

Here, we will start with the competition theory. In this model, the implicit system responds to variation in explicit strategy according to: $x_i^{ss} = p_i(r - x_e^{ss})$. Clearly, this predicts a negative relationship between implicit learning and explicit strategy (*Figure 5D*). Next, we note that total adaptation is given by $x_T^{ss} = x_i^{ss} + x_e^{ss}$. We can solve for $x_e^{ss}$ and substitute this into the model, yielding the following relationship between implicit learning and total adaptation: $x_i^{ss} = p_i(1 - p_i)^{-1}(r - x_T^{ss})$. This is the expression tested in Appendix 3 where we analyzed our data in *Figure 1* using steady-state implicit learning and total adaptation. We can rearrange this equation, solving for $x_T^{ss}$ yielding the dual expression: $x_T^{ss} = r + p_i^{-1}(p_i - 1)x_i^{ss}$. This expression is written within the inset in *Figure 5F*. Both equation variants show that in a stable learning system ($p_i < 1$) that implicit learning and total adaptation will exhibit a negative relationship. We can repeat this analysis, but this time solve for $x_i^{ss}$, and substitute into *Equation 4*, to obtain a relation between explicit learning and total adaptation. This yields $x_T^{ss} = p_i r + (1 - p_i)x_e^{ss}$. Note that this expression is provided in *Figure 5E*. Again, noting that $p_i < 1$, this predicts a positive relation between explicit strategy and total adaptation.

To summarize, the competition model makes three predictions about the pairwise relationships between implicit learning, explicit strategy, and total adaptation. First, as explicit strategies increase, this will tend to increase total adaptation (i.e. positive relation as in *Figure 5E*). As explicit strategy increases, the residual target error will decrease, leading to less implicit learning. This predicts that implicit learning will exhibit a negative correlation with both explicit strategy (*Figure 5D*) and total adaptation (*Figure 5F*).

## 7.2 SPE model predictions: explicit strategy responds to variations in implicit learning

Now, let us suppose we have the opposite scenario to the competition model. In an SPE model, implicit learning does not respond to explicit strategy. Suppose implicit learning varies randomly across subjects (due to inter-subject variability in implicit learning properties, e.g., error sensitivity) and explicit strategy responds to this variability in implicit learning. In this framework, competition occurs but with a reversed causal structure. Now, assuming $x_i^{ss}$ is due to an independent SPE learning mechanism, this will yield a residual target error of $r - x_i^{ss}$. A negative relationship between implicit and explicit learning occurs in the event that explicit strategies respond in proportion to this residual target error: $x_e^{ss} = p_e(r - x_i^{ss})$, where $p_e$ is an explicit learning gain. This equation is the same as the competition model in *Equation 4*, where the roles of $x_e^{ss}$ and $x_i^{ss}$ are reversed. Thus, similar relationships between $x_T^{ss}$ and each system occur. Assuming that $p_e$ is less than 1 (i.e. the explicit system does not overcompensate for the remaining error, yielding total learning > $r$) then the relationship between total adaptation and implicit learning will now be positive, with $x_T^{ss} = p_e r + (1 - p_e)x_i^{ss}$, and the relationship between total adaptation and explicit learning will now be negative: $x_T^{ss} = r + p_e^{-1}(p_e - 1) x_e^{ss}$. Note that these expressions are provided in *Figure 5B&C*.

To summarize, the SPE model makes three predictions about the pairwise relationships between implicit learning, explicit strategy, and total adaptation. First, as implicit learning increases, this will tend to increase total adaptation (i.e., positive relation as in *Figure 5C*). But as implicit learning increases, there is a smaller target error for the explicit system to correct, leading to less explicit strategy. This predicts that explicit strategy will exhibit a negative correlation with both implicit learning (*Figure 5A*) and total adaptation (*Figure 5B*). This provides a way to compare the competition and SPE models.

## 7.3 Simulating variations in implicit and explicit learning across participants

We constructed *Figure 5A–F* to provide more intuition on how to compare the competition and SPE model predictions described above. In these toy-simulations, we first fit $p_i$ and $p_e$ in the equations above to the implicit and explicit measures in the No PT Limit group in Exp. 3 yielding $p_i$ = 0.669 and $p_e$ = 0.689. These values are not important; the same qualitative behavior will occur provided they are between 0 and 1. We assumed that implicit learning varied across participants according to a normal distribution. For the distribution's mean we calculated the average implicit learning measured in the No PT Limit group. For the distribution's standard deviation, we used 4°. Then, we calculated explicit learning according to $x_e^{ss} = p_e(r - x_i^{ss})$ for each participant. We then simulated 'measurements' of implicit and explicit learning by adding a normal random variable with mean zero and standard deviation 2° to these simulated implicit and explicit learning measures. We simulated 250 participants in total.

Simulations for the competition theory were similar. Here, we simulated explicit strategies across participants according to a normal distribution. The mean was set equal to the average explicit strategy measured in the No PT Limit group. The standard deviation was set to 4°. To simulate implicit learning, we used the competition equation: $x_i^{ss} = p_i(r - x_e^{ss})$. We then added variability to these "true" values to obtain noisy implicit and explicit measures across 250 participants.

Results for these simulations are shown in *Figure 5A–F*. In Panels A-C, we show results for the 250 participants for the model where explicit systems respond to variability in an SPE-driven implicit system. In Panels D-F, we show simulations for the competition theory where implicit systems respond to variability in explicit strategy. In Panels A and D, we show the relationship between implicit and explicit learning. In Panels B and E, we show the relationship between total adaptation and explicit learning. In Panels C and F, we show the relationship between total adaptation and implicit learning. Red ellipses denote the 95% confidence ellipses for the 250 simulated participants.

## 7.4 Comparing pairwise implicit-explicit-total correlations between competition and SPE models

In this Appendix we show that both an SPE model and a target error learning model could exhibit negative participant-level correlations between implicit learning and explicit strategy. But their predictions diverge on the relationships between total adaptation and each individual learning system. Target error learning predicts a negative implicit-total correlation and positive explicit-total correlation. SPE learning predicts a positive implicit-total correlation and a negative explicit-total correlation. To test these predictions, we considered how total learning was related to implicit and explicit adaptation measured in the No PT Limit group in Exp. 3. Our observations closely agreed with the competition theory; greater explicit strategy was associated with greater total adaptation (*Figure 5G*, $\rho$ = 0.84, p < 0.001), whereas greater implicit learning was associated with lower total adaptation (*Figure 5H*, $\rho$ = −0.70, p < 0.001).

We repeated similar analyses across additional data sets that also measured implicit learning via exclusion (i.e. no aiming) trials: (1) the 60° rotation groups (combined across gradual and abrupt groups) in Experiment 1, (2) the 60° rotation groups reported by Maresch and colleagues (*Maresch et al., 2021*) (combined across the CR, IR-E, and IR-EI groups), and (3) the 60° rotation group described by Tsay and colleagues (*Tsay et al., 2021a*). We obtained the same result as in Experiment 3. Participants exhibited negative correlations between implicit learning and explicit strategy (*Figure 5—figure supplement 1G-I*), positive correlations between explicit strategy and total learning (*Figure 5—figure supplement 1D-F*) and negative correlations between implicit learning and total learning (*Figure 5—figure supplement 1A-C*). These additional studies also matched the competition theory's predictions.

## 7.5 Critical exceptions to these predictions

The competition theory predicts on average that implicit learning will exhibit a negative correlation with total adaptation (across individual participants). However, this prediction assumes that implicit learning is only driven by target errors, a condition we explore more completely in Part 3 of our Results. Second, it assumes that implicit learning properties ($a_i$ and $b_i$, summarized with the gain $p_i$ above) are identical across participants, an unlikely possibility. Variation in the implicit learning gain (e.g. Participant A has an implicit system that is more sensitive to error) will promote a positive correlation between implicit and total adaptation, that will weaken the negative correlations

we described above. Two examples where this appears to occur are shown in *Figure 5—figure supplement 2A*. Inter-subject variability in the implicit learning gain can dominate inter-subject variability in explicit strategy, which would lead to a positive relationship between implicit learning and total adaptation. It should be noted that the converse is not true in the independence model. SPE learning rules will always promote a positive relationship between implicit learning and total adaptation and will not show a negative correlation, despite inter-subject variability in implicit and explicit learning gains. A more thorough discussion on these matters is provided in Appendix 8.

# Appendix 8

In Appendix 7, we detail how an SPE independence model, and a target error competition theory predict implicit and explicit learning should vary across participants. To review, the SPE model predicts (1) positive correlations between implicit learning and total adaptation, and (2) negative correlations between explicit strategy and total adaptation. On the other hand, the competition theory predicts (1) negative correlations between implicit learning and total adaptation, and (2) positive correlations between explicit strategy and total adaptation. We noted several datasets that supported the competition theory: our data in Experiment 3 (*Figure 5G&H*), experiments conducted by *Maresch et al., 2021* (*Figure 5—figure supplement 1A,D&G*), our data in Experiment 1 (*Figure 5—figure supplement 1B,E&H*), and data collected by *Tsay et al., 2021a* (*Figure 5—figure supplement 1C,F&I*). Here, we detail a critical nuance in the competition theory's predictions that may result in little to no correlation between implicit learning and total adaptation.

## 8.1 Subject-to-subject correlations in implicit learning within the competition theory

The competition theory (i.e. target error learning model) will not always produce a negative relationship between implicit learning and total adaptation. In Appendix 7.1, we explained that the competition theory, $x_i^{ss} = p_i (r - x_e^{ss})$, does on average predict a negative correlation between implicit learning and total adaptation. Let us consider again why this occurs. Suppose two Participants *A* and *B* have identical implicit learning systems, but Participant *A* has superior explicit strategy. Overall, this means Participant *A* will adapt more to the perturbation. However, their greater strategy will create a smaller driving force for the implicit system, yielding less implicit learning. Thus, total adaptation is positively correlated with explicit strategy, but negatively correlated with implicit learning.

To restate this idea, in a target error model, between-subject variation in explicit strategy creates a negative relationship between implicit learning and total adaptation. However, these predictions rely on a key assumption; implicit learning properties must be the same across all participants to yield negative correlations. In other words, in our Participants *A* and *B* example, both participants were assumed to have the same $p_i$ parameter, a term that depends on implicit error sensitivity and retention. Between-subject variation in these implicit learning properties, however, will promote a positive relationship between total adaptation and implicit learning. Thus, it is entirely possible that the negative correlations promoted by between-subjects explicit variability can be negated by the positive correlations promoted by between-subjects implicit variability, yielding no correlation in some instances.

To illustrate this, consider the toy simulation in *Figure 5—figure supplement 4A*. At left, we simulate total implicit learning using the competition equation ($p_i$ = 0.8) across 35 participants adapting to a 30° rotation, whose explicit strategies vary according to a normal distribution (mean = 12°, S.D. = 4°). Note the strong negative relationship between implicit learning and total adaptation. At right, we show the same data (same explicit strategies) but introduce variability in implicit learning ($p_i$ in the model is varied according to a normal distribution with mean = 0.8, and S.D. = 0.1). Even though these data arise from the same competition equation, adding between-subject variation in implicit learning properties yields zero correlation between implicit learning and total adaptation (p = 0.199, R² = 0.05).

The competition equation predicts that the correlation between implicit learning and total adaptation is uniquely susceptible to contamination with between-subject implicit variability. That is, while the correlation between implicit learning and total adaptation (*Figure 5—figure supplement 4A*, right) was not statistically significant, the same simulated data exhibited a strong positive correlation between explicit strategy and total adaptation (*Figure 5—figure supplement 4B* right; p < 0.001, R² = 0.42), and a strong negative correlation between implicit learning and explicit strategy (*Figure 5—figure supplement 4C*, right; p < 0.001, R² = 0.77).

Thus, with implicit variability the competition theory can simultaneously exhibit no correlation between implicit learning and total adaptation, a strong positive correlation between explicit strategy and total adaptation, and a strong negative correlation between implicit learning and explicit strategy.

To conclude, correlative phenomena in the competition theory represent a balance between negative correlations induced by between-subject explicit variability, and positive correlations induced by between-subject implicit variability. Observing negative correlations is a probabilistic

phenomenon. A given study can easily fail to yield a statistically significant correlation between total adaptation and implicit learning, yet still be governed by the competition equation. To maximize the probability of detecting a negative correlation between implicit learning and total adaptation, there are several critical factors that should be considered by the experimenter.

To describe these factors, we compare our data in Experiment 3 (*Figure 5G&H*) to experimental conditions where we detected no statistically significant correlation between implicit learning and total adaptation. These include the 30° rotation groups collected in *Tsay et al., 2021a* and Experiment 1 (*Figure 5—figure supplement 2A*, middle and right). These studies used similar experimental procedures, yet only our data in Exp. 3 yielded a statistically significant correlation between implicit learning and total adaptation. Here, we describe four key factors that may have played a role in these differences. For each factor, we will perform simulations using the competition equation. Factors 1 and 2 deal with statistical power. Factors 3 and 4 deal with how changes in explicit strategy use alter the ability to measure correlations between implicit learning and total adaptation.

## 8.2 Factor 1. Statistical power: total number of trials

Because correlations between implicit learning and total adaptation are a balance between two opposing variability sources, high statistical power will increase one's ability to detect them in an experiment. One simple way to increase this power, is to increase the total number of trials used to measure total adaptation and implicit learning. That is, each reaching movement is corrupted by motor variability. To better estimate total adaptation and implicit learning, averaging over more trials lessens the effect of trial-to-trial reach variability on subject-to-subject correlations. This can be especially problematic for the number of aftereffect trials used to measure implicit learning, which remain limited in many studies.

Consider the simulations in *Figure 5—figure supplement 2B*. These simulations show a power analysis where we vary the total number of aftereffect trials in simulation, to detect the probability that an experiment with 30 participants will yield a statistically significant correlation between implicit and total adaptation. Here, implicit learning is set by the competition equation. All simulation parameters are held still (e.g. explicit parameter variability, implicit parameter variability, mean explicit strategy; see Appendix 8.8 below) except the total number of aftereffect trials used to calculate implicit learning. That is, we average over simulated trials to calculate total learning, explicit strategy, and implicit learning. Each simulated trial differs due to motor execution noise (i.e. varied according to a normal distribution). We repeat each simulation 40,000 times with 30 participants in each simulation, and calculate the total fraction of iterations where there was a statistically significant negative correlation between total adaptation and implicit learning (*Figure 5—figure supplement 2B*, red, left), no statistically significant correlation between total adaptation and implicit learning (*Figure 5—figure supplement 2B*, black, left), and a positive statistically significant correlation between these two variables (*Figure 5—figure supplement 2B*, green, left).

This power analysis qualitatively demonstrates that increasing the number of aftereffect trials greatly improves one's ability to detect a negative statistically significant correlation between total adaptation and implicit learning. We should note that our study (Experiment 3) is an outlier, in that we used a very large number of no feedback (and no aiming) trials to measure implicit learning: 80 trials. In cases where we did not detect a statistically significant correlation, the total aftereffect trial count was much smaller: Tsay et al. (2021) used only 20 trials to measure the implicit aftereffect and Exp. 1 used only 18 trials (*Figure 5—figure supplement 2B*, right). Thus, Exp. 3 was more likely to produce a negative correlation between implicit learning and total adaptation, given this experimental factor.

## 8.3 Factor 2. Statistical power: motor variability

The second factor that plays an important role in measuring subject-to-subject correlations, is also related to statistical power: motor variability. Like trial count (Factor 1), the more variable a participant's reaching movements are, the poorer one's estimate for total learning and implicit learning. To show this, we repeated our power analysis process described above (using the competition model), but this time held all simulation parameters constant, except trial-to-trial variability in executing a movement. We sampled this motor execution noise parameter for each participant; some simulated subjects had higher trial-to-trial variability than others. We gradually increased the mean motor noise

parameter across participants, as well as the variation in motor noise across participants. Results are shown in *Figure 5—figure supplement 2C*.

Motor execution noise plays a strong role in detecting statistically significant negative correlations between implicit learning and total adaptation (*Figure 5—figure supplement 2C*, left, red); as motor execution noise increases, the probability of detecting a statistically significant correlation falls sharply. Therefore, studies where subjects have smaller trial-to-trial variability in reaching movements, will be more likely to detect negative correlations between total adaptation and implicit learning. For example, we calculated the trial-by-trial variability in reach angle during the no feedback periods in our data (Exp. 3) as well as the 30° rotation datasets we described above (*Tsay et al., 2021a* and Experiment 1). We used this period so that trial-to-trial volatility in explicit strategy did not corrupt our estimate of motor variability (i.e. trial-to-trial variability is much larger during asymptotic behavior).

As shown in *Figure 5—figure supplement 2C* at right, participants in Experiment 3 exhibited smaller trial-by-trial reach variability (one-way ANOVA, $F = 6.84$, $p = 0.002$) than both the Tsay dataset (post-hoc test: $p = 0.003$) as well as Experiment 1 (post-hoc text: $p = 0.015$). Thus, Exp. 3 was more likely to produce a negative correlation between implicit learning and total adaptation, given this experimental factor. In addition, it should be noted that motor noise variability (Factor 2) will act synergistically with limited aftereffect trials (Factor 1) to impair one's ability to detect accurate implicit learning measures.

While it may be difficult to control motor noise, experimenters should consider the following parameters: (1) movement displacement, (2) the type of experimental apparatus (laptop vs. robot vs. tablet), (3) the speed of the reaching or shooting movements, and (4) target location. These experimental conditions will alter reaching variability and may improve one's ability to detect negative correlations between total adaptation and implicit learning.

## 8.4 Factor 3: explicit strategy use

Factors 3 and 4 relate less to statistical power, and more to the variability sources that underly subject-to-subject differences in implicit and explicit learning. A critical factor that determines one's probability of detecting negative correlations between implicit learning and total adaptation, is total explicit strategy use. To detect how overall strategy use affects the ability to obtain statistically significant correlations, we again used our power analyses. This time, we repeated our power analysis procedure, but varied the mean of the normal distribution used to simulate variable explicit strategies; we gradually increased the mean strategy use across our simulations, in the case where participants adapted to a 30° rotation. All other simulation parameters remained constant across simulations (n = 30 in each simulation, 40,000 iterations for each explicit strategy level). The results are shown in *Figure 5—figure supplement 2E*, at left.

Strategy use strongly affects one's ability to detect negative statistically significant correlations between implicit learning and total adaptation (*Figure 5—figure supplement 2E*, left, red). When participants use little explicit strategy on average, it is more difficult to obtain a statistically significant implicit-total adaptation correlation. In other words, at a given rotation size, studies where participants use greater strategies are more likely to yield a negative relationship between implicit learning and total adaptation. Comparing participants in Experiment 3 to the Tsay dataset and Experiment 1 (*Figure 5—figure supplement 2E*, right), we noted that participants in the Tsay dataset exhibited large reductions in explicit strategy use (one-way ANOVA, $F = 11.09$, $p < 0.001$; post-hoc tests had $p < 0.001$ for Experiment 3 vs. Tsay and Experiment 1 vs. Tsay). Thus, participants in the Tsay experiment were least likely to exhibit negative correlations between implicit learning and total adaptation, according to this experimental factor.

## 8.5 Factor 4: Between-subject variability in explicit strategy use

Recall that the relationship between implicit learning and total adaptation is a balance between variability sources: positive correlations induced by between-subjects implicit variability, and negative correlations induced by between-subjects explicit variability. Thus, more variability in explicit strategy increases how likely one is to detect a negative correlation between implicit and total adaptation. To demonstrate this, we performed a final power analysis. All parameters were constant across simulations, except variability in strategy use. For each simulation (n = 30) we sampled strategies from a normal distribution; we gradually increased the SD of this normal distribution across simulations

(40,000 simulations for each level) while holding mean explicit strategy constant. The results are shown in *Figure 5—figure supplement 2D*, at left.

These simulations demonstrated two critical properties. First, as subject-to-subject variability in strategy use increases, so too does the likelihood of detecting a negative relationship between implicit learning and total adaptation (*Figure 5—figure supplement 2D*, left, red). Second, when between-subject explicit variability is very low, there is even a chance of detecting positive correlations between total adaptation and implicit learning (*Figure 5—figure supplement 2D*, left, green) even in the competition theory. This key point should be kept in mind when experiments use conditions where strategy use is minimal across participants (e.g. exceedingly gradual rotations; very small rotations, etc.).

Along these lines, we should note that between-subject variability in explicit strategy use was greatest in Experiment 3. As compared to the Tsay dataset and Experiment 1, explicit variability was 32% and 72% greater in Experiment 3, respectively (*Figure 5—figure supplement 2D*, right). Therefore again, Experiment 3 was most likely to produce a negative correlation between implicit learning and total adaptation.

## 8.6 Unique susceptibility in the correlation between implicit learning and total adaptation

It is important to reiterate that with target error learning, negative correlations between implicit learning and total adaptation are uniquely challenging to detect. That is, there is more power to detect positive correlations between explicit strategy and total adaptation. For example, though we did not detect a negative relationship between implicit learning and total adaptation in the 30° conditions tested by *Tsay et al., 2021a* and in Experiment 1, we did detect a positive correlation between explicit strategy and total adaptation in these experiments (*Figure 5—figure supplement 3A and B*).

*Figure 5—figure supplement 3* (panels C-F), again shows the power analyses on Factors 1-4 illustrated in *Figure 5—figure supplement 2*, but this time investigates the correlation between explicit strategy and total adaptation. Across all 4 factors, the power analyses demonstrated that experiments should yield greater probability of detecting positive correlations between explicit strategy and total adaptation (*Figure 5—figure supplement 3C-F*, green curves at top), than negative correlations between implicit learning and total adaptation (*Figure 5—figure supplement 3C-F*, red curves at top). These data are recapitulated in the simulated $R^2$ statistic across the two correlations (*Figure 5—figure supplement 3C-F*, bottom row); the correlation between total adaptation and explicit strategy was greater than the correlation between total adaptation and implicit learning.

## 8.7 Appendix 8 summary

Here we explained processes that impact the correlation between implicit learning and total adaptation in the competition theory. Between-subject variability in explicit strategy and implicit learning properties promote positive and negative correlations between implicit learning and total adaptation, respectively. These opposing factors make it possible that correlations between implicit learning may be weak or absent in an experiment. We explored four key experimental factors that researchers should consider in their data sets to maximize the chance of detecting negative correlations between implicit learning and total adaptation. However, this is by no means a complete list. For example, greater SPE learning will drastically undermine the negative correlations between implicit learning and total adaptation produced by target error learning. Thus, we expect that conditions which use multiple visual landmarks (e.g., aiming targets) are unlikely to show negative correlations between implicit learning and total adaptation.

## 8.8 Appendix 8 methods

Here we analyzed data collected in Experiment 1, Experiment 3, and Tsay et al. (2021). Implicit and explicit learning measures were calculated as reported in the Methods section in our main text. These implicit and explicit learning measures were used to calculate the correlations shown in *Figure 5—figure supplement 2A* and *Figure 5—figure supplement 3A&B*. In addition, the explicit measures were used to calculate the strategy use in *Figure 5—figure supplement 2E*. Each dot in the right-most inset represents an individual participant. Variations in explicit strategy across experiments were assessed with a one-way ANOVA, with Bonferroni-corrected post-hoc tests. In addition, *Figure 5—figure supplement 2D* (at right) shows the std. dev. in explicit strategy across

participants within the three experimental conditions. In *Figure 5—figure supplement 2C*, we estimated motor variability within individual participants. To do this we calculated the standard deviation in the reach angle across trials in the no-aiming period at the end of each experiment. We chose this period to prevent volatility in explicit strategy from inflating our motor variability measure. Each dot in the right-most inset shows the reach angle standard deviation for a single participant. We assessed differences in motor variability across the three experiments using a one-way ANOVA, with Bonferroni-corrected post-hoc tests.

In *Figure 5—figure supplement 4*, we provide toy simulations to illustrate how variation in implicit learning properties alters pairwise relationships between implicit learning, explicit strategy, and total adaptation. For the left-most inset in panels A, B, and C, we simulated a condition with no variability in implicit learning properties. That is, we used the competition equation to simulate implicit learning, but held $a_i$ and $b_i$ constant across all participants (each individual dot in the panel). We chose $a_i$ and $b_i$ so that the implicit learning gain, $p_i$, was equal to 0.8. We simulated 35 participants adapting to a 30° rotation. Explicit strategy was sampled for each participant using a normal distribution with a mean of 12° and a standard deviation of 4°. The right-most inset in panels A, B, and C, use the exact same explicit strategies. However, here we allow $p_i$ (i.e., implicit learning properties) to vary across participants. To simulate this variation, we sample $p_i$ according to a normal distribution with a mean of 0.8 and a standard deviation of 0.1.

Finally, *Figure 5—figure supplement 2* and *Figure 5—figure supplement 3* show four power analyses. The power analyses were the same across these two figures, only, *Figure 5—figure supplement 2* focuses on how implicit learning relates to total adaptation, and *Figure 5—figure supplement 3* considers how explicit strategy relates to total adaptation. In these power analyses, there are several parameters. First, implicit error sensitivity was uniformly sampled between 0.9 and 0.95. Implicit error sensitivity was uniformly sampled between 0.2 and 0.3. The rotation size was always 30°. Other simulation parameters varied across each power analysis. For each power analysis, there was one parameter that varied across simulations, but all other parameters were fixed to default values. The default values were as follows. Explicit learning was sampled for each participant using a normal distribution with a mean of 10° and a standard deviation of 6°. The total number of trials used to measure total adaptation, implicit learning, and explicit learning was equal to 40. Motor variability had a mean of 12° across participants, with a standard deviation of 6°.

Power analyses in *Figure 5—figure supplement 2B* and *Figure 5—figure supplement 3C* used the default parameter values but varied the total number of probe trials used to measure implicit and explicit learning between 1 and 80. Power analyses in *Figure 5—figure supplement 2C* and *Figure 5—figure supplement 3D* used the default parameter values but varied the average motor variability between 5° and 20°, and the standard deviation in motor variability between 2° and 10°. As mean motor variability increased across simulations, so did the subject-level standard deviation. Power analyses in *Figure 5—figure supplement 2D* and *Figure 5—figure supplement 3E* used the default parameter values but varied the standard deviation in strategy use between participants between 0.1° and 8°. Finally, power analyses in *Figure 5—figure supplement 2E* and *Figure 5—figure supplement 3F* used the default parameters values but varied average strategy use between 0° and 20°.

In these power analyses, the parameter of interest was varied linearly between its two extreme values. For each value we conducted 40,000 simulations, each time sampling random variables for 30 participants according to the distributions noted above. Across these simulations we calculated the probability that a negative statistically significant relationship occurred between implicit learning and total adaptation (red lines in *Figure 5—figure supplement 2* and *Figure 5—figure supplement 3*), a positive statistically significant relationship occurred between implicit learning and total adaptation (green lines in *Figure 5—figure supplement 2*), no statistically significant relationship occurred between implicit learning and total adaptation (black lines in *Figure 5—figure supplement 2*), and a positive statistically significant relationship occurred between explicit learning and total adaptation (green lines in *Figure 5—figure supplement 3*). Statistically significant relationships were detected using a linear regression across the 30 participants in each simulation ($P < 0.05$). The bottom row in *Figure 5—figure supplement 3*, shows the average $R^2$ statistic for these linear regressions.

