## [Editor Report]

The interaction between implicit and explicit processes is central for motor learning. The present study builds upon diverse and sometimes seemingly conflicting data sets to propose a computational model, delineating a competing relationship between the explicit and implicit learning process during motor adaptation. The model provides a number of conceptual insights about the nature of error-based learning, not just for researchers in sensorimotor learning but also for those studying human learning in general.

---

## [Decision Letter]

[Editors’ note: after the initial submission, the editors requested that the authors prepare an action plan. This action plan was to detail how the reviewer concerns could be addressed in a revised submission. What follows is the initial editorial request for an action plan.]

Thank you for sending your article entitled "Competition between parallel sensorimotor learning systems" for peer review at eLife. Your article is being evaluated by 3 peer reviewers, and the evaluation is being overseen by a Reviewing Editor and Michael Frank as the Senior Editor. The reviewers have opted to remain anonymous.

Given the list of essential revisions, including new experiments, the editors and reviewers invite you to respond as soon as you can with an action plan for the completion of the additional work. We expect a revision plan that under normal circumstances can be accomplished within two months, although we understand that in reality revisions will take longer at the moment. We plan to share your responses with the reviewers and then advise further with a formal decision. After full discussion, all reviewers agree that this work if accepted would have a significant impact on the field. However, the reviewers raised a list of major concerns over the proposed model and its data handling. In particular, a lack of strong alternative models, a possible lack of explanatory power for outstanding findings in the field, and a strong reliance on modeling assumptions make it hard to accept the paper/model as it currently stands. We hope the authors can adequately address the following major concerns. [Editors’ note, the major concerns referenced here are detailed later in this decision narrative].

[Editors’ note: the authors provided an initial action plan in response to reviewer comments. Following the initial action plan, the authors were provided additional reviewer comments. These concerns were addressed in a second action plan. Additional reviewer comments were provided after the second action plan. During the composition of a third action plan, a decision was made to reject the manuscript.]

[Editors’ note: the authors submitted for reconsideration following the decision after peer review. What follows is the decision letter after the first round of review.]

Thank you for resubmitting your work entitled "Competition between parallel sensorimotor learning systems" for consideration by *eLife*. Your article has been reviewed by 3 peer reviewers, one of whom is a member of our Board of Reviewing Editors, and the evaluation has been overseen by Michael J Frank as the Senior Editor. The reviewers have opted to remain anonymous.

All the reviewers concluded that the study is not ready for publishing in *eLife*. The primary concern is that the direct evidence for supporting the competition model is weak or simply lacking. In the previous manuscript, the only solid, quantitative evidence is the negative correlation between explicit and implicit learning (Figure 3H). But as we mentioned in the last letter decision, this negative correlation can also be obtained by the alternative (and mainstream) model based on both target error and sensory prediction error. You provided a piece of new but essential evidence to differentiate these two models in the revision plan. Still, this new result (i.e., the negative correlation between total learning and implicit learning) is at odds with multiple datasets that have been published. We find this will continue to be a problem even we provide you more time to revise the paper or continue to collect more data as listed in the revision plan. The submission has been on discussion for a prolonged period as the reviewers hope to see solid evidence for the conceptual step-up that you presented in the manuscript. However, we are regretful that the study is not ready for publishing as it currently stands.

[Editors’ note: the authors appealed the editorial decision. The appeal was granted and the authors were invited to submit a revised manuscript and response to reviewer comments. What follows are the reviewer comments provided after the initial submission of the manuscript.]

Concerns:

1. There is no clear testing of alternative models. After Figure 1 and 2, the paper goes on with the competition model only. The independent model will never be able to explain differences in implicit learning for a given visuomotor rotation (VMR) size, so it is not a credible alternative to the competition model in its current form. It seems possible that extensions of that model based on the generalization of implicit learning (Day et al. 2016, McDougle et al. 2017) or task-dependent differences in error sensitivity could explain most of the prominent findings reported here. We hope the authors can show more model comparison results to support their model.

2. The measurement of implicit learning is compromised by not asking participants to stop aiming upon entering the washout period. This happened at three places in the study, once in Experiment 2, once in the re-used data by Fernandez-Ruiz et al., 2011, and once in the re-used data by Mazzoni and Krakauer 2006 (no-instruction group). We would like to know how the conclusion is impacted by the ill-estimated implicit learning.

3. All the new experiments limit preparation time (PT) to eliminate explicit learning. In fact, the results are based on the assumption that shortened PT leads to implicit-only learning (e.g., Figure 4). Is the manipulation effective, as claimed? This paradigm needs to be validated using independent measures, e.g., by direct measurement of implicit and/or explicit strategy.

4. Related to point 3, it is noted that the aiming report produced a different implicit learning estimate than a direct probe in Exp1 (Figure S5E). However, Figure 3 only reported Exp1 data with the probe estimate. What would the data look like with report-estimated implicit learning? Will it show the same negative correlation results with good model matches (Figure 3H)?

5. Figure 3H is one of the rare pieces of direct evidence to support the model, but we find the estimated slope parameter is based on a B value of 0.35, which is unusually high. How is it obtained? Why is it larger than most previous studies have suggested? In fact, this value is even larger than the learning rate estimated in Exp 3 (Figure 6C). Is it related to the small sample size (see below)?

6. The direct evidence supporting a competition of implicit and explicit learning is still weak. Most data presented here are about the steady-state learning amplitude; the authors suggest that steady-state implicit learning is proportional to the size of the rotation (Equation 5). But this is directly contradictory to the experimental finding that implicit learning saturates quickly with rotation size (Morehead et al., 2017; Kim et al., 2018). This prominent finding has been dismissed by the proposed model. On the other hand, one classical finding for conventional VMR is that explicit learning would gradually decrease after its initial spike, while implicit learning would gradually increase. The competition model appears unable to account for this stereotypical interaction. How can we account for this simple pattern with the new model or its possible extensions?

7. General comments about the data:

Experiments 3 and 4 (Figures 6 and 7) do not contribute to the theorization of competition between implicit and explicit learning. The authors appear to use them to show that implicit learning properties (error sensitivity) can be modified, but then this conclusion critically depends on the assumption that the limit-PT paradigm produces implicit-only learning, which is yet to be validated.

Figure 4 is not a piece of supporting evidence for the competition model since those obtained parameters are based on assumptions that the competition model holds and that the limit-PT condition "only" has implicit learning.

8. Concerns about data handling:

All the experiments have a limited number of participants. This is problematic for Exp 1 and 2, which have correlation analysis. Note Fig 3H contains a critical, quantitative prediction of the model, but it is based on a correlation analysis of n=9. This is an unacceptably small sample. Experiment 3 only had 10 participants, but its savings result (argued as purely implicit learning-driven) is novel and the first of its kind.

Experiment 2: This experiment tried to decouple implicit and explicit learning, but it is still problematic. If the authors believe that the amount of implicit adaptation follows a state-space model, then the measure of early implicit is correlated to the amount of late implicit because Implicit_late = (ai)^N * Implicit_early where N is the number of trials between early and late. Therefore the two measures are not properly decoupled. To decouple them, the authors should use two separate ways of measuring implicit and explicit. To measure explicit, they could use aim report (Taylor, Krakauer and Ivry 2011), and to measure implicit independently, they could use the after-effect by asking the participants to stop aiming as done here. They actually have done so in experiment 1 but did not use these values.

Figure 6: To evaluate the saving effect across conditions and experiments, using the interaction effect of ANOVA shall be desired.

Figure 8: The hypothesis that they did use an explicit strategy could explain why the difference between the two groups rapidly vanishes. Also, it is unclear whether the data from Taylor and Ivry (2011) are in favor of one of the models as the separate and shared error models are not compared.

*Reviewer #1:*

The present study by Albert and colleagues investigates how implicit learning and explicit learning interact during motor adaptation. This topic is under heated debate in the sensorimotor adaptation area in recent years, spurring diverse behavioral findings that warrant a unified computational model. The study is to fulfill this goal. It proposes that both implicit and explicit adaptation processes are based on a common error source (i.e., target error). This competition leads to different behavioral patterns in diverse task paradigms.

I find this study a timely and essential work for the field. It makes two novel contributions. First, when dissecting the contribution of explicit and implicit learning, the current study highlights the importance of distinguishing apparent learning size and covert changes in learning parameters. For example, the overt size of implicit learning could decrease while the related learning parameters (e.g., implicit error sensitivity) remain the same or even increase. Many researchers for long have overlooked this dissociation. Second, the current study also emphasizes the role of target error in typical perturbation paradigms. This is an excellent waking call since the predominant view now is that sensory prediction error is for implicit learning and target error for explicit learning.

Given that the paper aims to use a unified model to explain different phenomena, it mixes results from previous work and four new experiments. The paper's presentation can be improved by reducing the use of jargon and making straightforward claims about what the new experiments produce and what the previous studies produce. I will give a list of things to work on later (minor concerns).

Furthermore, my major concern is whether the property of the implicit learning process (e.g., error sensitivity) can be shown subjective to changes without making model assumptions.

My major concern is whether we can provide concrete evidence that the implicit learning properties (error sensitivity and retention) can be modified. Even though the authors claim that the error sensitivity of implicit learning can be changed, and the change subsequently leads to savings and interference (e.g., Figures 6 and 7), I find that the evidence is still indirect, contingent on model assumptions. Here is a list of results that concern error sensitivity changes.

1. Figures 1 and 2: Instruction is assumed to leave implicit learning parameters unchanged.

2. Figure 4: It appears that the implicit error sensitivity increases during relearning. However, Fig4D cannot be taken as supporting evidence. How the model is constructed (both implicit and explicit learning are based on target error) and what assumptions are made (Low RT = implicit learning, High RT = explicit + implicit) determine that implicit learning's error sensitivity must increase. In other words, the change in error sensitivity resulted from model assumptions; whether implicit error sensitivity by itself changes cannot be independently verified by the data.

3. Figure 6: With limited PT, savings still exists with an increased implicit error sensitivity. Again, this result also relies on the assumption that limited PT leads to implicit-only learning. Only with this assumption, the error sensitivity can be calculated as such.

4. Figure 7: with limited PT anterograde interference persists, appearing to show a reduced implicit error sensitivity. Again, this is based on the assumption that the limited PT condition leads to implicit-only learning.

Across all the manipulations, I still cannot find an independent piece of evidence that task manipulations can modify learning parameters of implicit learning without resorting to model assumptions. I would like to know what the authors take on this critical issue. Is it related to how error sensitivity is defined?

*Reviewer #2:*

The paper provides a thorough computational test of the idea that implicit adaptation to rotation of visual feedback of movement is driven by the same errors that lead to explicit adaptation (or strategic re-aiming movement direction), and that these implicit and explicit adaptive processes are therefore in competition with one another. The results are incompatible with previous suggestions that explicit adaptation is driven by task errors (i.e. discrepancy between cursor direction and target direction), and implicit adaptation is driven by sensory prediction errors (i.e. discrepancy between cursor direction and intended movement direction). The paper begins by describing these alternative ideas via state-space models of trial by trial adaptation, and then tests the models by fitting them both to published data and to new data collected to test model predictions.

The competitive model accounts for the balance of implicit-explicit adaptations observed previously when participants were instructed how to counter the visual rotation to augment explicit learning across multiple visual rotation sizes (20, 40 and 60 degrees; Neville and Cressman, 2018). It also fits previous data in which the rotation was introduced gradually to augment implicit learning (Saijo and Gomi, 2010). Conversely, a model based on independent adaptation to target errors and sensory prediction errors could not reproduce the previous results. The competitive model also accounts for individual participant differences in implicit-explicit adaptation. Previous work showed that people who increased their movement preparation time when faced with rotated feedback had smaller implicit reach aftereffects, suggesting that greater explicit adaptation led to smaller implicit learning (Fernandes-Ruis et al., 2011). Here the authors replicated this general effect, but also measured both implicit and explicit learning under experimental manipulation of preparation time. Their model predicted the observed inverse relationship between implicit and explicit adaptation across participants.

The authors then turned their attention to the issue of persistent sensorimotor memory – both in terms of savings (or benefit from previous exposure to a given visual rotation) and interference (or impaired performance due to exposure to an opposite visual rotation). This is a topic that has a long history of controversy, and there are conflicting reports about whether or not savings and interference rely predominantly on implicit or explicit learning. The competition model was able to account for some of these unresolved issues, by revealing the potential for a paradoxical situation in which the error sensitivity of an implicit learning process could increase without observable increase in implicit learning rate (or even a reduction in implicit learning rate). The authors showed that this paradox was likely at play in a previous paper that concluded that saving is entirely explicit (Haith et al., 2015), and then ran a new experiment with reduced preparation time to confirm some recent reports that implicit learning can result in savings. They used a similar approach to show that long-lasting interference can be induced for implicit learning.

Finally, the authors considered data that has long been cited to provide evidence that implicit adaptation is obligatory and driven by sensory prediction error (Mazzoni and Krakauer, 2006). In this previous paper, participants were instructed to aim to a secondary target when the visual feedback was rotated so that the cursor error towards the primary target would immediately be cancelled. Surprisingly, the participants' reach directions drifted even further away from the primary target over time – presumably in an attempt to correct the discrepancy between their intended movement direction and the observed movement direction. However, a competition model involving simultaneous adaptation to target errors from the primary target and opposing target errors from the secondary target offers an alternative explanation. The current authors show that such a model can capture the implicit "drift" in reach direction, suggesting that two competing target errors rather than a sensory prediction error can account for the data. This conclusion is consistent with more recent work by Taylor and Ivry (2011), which showed that reach directions did not drift in the absence of a second target, when participants were coached to immediately cancel rotation errors in advance. The reason for small implicit aftereffect reported by Taylor and Ivry (2011) is open for interpretation. The authors of the current paper suggest that adaptation to sensory prediction errors could underlie the effect, but an alternative possibility is that there is an adaptive mechanism based on reinforcement of successful action.

Overall, this paper provides important new insights into sensorimotor learning. Although there are some conceptual issues that require further tightening – including around the issue of error sensitivity versus observed adaptation, and the issue of whether or not there is evidence that sensory prediction errors drive visuomotor rotation – I think that the paper will be highly influential in the field of motor control.

Line 19: I think it is possible that more than 2 adaptive processes contribute to sensorimotor adaptation – perhaps add "at least".

Lines 24-27: This section does not refer to specific context or evidence and is consequently obtuse to me.

Line 52: Again – perhaps add "at least".

Line 68: I am not really sure what you are getting at by "implicit learning properties" here. Can you clarify?

Line 203: I think the model assumption that there was zero explicit aiming during washout is highly questionable here. The Morehead study showed early washout errors of less than 10 degrees, whereas errors were ~ 20 degrees in the abrupt and ~35 degrees in the gradual group for Saijo and Gomi. I think it highly likely that participants re-aimed in the opposite direction during washout in this study – what effect would this have on the model conclusions?

Line 382: It would be helpful to explicitly define what you mean by learning, adaptation, error sensitivity, true increases and true decreases in this discussion – the terminology and usage are currently imprecise. For example, it seems to me that according to the zones defined "truth" refers to error sensitivity, and "perception" refers to observed behaviour (or modelled adaptation state), but this is not explicitly explained in the text. This raises an interesting philosophical question – is it the error sensitivity or the final adaptation state associated with any given process that "truly" reflects the learning, savings or interference experienced by that process? Some consideration of this issue would enhance the paper in my opinion.

Figure 5 label: Descriptions for C and D appear inadvertently reversed – C shows effect of enhancing explicit learning and D shows effect of suppressing explicit learning.

Line 438: The method of comparing learning rate needs justification (i.e. comparing from 0 error in both cases so that there is no confound due to the retention factor).

Line 539: I am not sure if I agree that the implicit aftereffect in the no target group need reflect error-based adaptation to sensory prediction error. It could result from a form of associative learning – where a particular reach direction is associated with successful target acquisition for each target. The presence of an adaptive response to SPE does not fit with any of the other simulations in the paper, so it seems odd to insist that it remains here. Can you fit a dual error model to the Taylor and Ivry (single-target) data? I suspect it would not work, as the SPE should cause non-zero drift (i.e. shift the reach direction away from the primary target).

Line 556: Be precise here – do you really mean SPE? It seems as though you only provided quantitative evidence of a competition between errors when there were 2 physical targets.

Line 606: Is it necessarily the explicit response that is cached? I think of this more as the development of an association between action and reward – irrespective of what adaptive process resulted in task success. It would be nice to know what happened to aftereffects after blocks 1 and 2 in study 3. An associative effect might be switched on and off more easily – so if a mechanism of that kind were at play, I would predict a reduced aftereffect as in Huberdeau et al.

Line 720: Ref 12 IS the Mazzoni and Krakauer study, ref 11 involves multiple physical targets, and ref 21 is the Taylor and Ivry paper considered above and in the next point. None of these papers therefore require an explanation based on SPE. However, ref #17 appears to provide a more compelling contradiction to the notion that target errors drive all forms of adaptation – as people adapt to correct a rotation even when there is never a target error (because the cursor jumps to match the cursor direction). A possible explanation that does not involve SPE might be that people retain a memory of the initial target location and detect a "target error" with respect to that location.

Line 737: I don't agree with this – the observation of an after-effect with only 1 physical target but instructed re-aiming so that there was 0 target error, certainly implies some other process besides a visual target error as a driver of implicit learning. However, as argued above, such a process need not be driven by SPE. Model-free processes could be at play…

Line 791: Some more detail on how timing accuracy was assured in the remote experiments is needed. The mouse sample rate can be approximately 250 Hz, but my experience with it is that there can be occasional long delays between samples on some systems. The delay between commands to print to the screen and the physical appearance on the screen can also be long and variable – depending on the software and hardware involved.

Line 861: How did you specify or vary the retention parameters to create these error sensitivity maps?

Line 865: Should not the subscripts here be "e" and "I" rather than "f" and "s"?

Line 1009: Why were data sometimes extracted using a drawing package (presumably by eye? – please provide further details), and sometimes using GRABIT in Matlab?

Line 1073: More detail about the timing accuracy and kinematics of movement are required.

Line 1148: Again – should not the subscripts here be e and I rather than f and s?

*Reviewer #3:*

In this paper, Albert and colleagues explore the role of target error and sensory prediction error on motor adaptation. They suggest that both implicit and explicit adaptation would be driven by target error. In addition, implicit adaptation is also influenced by sensory prediction error.

While I appreciate the effort that the authors have done to come up with at theory that could account for many studies, there is not a single figure/result that does not suffer from main limitations as I highlight in the major comments below. Overall, the main limitations are:

I believe that the authors neglect some very relevant papers that contradict their theory and model.

– They did not take into account some results on the topic such as Day et al. 2016 and McDougle et al. 2017. It is true that they acknowledge it and discuss it but I am not convinced by their arguments (more on this topic in a major comment about the discussion below).

– They did not take into account the fact that there is no proof that limiting RT is a good way to suppress the explicit component of adaptation. When the number of targets is limited, these explicit responses can then be cached without any RT cost. McDougle and Taylor, Nat Com (2019) demonstrated this for 2 targets (here the authors used only 4 targets). There exist no other papers that proof that limiting RT would suppress the explicit strategy as claimed by the authors. To do so, one needs to limit PT and to measure the explicit or the implicit component. The authors should prove that this assumption holds because it is instrumental to the whole paper. The authors acknowledge this limitation in their discussion but dismiss it quite rapidly. This manipulation is so instrumental to the paper that it needs to be proven, not argued (more on this topic in a major comment about the discussion below).

– They did not take into account the fact that the after-effect is a measure of implicit adaptation if and only if the participants are told to abandon any explicit strategy before entering the washout period (as done in Taylor, Krakauer and Ivry, 2011). This has important consequences for their interpretation of older studies because it was then never asked to participants to stop aiming before entering the washout period as the authors did in experiment 2.

There are also major problems with statistics/design:

– I disagree with the authors that 10-20 people per group (in their justification of the sample size in the second document) is standard for motor adaptation experiments. It is standard for their own laboratories but not for the field anymore. They even did not reach N=10 for some of the reported experiments (one group in experiment 1). Furthermore, this sample size is excessively low to obtain reliable estimation of correlations. Small N correlation cannot be trusted: https://garstats.wordpress.com/2018/06/01/smallncorr/

– Justification of sample size is missing. What was the criteria to stop data collection? Why is the number of participants per group so variable (N=9 and N=13 for experiment 1, N=17 for experiment 2, N=10 for experiment 3, N=10 ? per group for experiment 4). Optional stopping is problematic when linked to data peeking (Armitage, McPherson and Rowe, Journal of the Royal Statistical Society. Series A (General), 1969). It is unclear how optional stopping influences the outcome of the statistical tests of this paper.

– The authors should follow the best practices and add the individual data to all their bar graphs (Rousselet et al. EJN 2016).

– Missing interactions (Nieuwenhuis et al., Nature Neuro 2011), misinterpretation of non-significant p-values (Altman 1995 https://www.bmj.com/content/311/7003/485)

Given these main limitations, I don't think that the authors have a convincing case in favor of their model and I don't think that any of these results actually support the error competition model.

Detailed major comments per equation/figure/section:

Equation 2: the authors note that the sensory prediction error is "anchored to the aim location " (line 104-105) which is exactly what Day et al. 2016 and McDougle et al. 2017 demonstrated. Yet, they did not fully take into account the implication of this statement. If it is so, it means that the optimal location to determine the extent of implicit motor adaptation is the aim location and not the target. Indeed, if the SPE is measured with respect to the aim location and is linked to the implicit system, it means that implicit adaptation will be maximum at the aim location and, because of local generalization, the amount of implicit adaptation will decay gradually when one wants to measure this system away of that optimal location (Figure 7 of Day et al). This means that, the further the aiming direction is from the target, the smaller the amount of implicit adaptation measured at the target location will be. This will result in an artificial negative correlation between the explicit and implicit system without having to relate to a common error source.

Equation 3: the authors do not take into account results from their lab (Marko et al., Herzfeld) and from others (Wei and Kording) that show that the sensitivity to error depends on the size of the rotation.

Equation 5: In this equation, the authors suggest that the steady-state amount of implicit adaptation is directly proportional to the size of the rotation. Thanks to a paradigm similar to Mazzoni and Krakauer 2006, the team of Rich Ivry has demonstrated that the implicit response saturates very quickly with perturbation size (Kim et al., communications Biology, 2018, see their Figure 1).

Data from Cressman, Figure 1: Following Day et al., one should expect that, when the explicit strategy is larger (instruction group in Cressman et al. compared to the no-instruction group), the authors are measuring the amount of implicit adaptation further from aiming direction where it is maximum. As a result, the amount of implicit adaptation appears smaller in the instruction group simply because they were aiming more than the no-instruction group (Figure 1G).

– Figure 1H: the absence of increase in implicit adaptation is not due to competition as claimed by the authors but is due to the saturation of the implicit response with increasing rotation size (Kim et al. 2018). It saturates at around 20{degree sign} for all rotations larger than 6{degree sign}.

Data from Saijo and Gomi: Here the authors interpret the after-effect as a measure of implicit adaptation but it is not as the participants were not told that the perturbation would be switched off.

– Even if these after-effects did represent implicit adaptation, these results could be explained by Kim et al. 2018 and Day et al. 2016. For a 60{degree sign} rotation, the implicit component of adaptation will saturate around 15-20{degree sign} (like for any large perturbation). The explicit component has to compensate for that but it does a better job in the abrupt condition than in the gradual condition where some target error remains. Given that the aiming direction is larger in the abrupt case than in the gradual case, the amount of implicit adaptation is again measured further away from its optimal location in the abrupt case than in the gradual case (Day et al. 2016 and McDougle et al. 2017).

– There is no proof that introducing a perturbation gradually suppresses explicit learning (line 191). The authors did not provide a citation for that and I don't think there is one. People confound awareness and explicit (also valid for line 212). Rather, given that the implicit system saturates at around 20{degree sign} for large rotation, I would expect that the re-aiming accounts for 20{degree sign} of the ~40{degree sign} of adaptation in the gradual condition.

Line 205-215: The chosen parameters appear very subjective. The authors should perform a sensitivity analyses to demonstrate that their conclusions do not depend on these specific parameters.

Figure 3: The after-effect in the study of Fernandez-Ruiz et al. does not solely represent the implicit adaptation component as these researchers did not tell their participants that they should stop aiming during the washout.

– Correlations based on N=9 should not be considered as meaningful. Such correlation is subject to the statistical significance fallacy: it can only be true if it is significant with N=9 while this represents a fallacy (Button et al. 2013).

Experiment 1 of the authors suffer from several limitations:

– small sample size (N=9 for one group). Why is the sample size different between the two groups?

– Limiting RT does not abolish the explicit component of adaptation (Line 1027: where is the evidence that limit PT is effective in abolishing explicit re-aiming?). Haith and colleagues limited reaction time on a small subset of trials. If the authors want to use this manipulation to limit re-aiming, they should first demonstrate that this manipulation is effective in doing so (other authors that have used this manipulation but have failed to do validate it first). I wonder why the authors did not measure the implicit component for their limit PT group like they did in the No PT limit group. That is required to validate their manipulation.

– The negative correlation from Figure 3H can be explained by the fact that the SPE is anchored at the aiming direction and that, the larger the aiming direction is, the further the authors are measuring implicit adaptation away from its optimal location.

– The difference between the PT limit and NO PT limit is not very convincing. First, the difference is barely significant (line 268). Why did the authors use the last 10 epochs for experiment 1 and the last 15 for experiment 2? This looks like a post-hoc decision to me and the authors should motivate their choice and should demonstrate that their results hold for different choice of epochs (last 5, 10, 15 and 20) to demonstrate the robustness of their results. Second, the degrees of freedom of the t-test (line 268) does not match the number of participants (9 vs. 13 but t(30)?)

– Why did the authors measure the explicit strategy via report (Figure S5E) while they don't use those values for the correlations? This looks like a post-hoc decision to me.

Experiment 2:

– This experiment is based on a small sample size to test for correlations (N=17). What was the objective criterion used to stop data collection at N=17 and not at another sample size? This should be reported.

– This experiment does not decouple implicit and explicit in contrast to what the authors pretend. If the authors believe that the amount of implicit adaptation follows a state-space model, then the measure of early implicit is correlated to the amount of late implicit because Implicit_late = (ai)^N * Implicit_early where N is the number of trials between early and late. Therefore the two measures are not properly decoupled. To decouple them, the authors should use two separate ways of measuring implicit and explicit. To measure explicit, they could use aim report (Taylor, Krakauer and Ivry 2011) and to measure implicit independently, they could use the after-effect by asking the participants to stop aiming as done here. They actually have done so in experiment 1 but did not use these values.

Figure 4:

– Data from the last panel of Figure 4B (line 321-333) should be analyzed with a 2x2 ANOVA and not by a t-test if the authors want to make the point that the learning differences were higher in high PT than low PT trials (Nieuwenhuis et al., Nature Neuroscience, 2011).

– Lines 329-330: the authors should demonstrate that the explicit strategy is actually suppressed by measuring it via report. It is possible that limiting PT reduces the explicit strategy but it might not suppress it. Therefore, any remaining amount of explicit strategy could be subject to savings.

– Coltman and Gribble (2019) demonstrated that fitting state-space models to individual subject's data was highly unreliable and that a bootstrap procedure should be preferred. Lines 344-353: the authors should replace their t-tests with permutation tests in order to avoid fitting the model to individual subject's data.

– It is unclear how the error sensitivity parameters analysed in Figure 4D were obtained. I can follow in the Results section but there is basically nothing on this in the methods. This needs to be expanded.

– The authors make the assumption that the amount of implicit adaptation is the same for the high PT target and for the low PT target. What is the evidence that this assumption is reasonable? Those two targets are far apart while implicit adaptation only generalizes locally. Furthermore, low PT target is visited 4 times less frequently than the high PT target. The authors should redo the experiment and should measure the implicit component for the high PT trials to make sure that it is related to the implicit component for the low PT trials.

– I don't understand why the authors illustrated the results from line 344-349 on Figure 4D and not the results of the following lines, which are also plausible. By doing so, the authors biased the results in favor of their preferred hypothesis. This figure is by no means a proof that the competition hypothesis is true. It shows that "if" the competition hypothesis is true, then there are surprising results ahead. The authors should do a better job at explaining that both models provide different interpretation of the data.

Figure 5: This figure also represent a biased view of the results (like Figure 4D). The competition hypothesis is presented in details while the alternative hypothesis is missing. How would the competition map look like with separate errors, especially when taking into account that the SPE (and implicit adaptation) is anchored at the aiming direction (generalization)?

Figure 6:

This figure is compatible with the fact that limiting PT on all trials is not efficient to suppress explicit adaptation, at least less so than doing it on 20% of the trials (see McDougle et al. 2019 for why this is the case) and that the remaining explicit adaptation leads to savings.

– I don't understand why the authors did not measure implicit (and therefore explicit adaptation) directly in these experiment like they did in experiment 2. This would have given the authors a direct readout of the implicit component of adaptation and would have validated the fact that limiting PT might be a good way to suppress explicit adaptation. Again, this proof is missing in the paper.

– Experiment 3 is based on a very limited number of participants (N=10!). No individual data are presented.

– Data from Figure 6C should be analyzed with an ANOVA and an interaction between the factor experiment (Haith vs. experiment 3) and block (1 vs. 2) should be demonstrated to support such conclusion (Nieuwenhuis et al. 2011).

Figure 7:

The data contained in this figure suffer from the same limitations as in the previous graphs: limiting PT does not exclude explicit strategy, sample size is small (N=10 per group), no direct measure of implicit or explicit is provided.

– In addition, no statistical tests are provided beyond the two stars on the graph. The data should be analyzed with an ANOVA (Nieuwenhuis et al. 2011).

– The number of participants per group is never provided (N=20 for both groups together).

– It is unclear to me how this result contributes to the dissociation between the separate and shared error models.

Figure 8:

The whole explanation here seems post-hoc because none of the two models actually account for this data. The authors had to adapt the model to account for this data. Note that despite that, the model would fail to explain the data from Kim et al. 2018 that represents a very similar task manipulation.

– Line 463-467: the authors claim equivalence based on a non-significant p-value (Altman 1995). Given the small effect size, they don’t have any power to detect effects of small or medium size. They cannot conclude that there is no difference. They can only conclude that they don’t have enough power to detect a difference. As a result, it does NOT suggest that implicit adaptation was unaltered by the changes in explicit strategy.

– In Mazzoni and Krakauer, the aiming direction was neither controlled nor measured. As a result, given the appearance of a target error with training, it is possible that the participants aimed in the direction opposite to the target error in order to reduce it. This would have reduced the apparent increase in implicit adaptation. The authors argue against this possibility based on the 47.8{degree sign} change in hand angle due to the instruction to stop aiming. I remained unconvinced by this argument as I would like to get more info about this change (Mean, SD, individual data). Furthermore, it is unclear what the actual instructions were. Asking to stop aiming or asking to bring one’s invisible hand on the primary target will have different effects on the change in hand angle.

– The data during the washout period suffers from the fact that the participants from the no-strategy group were not told to stop aiming. The hypothesis that they did use an explicit strategy could explain why the difference between the two groups rapidly vanishes. In other words, if the authors want to use this experiment to demonstrate support any of their hypotheses, they should redo it properly by telling the participants to stop using any explicit strategy at the start of the washout period to make sure that the after-effect is devoid of any explicit strategy.

– It is unclear whether the data from Taylor and Ivry (2011) are in favor of one of the models as the separate and shared error models are not compared.

Discussion: it is important and positive that the authors discuss the limitation of their approach but I feel that they dismiss potential limitations rather quickly even though these are critical for their conclusions. They need to provide new data to prove those points rather than arguments.

On limiting PT (lines 605-615):

The authors used three different arguments to support the fact that limiting PT suppresses explicit strategy.

– Their first argument is that Haith did not observe savings in low PT trials. This is true but Haith only used low PT trials (with a change in target) on 20% of the trials. Restricting RT together with a switch in target location is probably instrumental in making the caching of the response harder. This is very different in the experiments done by the authors. In addition, one could argue that Haith et al. did not find evidence for savings but that these authors had limited power to detect a small or medium effect size (their N=12 per group). I agree that savings in low PT trials is smaller than in high PT trials but is savings completely absent in low PT trials? Figure 6H shows that learning is slightly better on block 2 compared to block 1. N=12 is clearly insufficient to detect such a small difference.

– Their second argument is that they used four different targets. McDougle et al. demonstrated caching of explicit strategy without RT costs for two targets and impossibility to do so for 12 targets. The authors could use the experimental design of McDougle if they wanted to prove that caching explicit strategies is impossible for four targets. I don't see why you could cache strategies without RT cost for 2 targets but not for 4 targets. This argument in not convincing.

– The last argument of the authors is that they imposed even shorter latencies (200ms) than Haith (300ms). Yet, if one can cache explicit strategies without reaction time cost, it does not matter whether a limit of 200 or 300ms is imposed as there is no RT cost.

On Generalization (line 678-702).

How much does the amount of implicit adaptation decays with increasing aiming direction? Above, I argued that the data from Day et al. would predict a negative correlation between the explicit and the implicit components of adaptation and a decrease in implicit adaptation with increasing rotation size. The authors clearly disagree on the basis of four arguments. None of them convinced me.

– First, they estimate this decrease to be only 5{degree sign} based on these two papers (FigS5A and B but 5C shows ~10{degree sign}). This seems to be a very conservative estimate as Day et al. reported a 10{degree sign} reduction in after-effects for 40{degree sign} of aiming direction (see their Figure 7). An explicit component of 40{degree sign} was measured by Neville and Cressman for a 60{degree sign} rotation. The 10{degree sign} reduction based on a 40{degree sign} explicit strategy fits perfectly with data from Figure 1G (black bars) and Figure 2F. Off course, the experimental conditions will influence this generalization effect but this should push the authors to investigate this possibility rather than to dismiss it because the values do not precisely match. How close should it match to be accepted?

– Second, it is unclear how this generalization changes with the number of targets (argument on lines 687-689). This has never been studied and cannot be used as an argument based on further assumptions. Furthermore, I am not sure that the generalization would be so different for 2 or 4 targets.

– Third, the authors measured the explicit strategy in experiment 1 via report in a very different way than what is usually used by the authors as the participants do not have to make use of them. It seems to be suboptimal as the authors did not use them for their correlation on Figure 3H and the difference reported in Figure S5E is tiny (no stats are provided) but is based on a very limited number of participants with no individual data to be seen. If it is suboptimal for Figure 3H why is it sufficient as an argument?

– Fourth, when interpreting the data of Neville and Cressman (line 690-692), the authors mention that there were no differences between the three targets even though two of them corresponded to aim directions for other targets. As far as I can tell, the absence of difference in implicit adaptation across the three targets is not mentioned in the paper by Neville and Cressman as they collapsed the data across the three targets for their statistical analyses throughout the paper. In addition, I don't understand why we should expect a difference between the three targets. If the SPE and the implicit process are anchored to the aiming direction and given that the aiming direction is different for the three targets, I would not expect that the aiming direction of a visible target would be influenced by the fact that, for some participants, this aiming direction corresponds to the location of an invisible target.

– Finally, the authors argue here about the size of the influence of the generalization effect on the amount of implicit adaptation. They never challenge the fact that the anchoring of implicit adaptation on the aiming direction and the presence of a generalization effect (independently of its size) leads to a negative correlation between the implicit and explicit component of adaptations (their Figure 3) without any need for the competition model.

Final recommendations

– The authors should perform an unbiased analysis of a model that include separate error sources, the generalization effect and a saturation of implicit adaptation with increasing rotation size. In my opinion, such model would account for almost all of the presented results.

– They should redo all the experiments based on limited preparation time and should include direct measures of implicit or/and explicit strategies (for validation purposes). This would require larger group size.

– They should replicate the experiments where they need a measure of after-effect devoid of any explicit strategies as this has only become standard recently (experiment for Figure 2 and Figure 8). Not that for Figure 8, they might want to measure the explicit aim during the adaptation period as well.

[Editors’ note: further revisions were suggested prior to acceptance, as described below.]

Thank you for resubmitting your work entitled "Competition between parallel sensorimotor learning systems" for further consideration by *eLife*. Your revised article has been evaluated by Michael Frank (Senior Editor) and a Reviewing Editor.

The manuscript has been improved but there are some remaining issues that need to be addressed, as outlined below:

One review raised the concern about Experiment 1, which presented perturbations with increasing rotation size and elicited larger implicit learning than the abrupt perturbation condition. However, this design confounded the condition/trial order and the perturbation size. The other concern is the newly added non-monotonicity data set from Tsay's study. On the one hand, the current paper states that the proposed model might not apply to error clamp learning; on the other hand, this part of the results was "predicted" by the model squarely. Thus, can it count as evidence that the proposed model is parsimonious for all visuomotor rotation paradigms? This message must be clearly stated with a special reference to error-clamp learning.

Please find the two reviewers' comments and recommendations below, and I hope this will be helpful for revising the paper.

*Reviewer #1:*

The present study investigates how implicit and explicit learning interacts during sensorimotor adaptation, especially the role of performance or target error during this interaction. It is timely for the area that needs a more mechanistic model to explain diverse findings accumulated in recent years. The revision has addressed previous major concerns and provided extra data that supports the idea that implicit and explicit processes compete for a common target error for a large proportion of studies in the area. The paper is thoughtfully organized and convincingly presented (though a bit too long), including a variety of data sets.

As a repeal submission, the current work has successfully addressed previous major concerns:

1) Direct evidence for supporting the competition model is lacking.

The revision added new evidence that total learning is negatively correlated with implicit learning but positively correlated with explicit learning (Figure 4G and H). This part of the data argues against the alternative SPE model and provides direct support to the competition model. The added appendix 3 also includes other studies' data sets to strengthen the supports.

Furthermore, Experiment1 is new with an incremental perturbation to decrease rotation awareness and thus provides new support for the model. This also helps to address the previous concern that the data from Saijo and Gomi is not clean enough due to a lack of stop-aiming instruction during the measurement of implicit learning. The other not-so-clean data set from Fernadez-Ruiz is removed from the main text in the revision. Putting together, these new data sets and analysis results constitute a rich set of direct supports that address the biggest concern in the previous submission.

2) Lack of testing alternative models.

The revision tested an alternative idea that explicit learning compensates for the variation in implicit learning instead of the other way around as in their competition model (Figure 4). It also tested an alternative model based on the generalization of implicit learning anchored on the aiming direction (Figure 5). The evidence is clear: both alternative models failed to capture the data.

3) The limited preparation time appears not to exclude explicit learning.

This concern was raised by all previous reviewers. The reviewers also pointed out a possible indicator of explicit learning, i.e., a small spurious drop in reaching angle at the beginning of the no-aiming washout phase. The revision provided a new experiment to show that the small drop is most likely due to a time-dependent decay of implicit learning during the 30s instruction period. This new data set (Exp 3) is indeed convincing and important. I am impressed with the authors' thoughtful effort to explain this subtle and tricky issue.

In sum, I believe the revision did an excellent job of addressing all previous major concerns with more data, more thorough analysis, and more model comparisons. Here I suggest a few minor changes for this revision:

Line 59: the cited papers (17-21) did not all suggest savings resulted from changes of the implicit learning rate, as implied by the context here.

Some references are not in the right format: ref1 states "Journal of Neuroscience," while refs2 and 4 state "The Journal of neuroscience : the official journal of the Society for Neuroscience." Possibly caused by downloads from google scholar. Please keep it consistent.

Line66: replace the semicolon with a comma. Also, is citation 9 about showing a dominant role of SPE?

Figure 1H: it appears the no-aiming trials were at the end of each stepwise period, but in the Methods, it is stated that these trials appeared twice (at the middle and the end). Which version is true?

L1331: why use bootstrapping here? Why not simply fit the model to individual subjects (given n = 37 here)? Interestingly, when comparing 60-degree stepwise vs. 60-degree abrupt, a direct fit to individuals was used… In the Results, it is stated that a single parameter can parsimoniously "predict" the data. But in the Methods, it is clear that all data were used to "fit" the parameter p_i. Can we call this a prediction, strictly? Is it possible to fit a partial data set to show the same results? Not through bootstrapping but through using, say, two stepwise phases? This type of prediction has been done for Exp2, but not here.

Line 195: the authors ruled out the potential bias caused by the interdependence between implicit and explicit learning as one was obtained from subtracting the other from the total learning. By rearranging the model (Equation4) to have the implicit learning as a function of total learning (as opposed to explicit learning), and by showing the model prediction would not change after this re-arrangement, the authors tried to argue that the modeling results are not free from the inter-dependence of two types of learning. I think that this argument aims for the wrong target. The real concern is that implicit and explicit learning are not independently measured, which is true no matter how the model equation is arranged. The prediction of the rearranged model, of course, would provide the same model prediction since this does not change the model at all, given the inter-dependence of variables. All the data in Figure 1 have this problem. It is a data problem, not a model problem. And, the data refute the alternative, independent model, that is. There is no need to re-arrange the model without solving the problem in data.

The authors added the data set from Tsay et al. to Part 1, which shows a non-monotonic change of implicit learning with increasing rotation. However, this is only one version of their data set obtained by the online platform; the other version of the same experiment conducted in person shows a different picture with constant implicit learning over different rotations. Any particular reason to report one version instead of the other? Here owes an explanation to the readers.

Line 203: dual model -> competition model

Line463: "that the x-axis in Figures 5A-C slightly overestimates explicit strategy", the axis overestimates…

Line 496: this paragraph starts with a why question (why the competition model works better), but it does not address the question but shows it works better.

Figure 6C: not clear what the right panel is without axis labeling and a detailed description in the caption.

Line 563: the family dinner story is interesting and naughty, but our readers might not need it, especially when the paper is already long. I suggest removing it and starting this section with the burning question of why the implicit error sensitivity changes without the changes of implicit learning size.

Line 589: what is Timepoint 1 and 2, never mentioned before or in the figure. It is also confusing with a capital T.

Figure 7C: it would make much better sense to plot total learning in C along with the implicit and explicit learning.

L595: the data presented before in Figures2 are explained in this map illustration. Enhancing or suppressing explicit learning has been conceptualized as moving along the y-axis without changing the implicit error sensitivity. Retrospectively, this is also the case when the model is fitted to these data by assuming a constant implicit learning rate in previous figures. Is there any evidence to support this assumption for model fitting, or can we safely claim varying implicit learning rates could not account for the data better than otherwise?

L685: we observe implicit learning is suppressed as a result of anterograde interference. Without preparation time constraints, the impairment in implicit learning is less. Is there any way to compare their respectively implicit error sensitivity? This would give us some information about how explicit learning compensates.

Figure 10G: the conceptual framework is speculative. It is fine to discuss the possible neurophysiological underpinnings in the Discussion as it currently stands. But we are better off removing it from the Results.

*Reviewer #4:*

In this paper, Albert et al. test a novel model where explicit and implicit motor adaptation processes share an error signal. Under this error sharing scheme, the two systems compete – that is, the more that explicit learning contributes to behavior, the less that implicit adaptation does. The authors attempt to demonstrate this effect over a variety of new experiments and a comprehensive re-analysis of older experiments. They contend that the popular model of SPEs exclusively driving implicit adaptation (and implicit/explicit independence) does not account for these results. Once target error sensitivity is included into the model, the resulting competition process allows the model to fit a variety of seemingly disparate results. Overall, the competition model is argued to be the correct model of explicit/implicit interactions during visuomotor learning.

I'm of two minds on this paper. On the positive side, this paper has compelling ideas, a laudable breadth and amount of data/analyses, and several strong results (mainly the reduced-PT results). It is important for the motor learning field to start developing a synthesis of the 'Library of Babel' of the adaptation literature, as is attempted here and elsewhere (e.g., D. Wolpert lab's 'COIN' model). On the negative side, the empirical support feels a bit like a patch-work – some experiments have clear flaws (e.g., Exp 1, see below), others are considered in a vacuum that dismisses previous work (e.g., nonmonotonicity effect), and many leftover mysteries are treated in the Discussion section rather than dealt with in targeted experiments. While some of the responses to the previous reviewers are effective (e.g., showing that reduced PT can block strategies), others are not (e.g., all of Exp 1, squaring certain key findings with other published conflicting results, the treatment of generalization). The overall effect is somewhat muddy – a genuinely interesting idea worth pursuing, but unclear if the burden of proof is met.

(1) The stepwise condition in Exp 1 is critically flawed. Rotation size is confounded with time. It is unclear – and unlikely – that implicit learning has reached an asymptote so quickly. Thus, the scaling effect is at best significantly confounded, and at worst nearly completely confounded. I think that this flaw also injects uncertainty into further analyses that use this experiment (e.g., Figure 5).

(2) It could be argued that the direct between-condition comparison of the 60° blocks in Exp 1 rescues the flaw mentioned above, in that the number of completed trials is matched. However, plan- or movement-based generalization (Gonzalez Castro et al., 2011) artifacts, which would boost adaptation at the target for the stepwise condition relative to the abrupt one, are one way (perhaps among others) to close some of that gap. With reasonable assumptions about the range of implicit learning rates that the authors themselves make, and an implicit generalization function similar to previous papers that isolate implicit adaptation (e.g., σ around ~30°), a similar gap could probably be produced by a movement- or plan-based generalization model. [I note here that Day et al., 2016 is not, in my view, a usable data set for extracting a generalization function, see point 4 below.]

(3) The nonmonotonicity result requires disregarding the results of Morehead at al., 2017, and essentially, according to the rebuttal, an entire method (invariant clamp). The authors do mention and discuss that paper and that method, which is welcome. However, the authors claim that "We are not sure that the implicit properties observed in an invariant error context apply to the conditions we consider in our manuscript." This is curious and raises several crucial parsimony issues, for instance: the results from the Tsay study are fully consistent with Morehead 2017. First, attenuated implicit adaptation to small rotations (not clamps; Morehead Figure 5) could be attributed to error cancellation (assuming aiming has become negligible to nonexistent, or never happened in the first place). This (and/or something like the independence model) thus may explain the attenuated 15° result in Figure 1N. Second, and more importantly, the drop-off of several degrees of adaptation from 30°/60° to 90° in Figure 1N is eerily similar to that seen in Morehead '17. Here's what we're left with: An odd coincidence whereby another method predicts essentially these exact results but is simultaneously (vaguely) not applicable. If the authors agree that invariant clamps do limit explicit learning contributions (see Tsay et al. 2020), it would seem that similar nonmonotonicity being present in both rotations and invariant error-clamps works directly against the competition model. Moreover, it could be argued that the weakened 90˚ adaptation in clamps (Morehead) explains why people aim more during 90° rotations. A clear reason, preferably with empirical support, for why various inconvenient results from the invariant clamp literature (see point 6 for another example) are either erroneous, or different enough in kind to essentially be dismissed, is needed.

(4) The treatment given to non-target based generalization (i.e., plan/aim) is useful and a nice revision w/r/t previous reviewer comments. However, there are remaining issues that muddy the waters. First, it should be noted that Day et al. did not counterbalance the rotation sign. This might seem like a nitpick, but it is well known that intrinsic biases will significantly contaminate VMR data, especially in e.g. single target studies. I would thus not rely on the Day generalization function as a reasonable point of comparison, especially using linear regression on what is a Gaussian/cosine-like function. It appears that the generalization explanation is somewhat less handicapped if more reasonable generalization parameterizations are considered. Admittedly, they would still likely produce, quantitatively, too slow of a drop-off relative to the competition model for explaining Exps 2 and 3. This is a quantitative difference, not a qualitative one. W/r/t point 1 above, the use of Exp 1 is an additional problematic aspect of the generalization comparison (Figure 5, lower panels). All in all, I think the authors do make a solid case that the generalization explanation is not a clear winner; but, if it is acknowledged that it can contribute to the negative correlation, and is parameterized without using Day et al. and linear assumptions, I'd expect the amount of effect left over to be smaller than depicted in the current paper. If the answer comes down to a few degrees of difference when it is known that different methods of measuring implicit learning produce differences well beyond that range (Maresch), this key result becomes less convincing. Indeed, the authors acknowledge in the Discussion a range of 22°-45° of implicit learning seen across studies.

(5) I may be missing something here, but is the error sensitivity finding reported in Figure 6 not circular? No savings is seen in the raw data itself, but if the proposed model is used, a sensitivity effect is recovered. This wasn't clear to me as presented.

(6) The attenuation (anti-savings) findings of Avraham et al. 2021 are not sufficiently explained by this model.

(7) Have the authors considered context and inference effects (Heald et al. 2020) as possible drivers of several of the presented results?

---

## [Author Response]

[Editors’ note: The authors appealed the original decision. What follows is the authors’ response to the first round of review.]

Concerns:1. There is no clear testing of alternative models. After Figure 1 and 2, the paper goes on with the competition model only. The independent model will never be able to explain differences in implicit learning for a given visuomotor rotation (VMR) size, so it is not a credible alternative to the competition model in its current form. It seems possible that extensions of that model based on the generalization of implicit learning (Day et al. 2106, McDougle et al. 2017) or task-dependent differences in error sensitivity could explain most of the prominent findings reported here. We hope the authors can show more model comparison results to support their model.

Two studies have measured generalization in rotation tasks with instructions to aim directly at the target (Day et al., 2016; McDougle et al., 2017) and have demonstrated that learning generalizes around one’s aiming direction, as opposed to the target direction. As the reviewers note, this generalization rule can contribute to the negative correlation we observed between explicit learning and exclusion-based implicit learning. To address this, in our revised manuscript we directly compare the competition model to an alternative SPE generalization model. Our new analysis is documented in Figure 4 in the revised manuscript. First, we empirically compare the relationship between implicit and explicit learning in our data to generalization curves measured in past studies. In Figures 4A-B we overlay data that we collected in Experiments 2 and 3 with generalization curves measured by Krakauer et al. (2000) and Day et al. (2016). In this way, we test whether generalization is consistent with the correlations between implicit and explicit learning that we measured in our experiments.

In Figures 4B,C, we show dimensioned (i.e., in degrees) relationships between implicit and explicit learning. In Figure 4A, we show a dimensionless implicit measure in which implicit learning is normalized to its “zero strategy” level. To estimate this value, we used the y-intercepts in the linear regressions labeled “data” in Figures 3G and Q. Most notably, the magenta line (Day 1T) shows the aim-based generalization curve measured by Day et al. (2016), where assays were used to tease apart implicit and explicit learning.

These new results demonstrate that implicit learning in Experiments. 2 and 3 (black and brown points) declined about 300% more rapidly with increases in explicit strategy, than predicted by generalization measured by Day et al. (2016). Two empirical considerations would also suggest that the discrepancy between our data and the Day et al. (2006) generalization curve is even larger than that observed in Figures 4A-B.

1. Day et al. (2006) use 1 adaptation target, whereas our tasks used 4 targets. Krakauer et al. (2000) demonstrated that the generalization curve widens with increases in target number (see Figure 4AC). Thus, the Day et al. (2006) study data likely overestimate the extent of generalization-based decay in implicit learning expected in our experimental conditions.

Our “explicit angle” in Figures 4A-B is calculated by subtracting total adaptation and exclusion-based implicit learning. Thus, under the plan-based generalization hypothesis, this measure would overestimate explicit strategy. A generalization-based correction would “contract” our data along the x-axis in Figures 4C-E, increasing the disconnect between our results and the generalization curves. To demonstrate this, we conducted a control analysis where we corrected our explicit measures according to the Day et al. generalization curve (see Appendix 4 in revised manuscript). Our results are shown in Figure . As predicted, correcting explicit measures yielded a poorer match between the data and generalization hypothesis (see discrepancy below each inset).

These analyses show that the alternative hypothesis of plan-based generalization is inconsistent with our measured data, though it could make minor contributions to the relationships we report between implicit and explicit learning (analysis shown on Section 2.2., Lines 442-468 in revised manuscript).

These analyses have relied on an empirical comparison between our data and past generalization studies. But we can also demonstrate mathematically that the implicit and explicit learning patterns we measured are inconsistent with the generalization hypothesis. First, recall that the competition equation states that steady-state implicit learning varies inversely with explicit strategy according to (Equation 4):xiss=bi1−ai+bi(r−xess)

We can condense this equation by defining an implicit proportionality constant, *p*:xiss=pr−pxess, where p=bi1−ai+bi

These new data provide a way to directly compare the competition model to an SPE generalization model. The key prediction, again, is to test how the relationship between implicit and explicit learning varies with the rotation’s magnitude. To do this, we calculated the implicit proportionality constant (*p* above) in the competition model and the SPE generalization model, that best matched the measured implicit-explicit relationship. We calculated this value in the 60° learning block alone, holding out all other rotation sizes. We then used this gain to predict the implicit-explicit relationship across the held-out rotation sizes. The competition model is shown in the solid black line in Figure 5D. The Day et al. (2016) generalization model is shown in the gray line. The total prediction error across the held-out 15°, 30°, and 45° rotation periods was two times larger in the SPE generalization model (Figure 4E, repeated-measures ANOVA, F(2,35)=38.7, p<0.001, *η_p_^2^*=0.689; post-hoc comparisons all p<0.001). Issues with the SPE generalization model were not caused by misestimating the generalization gain, *m*. We fit the SPE generalization model again this time allowing both π and *m* to vary to best capture behavior in the 60° period (Figure 4D, SPE gen. best B4). This optimized model generalized very poorly to the held-out data, yielding a prediction error three times larger than the competition model (Figure45D, SPE gen best B4).

To understand why the competition model yielded superior predictions, we fit separate linear regressions to the implicit-explicit relationship measured during each rotation period. The regression slopes and 95% CIs are shown in Figure 4F (data). Remarkably, the measured implicit-explicit slope appeared to be constant across all rotation magnitudes in agreement with the competition theory (Figure 4H, competition). These measurements sharply contrasted with the SPE generalization model, which predicted that the regression slope would decline as the rotation magnitude decreased.

In summary, data in Experiment 1 were poorly described by an SPE learning model with aim-based generalization. While generalization may add to the correlations between implicit and explicit learning, its contribution is small relative to the competition theory. New analyses are described in Section 2.2 in the revised paper.

These analyses all considered how a negative relationship between implicit and explicit learning could emerge due to generalization (as opposed to competition). In our revised work, we consider an additional mechanism by which this inverse relationship could occur. Suppose that implicit learning is driven solely by SPEs as in earlier models. In this case, implicit learning should be immune to changes in explicit strategy. However, a participant that has a better implicit learning system will require less explicit re-aiming to reach a desired adaptation level. In other words, individuals with large SPE-driven learning may use less explicit strategy relative to those with less SPE-driven implicit learning. Like the competition model, this scenario would also yield a negative relationship between implicit and explicit learning, due to the way explicit strategies respond to variation in the implicit system. In other words, the competition equation supposes that the implicit-explicit relationship arises because the implicit system responds to variability in explicit strategy. An SPE learning model supposes that the implicit-explicit relationship arises because the explicit system responds to variability in implicit learning.

To model the latter possibility, suppose that the implicit system responds solely to an SPE. Total implicit learning is given by (Equation 5), and can be re-written as *x_i_^ss^* = *p_i_r*, where π is a gain that depends on implicit learning properties (*a_i_* and *b_i_*). Thus, implicit learning is centered on *p_i_r*, but varies according to a normal distribution: *x_i_^ss^* = *N*(*p_i_r*,*σ_i_^2^*) where *σ_i_* represents the standard deviation in implicit learning. The target error that remains to drive explicit strategy is equal to the rotation minus implicit adaptation: *r* – *x_i_^ss^*. A negative relationship between implicit and explicit strategy will occur when explicit strategy responds in proportion to this error, *x_e_^ss^* = *p_e_*(*r* – *x_i_^ss^*) where *p_e_* is the explicit system’s response gain.

Overall, this model predicts important pairwise relationships between implicit learning, explicit strategy, and total adaptation (equations provided on Lines 1577-1620 in paper, and also in Figure 4). First, as noted implicit learning and explicit adaptation will show a negative relationship. Second, increases in implicit learning will tend to increase total adaptation. These increases in implicit learning will leave smaller errors to drive explicit learning, resulting in a negative relationship between explicit strategy and total learning. Figures 5A-C illustrate these pairwise relationships in our revised manuscript.

How do these relationships compare to the competition equation, in which implicit learning is driven by target errors? Suppose that participants choose a strategy that scales with the rotation’s size according to *p_e_r* (*p_e_* is the explicit gain) but varies across participants (standard deviation, *σ_e_*). Thus, *x_e_^ss^* is given by a normal distribution *N*(*p_e_r*,*σ_e_^2^*). The target error which drives implicit learning is equal to the rotation minus explicit strategy. The competition theory proposes that the implicit system is driven in proportion to this error according to: *x_i_^ss^* = *p_i_*(*r* – *x_e_^ss^*), where π is an implicit learning gain that depends on implicit learning parameters *a_i_* and *b_i_* (see Equation 4). This model will produce a negative correlation between implicit learning and explicit strategy. People that use a larger strategy will exhibit greater total learning. Simultaneously, increases in strategy will leave smaller errors to drive implicit learning, resulting in a negative relationship between implicit learning and total learning. These 3 predictions are illustrated in Figures 5D-F in our revised manuscript; these relationships correspond to linear equations which are provided on Lines 1577-1620 in paper, and also in Figure 5.

In sum, both an SPE learning model and a target error learning model could exhibit negative participant level correlations between implicit learning and explicit learning (Figures 5A and 5D). However, they make opposing predictions concerning the relationships between total adaptation and each individual learning system. To test these predictions, we considered how total learning was related to implicit and explicit adaptation measured in the No PT Limit group in Experiment 3. These data are now show in Figures 5GandH in our revised work.

Our observations closely agreed with the competition theory; increases in explicit strategy led to increases in total adaptation (Figure 5G, *ρ*=0.84, p<0.001), whereas increases in implicit learning were associated with decreases in total adaptation (Figure 5H, *ρ*=-0.70, p<0.001). We repeated similar analyses across additional data sets which also measured implicit learning via exclusion (i.e., no aiming) trials: (1) the 60° rotation condition (combined across gradual and abrupt groups) in Experiment 1, (2) the 60° rotation groups reported by Maresch et al. (2021), and (3) the 60° rotation group described by Tsay et al. (2021). We obtained the same result as in Experiment 3. Participants exhibited negative correlations between implicit learning and explicit strategy, positive correlations between explicit strategy and total adaptation, and negative correlation between implicit learning and total adaptation. These additional results are reported in Figure 5-Supplement 1 in the revised manuscript.

Thus, these additional studies also matched the competition model’s predictions, but rejected the SPE learning model on two accounts: (1) the relationship between implicit learning and total adaptation was negative, not positive as predicted by the SPE model, and (2) the relationship between explicit learning and total adaptation was positive, not negative as predicted by the SPE model.

Our revised manuscript notes an important caveat with this last analysis. Indeed, the competition model predicts on average that implicit learning will exhibit a negative correlation with total adaptation (across individual participants). However, this prediction assumes that implicit learning properties (retention and error sensitivity, the gain p_i_ above) are identical across participants, an unlikely possibility. Variation in the implicit learning gain (e.g., Participant A has an implicit system that is more sensitive to error) will promote a positive trend between implicit learning and total adaptation, that will weaken the negative correlations generated by implicit-explicit competition. Thus, pairwise correlations between implicit learning, explicit strategy, and total adaptation will vary probabilistically across experiments. While it is critical the reader keep this in mind, a complete investigation of these second-order phenomena is complex and beyond the scope of our work. For this reason, we treat these issues in Appendix 3 in the revised manuscript. Appendix 3 describes several experimental conditions (e.g., variation in explicit strategy across participants, average explicit learning) that will alter the participant-level relationship between implicit and explicit learning.

Summary

In our revised manuscript we compare the competition theory to two alternate models that also yield a negative participant-level relationship between implicit learning and explicit strategy. The first alternate model supposes that implicit learning is driven by SPEs, but is altered by plan-based generalization. The second alternate model supposes that implicit learning is driven by SPEs, but explicit strategies respond to variability in implicit learning across participants.

The generalization model, however, did not match our data:

1. Implicit learning declined 300% more rapidly than predicted by generalization (Figures 4A-C).

2. The implicit-explicit learning relationship did not accurately generalize across rotation sizes in the SPE generalization model (Figures 4DandE).

3. The gain relating implicit and explicit learning remained the same across rotation sizes, rejecting the SPE generalization model, but supporting the competition theory (Figure 4F).

The alternate model where explicit systems respond to subject-to-subject variability in SPE-driven implicit learning did not match our data:

1. We observed a negative relationship between implicit learning and total adaptation in Experiment 3 (Figure 4H) and 3 additional datasets (Figure 4-Supplement 1) as predicted by the competition theory, but in contrast to the SPE learning model.

2. We observed a positive relationship between implicit learning and total adaptation in Experiment 3 (Figure 4G) and 3 additional datasets (Figure 4-Supplement 1) as predicted by the competition theory, but in contrast to the SPE learning model.

Given the importance of these analyses, we devote an entire section (Part 2) of our revised manuscript to comparisons with alternate SPE learning models (Lines 374-504 in revised manuscript). We also describe second-order phenomena in Appendix 3 of our revised manuscript (Lines 2167-2414).

2. The measurement of implicit learning is compromised by not asking participants to stop aiming upon entering the washout period. This happened at three places in the study, once in Experiment 2, once in the re-used data by Fernadez-Ruiz et al., 2011, and once in the re-used data by Mazzoni and Krakauer 2006 (no-instruction group). We would like to know how the conclusion is impacted by the ill-estimated implicit learning.

We agree with the reviewers’ concerns, with one clarification: in Experiment 2 we did instruct participants to stop aiming prior to the no feedback washout period, so this criticism is not relevant to Experiment 2 (note Experiment 2 is now Experiment 3 in the revised work). With that said, this criticism would apply to the Saijo and Gomi (2010) analysis in our original manuscript, which was not noted in the reviewers’ concern. Here, we will address each limitation noted by the reviewers in turn.

Saijo and Gomi (2010)

Let us begin with the Saijo and Gomi (2010) analysis. In our original manuscript, we used conditions tested by Saijo and Gomi to investigate one of the competition theory’s predictions: decreasing explicit strategy for a given rotation size will increase implicit learning. Specifically, we analyzed whether gradual learning suppressed explicit strategy in this study, thus facilitating greater implicit adaptation. While our analysis was consistent with this hypothesis, as noted by the reviewers, there is a limitation; washout trials were used to estimate implicit learning, though participants were not directly instructed to stop aiming. This represents a potential error source. For this reason, in our revised manuscript, we have collected new data to test the competition model’s prediction.

These new data also examine gradual and abrupt rotations. In Experiment 1 (new data in the paper), participants were exposed to a 60° rotation, either abruptly (n=36), or in a stepwise manner (n=37) where the rotation magnitude increased by 15° across 4 distinct learning blocks (Figure 2D). Unlike Saijo and Gomi, implicit learning was measured via exclusion during each learning period by instructing participants to aim directly towards the target. As we hypothesized, in Figure 2F, we now demonstrate that stepwise rotation onset (which yields smaller target errors) muted the explicit response to the rotation (compared to abrupt learning). The competition model predicts that decreases in explicit strategy should facilitate greater implicit adaptation. To test this prediction, we compared implicit learning across the gradual and abrupt groups during the fourth learning block, where both groups experienced the 60° rotation size (Figures 2E and G).

Consistent with our hypothesis, participants in the stepwise condition exhibited a 10° reduction in explicit re-aiming (Figure 2F, two-sample t-test, t(71)=4.97, p<0.001, d=1.16), but a concomitant 80% increase in their implicit recalibration (Figure 2G, data, two-sample t-test, t(71)=6.4, p<0.001, d=1.5). To test whether these changes in implicit learning matched the competition model, we fit the independence Equation (Equation 5) and the competition Equation (Equation 4) to the implicit and explicit reach angles measured in Blocks 14, across the stepwise and abrupt conditions, while holding implicit learning parameters constant. In other words, we asked whether the same parameters (*a_i_* and *b_i_*) could parsimoniously explain the implicit learning patterns measured across all 5 conditions (all 4 stepwise rotation sizes plus the abrupt condition). As expected, the competition model predicted that implicit learning would increase in the stepwise group (Figure 2G, comp., 2-sample t-test, t(71)=4.97, p<0.001), unlike the SPE-only learning model (Figure 2G, indep.).

Thus, in our revised manuscript we present new data that confirm the hypothesis we initially explored in the Saijo and Gomi (2010) dataset: decreasing explicit strategy enhances implicit learning. These new data do not present the same limitation noted by the reviewers. Lastly, it is critical to note that while stepwise participants showed greater implicit learning, their total adaptation was approximately 4° lower than the abrupt group (Figure 2E, right-most gray area (last 20 trials); two-sample t-test, t(71)=3.33, p=0.001, d=0.78). This surprising phenomenon is predicted by the competition equation. When strategies increase in the abrupt rotation group, this will tend to increase total adaptation. However, larger strategies leave smaller errors to drive implicit learning. Hence, greater adaptation will be associated with larger strategies, but less implicit learning. Indeed, the competition model predicted 53.47° total learning in the abrupt group but only 50.42° in the stepwise group. Recall that, we described this paradoxical phenomenon at the individual-participant level in Point 1 above. Note that this pattern was also observed by Saijo and Gomi. Surprisingly, when participants were exposed to a 60° rotation in a stepwise manner, total adaptation dropped over 10°, whereas the aftereffect exhibited during the washout period nearly doubled.

Summary

In our revised paper we have removed this Saijo and Gomi analysis from the main figures, and instead use this as supportive data in Figure 2-Supplement 2, and also in Appendix 2. We now state, “It is interesting to note that these implicit learning patterns are broadly consistent with the observation that gradual rotations improve procedural learning^34,43^, although these earlier studies did not properly tease apart implicit and explicit adaptation (see the Saijo and Gomi analysis described in Appendix 2).”

We have replaced this analysis with our new data in Experiment 1. In these new data, abrupt and gradual learning was compared using the appropriate implicit learning measures. These new data are included in both Figures 1 and 2 in the revised manuscript. Importantly, these new data point to the same conclusions we reached in our initial Saijo and Gomi analysis.

Mazzoni and Krakauer (2006)

The reviewers note that the uninstructed group in Mazzoni and Krakauer was not told to stop aiming during the washout period. We agree with the potential error source. However, it is important to note that these data were not included to directly support the competition model, but as a way to demonstrate its limitations. For example, in Mazzoni and Krakauer (2006) our simple target error model cannot reproduce the measured data. That is, in the instructed group implicit learning continued despite eliminating the primary target error. Thus, in more complicated scenarios where multiple visual landmarks are used during adaptation, the learning rules must be more sophisticated than the simple competition theory. Here we showed that one possibility is that both the primary target and the aiming target drive implicit adaptation via two simultaneous target errors. Our goal by presenting these data (and similar data collected by Taylor and Ivry, 2011) was to emphasize that our work is not meant to support an either-or hypothesis between target error and SPE learning, but rather the more pluralistic conclusion that both error sources contribute to implicit learning in a condition-dependent manner.

In our revised manuscript, we now better appreciate this conclusion and its potential limitation in the following passage: “It is important, however, to note a limitation in these analyses. Our earlier study did not employ the standard conditions used to measure implicit aftereffects: i.e., instructing participants to aim directly at the target, and also removing any visual feedback. Thus, the proposed dual-error model relies on the assumption that differences in washout were primarily related to the implicit system. These assumptions need to be tested more completely in future experiments.

In summary, the conditions tested by Mazzoni and Krakauer show that the simplistic idea that adaptation is driven by only one target error, or only one SPE, cannot be true in general^54^. We propose a new hypothesis that when people move a cursor to one visual target, while aiming at another visual target, each target may partly contribute to implicit learning. When these two error sources conflict with one another, the implicit learning system may exhibit an attenuation in total adaptation. Thus, implicit learning modules may compete with one another when presented with opposing errors.”

Fernandez-Ruiz et al. (2011)

As noted by the reviewers, our original manuscript also reported data from a study by Fernandez-Ruiz et al. (2011). Here, we showed that participants in this study who increased their preparation time more upon rotation onset tended to exhibit a smaller aftereffect. We used these data to test the idea that when a participant uses a larger strategy, they inadvertently suppress their own implicit learning. However, we agree with the reviewers that this analysis is problematic because participants were not instructed to stop aiming in this earlier study. Thus, we have removed these data in the revised manuscript. This change has little to no effect on the manuscript, given that individual-level correlations between implicit and explicit learning are tested in Experiments 1, 2, and 3 in the revised paper (Figures 3-5), and are also reported from earlier works (Maresch et al., 2021; Tsay et al., 2021) in Figure 4-Supplements 1GandI, where implicit learning was more appropriately measured via exclusion. Note that all 5 studies support the same ideas suggested by the Fernandez-Ruiz et al. study: increases in explicit strategy suppress implicit learning.

3. All the new experiments limit preparation time (PT) to eliminate explicit learning. In fact, the results are based on the assumption that shortened PT leads to implicit-only learning (e.g., Figure 4). Is the manipulation effective, as claimed? This paradigm needs to be validated using independent measures, e.g., by direct measurement of implicit and/or explicit strategy.

We agree that it is important to test how effectively this condition limits explicit strategy use. Therefore, we have performed two additional control studies to measure explicit strategy in the limited PT condition. First, we added a limited preparation time (Limit PT) group to our laptop-based study in Experiment 3 (Experiment 2 in original manuscript). In Experiment 3, participants in the Limit PT group (n=21) adapted to a 30° rotation but under a limited preparation time condition. As for the Limit PT group in Experiment 2 (Experiment 1 in original paper), we imposed a bound on reaction time to suppress movement preparation time. However, unlike Experiment 2, once the rotation ended, participants were told to stop re-aiming. This permitted us to examine whether limiting preparation time suppressed explicit strategy as intended. Our analysis of these new data is shown in Figures 3K-Q.

In Experiment 3, we now compare two groups: one where participants had no preparation time limit (No PT Limit, Figures 3H-J) and one where an upper bound was placed on preparation time (Limit PT, Figures 3K-M). We measured early and late implicit learning measures over a 20-cycle no feedback and no aiming period at the end of the experiment. The voluntary decrease in reach angle over this no aiming period revealed each participant’s explicit strategy (Figure 3N). When no reaction time limit was imposed (No PT Limit), reaiming totaled approximately 11.86° (Figure 3N, E3, black), and did not differ statistically across Experiments 2 and 3 (t(42)=0.50, p=0.621). Recall that Experiment 2 tested participants using a robotic manipulandum and Experiment 3 tested participants in a similar laptop-based paradigm. As in earlier reports, limiting reaction time dramatically suppressed explicit strategy, yielding only 2.09° of re-aiming (Figure 3N, E3, red). Therefore, our new control dataset suggests that our limited reaction time technique was highly effective at suppressing explicit strategy as initially claimed in our original manuscript.

Since we have now demonstrated that limiting PT is effective at suppressing explicit strategy, we use our new Limit PT group in Experiment 3, to corroborate several key competition model predictions. First, consistent with the competition model, suppressing explicit strategy increased implicit learning by approximately 40% (Figure 3O, No PT Limit vs. Limit PT, two-sample t-test, t(54)=3.56, p<0.001, d=0.98). Second, we used the Limit PT group’s behavior in Experiment 3 to estimate implicit learning parameters (*a_i_* and *b_i_*) as we did in Experiment 1 (now Experiment 2) in our original manuscript. Briefly, the implicit retention factor was estimated during the no feedback and no aiming period, and the implicit error sensitivity was estimated via trial-to-trial changes in reach angle during the rotation period. We used these parameters to specify the unknown implicit learning gain in the competition model (*p_i_* described in Point 1 above). Limit PT group behavior predicted that implicit and explicit learning should be related by the following competition model: *x_i_* = 0.63(30 – *x_e_*). As in Experiment 2, we observed a striking correspondence between this model prediction (Figure 3Q, bottom, model) and the actual implicit-explicit relationship measured across participants in the No PT Limit group (Figure 3Q, bottom, points). The slope and bias predicted by Equation 4 (-0.67 and 20.2°, respectively) differed from the measured linear regression by less than 8% (Figure 3Q, bottom brown line, R^2^=0.78; slope is -0.63 with 95% CI [-0.74, -0.51] and intercept is 19.7° with 95% CI [18.2°, 21.3°]).

With that said, our new data alone suggest that while explicit strategies are strongly suppressed by limiting preparation time, they are not entirely eliminated; when we limited preparation time in Experiment 3, we observed that participants still exhibited a small decrease (2.09°) in reach angle when we instructed them to aim their hand straight to the target (Figure 3L, no aiming; Figure 3N, E3, red). This ‘cached’ explicit strategy, while small, may have contributed to the 8° reach angle change measured early during the second rotation in our savings experiment (Experiment 4, Figure 8C in revised paper).

For this reason, we consider another important phenomenon in our revised manuscript: time-dependent decay in implicit learning. That is, the 2° decrease in reach angle we observed when participants were told to stop aiming in the PT Limit group in Experiment 3 may be due to time-based decay in implicit learning over the 30 second instruction period, as opposed to a voluntary reduction in strategy. To test this possibility, we tested another limited preparation group (n=12, Figure 8-Supplement 1A, decay-only, black). But this time, participants were instructed that the experiment’s disturbance was still on, and that they should continue to move the ‘imagined’ cursor through the target during the terminal no feedback period. Still, reach angles decreased by approximately 2.1° (Figure 8-Supplement 1B, black). Indeed, we detected no statistically significant difference between the change in reach angle in this decay-only condition, and the Limit PT group in Experiment 3 (Figure 8-Supplement 1B; two-sample t-test, t(31)=0.016, p=0.987).

This control experiment suggested that residual ‘explicit strategies’ we measured in the Limit PT condition, were in actuality caused by time-dependent decay in implicit learning. Thus, our Limit PT protocol appears to almost entirely eliminate explicit strategy. This additional analysis lends further credence to the hypothesis that savings in Experiment 4, was primarily due to changes in the implicit system rather than cached explicit strategies (and also that the impairment in learning observed in Experiment 5, was driven by the implicit system).

Summary

We collected additional experimental conditions in Experiment 3 in the revised manuscript (a Limit PT group and a decay-only group). Data in the Limit PT condition suggested that explicit strategies are suppressed to approximately 2° by our preparation time limit (compared to about 12° under normal conditions). Data in the decay-only group suggest that this 2° change in reach angle was not due to a cached explicit strategy but rather time-dependent implicit decay. Together, these conditions demonstrate that limiting reaction time in our experiments prevents the caching of explicit strategies; our limited preparation time measures do indeed reflect implicit learning.

4. Related to point 3, it is noted that the aiming report produced a different implicit learning estimate than a direct probe in Exp1 (Figure S5E). However, Figure 3 only reported Exp1 data with the probe estimate. What would the data look like with report-estimated implicit learning? Will it show the same negative correlation results with good model matches (Figure 3H)?

We ended Experiment 2 (Experiment 1 in the original manuscript) by asking participants to verbally report where they remembered aiming in order to hit the target. While many studies use similar reports to tease apart implicit and explicit strategy (e.g., McDougle et al., 2015), our probe had important limitations. Normally, participants are probed at various points throughout adaptation by asking them to report where they plan to aim on a given trial, and then allowing them to execute their reaching movement (thus they are able to continuously evaluate and familiarize themselves with their aiming strategy over time). However, we did not use this reporting paradigm given recent evidence (Maresch et al., 2020) that reporting explicit strategies increases explicit strategy use, which we expected would have the undesirable consequence of suppressing implicit learning (given the competition theory). Thus, in Experiment 2, reporting was measured only once; participants tried to recall where they were aiming throughout the experiment.

This methodology may have reduced the reliability in this measure of explicit adaptation. As stated in our Methods, we observed that participants were prone to incorrectly reporting their aiming direction, with several reported angles in the perturbation’s direction, rather than the compensatory direction. In fact, 25% of participant responses were reported in the incorrect direction. Thus, as outlined in our Methods, we chose to take the absolute value of these measures, given recent evidence that strategies are prone to sign errors (McDougle and Taylor, 2019). That is, we assumed that participants remembered their aiming magnitude, but misreported the orientation. In sum, we suspect that our report-based strategy measures are prone to inaccuracies and opted to interpret them cautiously and sparingly in our original manuscript.

Nevertheless, in the revised manuscript, we now also report the individual-level relationships between report-based implicit learning and report-based explicit strategy in Experiment 2 (previously Experiment 1). These are now illustrated in Figure 3-Supplement 2C. While reported explicit strategies were on average greater than our probe-based measure, and report-based implicit learning was smaller than our probe-based measure (Figures 3-Supplement 2AandB; paired t-test, t(8)=2.59, p=0.032, d=0.7), the report-based measures exhibited a strong correlation which aligned with competition model’s prediction (Figure 3-Supplement 2C; R^2^=0.95; slope is -0.93 with 95% CI [-1.11, -0.75] and intercept is 25.51° with 95% CI [22.69°, 28.34°]).

In summary, we have now added report-based implicit learning and explicit learning measurements to our revised manuscript. We show that these measures also exhibit a strong negative correlation with good agreement to the competition model. However, we still feel that it is important to question the reliability of these report-based measures (given that they were collected only once at the end of the experiment, when no longer in a ‘reaching context’). Our new analysis is described on Lines 319-327 in the paper.

5. Figure 3H is one of the rare pieces of direct evidence to support the model, but we find the estimated slope parameter is based on a B value of 0.35, which is unusually high. How is it obtained? Why is it larger than most previous studies have suggested? In fact, this value is even larger than the learning rate estimated in Exp 3 (Figure 6C). Is it related to the small sample size (see below)?

We can see where there may be confusion concerning how we estimated implicit error sensitivity, *b_i_* in Experiment 1 (now Experiment 2 in the revised manuscript). Note that we use the same procedures to estimate error sensitivity in Experiment 3 (newly collected Limit PT group in laptop-based study). We can explain why the error sensitivity we estimate is appropriate. In Figure 3H (now Figure 3G), as well as throughout our paper, we attempt to understand asymptotic behavior. Therefore, we require error sensitivity during the asymptotic learning phase. In other words, steady-state adaptation will depend on asymptotic implicit error sensitivity, not initial error sensitivity. This subtlety is important because as we and others have shown, error sensitivity increases as errors get smaller (e.g., Marko et al., 2012; Kim et al., 2018, Albert et al., 2021). This was indeed the case in the Limit PT condition in Experiment 2 (previously Experiment 1), as shown in Figure 3-Supplement 1A in the revised manuscript.

To calculate error sensitivity in each cycle, we used its empirical definition (Equation 14 in revised manuscript). To determine the terminal error sensitivity during the adaptation period, we averaged error sensitivity over the last 5 rotation cycles (last 5 cycles in S3-1A). This error sensitivity was approximately 0.346, as shown by the horizontal dashed line in S3-1A. How reasonable is this value? The corresponding terminal error in the Limit PT condition in Experiment 2 (previously Experiment 1) was approximately 7.5° as shown in Figure 3-Supplement 1B (blue horizontal line). Recently Kim et al. (2018) calculated implicit error sensitivity as a function of error size; we show these curves in Figure 3-Supplement 1C in the revised manuscript. The blue line shows the terminal error in the Limit PT condition. As we can see, these earlier works suggested that for this particular error size, implicit error sensitivity varies somewhere between 0.25 and 0.35. Thus, our reported value of 0.346, does indeed appear reasonable. We thank the reviewers for noting this potential point of confusion and now reference our analysis on Lines 304-306 in the revised manuscript. We repeat this analysis for Experiment 3 (Limit PT group in newly added laptop-based study) in Figure 3-Supplements 1D-F.

Now we should address the inquiry about why this value is larger than the error sensitivity estimated in our Haith et al. (2015) state-space model. This question is actually common to most studies (including our own) in the literature that use state-space models to interrogate sensorimotor adaptation (e.g., Smith et al., 2006; Mawase et al., 2014; Galea et al. 2015; McDougle et al., 2015; Coltman et al., 2019, etc.). This issue arises because in our Haith et al. (2015) model, error sensitivity remains constant within a learning block. Clearly, we know this is not the case: within a single perturbation exposure not only does error sensitivity vary with error size (Figure 3-Supplement 1A, Marko et al., 2012; Kim et al., 2018, Albert et al., 2021), but it also changes with one’s error history (Herzfeld et al., 2014; Gonzalez-Castro and Hadjiosif et al., 2014; Albert et al., 2021). Our simple model does not describe these within-block changes, but instead permits savings by supposing that error sensitivity can differ across Blocks 1 and 2.

Even with this simplification, the model possesses 6 free parameters: *b_i_* in Block 1, *b_i_* in Block 2, *b_e_* in Block 1, *b_e_* in Block 2, *a_i_*, and *a_e_*. If we were to allow error sensitivity to change within each block, the simplest choice would require at least 2-4 free parameters: rate parameters describing how *b_i_* and *b_e_* change within each block. Thus, a simple model of within-block and between-block changes in error sensitivity, would likely require a state-space model with at least 8-10 free parameters. As these models become more complex, the likelihood that they overfit the data increases significantly.

Thus, the common choice is to simplify this complexity, by not permitting *b* to change within a block. This assumption will cause the model to identify an error sensitivity that is intermediate between its initial value in the block, and its terminal value within a block. Of course, this is an abstraction of reality because *b* is not constant. Nevertheless, this common assumption still has value, as it provides a means of testing if some “average” error sensitivity differs across two different perturbation exposures. However, because error sensitivity increases over the block, this “average” error sensitivity parameter will be smaller than the terminal error sensitivity reached later in the block. Thus, this is the reason why when we estimate *b* late during the Limit PT condition, we obtained a sensitivity near 35%, but when we fit a model to the entire exposure period in Haith et al. (2015), the average error sensitivity identified by the model falls far short of this value.

While using a constant *b_i_* value in our Haith et al. (2015) model represents a potential limitation, a more sophisticated state-space model is not needed for our purposes. Here, we compare an SPE model and a TE model, in order to demonstrate the possibility that some “average” implicit error sensitivity is greater during the second rotation. Our main purpose is to raise an alternate hypothesis to our original conclusion in Haith et al., and recent evidence that implicit learning processes do not exhibit savings. Also note that our choice to fix *b_i_* to a constant value within a block is consistent with past studies that have used statespace models to document changes in error sensitivity across perturbation exposures (e.g., Lerner and Albert et al., 2020; Coltman et al., 2019; Mawase et al., 2014).

In summary, we hope that this discussion helps to clarify the error sensitivity estimates in our work. In our revised manuscript we have now added Figure 3-Supplement 1 to show that our asymptotic error sensitivity estimates are consistent with past literature.

6a. The direct evidence supporting a competition of implicit and explicit learning is still weak.

First, we humbly submit that our work provides considerable support for a competition between implicit and explicit learning. Given the large amount of data added to the revised manuscript it seems appropriate to begin with a brief summary of the analyses that support the competition theory over the standard SPEonly learning hypothesis.

A. Saturated learning with increasing rotation size in some experiments, yet an increasing response in others (Figure 1 in revised manuscript), which is inconsistent with generalized SPE-learning.

B. Quadratic responses to rotation size which are inconsistent with generalized SPE-learning (new analysis in Figure 1 in revised manuscript, described more below)

C. Decreases in implicit adaptation due to coaching, with a simultaneous insensitivity to rotation size (Figure 2 in revised manuscript), whose co-occurrence is unexplained by generalized SPE-learning.

D. Increases in implicit learning driven by gradual perturbation onsets (Figure 1 in revised paper) whose subject-to-subject properties are inconsistent with generalization of SPE learning (Figure 5 in revised paper along with several supplements).

E. Increases in implicit learning due to suppressed explicit strategy, consistent with the competition model (Figure 3O in revised manuscript).

F. Negative subject-level correlations between implicit and explicit learning, with properties that are quantitatively predicted by the competition model (Figures 3GandQ).

G. Negative correlations between implicit-total learning that support the competition theory and disprove an SPE-only model (Figure 4H).

H. Positive correlations between explicit-total totaling that support the competition theory and disprove an SPE-only model (Figure 4G).

I. Negative implicit-explicit correlations (Figures 3G, P, and Q) that we have now detected across three additional studies (Figure 4-Supplements 1G-I), whose properties are inconsistent with a generalized SPE learning model (Figure 5).

The totality of this evidence cannot be reconciled by the standard model where implicit learning is solely driven by SPEs. With that said, we wish to restate that despite these points, our paper is intended to show that implicit target error learning exists in addition to, not instead of, the implicit sensory prediction error system. We hope that this pluralistic intention is better emphasized in our revised paper (see for example our text on Lines 787-790 and 1036-1103).

6b. Most data presented here are about the steady-state learning amplitude; the authors suggest that steady-state implicit learning is proportional to the size of the rotation (Equation5). But this is directly contradictory to the experimental finding that implicit learning saturates quickly with rotation size (Morehead et al., 2017; Kim et al., 2018). This prominent finding has been dismissed by the proposed model.

These concerns are critical. We agree with the reviewers that Morehead et al., 2017 and Kim et al., 2018 suggest that implicit learning saturates quickly with rotation size. With that said, these studies both use invariant error-clamp perturbations, not standard visuomotor rotations. We are not sure that the implicit properties observed in an invariant error context apply to the conditions we consider in our manuscript. In our revised manuscript, we consider variations in steady-state implicit learning across two new data sets: (1) stepwise adaptation in Experiment 1 and (2) non-monotonic implicit responses in Tsay et al., 2021. As we will demonstrate below, our data and re-analysis strongly indicate that the implicit system does not always saturate with changes in rotation size. Rather, the implicit system exhibits a complicated response to both rotation size and explicit strategies, which together yield three implicit learning phenotypes:

1. Saturation in steady-state implicit learning despite increasing rotation size (analysis reported in original paper using Neville and Cressman, 2018).

2. Scaling in steady-state implicit learning with increasing rotation size (new analysis using stepwise learning in Experiment 1).

3. Non-monotonic (quadratic) steady-state implicit behavior due to increases in rotation magnitude (new analysis added which uses data from Tsay et al., 2021).

We explore these three phenotypes in Figure 1 in the revised paper. Below, we provide an overview of these phenotypes, and make references to our revised text and Figure 1 where appropriate.

To begin with, we discuss cases where implicit learning appears to remain constant over large changes in rotation magnitude. This analysis is the same as the Neville and Cressman (2018) analysis described in our original manuscript (though the way we present certain aspects has been updated). Recall that Neville and Cressman (2018) examined how both implicit and explicit systems responded to changes in rotation size.

Participants adapted to a 20° (n=11), 40° (n=10), or 60° (n=10) visuomotor rotation (Figure 1C). Figures 1D and 1E show exclusion-based measures for implicit and explicit learning across all 3 rotations. Unsurprisingly, strategies increased proportionally with the rotation’s size (Figure 1E). On the other hand, implicit learning was insensitive to the rotation magnitude; it reached only 10° and remained constant despite tripling the perturbation’s size (Figure 1D).

As the reviewers note, on its face, this implicit saturation appears in line with an upper bound on implicit learning, as in Morehead et al., 2017 and Kim et al., 2018 (with the exception that here implicit learning was limited to 10° which falls short of the 15-25° limits observed in the invariant error-clamp paradigm).

Here, however, we propose another way that implicit learning might exhibit this saturation phenotype: not due to an upper bound on its compensation, but instead, competition.

In the competition model, the implicit system is driven by the residual error between the rotation and explicit strategy (*r* – *x_e_^ss^* in Equation 4). As a result, this model predicts that when an increase in rotation magnitude is matched by an equal change in explicit strategy (Figure 1-Supplement 1A, same), the implicit learning system’s driving force will remain constant (Figure 1-Supplement 1B, same). This constant driving input leads to a phenotype where implicit learning appears to “saturate” with increases in rotation size (Figure 1-Supplement 1C, same).

To investigate this possibility, we measured the rate at which explicit strategy increased with rotation size in the measured behavior. Rapid increases in explicit strategy (Figure 1E) limited changes in the competitive driving force; whereas the rotation increased by 40°, the driving force changed by less than 2.5° (Figure 1F). Thus, the competition model (Figure 1G, competition) yielded a close correspondence to the measured data (Figure 1G, data); when the driving input remains the same across rotation sizes, implicit learning will remain the same.

This model suggested that the implicit system’s response saturated not due to an intrinsic upper limit in implicit learning, but due to a constant driving input. The key prediction is that the implicit system should be ‘released’ from this saturation phenotype by weakening the explicit system’s response to the rotation. That is, when a change in rotation size is accompanied by a smaller change in explicit strategy (Figure 1-Supplement 1A, slower), the competitive driving input to the implicit system should increase (Figure 1Supplement 1B, slower). In the revised manuscript, we have added a new dataset (n=37) to test this idea. In this new experiment, we probed implicit and explicit learning across similar rotation sizes (15-60°) but using a stepwise perturbation. In Experiment 1, participants (n=37) adapted to a stepwise perturbation which started at 15°, but increased to 60° in 15° increments (Figure 1H). Towards the end of each learning block, we assessed implicit and explicit adaptation by instructing participants to aim directly to the target (Figure 1H, implicit learning shown in black shaded regions).

Stepwise rotation onset muted the explicit system’s gain, relative to the abrupt rotation conditions used by Neville and Cressman (2018); explicit strategies increased with a 94.9% gain (change in strategy divided by change in rotation) across the abrupt groups in Figure 1E, but only a 55.5% gain in the stepwise condition shown in Figure 1J. This suppression in the explicit response caused the competition model’s implicit driving input (Figure 1K) to increase with rotation magnitude. Remarkably, this increasing driving input predicted a “scaling” phenotype in steady-state learning (Figure 1L, competition) that precisely matched the measured implicit response (Figures 1I; 1L, data).

Thus, steady-state implicit learning can exhibit a saturation phenotype (Figure 1G) and a scaling phenotype (Figure 1L), as predicted by the competition model. But recent work in Tsay et al., 2021, suggests a third steady-state implicit phenotype: non-monotonicity. We have now added an analysis of these data to the revised manuscript. In this study, the authors probed a wider range in rotation size, 15° to 90° (Figure 1M). A terminal no aiming period was used to measure steady-state implicit learning levels (n=25/group). Curiously, whereas implicit learning increased across the 15° and 30° rotations, it remained the same in the 60° rotation group, and then decreased in the 90° rotation group (Figure 1N).

To determine whether this non-monotonicity could be captured by the competition model, we considered again how explicit re-aiming increased with rotation size (Figure 1O). We observed an intriguing pattern. When the rotation increased from 15° to 30°, explicit strategy responded with a very low gain (4.5%, change in strategy divided by change in rotation). An increase in rotation size to 60° was associated with a medium-sized gain (80.1%). The last increase to 90° caused a marked change in the explicit system: a 53.3° increase in explicit strategy (177.7% gain). Thus, explicit strategy increased more than the rotation had. Critically, this condition produces a decrease in the implicit driving input in the competition model (Figure 1-Supplement 1, faster). Overall, this large modulation in explicit learning gain (4.5% to 80.1% to 177.7%) yielded non-monotonic behavior in the implicit driving input (Figure 1P) which increased between 15° and 30°, changed little between 30° and 60°, and then dropped between 60° and 90°. As a result, the competition model (Figure 1Q, competition) exhibited a non-monotonic envelope, which closely tracked the measured implicit steady-states (Figure 1Q, data).

Together, these studies demonstrate that the implicit system exhibits at least 3 steady-state phenomena: saturation, scaling, and non-monotonicity. In abrupt conditions, the implicit response saturated. In a more gradual condition, the implicit response scaled. When the rotation increased to an orthogonal extreme (90° rotation), implicit learning showed a non-monotonic decrease. A model where implicit learning is driven solely by SPEs could only produce the scaling phenotype (Figures 1G, 1L, and 1Q, independence, not shown here but provided in the revised manuscript). The competition model, however, predicted that the implicit system should respond not only to the rotation, but also changes in explicit strategy. This additional dimension yielded a match to the data across all 3 implicit learning phenotypes (Figures 1G, 1L, and 1Q, competition). Thus, the implicit system appeared to compete with explicit strategy, in their shared response to target error.

In sum, we now add two additional datasets to the revised manuscript: data collected using a stepwise rotation in Experiment 1, and data recently collected by Tsay et al., 2021. Collectively, these new data show that steady-state implicit learning is complex, exhibiting at least three contrasting phenotypes: saturation, scaling, and non-monotonicity. Remarkably, the competition theory provides a way to account for each of these patterns. Thus, the competition model does accurately describe how implicit learning varies due to both changes in rotation size and explicit strategy, at least in standard rotation paradigms. However, these rules do not explain implicit behavior in invariant error-clamp paradigms. In the revised manuscript, we delve further into a comparison between standard rotation learning and invariant error learning in our Discussion (Lines 911-941). The relevant passage is provided below:

“Although the implicit system is altered in many experimental conditions, one commonly observed phenomenon is its invariant response to changes in rotation size^3,31,35,36,40^. For example, in the Neville and Cressman^31^ dataset examined in Figure 1, total implicit learning remained constant despite tripling the rotation’s magnitude. While this saturation in implicit learning is sometimes interpreted as a restriction in implicit adaptability, this rotation-insensitivity may have another cause entirely: competition. That is, when rotations increase in magnitude, rapid scaling in the explicit response may prevent increases in total implicit adaptation. Critically, in the competition theory, implicit learning is driven not by the rotation, but by the residual error that remains between the rotation and explicit strategy. Thus, when we used gradual rotations to reduce explicit adaptation (Experiment 1), prior invariance in the implicit response was lifted: as the rotation increased, so too did implicit learning^37^ (Figure 1I). The competition theory could readily describe these two implicit learning phenotypes: saturation and scaling (Figures 1GandL). Furthermore, it also provided insight as to why implicit learning can even exhibit a non-monotonic response, as in Tsay et al. (2021)^36^. All in all, our data suggest that implicit insensitivity to rotation size is not due to a limitation in implicit learning, but rather a suppression created by competition with explicit strategy.

With that said, this competitive saturation in implicit adaptation should not be conflated with the upper limits in implicit adaptation that have been measured in response to invariant errors^3,11,40^. In this latter condition, implicit adaptation reaches a ceiling whose value varies somewhere between 15 degrees^3^ and 25 degrees^40^ across studies. In these experiments, participants adapt to an error that remains constant and is not coupled to the reach angle (thus, the competition theory cannot apply). While the state-space model naturally predicts that total adaptation can exceed the error size which drives learning in this errorclamp condition (as is observed in response to small error-clamps), it cannot explain why asymptotic learning is insensitive to the error’s magnitude. One idea is that proprioceptive signals^40,83^ may eventually outweigh the irrelevant visual errors in the clamped-error condition, thus prematurely halting adaptation. Another possibility is that implicit learning obeys the state-space competitive model described here, but only up until a ceiling that limits total possible implicit corrections. Indeed, in Experiment 1, we produced scaling in the implicit system in response to rotation size, but never evoked more than about 22° of implicit learning. However, when we used similar gradual conditions in the past to probe implicit learning^37^, we observed about 32° implicit learning in response to a 70° rotation. Further, in a recent study by Maresch and colleagues^47^, where strategies were probed only intermittently, implicit learning reached nearly 45°. Thus, there remains much work to be done to better understand variations in implicit learning across errorclamp conditions, and standard rotation conditions.”

6c. On the other hand, one classical finding for conventional VMR is that explicit learning would gradually decrease after its initial spike, while implicit learning would gradually increase. The competition model appears unable to account for this stereotypical interaction. How can we account for this simple pattern with the new model or its possible extensions?

As the reviewers note, some studies have observed an initial spike in explicit strategy which then gradually declines over adaptation (e.g., McDougle et al., 2015; some groups in Yin and Wei, 2020) – though it should be noted that in other cases, this does not appear to occur (e.g., most cases in Morehead et al., 2015; multi-day training in Wilterson et al., 2021). While we opted not to investigate these dynamic phenomena in our paper, it should be noted that the competition model can produce this explicit phenotype.

In the simplest case where adaptation depends solely on a target error shared between the implicit and explicit systems, this gives rise to the competitive state-space system given by:xi(n+1)=aixi(n)+bietarget(n)xe(n+1)=aexe(n)+beetarget(n)

This is the competition model we use in our Haith et al. (2015) simulations. In this model, the rise and fall in explicit strategy can occur due to the way that both systems interact through *e*_target_ in these equations. An example can even be observed in our manuscript, in our Haith et al. (2015) analysis (Figure 6D).

Note how explicit adaptation predicted by the competition model (magenta curves) rises during the first 20 rotation trials on both Day 1 and Day 2, and then declines over the rest of adaptation. Thus, clearly the competition model naturally describes such waxing and waning in the explicit response; high explicit error sensitivity initially causes increases in explicit learning, but then as the implicit process develops, errors are siphoned away from the explicit system causing it to gradually decline.

Altogether, these simulations illustrate that the competition model can indeed account for the explicit and implicit phenotypes described by the reviewers. With that said, there is another potential culprit in the rise-and-fall explicit learning phenotype. In some cases where aiming landmarks are provided during adaptation, target errors can be eliminated, but implicit learning persists with a concomitant decrease in explicit strategy (e.g., Taylor et al., 2014; McDougle et al., 2011). We suspect in these cases that persistent implicit learning is due to errors between the cursor and the aiming landmarks, as demonstrated by Taylor and Ivry (2011). Thus, in order to maintain the cursor on the target without overcompensating for the rotation, explicit strategies must decline.

In sum, in the revised manuscript we note that the competition model can produce the implicit-explicit learning phenotype noted by the reviewers: rising-and-falling explicit strategy accompanied by persistent implicit learning, as demonstrated in Figure 5D. However, we also recognize that in aiming landmark studies, this behavior can be driven by a second error (an SPE and/or target error between the cursor and aiming landmark). In our revised paper, we have added a passage concerning these matters:

“…For example, early during learning, it is common that explicit strategies increase, peak, and then decline. That is, when errors are initially large, strategies increase rapidly. But as implicit learning builds, the explicit system’s response can decline in a compensatory manner^1,9,12^. This dynamic phenomenon can also occur in the competition theory, where both implicit and explicit systems respond to target error (Figure 6D). But in many cases, a second error source may drive this behavioral phenotype. That is, in cases with aiming landmarks ^1,9,12^, errors between the cursor and primary target can be eliminated, but implicit learning persists. This implicit learning is likely driven by the SPEs and target errors that remain between the cursor and the aiming landmark, as in Taylor and Ivry (2011)^9^. This persistent implicit adaptation must be counteracted by decreasing explicit strategy to avoid overcompensation. In sum, competition between implicit learning and explicit strategy is complex. Both systems can respond to one another in ways that change with experimental conditions.”

7. General comments about the data:Experiments 3 and 4 (Figures 6 and 7) do not contribute to the theorization of competition between implicit and explicit learning. The authors appear to use them to show that implicit learning properties (error sensitivity) can be modified, but then this conclusion critically depends on the assumption that the limit-PT paradigm produces implicit-only learning, which is yet to be validated.Figure 4 is not a piece of supporting evidence for the competition model since those obtained parameters are based on assumptions that the competition model holds and that the limit-PT condition "only" has implicit learning.

We agree with these points. Yes, we do not intend for Experiments 3 and 4 (now Experiments 4 and 5) to directly contribute to the competition theory’s experimental evidence. We have made changes to our revised manuscript to make this clearer. First, our Results are now divided in 4 sections. Sections 1 and 2 present evidence that supports the competition theory. The savings and interference studies are included in a separate section (Part 3). We expect that these analyses will be important to the reader, as they provide new ideas about how the implicit system may exhibit changes that are masked by explicit strategy.

Secondly, we have now added new analysis to the Haith et al. (2015) dataset, so that it better recognizes and contrasts competition and independence model predictions. In the original Figure 4, we solely fit the competition model to the Haith et al. dataset and highlighted the implicit and explicit error sensitivities predicted by the model. In the revised manuscript, we now also fit the independence model to the Haith et al. dataset and include its error sensitivity estimates in Figure 6 (previously Figure 4) as well. We hope that this inclusion better illustrates that these model fits are not meant as evidence for one model or the other, but rather two contrasting interpretations of the same data.

In the revised paper we explain the competition model predicts that both the implicit system and explicit system exhibited a statistically significant error sensitivity increase (Figure 6D, right; two-way rm-ANOVA, within-subject effect of exposure number, F(1,13)=10.14, p=0.007, η_p_^2^=0.438; within-subject effect of preparation time, F(1,13)=0.051, p=0.824, η_p_^2^=0.004; exposure by preparation interaction, F(1,13)=1.24, p=0.285). The independence model suggested only the explicit system exhibited a statistically significant increase in error sensitivity (Figure 5E; 2-way rm-ANOVA, learning process by exposure number interaction, F(1,13)=7.016, p=0.02; significant interaction followed by 1-way rm-ANOVA across exposures: explicit system with F(1,13)=9.518, p=0.009, η_p_^2^=0.423; implicit system with F(1,13)=2.328, p=0.151, η_p_^2^=0.152).

In the revised manuscript we conclude these results with: “In summary, when we reanalyzed our earlier data, the competition and independence theories suggested that our data could be explained by two contrasting hypothetical outcomes. If we assumed that implicit and explicit systems were independent, then only explicit learning contributed to savings, as we concluded in our original report. However, if we assumed that the implicit and explicit systems learned from the same error (competition model), then both implicit and explicit systems contributed to savings. Which interpretation is more parsimonious with measured behavior?”

We hope that these additions and revisions to the manuscript better illustrate that our analysis is not meant to support one model over the other, but rather to illustrate their contrasting implications about the implicit system.

Finally, the reviewers also note in their concern, that our analyses in Figures 7, 8, and 9 (previously Figures 4, 6, and 7) depend on the assumption that limited preparation time trials isolate implicit learning. We responded to similar concerns in Point 3 above. We will reiterate this discussion here. In the revised paper, we have added 2 additional experiments to confirm that the limited preparation time conditions we use in our studies accurately isolate implicit learning. First, we have now added a limited preparation time (Limit PT) group to our laptop-based study in Experiment 3 (Experiment 2 in original manuscript). In Experiment 3, participants in the Limit PT group (n=21) adapted to a 30° rotation but under a limited preparation time condition. As for the Limit PT group in Experiment 2, we imposed a strict bound on reaction time to suppress movement preparation time. However, unlike Experiment 2 (Experiment 1 in original manuscript) once the rotation ended, participants were told to stop re-aiming. This no-aiming instruction allowed us to examine whether limiting preparation time suppressed explicit strategy as intended. Our analysis of these new data is shown in Figures 3K-N.

In Experiment 3, we now compare two groups: one where participants had no preparation time limit (No PT Limit, Figures 3H-J) and one where an upper bound was placed on preparation time (Limit PT, Figures 3K-M). We measured early and late implicit learning measures over a 20-cycle no feedback and no aiming period at the end of the experiment. The voluntary decrease in reach angle over this no aiming period revealed each participant’s explicit strategy (Figure 3N). When no reaction time limit was imposed (No PT Limit), reaiming totaled approximately 11.86° (Figure 3N, E3, black), and did not differ statistically across Experiments 2 and 3 (t(42)=0.50, p=0.621). Recall that Experiment 2 tested participants with a robotic arm and Experiment 3 tested participants in a similar laptop-based paradigm. As in earlier reports, limiting reaction time dramatically suppressed explicit strategy, yielding only 2.09° of re-aiming (Figure 3N, E3, red).

Therefore, our new control dataset suggest that our limited reaction time technique was highly effective at suppressing explicit strategy as initially claimed in our original manuscript. However, this experiment alone suggests that while explicit strategies are strongly suppressed by limiting preparation time, they are not entirely eliminated; when we limited preparation time in Experiment 3, we observed that participants still exhibited a small decrease (2.09°) in reach angle when we instructed them to aim their hand straight to the target (Figure 3L, no aiming; Figure 3N, E3, red). This ‘cached’ explicit strategy, while small, may have contributed to the 8° reach angle change measured early during the second rotation in our savings experiment (Experiment 4, Figure 7C in revised paper).

For this reason, it was critical to consider whether this 2° change in reach angle was indeed due to explicit strategy or instead caused by time-based decay in implicit learning over the 30 sec instruction period. To test this possibility, we collected another limited preparation group (n=12, Figure 7-Supplement 1A, decay-only, black). But this time, participants were instructed that the experiment’s disturbance was still on, and that they should continue to move the ‘imagined’ cursor to the target during the no feedback period. Even though participants were instructed to continue using their strategy, their reach angles decreased by approximately 2.1° (Figure 7-Supplement 1, black decay-only). Indeed, we detected no statistically significant difference between the change in reach angle in this decay-only condition, and the Limit PT group in Experiment 3 (Figure 7-Supplement 1B; two-sample t-test, t(31)=0.016, p=0.987).

This control experiment suggested that residual ‘explicit strategies’ we measured in the Limit PT condition, were caused by time-dependent decay in implicit learning. Thus, our Limit PT exhibits a near-complete suppression of explicit strategy.

Summary

In our revised manuscript, we have added the independence model’s predictions to Figure 6 (previously Figure 4) and have altered our text to better explain that our Haith et al. analysis is not intended to support the competition model, but to compare each models’ implications. In addition, we add two new experiments to corroborate our assumption that limited preparation time trials reveal implicit learning in our tasks. Data in the Limit PT condition in Experiment 3 suggest that explicit strategies are suppressed to approximately 2° by our preparation time limit (compared to about 12° under normal conditions). Data in the decay-only group suggest that this 2° change in reach angle was not due to a cached explicit strategy but rather time dependent decay in implicit learning. Together, these conditions demonstrate that limiting reaction time in our experiments almost completely prevents the caching of explicit strategies; our limited preparation time measures do indeed reflect implicit learning.

8a. Concerns about data handling:All the experiments have a limited number of participants. This is problematic for Exp1 and 2, which have correlation analysis. Note Fig3H contains a critical, quantitative prediction of the model, but it is based on a correlation analysis of n=9. This is an unacceptably small sample. Experiment 3 only had 10 participants, but its savings result (argued as purely implicit learning-driven) is novel and the first of its kind.

We agree with these points. Yes, we do not intend for Experiments 3 and 4 (now Experiments 4 and 5) to directly contribute to the competition theory’s experimental evidence. We have made changes to our revised manuscript to make this clearer. First, our Results are now divided in 4 sections. Sections 1 and 2 present evidence that supports the competition theory. The savings and interference studies are included in a separate section (Part 3). We expect that these analyses will be important to the reader, as they provide new ideas about how the implicit system may exhibit changes that are masked by explicit strategy.

Secondly, we have now added new analysis to the Haith et al. (2015) dataset, so that it better recognizes and contrasts competition and independence model predictions. In the original Figure 4, we solely fit the competition model to the Haith et al. dataset and highlighted the implicit and explicit error sensitivities predicted by the model. In the revised manuscript, we now also fit the independence model to the Haith et al. dataset and include its error sensitivity estimates in Figure 6 (previously Figure 4) as well. We hope that this inclusion better illustrates that these model fits are not meant as evidence for one model or the other, but rather two contrasting interpretations of the same data. Below we reproduce the relevant changes to Figure 6 (previously Figure 4).

In the revised paper we explain the competition model predicts that both the implicit system and explicit system exhibited a statistically significant error sensitivity increase (Figure 6D, right; two-way rm-ANOVA, within-subject effect of exposure number, F(1,13)=10.14, p=0.007, η_p_^2^=0.438; within-subject effect of preparation time, F(1,13)=0.051, p=0.824, η_p_^2^=0.004; exposure by preparation interaction, F(1,13)=1.24, p=0.285). The independence model suggested only the explicit system exhibited a statistically significant increase in error sensitivity (Figure 5E; 2-way rm-ANOVA, learning process by exposure number interaction, F(1,13)=7.016, p=0.02; significant interaction followed by 1-way rm-ANOVA across exposures: explicit system with F(1,13)=9.518, p=0.009, η_p_^2^=0.423; implicit system with F(1,13)=2.328, p=0.151, η_p_^2^=0.152).

In the revised manuscript we conclude these results with: “In summary, when we reanalyzed our earlier data, the competition and independence theories suggested that our data could be explained by two contrasting hypothetical outcomes. If we assumed that implicit and explicit systems were independent, then only explicit learning contributed to savings, as we concluded in our original report. However, if we assumed that the implicit and explicit systems learned from the same error (competition model), then both implicit and explicit systems contributed to savings. Which interpretation is more parsimonious with measured behavior?”

We hope that these additions and revisions to the manuscript better illustrate that our analysis is not meant to support one model over the other, but rather to illustrate their contrasting implications about the implicit system.

Finally, the reviewers also note in their concern, that our analyses in Figures 7, 8, and 9 (previously Figures 4, 6, and 7) depend on the assumption that limited preparation time trials isolate implicit learning. We responded to similar concerns in Point 3 above. We will reiterate this discussion here. In the revised paper, we have added 2 additional experiments to confirm that the limited preparation time conditions we use in our studies accurately isolate implicit learning. First, we have now added a limited preparation time (Limit PT) group to our laptop-based study in Experiment 3 (Experiment 2 in original manuscript). In Experiment 3, participants in the Limit PT group (n=21) adapted to a 30° rotation but under a limited preparation time condition. As for the Limit PT group in Experiment 2, we imposed a strict bound on reaction time to suppress movement preparation time. However, unlike Experiment 2 (Experiment 1 in original manuscript) once the rotation ended, participants were told to stop re-aiming. This no-aiming instruction allowed us to examine whether limiting preparation time suppressed explicit strategy as intended. Our analysis of these new data is shown in Figures 3K-N.

In Experiment 3, we now compare two groups: one where participants had no preparation time limit (No PT Limit, Figures 3H-J) and one where an upper bound was placed on preparation time (Limit PT, Figures 3K-M). We measured early and late implicit learning measures over a 20-cycle no feedback and no aiming period at the end of the experiment. The voluntary decrease in reach angle over this no aiming period revealed each participant’s explicit strategy (Figure 3N). When no reaction time limit was imposed (No PT Limit), reaiming totaled approximately 11.86° (Figure 3N, E3, black), and did not differ statistically across Experiments 2 and 3 (t(42)=0.50, p=0.621). Recall that Experiment 2 tested participants with a robotic arm and Experiment 3 tested participants in a similar laptop-based paradigm. As in earlier reports, limiting reaction time dramatically suppressed explicit strategy, yielding only 2.09° of re-aiming (Figure 3N, E3, red).

Therefore, our new control dataset suggest that our limited reaction time technique was highly effective at suppressing explicit strategy as initially claimed in our original manuscript. However, this experiment alone suggests that while explicit strategies are strongly suppressed by limiting preparation time, they are not entirely eliminated; when we limited preparation time in Experiment 3, we observed that participants still exhibited a small decrease (2.09°) in reach angle when we instructed them to aim their hand straight to the target (Figure 3L, no aiming; Figure 3N, E3, red). This ‘cached’ explicit strategy, while small, may have contributed to the 8° reach angle change measured early during the second rotation in our savings experiment (Experiment 4, Figure 7C in revised paper).

For this reason, it was critical to consider whether this 2° change in reach angle was indeed due to explicit strategy or instead caused by time-based decay in implicit learning over the 30 sec instruction period. To test this possibility, we collected another limited preparation group (n=12, Figure 7-Supplement 1A, decay-only, black). But this time, participants were instructed that the experiment’s disturbance was still on, and that they should continue to move the ‘imagined’ cursor to the target during the no feedback period. Even though participants were instructed to continue using their strategy, their reach angles decreased by approximately 2.1° (Figure 7-Supplement 1, black decay-only). Indeed, we detected no statistically significant difference between the change in reach angle in this decay-only condition, and the Limit PT group in Experiment 3 (Figure 7-Supplement 1B; two-sample t-test, t(31)=0.016, p=0.987).

This control experiment suggested that residual ‘explicit strategies’ we measured in the Limit PT condition, were caused by time-dependent decay in implicit learning. Thus, our Limit PT exhibits a near-complete suppression of explicit strategy.

Summary

In our revised manuscript, we have added the independence model’s predictions to Figure 6 (previously Figure 4) and have altered our text to better explain that our Haith et al. analysis is not intended to support the competition model, but to compare each models’ implications. In addition, we add two new experiments to corroborate our assumption that limited preparation time trials reveal implicit learning in our tasks. Data in the Limit PT condition in Experiment 3 suggest that explicit strategies are suppressed to approximately 2° by our preparation time limit (compared to about 12° under normal conditions). Data in the decay-only group suggest that this 2° change in reach angle was not due to a cached explicit strategy but rather timedependent decay in implicit learning. Together, these conditions demonstrate that limiting reaction time in our experiments almost completely prevents the caching of explicit strategies; our limited preparation time measures do indeed reflect implicit learning.

8b. Experiment 2: This experiment tried to decouple implicit and explicit learning, but it is still problematic. If the authors believe that the amount of implicit adaptation follows a state-space model, then the measure of early implicit is correlated to the amount of late implicit because Implicit_late = (ai)^N * Implicit_early where N is the number of trials between early and late. Therefore the two measures are not properly decoupled. To decouple them, the authors should use two separate ways of measuring implicit and explicit. To measure explicit, they could use aim report (Taylor, Krakauer and Ivry 2011), and to measure implicit independently, they could use the after-effect by asking the participants to stop aiming as done here. They actually have done so in experiment 1 but did not use these values.

One reason we conducted Experiment 2 (now Experiment 3) was to examine whether the implicit-explicit relationships were due to spurious correlations (e.g., implicit is compared to explicit, which is total learning minus implicit). While we appreciate the reviewers’ suggestion, we still maintain that the late implicit learning measure in Experiment 2 eliminates spurious correlation as intended.

To illustrate this, imagine that implicit and explicit learning have no relationship. Let’s represent these as random variables I and E. Supposing I and E are independent this implies that cov(I,E) = 0. Now let’s introduce a third random variable, T, which represents total learning. Suppose that we measure total adaptation on the last several cycles. This total adaptation will be equal to the sum of I and E. So total adaptation, T is given by T = I + E. Now, we the experimenter cannot measure I, E, and T. We can only measure these variables corrupted by motor execution noise, M. A reasonable assumption is that M is a zero-mean random variable, that is independently distributed across each reach.cov(Imeasure,Eestimate)=cov(I+M2,E+M1−M2)=var(M2)

At the end of adaptation, the measured total adaptation is thus T_measure_ = T + M_1_ = I + E + M_1_. Now, on the next cycle, we do exclusion trials. These reveal I, but this is also corrupted by motor execution noise. Thus, the I that we measure is given by: I_measure_ = I + M_2_. Now, in Experiment 1, we derived E by subtracting I_measure_ from T_measure_. This is given by: E_estimate_ = T_measure_ – I_measure_ = I + E + M_1_ – (I + M_2_) = E + M_1_ – M_2_. Now, we can see the issue when we correlate I_measure_ with E_estimate_:This assumes as we said that I and E are independent.

This clearly illustrates the issue of spurious correlation. Even though I and E are independent, the way we measured I and E results in a non-zero misleading correlation.

Our late implicit measure in Experiment 3 prevents this spurious correlation. Suppose we measure I again on the second cycle in the no-aiming aftereffect period. As noted in the criticism, I here will relate to the previous I, but will be scaled by the retention factor *a*. Thus, the second I that we measure is given by: I_2_ = *a*I + M_3_ where M_3_ is now motor execution noise on the 2^nd^ no-aiming cycle. Now let us determine the covariance between I_2_ and E_estimate_: cov(I_2_, E_estimate_) = cov(*a*I + M_3_, E + M_1_ – M_2_). Noting that I, E, M_1_, M_2_, and M_3_ are all independent, this covariance will be equal to 0.

What does this mean? Even though E_estimate_ is based on the initial ‘early I’ measurement, the second ‘late I’ that we measure (even though it is related to the first I via *a*) is no longer correlated with E_estimate_. This is the type of spurious correlation we avoid with our late implicit measure in Experiment 2. Thus, the only way for E_estimate_ to be correlated with any later I measurement (apart from the initial one) requires that the original E and I are not in fact independent (as posited by the competition model).

In our revised manuscript we have increased the number of participants in Experiment 3 to n=35. Below, we show the early implicit-explicit relationship in Figure 3Q and the decoupled late implicit-explicit relationship in Figure 3P in the revised manuscript.

Second, though our late implicit measure does decouple implicit and explicit learning we have added the report-based measures requested by the reviewers to the paper. We also check the relationship between implicit and explicit learning using the aiming reports collected in the experiment. This correlation is now provided in Figure 3-Supplement 2C. In addition, we compare report and exclusion-based implicit and explicit measures in Figure 3-Supplements 2AandB. Note, we are not highly confident in these report measures, given that we collected them only one time and 25% were reported in the incorrect direction (see Point 4).

Thirdly, the revised paper includes new analyses that corroborate the competition model while avoiding spurious implicit-explicit correlations. Recall that spurious correlations can arise between exclusion-based implicit learning and explicit strategy. To avoid this potential issue, the competition model can be restated in another way. As we discussed in Point 1 above, the competition model predicts that *x_i_*^ss^ = *p_i_*(*r* – *x_e_^ss^*), where π is a gain that depends on implicit error sensitivity and retention. In our manuscript we investigate whether *x_i_^ss^* and *x_e_^ss^* vary according to this relationship. Note, however, that total adaptation is equal to *x_T_^ss^* = *x_i_^ss^* + *x_e_^ss^*. By combining these two equations, we can relate implicit learning to total adaptation via the equation: *x_T_^ss^* = *r* + (*p_i_* – 1)*p_i_*^-1^*x_i_^ss^* (see Lines 1576-1620 in revised paper). This equation predicts that total learning will be negatively correlated to implicit learning. This counterintuitive relationship, that greater overall adaptation will be supported by reduced implicit learning, is opposite to that predicted by an SPE learning model like the independence equation. Supposing that implicit learning responds only to SPE, but strategies respond to variations in implicit learning yields: *x_e_^ss^* = *p_e_*(*r* – *x_i_^ss^*) where *p_e_* is the gain in the explicit response. Combining this with *x_T_^ss^* = *x_i_^ss^* + *x_e_^ss^* yields a positive linear relationship between total adaptation and implicit learning: *x_T_^ss^* = *p_e_r* + (1 – *p_e_*)*x_i_^ss^*.

In sum the critical idea is that we can compare total adaptation and implicit learning. There is no spurious correlation between these quantities, because they are calculated on distinct trials. A negative correlation supports the competition model. A positive correlation supports the independence model. In the revised manuscript we analyzed this relationship in Experiment 3, as well as Experiment 1, new data from Maresch et al., 2021, and new data from Tsay et al., 2021. These are shown in Figures 4H and Figure 4-Supplements 1A-C.

Lastly, it should be noted that these analyses occur at the subject-level but can also be done to analyze group-level effects. In particular, we use the equations above in order to determine whether the implicit learning phenotypes examined in Point 6B (saturation, scaling, and non-monotonicity) also match the competition model. To this end, the relationships between implicit learning and total adaptation can be restated with algebraic manipulations. For the competition model we obtain: *x_i_^ss^* = *p_i_*(1 – *p_i_*)^-1^(*r* – *x_T_^ss^*). In words, this equation provides a way to predict implicit learning using total adaptation. In the revised manuscript we analyze three implicit phenotypes: (1) saturation in Neville and Cressman, 2018, (2) scaling in Experiment 1, and (3) non-monotonicity in Tsay et al., 2021 (see Point 6B above for more details). In the Neville and Cressman dataset, we fit the above model to identify one π value that best predicts *x_i_^ss^* using *x_T_^ss^* across all 6 experimental groups (all 3 rotation magnitudes crossed with the 2 instruction conditions). In Experiment 1, we fit the above model to identify one π that best predicts *x_i_^ss^* using *x_T_^ss^* across 5 periods (all 4 rotation sizes in the stepwise group plus the last 60° learning period in the abrupt group). Finally, in Tsay et al., we fit the above model to identify one π that best predicts *x_i_^ss^* using *x_T_^ss^* across all 4 rotation sizes. The data are compared with model predictions in Figure 1-Supplement 2.

In black we show measured implicit learning. In blue we show implicit learning predicted using explicit strategy via the standard competition model: *x_i_^ss^* = *p_i_*(*r* – *x_e_^ss^*). In gray we show implicit learning predicted using total adaptation via the reworked competition model: *x_i_^ss^* = *p_i_*(1 – *p_i_*)^-1^(*r* – *x_T_^ss^*). The similarity between each models predictions confirms that the group-level implicit phenotypes predicted by the competition model are not due to a spurious correlation between exclusion-based implicit and explicit learning.

Summary

Spurious correlations are common in implicit-explicit analyses. This occurs in many studies, either when both processes are inferred using exclusion trials (where explicit learning is estimated as total learning minus implicit learning), or report-based measures (where implicit learning is estimated as total learning minus explicit strategy). We have now improved and added several analyses to the revised manuscript to corroborate our main results while avoiding sources of spurious correlation. These include:

1. In Experiment 3, we have increased the total participant count to n=35. In Figure 3P we show the relation between explicit strategy and ‘late’ implicit learning which are calculated on separate trials.

2. We now re-derive the competition model in a way that relates implicit learning to total adaptation as opposed to explicit strategy. This version avoids spurious correlations. We show the correlation between implicit learning and total adaptation at the individual participant-level in Experiment 3 (Figure 4H), Experiment 1 (Figure 4-Supplement 1B), Maresch et al., 2021 (Figure 4-Supplement 1A), and Tsay et al., 2021 (Figure 4-Supplement 1C).

3. Finally, we also use total adaptation to predict variations in implicit learning at the group-level in Experiment 1 (Figure 1-Supplement 2, Experiment 1 stepwise), Neville and Cressman, 2018 (Figure 1-Supplement 2, Neville and Cressman (2018)), and Tsay et al., 2021 (Figure 1-Supplement 2, Tsay et al., 2021).

Lastly, as requested, we also report the relationship between implicit learning and explicit strategy using the report measures (Figure 3-Supplement 2) collected in Experiment 2 (previously Experiment 1).

8c. Figure 6: To evaluate the saving effect across conditions and experiments, using the interaction effect of ANOVA shall be desired.

We appreciate this suggestion. We have now updated all relevant analyses in our revised manuscript. We used either a two-way ANOVA, two-way repeated measures ANOVA, or mixed-ANOVA depending on the experimental design. When interactions were statistically significant, we next measured simple main effects via one-way ANOVA. We outline this on Lines 1195-1207. All conclusions reached in the original manuscript remained the same using this updated statistical procedure. These include:

1. When measuring savings in Haith et al., we observed that the learning rate increased during the second exposure on high preparation time trials, but not on low preparation time trials (Figure 5B, right; two-way rm-ANOVA, prep. time by exposure number interaction, F(1,13)=5.29, p=0.039; significant interaction followed by one-way rm-ANOVA across Days 1 and 2: high preparation time with F(1,13)=6.53, p=0.024, η_p_^2^=0.335; low preparation time with F(1,13)=1.11, p=0.312, η_p_^2^=0.079). We corroborate this rate analysis by measuring early changes in reach angle (first 40 trials following rotation onset) across Days 1 and 2 (Figure 5C, left and middle). Only high preparation time trials exhibited a statistically significant increase in reach angles, consistent with savings (Figure 5C, right; two-way rm-ANOVA, preparation time by exposure interaction, F(1,13)=13.79, p=0.003; significant interaction followed by one-way rm-ANOVA across days: high preparation time with F(1,13)=11.84, p=0.004, η_p_^2^=0.477; low preparation time with F(1,13)=0.029, p=0.867, η_p_^2^=0.002).

2. When comparing implicit and explicit error sensitivities predicted by the competition model, the model predicted that both implicit system and explicit systems exhibited a statistically significant error sensitivity increase (Figure 5D, right; two-way rm-ANOVA, within-subject effect of exposure no., F(1,13)=10.14, p=0.007, η_p_^2^=0.438; within-subject effect of learning process, F(1,13)=0.051, p=0.824, η_p_^2^=0.004; exposure no. by learning process interaction, F(1,13)=1.24, p=0.285).

3. When comparing implicit and explicit error sensitivities predicted by the independence model, the model predicted that only the explicit system exhibited a statistically significant increase in error sensitivity (Figure 5E; two-way rm-ANOVA, learning process (implicit vs explicit) by exposure interaction, F(1,13)=7.016, p=0.02; significant interaction followed by one-way rm-ANOVA across exposures: explicit system F(1,13)=9.518, p=0.009, η_p_^2^=0.423; implicit system with F(1,13)=2.328, p=0.151, η_p_^2^=0.152).

4. When measuring savings under limited preparation time in Experiment 4, we still observed the opposite outcome to Haith et al., 2015. Notably, low preparation time learning rates increased by more than 80% in Experiment 4 (Figure 7C top; mixed-ANOVA exposure number by experiment type interaction, F(1,22)=5.993, p=0.023; significant interaction followed by one-way rm-ANOVA across exposures: Haith et al. with F(1,13)=1.109, p=0.312, η_p_^2^=0.079; Experiment 4 with F(1,9)=5.442, p=0.045, η_p_^2^=0.377). Statistically significant increases in reach angle were detected immediately following rotation onset in Experiment 4 (Figure 7B, bottom), but not our earlier data (Figure 7C, bottom; mixed-ANOVA exposure number by experiment interaction, F(1,22)=4.411, p=0.047; significant interaction followed by one-way rm-ANOVA across exposures: Haith et al. with F(1,13)=0.029, p=0.867, η_p_^2^=0.002; Experiment 4 with F(1,9)=11.275, p=0.008, η_p_^2^=0.556).

5. Lastly, we updated our anterograde interference analysis in Figure 8. While both low preparation time and high preparation time trials exhibited decreases in learning rate which improved with the passage of time (Figure 8C; two-way ANOVA, main effect of time delay, F(1,50)=5.643, p=0.021, η_p_^2^=0.101), these impairments were greatly exacerbated by limiting preparation time (Figure 8C; two-way ANOVA, main effect of preparation time, F(1,50)=11.747, p=0.001, η_p_^2^=0.19).

8d. Figure 8: The hypothesis that they did use an explicit strategy could explain why the difference between the two groups rapidly vanishes. Also, it is unclear whether the data from Taylor and Ivry (2011) are in favor of one of the models as the separate and shared error models are not compared.

This is a good criticism, similar to that raised in Point 2. Please note that Figure 9 was not included to directly test the competition model, but more so as a way to demonstrate its limitations. For example, in Mazzoni and Krakauer (2006) our simple target error model cannot reproduce the measured data; in the instructed group implicit learning continued despite eliminating the primary target error. The competition model would predict no implicit learning in this case. Thus, in more complicated scenarios where multiple visual landmarks are used during adaptation, the learning rules must be more sophisticated than the simple competition theory. We feel that it is critical that this message is communicated to the reader.

While we agree that the difference between the two groups in Mazzoni and Krakauer (2006) diminishes rapidly, it did not vanish entirely. In Author response image 1 we show the mean aftereffect on the last 10 trials of washout (trials 71-80). Note that a 3° difference (t-test over last 10 trials, t(9)=3.24, p=0.01) still persisted on these trials, consistent with a lingering difference in implicit learning across the two groups. There are at least 3 reasons why this difference may have quickly diminished: (1) implicit learning decays in both groups on each trial, (2) there is implicit learning in the opposite direction in response to the washout errors, and (3) the explicit system likely comes online to mitigate the ‘negative’ washout errors.

**Author response image 1. sa2fig1:** Measuring the aftereffect on the last 10 washout trials in Mazzoni and Krakauer, 2006.

Nevertheless, while it seems likely that the washout period reveals differences in implicit learning across the two groups, we agree that we cannot know this with certainty because participants were not told to stop aiming. The Mazzoni and Krakauer analysis, however, still shows limitations in the competition model and provides initial ideas about how learning models may be extended in the future. To reflect this point, this analysis now appears in Part 4 of the revised paper, which is titled: “*Limitations of the competition theory*”. In addition, we have revised the text to appropriately note the limitation in the Mazzoni and Krakauer analysis, reproduced below (Lines 752-762):

“It is important, however, to note a limitation in these analyses. Our earlier study did not employ the standard conditions used to measure implicit aftereffects: i.e., instructing participants to aim directly at the target, and also removing any visual feedback. Thus, the proposed dual-error model relies on the assumption that differences in washout were primarily related to the implicit system. These assumptions need to be tested more completely in future experiments.

In summary, the conditions tested by Mazzoni and Krakauer show that the simplistic idea that adaptation is driven by only one target error, or only one SPE, cannot be true in general^54^. We propose a new hypothesis that when people move a cursor to one visual target, while aiming at another visual target, each target may partly contribute to implicit learning. When these two error sources conflict with one another, the implicit learning system may exhibit an attenuation in total adaptation. Thus, implicit learning modules may compete with one another when presented with opposing errors.”

Finally, note that the Taylor and Ivry (2011) data are included in Part 4 of the revised paper. Similar to our Mazzoni and Krakauer analysis, our goal for this dataset, is to show that the competition theory is not a universal model that can be applied across all scenarios. In these data, even when no secondary aiming target is provided, implicit learning is evident in the aiming group where target error is zero. Thus, it would appear in the no aiming target group, that implicit learning was driven by an SPE. However, an SPE-only model cannot explain why adaptation is 3 times larger when the second target is provided. Thus, it seems in this case neither a target error nor an SPE alone drive adaptation. These data raise several questions. For instance, if SPE-learning occurs in Taylor and Ivry even without an aiming target, why did not we not detect it in our primary experiments? We think it is critical that the reader note these questions to inform new experiments in the future.

In our revised manuscript, we have expanded on the implications of both the Mazzoni and Krakauer, and Taylor and Ivry studies. These data suggest that both target errors and SPEs play a role in implicit learning, but that experimental contexts/conditions alter their contributions. One possibility is that SPE learning is highly sensitive to context. In cases where no aiming target is ever provided (as in all studies in our primary analyses), it may be minimal. Providing aiming targets in the past may invoke an implicit memory (like memory-guided saccade adaptation) that can drive ‘weaker’ SPE learning. Removing the aiming targets partway through the movement may provide a moderate memory that drives more SPE learning. And seeing the aiming target at all times may create the strongest memory driving the most SPE learning. An another possibility altogether is that the aftereffects noted in the no aiming target group in Taylor and Ivry reflect a reward-based and use-dependent memory: i.e., aiming to another location and receiving rewards may bias future reaching movements in that direction even without any visual target errors to drive learning. The relevant passage on these matters is provided below (Lines 1063-1090):

“…the nature of aim-cursor errors remains uncertain. For example, while this error source generates strong adaptation when the aim location coincides with a physical target (Figure 10H, instruction with target), implicit learning is observed even in the absence of a physical aiming landmark^9^ (Figure 10H, instruction without target), albeit to a smaller degree. This latter condition may implicate an SPE learning that does not require an aiming target. Thus, it may be that the aim-cursor error in Mazzoni and Krakauer is actually an SPE that is enhanced by the presence of a physical target. In this view, implicit learning is driven by a target error module and an SPE module that is enhanced by a visual target error^4,11,86^.

These various implicit learning modules are likely strongly dependent on experimental contexts, in ways we do not yet understand. For example, Taylor and Ivry (2011) would suggest that all experiments produce implicit some SPE learning, but less so in paradigms with no aiming targets. Yet, the competition equation accurately matched single-target behavior in Figures 1-9 without an SPE learning module. It is not clear why SPE learning would be absent in these experiments. One idea may be that the aftereffect observed by Taylor and Ivry (2011) in the absence of an aiming target, was actually a lingering associative motor memory that was reinforced by successfully hitting the target during the rotation period. Indeed, such a model-free learning mechanism^87^ should be included in a more complete implicit learning model. It is currently overlooked in error-based systems such as the competition and independence equations.

Another idea is that some SPE learning did occur in the no aiming target experiments we analyzed in Figures 1-9, but was overshadowed by the implicit system’s response to target error. A third possibility is that the SPE learning observed by Taylor and Ivry (2011) was contextually enhanced by participants implicitly recalling the aiming landmark locations (akin to memory-guided saccade adaptation) provided during the baseline period. This possibility would suggest SPEs vary along a complex spectrum: (1) never providing an aiming target causes little or no SPE learning (as in our experiments), (2) providing an aiming target during past training allows implicit recall that leads to small SPE learning, (3) providing an aiming target that disappears during the movement promotes better recall and leads to medium-sized SPE learning (i.e., the disappearing target condition in Taylor and Ivry), and (4) an aiming target that always remains visible leads to the largest SPE learning levels. This context-dependent SPE hypothesis may be related to recent work suggesting that both target errors and SPEs drive implicit learning, but implicit SPE learning is altered by distraction^54^.”

Reviewer #1:The present study by Albert and colleagues investigates how implicit learning and explicit learning interact during motor adaptation. This topic is under heated debate in the sensorimotor adaptation area in recent years, spurring diverse behavioral findings that warrant a unified computational model. The study is to fulfill this goal. It proposes that both implicit and explicit adaptation processes are based on a common error source (i.e., target error). This competition leads to different behavioral patterns in diverse task paradigms.I find this study a timely and essential work for the field. It makes two novel contributions. First, when dissecting the contribution of explicit and implicit learning, the current study highlights the importance of distinguishing apparent learning size and covert changes in learning parameters. For example, the overt size of implicit learning could decrease while the related learning parameters (e.g., implicit error sensitivity) remain the same or even increase. Many researchers for long have overlooked this dissociation. Second, the current study also emphasizes the role of target error in typical perturbation paradigms. This is an excellent waking call since the predominant view now is that sensory prediction error is for implicit learning and target error for explicit learning.Given that the paper aims to use a unified model to explain different phenomena, it mixes results from previous work and four new experiments. The paper's presentation can be improved by reducing the use of jargon and making straightforward claims about what the new experiments produce and what the previous studies produce. I will give a list of things to work on later (minor concerns).Furthermore, my major concern is whether the property of the implicit learning process (e.g., error sensitivity) can be shown subjective to changes without making model assumptions.My major concern is whether we can provide concrete evidence that the implicit learning properties (error sensitivity and retention) can be modified. Even though the authors claim that the error sensitivity of implicit learning can be changed, and the change subsequently leads to savings and interference (e.g., Figures 6 and 7), I find that the evidence is still indirect, contingent on model assumptions. Here is a list of results that concern error sensitivity changes.1.1. Figures 1 and 2: Instruction is assumed to leave implicit learning parameters unchanged.

Yes, we agree that this is an assumption, but feel that it is reasonable in this instance. To clarify, Figure 1 in the original manuscript concerned our Neville and Cressman (2018) analysis and Figure 2 concerned our Saijo and Gomi (2010) analysis. Instructions were only provided in the Neville and Cressman study, and so this concern seems more relevant for Figure 1 (Figures 2A-C in revised manuscript). Furthermore, our Neville and Cressman analysis explores (1) changes in rotation size, and (2) changes in strategy levels. Only the latter condition is relevant to this criticism.

In the Neville and Cressman study, instructions were given to some participants in a ‘strategy’ group. In this group, participants were briefed about the rotation with an image that depicted how feedback would be rotated, and how they could compensate for it.

In sum, participants were given instructions prior to the rotation. Once the rotation turned on, all subjects experienced the same trial structure, the same feedback, and no additional instructions. In our view, this intervention simply influences participants to use larger explicit strategies than they might have otherwise as is evident in their learning curves.

While we are not sure why changes in the participant’s aim direction would result in a change in implicit learning properties, the most likely mechanism in our view would be changes in implicit error sensitivity. Changes in implicit error sensitivity could potentially impact the competition model, where the implicit system’s learning gain is given by π = *b_i_*(1-*a_i_*+*b_i_*)^-1^. In other words, we might expect individuals with higher error sensitivity to have a larger implicit learning gain in the competition model. This is potentially relevant to the strategy conditions in Neville and Cressman; the no-instruction group learned much more slowly and thus, experienced larger errors throughout adaptation. These larger errors would result in a smaller error sensitivity, and therefore, a smaller gain in the competition model. This, however, would be puzzling, because the no-strategy group exhibited more implicit learning, not less, than the strategy group.

Nevertheless, whether implicit error sensitivity varies due to error size, or some other mechanism, has little impact on the model. To see this, recall that the ‘tunable’ component of the competition model is the implicit system’s learning gain, which depends on the retention factor and error sensitivity according to π = *b_i_*(1-*a_i_*+*b_i_*)^-1^. Fortunately, this implicit learning gain responds weakly to changes in error sensitivity (which appears in both the numerator and denominator). For example, let us suppose that participants in the strategy group exhibited an implicit error sensitivity of 0.3, but in the no-strategy group, implicit error sensitivity was only 0.2. For an implicit retention factor of 0.9565 (see Methods in revised paper), the nostrategy learning gain would be 0.821 and the strategy learning gain would be 0.873. Thus, even though implicit error sensitivity was 50% larger in the strategy participants, the competitive implicit learning gain would change only 6.3%. For an even more extreme case where implicit error sensitivity was double (0.4) in the strategy group, this would still only lead to a 9.8% change in the competitive implicit learning gain.

In sum, the competition model’s learning gain which governs implicit behavior is very robust to changes in implicit error sensitivity. Therefore, any changes in implicit error sensitivity that may have occurred across the instructed and uninstructed groups has little to no impact on model behavior. In other words, even moderate size differences in error sensitivity will have little to no effect on the competition model predictions in Figures 1 and 2 (in revised paper). We explain this on Lines 1946-1973 in the revised paper.

Despite this, we still felt it was important to check whether implicit learning properties may have differed across the strategy and no-strategy conditions, or in other words, high strategy use and low strategy use conditions. To do this, we examined implicit and explicit learning in Experiments 1-3 and measured whether the implicit-explicit relationship differed over low and high strategy domains. The idea here is that if implicit learning properties change with strategy, that we should detect structural changes in implicit behavior as we move from low strategy use to high strategy use. To do this, we divided our data in Experiments 1-3 into a low strategy (subjects whose strategy was less than the median) and a high strategy (subjects whose strategy was greater than the median) group. We then linearly regressed implicit learning onto strategy in the low strategy and high strategy groups separately. The domains for each analysis were separated by the dashed lines shown in Author response image 2.

**Author response image 2. sa2fig2:** Here we split the data in Experiments 1-3 into low and high strategy groups (based on the median strategy). We fit a regression to the ‘low-strategy’ group (blue line) and the ‘high-strategy’ group (magenta line). We compared the slope and intercept to see if there was a systematic difference in the regression’s gain and bias for low and high strategy cases.

Across the 6 conditions above, strategies were 135% larger in high strategy group participants than low strategy group participants. Implicit behavior, however, appeared insensitive to this large difference in strategy; overall, the slope and bias in the 6 implicit-explicit regressions differed by only 9.3% and 5.1%, respectively. In sum, even though strategy more than doubled in the high strategy group, the gain and bias in the implicit-explicit relationship changed less than 10%. This control analysis strongly suggests that the implicit system learns very similarly despite large changes in strategy.

Summary

We appreciate the reviewer’s concern that changes in instruction could alter implicit learning. We assume in our modeling that the implicit learning gain remains the same across instruction conditions. However, it is possible that implicit error sensitivity could differ across groups. Fortunately, the learning gain in the competition model is very insensitive to changes in implicit error sensitivity: even doubling error sensitivity increases the competition model’s learning gain by less than 10%. Indeed, the relationship between implicit and explicit learning exhibited little to no difference across participants with high and low strategy use. Given these concerns, we feel that our assumption in Neville and Cressman, that the implicit learning gain is the same across strategy conditions, is reasonable. Indeed, the data in Figures 2BandC (revised paper) well-match the competition model’s prediction even with a fixed learning gain. We should also note that overall, our paper examines many ways that implicit behavior exhibits competitive behavior that do not rely on examining changes in instruction: e.g., (1) steady-state responses to changes in rotation size (revised Figure 1, see Point 6B) in Neville and Cressman, Experiment 1, and Tsay et al., (2) changes in implicit learning with gradual vs. abrupt rotation onset (Figures 2D-G),(3) increases in implicit learning due to limits in preparation time (Figure 3O), and (4) many pairwise correlations between implicit and explicit learning, and total adaptation (Figure 4 as well as Figure 4-Supplement 1).

1.2. Figure 4: It appears that the implicit error sensitivity increases during relearning. However, Fig4D cannot be taken as supporting evidence. How the model is constructed (both implicit and explicit learning are based on target error) and what assumptions are made (Low RT = implicit learning, High RT = explicit + implicit) determine that implicit learning's error sensitivity must increase. In other words, the change in error sensitivity resulted from model assumptions; whether implicit error sensitivity by itself changes cannot be independently verified by the data.

The reviewer is correct. Please refer to Point 7 for a thorough response to these comments. To briefly summarize, it was not our intention to imply that the competition model results ‘proved’ that implicit error sensitivity increased in the Haith et al., 2015 experiment. Instead, we had intended to show that the competition model presents a hypothetical situation: that implicit error sensitivity can increase without any change in the implicit learning timecourse. This is significant because it means that additional experiments are needed to verify the conclusion that we reached in Haith et al., 2015: that savings is due to explicit strategy alone.

In the revised paper, we have made several changes to make these points clearer. Again, these are more thoroughly described in Point 7.

1. First, we now divide our results into 4 sections. In Part 1, we provide evidence for the competition theory. In Part 2, we compare the competition theory to alternative models. In Part 3, we discuss savings and interference. In Part 4, we discuss limitations in the competition model. It is our hope that by dividing our results into clearer sections, we better delineate their different purposes.

2. Secondly, we now show model parameters for the independence model in Figure 6 (previously Figure 4) in addition to the competition model. In this way, we hope to illustrate that neither model is the ‘true’ model: rather, they are models with contrasting predictions.

3. Thirdly, we have revised our text to reflect the hypothetical nature of these results. For instance, we now summarize our Haith et al., 2015 analysis with the following passage: “In summary, when we reanalyzed our earlier data, the competition and independence theories suggested that our data could be explained by two contrasting hypothetical outcomes. If we assumed that implicit and explicit systems were independent, then only explicit learning contributed to savings, as we concluded in our original report. However, if we assumed that the implicit and explicit systems learned from the same error (competition model), then both implicit and explicit systems contributed to savings. Which interpretation is more parsimonious with measured behavior?”

We hope that these additions and revisions to the manuscript better illustrate that our analysis is not meant to support one model over the other, but rather to illustrate their contrasting implications about the implicit system.

1.3. Figure 6: With limited PT, savings still exists with an increased implicit error sensitivity. Again, this result also relies on the assumption that limited PT leads to implicit-only learning. Only with this assumption, the error sensitivity can be calculated as such.

We completely agree. Please refer to Points 3and7 (in our response to the editorial summary) for a more thorough response to these comments. To briefly summarize, in the revised paper, we have added 2 additional experiments to confirm that the limited preparation time conditions we use in our studies accurately isolate implicit learning. Firstly, we directly measure explicit strategy under limited preparation time conditions. In our new Limit PT group in Experiment 3, participants adapted under limited preparation time, and are told to stop aiming at the end of the experiment. These new data are shown in Figures 3K-N.

When no reaction time limit was imposed (No PT Limit), re-aiming totaled approximately 11.86° (Figure 3N, E3, black), and did not differ statistically across Experiments 2 and 3 (t(42)=0.50, p=0.621). Recall that Experiment 2 tested participants with a robotic arm and Experiment 3 tested participants in a similar laptop-based paradigm.

As in earlier reports, limiting reaction time dramatically suppressed explicit strategy, yielding only 2.09° of re-aiming (Figure 3N, E3, red).

Therefore, our new data suggest that limiting preparation time is highly effect at suppressing strategy use. However, participants in the limited preparation time condition still exhibited a small decrease (2.09°) in reach angle when we instructed them to aim their hand straight to the target. This ‘cached’ strategy, while small, may have contributed to the 8° reach angle change measured early during the second rotation in our savings experiment (Experiment 4, Figure 8C in revised paper).

For this reason, we considered whether this 2° change in reach angle was indeed due to explicit strategy or instead caused by time-based decay in implicit learning over the 30 sec instruction period. To test this possibility, we tested another limited preparation group (n=12, Figure 8-Supplement 1A, decay-only, black). But this time, participants were instructed that the experiment’s disturbance was still on, and that they should continue to move the ‘imagined’ cursor to the target during the no feedback period. Even though subjects were instructed to continue using their strategy, their reach angles decreased by approximately 2.1° (Figure 8-Supplement 1, black decay-only). Indeed, we detected no statistically significant difference between the change in reach angle in this decay-only condition, and the Limit PT group in Experiment 3 (Figure 8-Supplement 1B; two-sample t-test, t(31)=0.016, p=0.987).

This control experiment suggested that residual ‘explicit strategies’ we measured in the Limit PT condition, were caused by time-dependent decay in implicit learning. Thus, our method for limiting preparation time does prevent the participant from expressing explicit strategies.

With regards to savings, the key prediction is that while no savings was observed on limited preparation time trials, by consistently limiting preparation time in Experiment 4 (previously Experiment 3), the implicit system may exhibit savings. In the revised manuscript we have improved our statistical methodology for comparing our new experiment with Haith et al. Using a mixed-ANOVA we determined that low preparation time learning rates increased by more than 80% in Experiment 4 (Figure 8C top; mixed-ANOVA exposure number by experiment type interaction, F(1,22)=5.993, p=0.023; significant interaction followed by one-way rmANOVA across exposures: Haith et al. with F(1,13)=1.109, p=0.312, η_p_^2^=0.079; Experiment 4 with F(1,9)=5.442, p=0.045, η_p_^2^=0.377). Statistically significant increases in reach angle were also detected immediately following rotation onset in Experiment 4 (Figure 7B, bottom), but not in Haith et al. (Figure 8C, bottom; mixed-ANOVA exposure number by experiment interaction, F(1,22)=4.411, p=0.047; significant interaction followed by one-way rm-ANOVA across exposures: Haith et al. with F(1,13)=0.029, p=0.867, η_p_^2^=0.002; Experiment 4 with F(1,9)=11.275, p=0.008, η_p_^2^=0.556).

Summary

In our revised manuscript, we add 2 experiments to corroborate our assumption that limited preparation time trials reveal implicit learning in our tasks. Data in the Limit PT condition in Experiment 3 suggest that explicit strategies are suppressed to approximately 2° by our preparation time limit (compared to about 12° under normal conditions). Data in the decay-only group suggest that this 2° change in reach angle was not due to a cached explicit strategy but rather time-dependent decay in implicit learning. Together, these conditions demonstrate that limiting reaction time in our experiments almost completely prevents the caching of explicit strategies; our limited preparation time measures do indeed reflect implicit learning. Therefore, we can study how the implicit system changes by limiting preparation time and measuring the learning curve on two consecutive exposures to that rotation. Our new statistical analysis suggested that these limited reaction time conditions produced savings that was not observed in Haith et al. In total, the data show that the implicit system does exhibit savings in Experiment 4 (previously Experiment 3), but its expression can be impeded by changes in the explicit system.

1.4. Figure 7: with limited PT anterograde interference persists, appearing to show a reduced implicit error sensitivity. Again, this is based on the assumption that the limited PT condition leads to implicit-only learning.

We again appreciate this concern. As we outline in Points 3, 7, and 1-3, we have collected two new datasets which show that limiting preparation time prevents explicit strategy use. This provides critical support that the sustained impairments in learning measured in Experiment 5 (previously Experiment 4) are dependent on an implicit system.

While on this topic, we should also note that we have made improvements to our statistical analyses of these data (as suggested by the reviewers). We now directly test how preparation time and time passage alter anterograde interference using a two-way ANOVA. Again, consistent with an impairment in implicit learning, we observed that anterograde interference worsened by limiting preparation time (Figure 9C; two-way ANOVA, main effect of preparation time, F(1,50)=11.747, p=0.001, η_p_^2^=0.19).

1.5. Across all the manipulations, I still cannot find an independent piece of evidence that task manipulations can modify learning parameters of implicit learning without resorting to model assumptions. I would like to know what the authors take on this critical issue. Is it related to how error sensitivity is defined?

We appreciate these questions. Please refer to Points 3, 7, 1-2, 1-3, and 1-4 for a detailed response to these concerns. Let us summarize and put these points into perspective.

Our paper has several components, which were not carefully delineated in our original manuscript. In our revised manuscript we now divide our Results into 4 separate sections. The initial 2 sections show direct evidence for the competition theory and compare it to alternate models. The fourth section shows limitations in the competition model. The concerns noted by the reviewer above, are addressed in Part 3.

Part 3 begins with our analysis of the Haith et al. (2015) dataset. In our revised manuscript we fit both the competition and independence models to these data. We assume that limited reaction time trials reveal implicit learning. Here we are not trying to claim that either model is ‘correct’, but rather, we demonstrate that each model offers a different interpretation of the same experimental data. Namely, the competition model suggests that savings is due to an increase in both implicit and explicit error sensitivity. However, the independent model suggests that only the explicit system contributes to savings. The reviewer is right that these results depend on the structure of each model. In the independence model, that implicit system is only driven by the rotation. Because implicit learning (i.e., limited preparation time angle) is the same across Exposures 1 and 2, this model is ‘forced’ to predict that implicit error sensitivity is unchanged (because the implicit system learns the same in each exposure). However, in the competition model, the implicit system is driven not solely by the rotation, but also responds to explicit strategy. Because aiming increases during the second exposure, the competition model holds that the implicit system should have a smaller driving force. Thus, this model is ‘forced’ to predict that implicit error sensitivity must increase during the second exposure; the smaller driving force must be counteracted by a larger error sensitivity in order to achieve the same implicit learning level.

In sum, this initial section does not show that the competition model, or the independence model is true. Rather, it shows that the model you choose has vastly different implications about the data. We hope that the revised text on Lines 510-560 better explains this analysis and the way it should be interpreted.

Now that we know that explicit strategies could hypothetically impede our ability to measures changes in implicit learning, how can we detect them? As in our original manuscript, Part 3 (Figure 7 in revised paper) uses simulations to demonstrate that implicit learning could be better examined after suppressing explicit strategies. In the rest of this section, we do this in two scenarios, savings and interference, and compare our results to past data (i.e., Haith et al., 2015 and Lerner et al., 2020).

In Experiment 4, we limit preparation time on every trial to suppress explicit strategy. As discussed in Points 3, 7, and 1-2, we have now confirmed that our limited preparation time methodology prevents explicit strategy use. This gives us the ability to detect changes in implicit learning using limited preparation time trials. In Experiment 4, we measure changes in limited preparation time learning in two ways: measuring learning rate (via an exponential curve) and measuring the average change in reach angle immediately following rotation onset. As suggested by the reviewers, we now show the individual participant results in revisions to Figure 8 (previously Figure 6).

As outlined in Point 1-3 above, we updated our statistical approach to detect whether limited preparation time learning exhibited a change in learning rate in Haith et al., as well as Experiment 4 (our new savings Experiment, previously Experiment 3). The mixed-ANOVA showed savings on limited preparation time trials both in our rate measure (Figure 8C top; mixed-ANOVA exposure no. by experiment type interaction, F(1,22)=5.993, p=0.023; significant interaction followed by one-way rm-ANOVA across exposures: Haith et al. with F(1,13)=1.109, p=0.312, η_p_^2^=0.079; Experiment 4 with F(1,9)=5.442, p=0.045, η_p_^2^=0.377) and in the average reach angle following rotation onset, but not in Haith et al., 2015 (Figure 8C, bottom; mixed-ANOVA exposure no. by experiment interaction, F(1,22)=4.411, p=0.047; interaction followed by 1-way rm-ANOVA across exposures: Haith et al. with F(1,13)=0.029, p=0.867, η_p_^2^=0.002; Experiment 4 with F(1,9)=11.275, p=0.008, η_p_^2^=0.556).

Let us add up these pieces. Savings occurred on limited preparation time trials in Experiment 4, not Haith et al. Limiting preparation time prevents explicit strategy use. Thus, the implicit system must have caused the savings in Experiment 4. The competition model suggests this savings was not detected in Haith et al., due to the large concomitant changes in the explicit system’s response to the rotation. Please note that these results do not have to do with how error sensitivity is defined. When analyzing savings in Figure 8, we are not using a model-based approach. Rather, we are simply measuring learning rate via averages (Figure 8C, bottom) and an exponential curve (Figure 8C, top).

Finally, similar arguments hold in our anterograde interference experiment (Experiment 5, previously Experiment 4). In this experiment we again limited preparation time. We now know this isolates the implicit system given our additional control experiments. We found that our paradigm produced a strong learning impairment that did not resolve even after 24 hours. In previous work, we found that interference had lifted after 24 hours. Thus, the stronger interference observed in Experiment 5 suggests that explicit strategies can mitigate the lingering deficits that occur in the implicit system. In our revised paper, we have now added participant results to Figure 9 (previously Figure 7). We have also updated our statistics. Consistent with our original conclusions, we found that both low preparation time (Experiment 5) and high preparation time (earlier Lerner et al., 2020 study) trials exhibited decreases in learning rate which improved with the passage of time (Figure 9C; two-way ANOVA, main effect of time delay, F(1,50)=5.643, p=0.021, η_p_^2^=0.101). These impairments were greatly exacerbated by limiting preparation time (Figure 9C; two-way ANOVA, main effect of preparation time, F(1,50)=11.747, p=0.001, η_p_^2^=0.19). Again, these data did not rely on using a model to estimate error sensitivity; our interference metrics were entirely empirical.

Summary

We think the reviewer’s line of inquiry is very important. We have made several additions to the revised manuscript to address them, and related concerns.

1. We have added two new experiments (Limit PT and decay-only groups in Experiment 3). Here we show that limiting preparation time isolates implicit learning. We more clearly detail for the reader why it is important to test the assumption that limiting preparation time suppresses explicit strategy.

2. We better delineate our various result sections and discuss which support the competition model and which are hypothetical predictions.

3. We now illustrate independence model results in Figure 6, in our Haith et al., analysis.

4. We now provide individual participant data in Figures 6, 8, and 9 to increase the transparency of our various Results sections.

5. We have updated our statistical methods, and now using two-way ANOVA and mixed ANOVA to better detect savings and interference in our studies.

We hope that these new experiments and analyses lend further credence to our hypothesis that implicit error sensitivity can change over time. We do not believe our experiments are a final conclusion to this controversial topic. Rather, we hope to provide enough preliminary evidence to motivate the need for future experiments: especially why our results contrast with implicit behavior in the invariant error clamp paradigm (Avraham et al., 2021), which we now discuss on Lines 879-890 in the revised paper.

Reviewer #2:The paper provides a thorough computational test of the idea that implicit adaptation to rotation of visual feedback of movement is driven by the same errors that lead to explicit adaptation (or strategic re-aiming movement direction), and that these implicit and explicit adaptive processes are therefore in competition with one another. The results are incompatible with previous suggestions that explicit adaptation is driven by task errors (i.e. discrepancy between cursor direction and target direction), and implicit adaptation is driven by sensory prediction errors (i.e. discrepancy between cursor direction and intended movement direction). The paper begins by describing these alternative ideas via state-space models of trial by trial adaptation, and then tests the models by fitting them both to published data and to new data collected to test model predictions.The competitive model accounts for the balance of implicit-explicit adaptations observed previously when participants were instructed how to counter the visual rotation to augment explicit learning across multiple visual rotation sizes (20, 40 and 60 degrees; Neville and Cressman, 2018). It also fits previous data in which the rotation was introduced gradually to augment implicit learning (Saijo and Gomi, 2010). Conversely, a model based on independent adaptation to target errors and sensory prediction errors could not reproduce the previous results. The competitive model also accounts for individual participant differences in implicit-explicit adaptation. Previous work showed that people who increased their movement preparation time when faced with rotated feedback had smaller implicit reach aftereffects, suggesting that greater explicit adaptation led to smaller implicit learning (Fernandes-Ruis et al., 2011). Here the authors replicated this general effect, but also measured both implicit and explicit learning under experimental manipulation of preparation time. Their model predicted the observed inverse relationship between implicit and explicit adaptation across participants.The authors then turned their attention to the issue of persistent sensorimotor memory – both in terms of savings (or benefit from previous exposure to a given visual rotation) and interference (or impaired performance due to exposure to an opposite visual rotation). This is a topic that has a long history of controversy, and there are conflicting reports about whether or not savings and interference rely predominantly on implicit or explicit learning. The competition model was able to account for some of these unresolved issues, by revealing the potential for a paradoxical situation in which the error sensitivity of an implicit learning process could increase without observable increase in implicit learning rate (or even a reduction in implicit learning rate). The authors showed that this paradox was likely at play in a previous paper that concluded that saving is entirely explicit (Haith et al., 2015), and then ran a new experiment with reduced preparation time to confirm some recent reports that implicit learning can result in savings. They used a similar approach to show that long-lasting interference can be induced for implicit learning.Finally, the authors considered data that has long been cited to provide evidence that implicit adaptation is obligatory and driven by sensory prediction error (Mazzoni and Krakauer, 2006). In this previous paper, participants were instructed to aim to a secondary target when the visual feedback was rotated so that the cursor error towards the primary target would immediately be cancelled. Surprisingly, the participants' reach directions drifted even further away from the primary target over time – presumably in an attempt to correct the discrepancy between their intended movement direction and the observed movement direction. However, a competition model involving simultaneous adaptation to target errors from the primary target and opposing target errors from the secondary target offers an alternative explanation. The current authors show that such a model can capture the implicit "drift" in reach direction, suggesting that two competing target errors rather than a sensory prediction error can account for the data. This conclusion is consistent with more recent work by Taylor and Ivry (2011), which showed that reach directions did not drift in the absence of a second target, when participants were coached to immediately cancel rotation errors in advance. The reason for small implicit aftereffect reported by Taylor and Ivry (2011) is open for interpretation. The authors of the current paper suggest that adaptation to sensory prediction errors could underlie the effect, but an alternative possibility is that there is an adaptive mechanism based on reinforcement of successful action.Overall, this paper provides important new insights into sensorimotor learning. Although there are some conceptual issues that require further tightening – including around the issue of error sensitivity versus observed adaptation, and the issue of whether or not there is evidence that sensory prediction errors drive visuomotor rotation – I think that the paper will be highly influential in the field of motor control.2.1 Line 19: I think it is possible that more than 2 adaptive processes contribute to sensorimotor adaptation – perhaps add "at least".

Agreed. The text now reads: “Sensorimotor learning relies on at least two parallel systems…”

2.2. Lines 24-27: This section does not refer to specific context or evidence and is consequently obtuse to me.

Agreed. We have amended our abstract. The relevant passage now explains by ‘context’, we mean the visual targets presented to the participant. The relevant passage now reads: “This shared error leads to competition such that an increase in the explicit system’s adaptive response siphons away resources that are need for adaptation of the implicit system, thus reducing its learning. As a result, asymptotic learning in the implicit system varies with experimental conditions, and strategies can mask changes in implicit error sensitivity. However, adaptation behavior becomes more complex when participants are presented with multiple visual landmarks, a situation which introduces learning from sensory prediction errors in addition to target errors.”

2.3. Line 52: Again – perhaps add "at least".

Agreed. The text now reads: “Multiple lines of evidence suggest that the brain engages at least two…”

2.4. Line 68: I am not really sure what you are getting at by "implicit learning properties" here. Can you clarify?

Agreed. We have added to this passage: “…directly changing implicit learning properties (e.g., its memory retention or sensitivity to error*)*”.

2.5. Line 203: I think the model assumption that there was zero explicit aiming during washout is highly questionable here. The Morehead study showed early washout errors of less than 10 degrees, whereas errors were ~ 20 degrees in the abrupt and ~35 degrees in the gradual group for Saijo and Gomi. I think it highly likely that participants re-aimed in the opposite direction during washout in this study – what effect would this have on the model conclusions?

We agree. It is possible and probable that participants re-aimed in the opposite direction during washout. The data and model may hint at this as shown in Author response image 3. For example, note the gradual group washout period (black arrow in Author response image 3). Here we can see that the washout proceeds at a slightly faster rate than the model fit. This could be due to explicit re-aiming in the opposite direction, which is not accounted for in the model.

**Author response image 3. sa2fig3:** Model fits to Saijo and Gomi (2010).

Nevertheless, we do not believe this re-aiming alters our primary conclusions:

1. Firstly, aftereffects remain large across all washout cycles measured (see Author response image 3), which to us, indicates that there is a strong lingering difference in the implicit aftereffects between the abrupt and gradual groups.

2. Secondly, suppose strategies rapidly switched direction during the washout period. Because the gradual participants experienced a larger error during washout, they would adopt a larger strategy to counter the error, relative to the abrupt group. This would lessen the differences between the abrupt and gradual groups, which exceeded 15° on the first washout cycle. In other words, this would mean that the ‘true’ difference in implicit learning is likely *larger* between the two groups than the washout aftereffects would suggest (which would only further support the hypothesis that the gradual condition led to increased implicit learning).

This being said, we do agree with the overall sentiment that our Saijo and Gomi (2010) analysis is limited. Because participants were not instructed to stop aiming, we cannot know with certainty that the reach angles during the washout period were solely due to implicit learning. We have moved this analysis to an appendix (Appendix 2) in our revised manuscript.

Second, we have added new data to our revised paper to investigate abrupt vs. gradual learning (Experiment 1) where implicit learning is more measured via no-aiming exclusion trials. We discuss these points in greater detail in our response to Point 2. We reproduce parts of this response below, for convenience.

Adapted from Point 2 response to editorial comment

In our revised manuscript, we have collected new data to test the competition model’s prediction. These new data use similar conditions studied by Saijo and Gomi. In Experiment 1 (new data in the paper), participants were exposed to a 60° rotation, either abruptly (n=36), or in a stepwise manner (n=37) where the perturbation magnitude increased by 15° across 4 distinct learning blocks (Figure 2D). Unlike Saijo and Gomi, implicit learning was measured via exclusion during each learning period by instructing participants to aim directly towards the target. As we hypothesized, in Figure 1J, we now demonstrate that stepwise rotation onset (which yields smaller target errors) muted the explicit response to the rotation (compared to abrupt learning). The competition model predicts that decreases in explicit strategy should facilitate greater implicit adaptation. To test this prediction, we compared implicit learning across the gradual and abrupt groups during the fourth learning block, where both groups experienced the 60° rotation size (Figure 2E).

2.6. Line 382: It would be helpful to explicitly define what you mean by learning, adaptation, error sensitivity, true increases and true decreases in this discussion – the terminology and usage are currently imprecise. For example, it seems to me that according to the zones defined "truth" refers to error sensitivity, and "perception" refers to observed behaviour (or modelled adaptation state), but this is not explicitly explained in the text. This raises an interesting philosophical question – is it the error sensitivity or the final adaptation state associated with any given process that "truly" reflects the learning, savings or interference experienced by that process? Some consideration of this issue would enhance the paper in my opinion.

This is a very insightful point. We agree that the words ‘true’ and ‘perceived’ imply that error sensitivity takes precedent over total adaptation. Rather, we want to know is whether changes in total adaptation ‘match’ changes in error sensitivity. Accordingly, we have revised our text and labels here. What we previously called a ‘True increase’ we now refer to as a ‘Matching increase’. What we previously called a ‘Perceived increase’ we now refer to as a ‘Mismatching increase’. What we previously called a ‘True decrease’, we now refer to as a ‘Matching decrease’. And finally, what we previously called a ‘Perceived decrease’ we now refer to as a ‘Mismatching decrease’. The revised text is now described on Lines 587-594 (and also Figure 7).

2.7. Figure 5 label: Descriptions for C and D appear inadvertently reversed – C shows effect of enhancing explicit learning and D shows effect of suppressing explicit learning.

Thank you for catching this, it is now fixed.

2.8. Line 438: The method of comparing learning rate needs justification (i.e. comparing from 0 error in both cases so that there is no confound due to the retention factor).

This is an excellent point. We repeated this analysis, but fit the reach angle only after the zero crossing during the second rotation exposure. To do this we started by fitting the exponential model to all the data. We used the model parameters to then estimate the zero-crossing point of the data, as the zero crossing point of the exponential fit (rounded to the nearest integer cycle). Our results were unchanged; two-way ANOVA exhibited a significant effect of both time between exposures and preparation time (two-way ANOVA, main effect of time delay, F(1,50)=4.23, p=0.045, η_p_^2^=0.067; main effect of preparation time, F(1,50)=8.303, p=0.006, η_p_^2^=0.132).

This control analysis is now described on Lines 679-682 in the revised paper: “This result was unrelated to initial differences in error across rotation exposures; we obtained analogous results (see Methods) when learning rate was calculated after the ‘zero-crossing’ in reach angle (two-way ANOVA, main effect of time delay, F(1,50)=4.23, p=0.045, η_p_^2^=0.067; main effect of prep. time, F(1,50)=8.303, p=0.006, η_p_^2^=0.132).”

2.9. Line 539: I am not sure if I agree that the implicit aftereffect in the no target group need reflect error-based adaptation to sensory prediction error. It could result from a form of associative learning – where a particular reach direction is associated with successful target acquisition for each target. The presence of an adaptive response to SPE does not fit with any of the other simulations in the paper, so it seems odd to insist that it remains here. Can you fit a dual error model to the Taylor and Ivry (single-target) data? I suspect it would not work, as the SPE should cause non-zero drift (i.e. shift the reach direction away from the primary target).

We agree that an associative learning mechanism may very well contribute to the implicit aftereffect, not solely in the Taylor and Ivry (2011) study, but in many motor control studies. In truth, we do not know the precise nature of the other forces driving implicit learning apart from visual target error. Your comment highlights a critical need for more research and modeling in this area.

An associative learning mechanism could very well produce an aftereffect in the no aiming target group, without a drift in the hand direction during the rotation period. A dual error model could, however, also produce the same result. The implicit system would respond to the SPE and drift, but this drift could be countered by a compensatory change in strategy. This would also cause an aftereffect, but no clear drift in hand direction during adaptation. A similar phenomenon has occurred in experiments that use aiming landmarks; for example, in Taylor et al. (2014), McDougle et al. (2015), and Day et al. (2016), target errors are rapidly eliminated, but implicit adaptation continues with a concomitant decrease in strategy.

In addition, there is another group tested by Taylor and Ivry which we did not describe in our manuscript. In the ‘disappearing target’ group, a secondary aiming location vanished mid-movement. This condition led to a medium-sized drift and aftereffect. To us, it would appear that this learning is due to an SPE, and that this SPE learning can be modulated by task conditions. Recent work by Tsay et al. (2021), also suggests that SPEs contribute to implicit learning, but can be modulated by disturbances to the target.

With that said, we completely agree that it appears contradictory that SPE learning should arise in Taylor and Ivry, but not the other data sets considered in our paper. We do not know why this is. In the revised manuscript however, we highlight this paradox, and suggest three potential hypotheses. First, there may have been SPE learning across the experiments we examined, but its magnitude was much smaller than target error learning (and thus may have gone undetected). Second, it is possible that what we call SPE learning is truly the associative process the reviewer describes, not learning due to error per se, but rather reinforcing actions that resulted in the desired outcome. The third possibility is that SPE learning is highly modulated by context, and thus, total SPE learning may vary substantially across experiments. For example, in Taylor and Ivry (2010), SPE learning may have been primed even in the no-aiming-target group due to an implicit recall of the aiming landmarks encountered during the baseline period. This idea would be akin to memory-guided saccade adaptation in the visual system. This may imply that reinforcement of the target location could modulate SPE learning along a gradient. When aiming targets are never provided during the task, there is little to no SPE learning (as in our primary data sets). When an aiming target is provided during initial training but not adaptation, there may be small SPE learning levels. Providing aiming targets that disappear during the movement would promote even better recall and promote moderate SPE driven learning. This would be equivalent to the disappearing target condition in Taylor and Ivry. Finally, always showing the aiming target would lead to the strongest SPE learning levels. This context-dependent SPE learning hypothesis could be related to recent work that shows both target errors and SPEs drive implicit learning, but the implicit SPE learning modulate is modulated by distraction.

In the revised manuscripts, we address these points in depth on Lines 1063-1090 in the revised paper. We provide the relevant passage below, for ease of reference:

“However, the nature of aim-cursor errors remains uncertain. For example, while this error source generates strong adaptation when the aim location coincides with a physical target (Figure 10H, instruction with target), implicit learning is observed even in the absence of a physical aiming landmark^9^ (Figure 10H, instruction without target), albeit to a smaller degree. This latter condition may implicate an SPE learning that does not require an aiming target. Thus, it may be that the aim-cursor error in Mazzoni and Krakauer is actually an SPE that is enhanced by the presence of a physical target. In this view, implicit learning is driven by a target error module and an SPE module that is enhanced by a visual target error^4,11,86^.

These various implicit learning modules are likely strongly dependent on experimental contexts, in ways we do not yet understand. For example, Taylor and Ivry (2011) would suggest that all experiments produce implicit some SPE learning, but less so in paradigms with no aiming targets. Yet, the competition equation accurately matched single-target behavior in Figures 1-9 without an SPE learning module. It is not clear why SPE learning would be absent in these experiments. One idea may be that the aftereffect observed by Taylor and Ivry (2011) in the absence of an aiming target, was actually a lingering associative motor memory that was reinforced by successfully hitting the target during the rotation period. Indeed, such a model-free learning mechanism^87^ should be included in a more complete implicit learning model. It is currently overlooked in error-based systems such as the competition and independence equations.

Another idea is that some SPE learning did occur in the no aiming target experiments we analyzed in Figures 1-9, but was overshadowed by the implicit system’s response to target error. A third possibility is that the SPE learning observed by Taylor and Ivry (2011) was contextually enhanced by participants implicitly recalling the aiming landmark locations (akin to memory-guided saccade adaptation) provided during the baseline period. This possibility would suggest SPEs vary along a complex spectrum: (1) never providing an aiming target causes little or no SPE learning (as in our experiments), (2) providing an aiming target during past training allows implicit recall that leads to small SPE learning, (3) providing an aiming target that disappears during the movement promotes better recall and leads to medium-sized SPE learning (i.e., the disappearing target condition in Taylor and Ivry), and (4) an aiming target that always remains visible leads to the largest SPE learning levels. This context-dependent SPE hypothesis may be related to recent work suggesting that both target errors and SPEs drive implicit learning, but implicit SPE learning is altered by distraction^54^.”

2.10. Line 556: Be precise here – do you really mean SPE? It seems as though you only provided quantitative evidence of a competition between errors when there were 2 physical targets.

We agree with this criticism. In this particular passage we have altered our language to indicate that ‘other target errors in the workspace’ can also drive implicit learning. This passage now reads: “The task-error driven implicit system likely exists in parallel with other implicit processes^4,11,56^. For example, in cases where primary target errors are eliminated, implicit adaptation persists (Figure 10). These residual changes are likely due to sensory prediction errors^2,4,9,11^ as well as other target errors that remain in the workspace (Figure 10H). When these error sources oppose one another, competition between parallel implicit learning modules may inhibit the overall implicit response (Figures 10A-C).”

We recognize this is still an oversimplification, per our response to Point 2-9 above. Thus, we also expand about the nature of SPEs and their possible confusion with associative learning in an expanded discussion on Lines 1063-1090 (see Point 2-9).

2.11. Line 606: Is it necessarily the explicit response that is cached? I think of this more as the development of an association between action and reward – irrespective of what adaptive process resulted in task success. It would be nice to know what happened to aftereffects after blocks 1 and 2 in study 3. An associative effect might be switched on and off more easily – so if a mechanism of that kind were at play, I would predict a reduced aftereffect as in Huberdeau et al.

We agree but want to make sure we understand your meaning. To us, it would seem in many cases that it is hard to dissociate the associative memory described by the reviewer, and a cached explicit strategy.

It seems possible that both memories could be contextually turned on and off. However, in Experiment 3 (now Experiment 4) we limit preparation time in order to suppress explicit strategy. Thus, rapidly ‘switching off’ part of the adapted response would likely implicate an associative memory that was formed independent of explicit caching.

While this is an intriguing possibility, unfortunately we cannot test it with our current data. There was no washout period at the end of the second rotation exposure. However, we think this is a fascinating point that should be noted to the reader. Thus, we have now added a discussion about associative memory to Lines 876-878 in our revised manuscript (reproduced below):

“…with multiple exposures to a rotation, explicit responses can be expressed at lower reaction times: a process termed caching^22,45^. Thus, changes in low preparation time adaptation commonly ascribed to the implicit system, may be contaminated by cached explicit strategies. This possibility seems unlikely to have altered our results. First, it is not clear why caching would occur in Experiment 4, but not our earlier study in Haith et al.^17^ (Figure 8); these earlier data implied that caching remains limited with only two exposures to a rotation (at least during the initial exposure to the second rotation over which savings was assessed). Nevertheless, to test the caching hypothesis, we measured explicit re-aiming under limited preparation time conditions in Experiment 3. We found that our method restricted explicit re-aiming to only 2°, compared to about 12° in the standard condition (Figure 3N). Moreover, this 2° decrement in reach angle was more likely due to temporal lability in implicit learning^31,72–76^ a similar 2° decrease in reach angle occurred over the 30 sec instruction period, even when participants were not told to stop aiming (Figure 8Supplement 1). Thus, while it appears that caching played little role in our results, our results should be taken cautiously. It is critical that future studies investigate how caching varies across experimental methodologies, and how cached strategies interact with implicit learning. In addition, such experiments should dissociate these cached explicit responses from associative implicit memories that may be rapidly instantiated in the appropriate context.”

2.12. Line 720: Ref 12 IS the Mazzoni and Krakauer study, ref 11 involves multiple physical targets, and ref 21 is the Taylor and Ivry paper considered above and in the next point. None of these papers therefore require an explanation based on SPE. However, ref #17 appears to provide a more compelling contradiction to the notion that target errors drive all forms of adaptation – as people adapt to correct a rotation even when there is never a target error (because the cursor jumps to match the cursor direction). A possible explanation that does not involve SPE might be that people retain a memory of the initial target location and detect a "target error" with respect to that location.

We agree this was a poorly constructed citation. We have removed the citation to Mazzoni and Krakauer here and added the citation to Reference 17. While we agree with your point about Reference 17, we also think it may be too early to say whether learning in this ‘jumping’ target case is due to a target error or an SPE. Furthermore, there are other scenarios like the disappearing target group in Taylor and Ivry, and data recently explored by Tsay et al. (2021) that suggest there is an implicit SPE learning system which might be modulated by task conditions. Overall, we think many experiments will be needed to better understand SPE learning, and we hope that our current results motivate their need. Please see our response to Point 2-9 above, where we describe revisions to the manuscript that highlight nuance in SPE learning, as well as other associative learning mechanisms that may contribute to processes labeled as SPE learning.

2.13. Line 737: I don't agree with this – the observation of an after-effect with only 1 physical target but instructed re-aiming so that there was 0 target error, certainly implies some other process besides a visual target error as a driver of implicit learning. However, as argued above, such a process need not be driven by SPE. Model-free processes could be at play…

We agree that an associative learning mechanism could also play a role here, as discussed in Point 2-9. We have now amended this passage to replace ‘strongly implicates an SPE learning mechanism’, to “may implicate an SPE learning that does not require an aiming target.” We also discuss associative learning alongside the SPE hypothesis in more complete detail on Lines 1075-1078 in the revised paper. Please see our response to Point 2-9 above for a more complete response to these concerns.

2.14. Line 791: Some more detail on how timing accuracy was assured in the remote experiments is needed. The mouse sample rate can be approximately 250 Hz, but my experience with it is that there can be occasional long delays between samples on some systems. The delay between commands to print to the screen and the physical appearance on the screen can also be long and variable – depending on the software and hardware involved.

We absolutely agree. In Experiments 2, 4, and 5 we measured the visual delay for our robotic manipulandum experiments. On average this delay is 55 ms. When we show reaction time (e.g., Figure 3) we correct these curves by subtracting the average delay. We alluded to screen delay in the original manuscript but did not report its magnitude. We have now added this to Lines 1440-1442.

We also have now measured visual delay across many laptops and operating systems (Windows and Mac) for our laptop-based experiment. This delay was on average 154 ms. We have updated our reaction time plots (e.g., Figure 3) to correct for this delay. In addition, we have added a passage to the text: “Finally, we used a separate procedure to estimate screen delay. To do this, participants were told to tap a circle that flashed on and off in a regular, repeating cycle. Participants were told to predict the appearance of the circle, and to tap exactly as the circle appeared. Because the stimulus was predictable, the difference between the appearance time, and the participant’s button press, revealed the system’s visual delay. The average visual delay we measured was 154 ms. This average value was subtracted out in the preparation times reported in Figures 3JandM, as well as Figure 7-Supplement 1.”

2.15. Line 861: How did you specify or vary the retention parameters to create these error sensitivity maps?

As noted in Point 2-16 below, we had a typographical error in the original manuscript (had replaced ‘e’ with ‘f’ and ‘i' with ‘s’ in our retention factor parameters). We intended to report that implicit and explicit retention factors were specified as: *a_i_* = 0.9829 and *a_e_* = 0.9278. These parameters were not varied in Figure 7 (previously Figure 5), only implicit and explicit error sensitivity.

As explained in response to Point 1-14 above, Figure 6 is ‘referenced’ to Haith et al. (2015). We used the retention factors identified by the competition model’s fit to the Day 1 learning curves. We now discuss this better on Lines 578-586 in the revised paper.

2.16. Line 865: Should not the subscripts here be "e" and "I" rather than "f" and "s"?

Thank you for catching this. We have fixed this error.

2.17. Line 1009: Why were data sometimes extracted using a drawing package (presumably by eye? – please provide further details), and sometimes using GRABIT in Matlab?

We used a two-step process when acquiring data from published figures. First, we attempted to open the figure in Adobe Illustrator. Depending on how these figures were constructed and embedded, occasionally the figure could be decomposed into its layers. This permitted us to use the *x* and *y* pixel values for each data point (which appeared as an object) to interpolate the necessary data from the figure.

However, in some cases, objects and layers could not be obtained in Illustrator. In these cases, we resorted to a second step. This is when we used the graphical utility GRABIT. GRABIT allows you to interpolate data at desired locations in an image. In this process we did our best to select the center of each data point in the image to estimate the data.

We now have added a section to our revised Methods that details this process on Lines 1133-1140.

2.18. Line 1073: More detail about the timing accuracy and kinematics of movement are required.

As detailed in our response to Point 2-14, we have measured visual delay across laptops and operating systems. On average this was 154 ms. We have adjusted our reaction time data in Figure 3 by this value, and also note this on Lines 1569-1574 in the revised manuscript.

Secondly, because we specified bounds on movement duration, movements were rapid and straight (like standard in-person studies). For reference we now show 2 representative participants (1 from the No PT Limit condition and 1 from the Limit PT condition) in Figure 3-Supplement 3 in the revised manuscript.

2.19. Line 1148: Again – should not the subscripts here be e and I rather than f and s?

Thank you for catching this. We have fixed this error.

Reviewer #3:In this paper, Albert and colleagues explore the role of target error and sensory prediction error on motor adaptation. They suggest that both implicit and explicit adaptation would be driven by target error. In addition, implicit adaptation is also influenced by sensory prediction error.While I appreciate the effort that the authors have done to come up with at theory that could account for many studies, there is not a single figure/result that does not suffer from main limitations as I highlight in the major comments below. Overall, the main limitations are:I believe that the authors neglect some very relevant papers that contradict their theory and model.3.1. They did not take into account some results on the topic such as Day et al. 2016 and McDougle et al. 2017. It is true that they acknowledge it and discuss it but I am not convinced by their arguments (more on this topic in a major comment about the discussion below).

We agree that a generalization analysis is important. Please see our response in Point 1 above and several points below. For convenience, we have adapted relevant passages from Point 1 below.

Adapted excerpts from Point 1

In our revised manuscript we directly compare the competition model to an alternative SPE generalization model. Our new analysis is documented in Figure 5 in the revised manuscript. First, we empirically compare the relationship between implicit and explicit learning in our data to the generalization curves measured in past studies. In Figures 5A-C we overlay data that we collected in Experiments 2 and 3 with generalization curves measured by Krakauer et al. (2000) and Day et al. (2016). In this way, we attempt to see whether past generalization curves empirically resemble our implicit-explicit learning measures.

[…]

Summary

In our revised manuscript we compare the competition theory to an SPE generalization model. This model did not match our data:

1. Implicit learning declined 300% more rapidly than predicted by generalization studies (Figures 4C-E).

2. The implicit-explicit learning relationship did not accurately generalize across rotation sizes in the SPE generalization model (Figure 4G).

3. The gain relating implicit and explicit learning remained the same across rotation sizes, rejecting the SPE generalization model, but supporting the competition theory (Figure 4H).

Given the importance of these analyses, we have devoted a section (Part 2) to them in our revised paper.

3.2. They did not take into account the fact that there is no proof that limiting RT is a good way to suppress the explicit component of adaptation. When the number of targets is limited, these explicit responses can then be cached without any RT cost. McDougle and Taylor, Nat Com (2019) demonstrated this for 2 targets (here the authors used only 4 targets). There exist no other papers that proof that limiting RT would suppress the explicit strategy as claimed by the authors. To do so, one needs to limit PT and to measure the explicit or the implicit component. The authors should prove that this assumption holds because it is instrumental to the whole paper. The authors acknowledge this limitation in their discussion but dismiss it quite rapidly. This manipulation is so instrumental to the paper that it needs to be proven, not argued (more on this topic in a major comment about the discussion below).

We appreciate this concern. We responded to this in Point 3 (in our response to editorial summary). For convenience, we provide excepts from this response below.

Adapted excerpts from Point 3 (editorial response)

We agree that it is important to test how effectively this condition limits explicit strategy use. Therefore, we have performed two additional control studies to measure explicit strategy in the limited PT condition. First, we have now added a limited preparation time (Limit PT) group to our laptop-based study in Experiment 3 (Experiment 2 in original manuscript). In Experiment 3, participants in the Limit PT group (n=21) adapted to a 30° rotation but under a limited preparation time condition. As for the Limit PT group in Experiment 2, we imposed a strict bound on reaction time to suppress movement preparation time. However, unlike Experiment 2 (Experiment 1 in original manuscript) once the rotation ended, participants were told to stop re-aiming. This permitted us to examine whether limiting preparation time suppressed explicit strategy as intended. Our analysis of these new data is shown in Figures 3K-Q.

In Experiment 3, we now compare two groups: one where participants had no preparation time limit (No PT Limit, Figures 3H-J) and one where an upper bound was placed on preparation time (Limit PT, Figures 3K-M). We measured early and late implicit learning measures over a 20-cycle no feedback and no aiming period at the end of the experiment. The voluntary decrease in reach angle over this no aiming period revealed each participant’s explicit strategy (Figure 3N). When no reaction time limit was imposed (No PT Limit), reaiming totaled approximately 11.86° (Figure 3N, E3, black), and did not differ statistically across Experiments 2 and 3 (t(42)=0.50, p=0.621). Recall that Experiment 2 tested participants using a robotic manipulandum and Experiment 3 tested participants in a similar laptop-based paradigm. As in earlier reports, limiting reaction time dramatically suppressed explicit strategy, yielding only 2.09° of re-aiming (Figure 3N, E3, red). Therefore, our new control dataset suggests that our limited reaction time technique was highly effective at suppressing explicit strategy as initially claimed in our original manuscript.

With that said, our new data alone suggest that while explicit strategies are strongly suppressed by limiting preparation time, they are not entirely eliminated; when we limited preparation time in Experiment 3, we observed that participants still exhibited a small decrease (2.09°) in reach angle when we instructed them to aim their hand straight to the target (Figure 3L, no aiming; Figure 3N, E3, red). This ‘cached’ explicit strategy, while small, may have contributed to the 8° reach angle change measured early during the second rotation in our savings experiment (Experiment 4, Figure 8C in revised paper).

For this reason, we consider another important phenomenon in our revised manuscript: time-dependent decay in implicit learning. That is, the 2° decrease in reach angle we observed when participants were told to stop aiming in the PT Limit group in Experiment 3 may be due to time-based decay in implicit learning over the 30 second instruction period, as opposed to a voluntary reduction in strategy. To test this possibility, we collected another limited preparation group (n=12, Figure 8-Supplement 1A, decay-only, black). But this time, participants were instructed that the experiment’s disturbance was still on, and that they should continue to move the ‘imagined’ cursor through the target during the terminal no feedback period. Still, reach angles decreased by approximately 2.1° (Figure 8-Supplement 1B, black). Indeed, we detected no statistically significant difference between the change in reach angle in this decay-only condition, and the Limit PT group in Experiment 3 (Figure 8-Supplement 1B; two-sample t-test, t(31)=0.016, p=0.987).

This control experiment suggested that residual ‘explicit strategies’ we measured in the Limit PT condition, were in actuality caused by time-dependent decay in implicit learning. Thus, our Limit PT protocol appears to eliminate explicit strategy. This additional analysis lends further credence to the hypothesis that savings in Experiment 4, was primarily due to changes in the implicit system rather than cached explicit strategies.

Summary

We agree that it is important to corroborate that our method for limiting reaction time suppresses explicit learning as intended. To this end we collected additional experimental conditions in Experiment 3 in the revised manuscript (a Limit PT group and a decay-only group). Data in the Limit PT condition suggest that explicit strategies are suppressed to approximately 2° by our preparation time limit (compared to about 12° under normal conditions). Data in the decay-only group suggest that this 2° change in reach angle was not due to a cached explicit strategy but rather time-dependent decay in implicit learning. Together, these conditions demonstrate that limiting reaction time in our experiments almost completely prevents the caching of explicit strategies; our limited preparation time measures do indeed reflect implicit learning. We use these data to corroborate key predictions of the competition theory in Figures 3Q (subject-to-subject correlations between implicit and explicit learning) and 8 (savings experiment) in the revised manuscript.

3.3. They did not take into account the fact that the after-effect is a measure of implicit adaptation if and only if the participants are told to abandon any explicit strategy before entering the washout period (as done in Taylor, Krakauer and Ivry, 2011). This has important consequences for their interpretation of older studies because it was then never asked to participants to stop aiming before entering the washout period as the authors did in experiment 2.

We appreciate this concern. We responded to this concern in Point 2 above. For convenience, we have adapted passages from Point 2 below.

Adapted excerpts from Point 2 (response to editorial summary)

This criticism applies to our Saijo and Gomi analysis, our Fernandez-Ruiz et al., analysis, and our Mazzoni and Krakauer analysis. We will address each in turn.

Saijo and Gomi (2010)

Let us begin with the Saijo and Gomi (2010) analysis. In our original manuscript, we used conditions tested by Saijo and Gomi to investigate one of the competition theory’s predictions: decreasing explicit strategy for a given rotation size will increase implicit learning. Specifically, we analyzed whether gradual learning suppressed explicit strategy in this study, thus facilitating greater implicit adaptation. While our analysis was consistent with this hypothesis, as noted by the reviewers, there is a limitation; washout trials were used to estimate implicit learning, though participants were not directly instructed to stop aiming. This represents a potential error source. For this reason, in our revised manuscript, we have collected new data to test the competition model’s prediction.

These new data also examine gradual and abrupt rotations. In Experiment 1 (new data in the paper), participants were exposed to a 60° rotation, either abruptly (n=36), or in a stepwise manner (n=37) where the rotation magnitude increased by 15° across 4 distinct learning blocks (Figure 2D). Unlike Saijo and Gomi, implicit learning was measured via exclusion during each learning period by instructing participants to aim directly towards the target. As we hypothesized, in Figure 2F, we now demonstrate that stepwise rotation onset (which yields smaller target errors) muted the explicit response to the rotation (compared to abrupt learning). The competition model predicts that decreases in explicit strategy should facilitate greater implicit adaptation. To test this prediction, we compared implicit learning across the gradual and abrupt groups during the fourth learning block, where both groups experienced the 60° rotation size (Figures 2EandG).

Consistent with our hypothesis, participants in the stepwise condition exhibited a 10° reduction in explicit re-aiming (Figure 2F, two-sample t-test, t(71)=4.97, p<0.001, d=1.16), but a concomitant 80% increase in their implicit recalibration (Figure 2G, data, two-sample t-test, t(71)=6.4, p<0.001, d=1.5). To test whether these changes in implicit learning matched the competition model, we fit the independence Equation (Equation 5) and the competition Equation (Equation 4) to the implicit and explicit reach angles measured in Blocks 14, across the stepwise and abrupt conditions, while holding implicit learning parameters constant. In other words, we asked whether the same parameters (*a_i_* and *b_i_*) could parsimoniously explain the implicit learning patterns measured across all 5 conditions (all 4 stepwise rotation sizes plus the abrupt condition). As expected, the competition model predicted that implicit learning would increase in the stepwise group (Figure 2G, comp., 2-sample t-test, t(71)=4.97, p<0.001), unlike the SPE-only learning model (Figure 2G, indep.).

Thus, in our revised manuscript we present new data that confirm the hypothesis we initially explored in the Saijo and Gomi (2010) dataset: decreasing explicit strategy enhances implicit learning. These new data do not present the same limitation noted by the reviewers. Lastly, it is critical to note that while stepwise participants showed greater implicit learning, their total adaptation was approximately 4° lower than the abrupt group (Figure 2E, right-most gray area (last 20 trials); two-sample t-test, t(71)=3.33, p=0.001, d=0.78). This surprising phenomenon is predicted by the competition equation. When strategies increase in the abrupt rotation group, this will tend to increase total adaptation. However, larger strategies leave smaller errors to drive implicit learning. Hence, greater adaptation will be associated with larger strategies, but less implicit learning. Indeed, the competition model predicted 53.47° total learning in the abrupt group but only 50.42° in the stepwise group. Recall that, we described this paradoxical phenomenon at the individual-participant level in Point 1 above. Note that this pattern was also observed by Saijo and Gomi. Surprisingly, when participants were exposed to a 60° rotation in a stepwise manner, total adaptation dropped over 10°, whereas the aftereffect exhibited during the washout period nearly doubled.

Summary

In our revised paper we have removed this Saijo and Gomi analysis from the main figures, and instead use this as supportive data in Figure 2-Supplement 2, and also in Appendix 2. We now state, “It is interesting to note that these implicit learning patterns are broadly consistent with the observation that gradual rotations improve procedural learning^34,43^, although these earlier studies did not properly tease apart implicit and explicit adaptation (see the Saijo and Gomi analysis described in Appendix 2).”

We have replaced this analysis with our new data in Experiment 1. In these new data, abrupt and gradual learning was compared using the appropriate implicit learning measures. These new data are included in both Figures 1 and 2 in the revised manuscript. Importantly, these new data point to the same conclusions we reached in our initial Saijo and Gomi analysis.

Mazzoni and Krakauer (2006)

The reviewers note that the uninstructed group in Mazzoni and Krakauer was not told to stop aiming during the washout period. We agree with the potential error source. However, it is important to note that these data were not included to directly support the competition model, but as a way to demonstrate its limitations. For example, in Mazzoni and Krakauer (2006) our simple target error model cannot reproduce the measured data. That is, in the instructed group implicit learning continued despite eliminating the primary target error. Thus, in more complicated scenarios where multiple visual landmarks are used during adaptation, the learning rules must be more sophisticated than the simple competition theory. Here we showed that one possibility is that both the primary target and the aiming target drive implicit adaptation via two simultaneous target errors. Our goal by presenting these data (and similar data collected by Taylor and Ivry, 2011) was to emphasize that our work is not meant to support an either-or hypothesis between target error and SPE learning, but rather the more pluralistic conclusion that both error sources contribute to implicit learning in a condition-dependent manner.

In our revised manuscript, we now better appreciate this conclusion and its potential limitation in the following passage: “It is important, however, to note a limitation in these analyses. Our earlier study did not employ the standard conditions used to measure implicit aftereffects: i.e., instructing participants to aim directly at the target, and also removing any visual feedback. Thus, the proposed dual-error model relies on the assumption that differences in washout were primarily related to the implicit system. These assumptions need to be tested more completely in future experiments.

In summary, the conditions tested by Mazzoni and Krakauer show that the simplistic idea that adaptation is driven by only one target error, or only one SPE, cannot be true in general^54^. We propose a new hypothesis that when people move a cursor to one visual target, while aiming at another visual target, each target may partly contribute to implicit learning. When these two error sources conflict with one another, the implicit learning system may exhibit an attenuation in total adaptation. Thus, implicit learning modules may compete with one another when presented with opposing errors.”

Fernandez-Ruiz et al. (2011)

As noted by the reviewers, our original manuscript also reported data from a study by Fernandez-Ruiz et al. (2011). Here, we showed that participants in this study who increased their preparation time more upon rotation onset tended to exhibit a smaller aftereffect. We used these data to test the idea that when a participant uses a larger strategy, they inadvertently suppress their own implicit learning. However, we agree with the reviewers that this analysis is problematic because participants were not instructed to stop aiming in this earlier study. Thus, we have removed these data in the revised manuscript. This change has little to no effect on the manuscript, given that individual-level correlations between implicit and explicit learning are tested in Experiments 1, 2, and 3 in the revised paper (Figures 3-5), and are also reported from earlier works (Maresch et al., 2021; Tsay et al., 2021) in Figure 4-Supplements 1GandI, where implicit learning was more appropriately measured via exclusion. Note that all 5 studies support the same ideas suggested by the Fernandez-Ruiz et al. study: increases in explicit strategy suppress implicit learning.

2.4. There are also major problems with statistics/design: I disagree with the authors that 10-20 people per group (in their justification of the sample size in the second document) is standard for motor adaptation experiments. It is standard for their own laboratories but not for the field anymore. They even did not reach N=10 for some of the reported experiments (one group in experiment 1). Furthermore, this sample size is excessively low to obtain reliable estimation of correlations. Small N correlation cannot be trusted: https://garstats.wordpress.com/2018/06/01/smallncorr/

We agree that some groups we collected are limited by small sample sizes. With that said, it is important to note that many recent published results are based on sample sizes similar to ours. For example, even in the critical literature you cite in your comments sample sizes were nearly 10 participants: (1) in Kim et al., (2018) there are 10 participants in each group. (2) In Morehead et al. (2017) there were 12 participants in each group. (3) In Day et al. (2016) there were 10 participants each group (7 different groups total).

That being said, we have made substantial additions to the manuscript to corroborate our main results in data sets that have larger samples. These new data sets include:

1. Experiment 1, added a stepwise learning group: n=37

2. Experiment 1, added an abrupt learning group: n=36

3. Experiment 3, no preparation time limit condition increased to n=35

4. Experiment 3, added a limited preparation time group: n=21

5. Figure 1, added new analysis of Tsay et al., 2021 (n=25/group)

6. Experiment 3, added a no-decay group: n=12

7. Figure 4-Supplement 2, added new analysis of Maresch et al., 2021 (n=40)

These new data corroborate and extend the conclusions we reached in our initial paper. Critically, we now reproduce our correlation analysis in Figure 3H (now Figure 3G) within our more highly powered experiments. For example, we show the same match between the competition model and data across participants in Experiment 3 (n=35, Figure 1Q). We also observe strong negative implicit-explicit correlations that agree with the competition model in the stepwise condition in Experiment 1 (n=37, Figure 4F). Similar correlations were also detected in the 60° rotation conditions in Maresch et al., 2021 (n=40, Figure 4-Supplement 2G) and Tsay et al., 2021 (Figure 4-Supplement 2I, n=25) data sets.

Finally, our revised paper illustrates that implicit and explicit correlations with total adaptation also exhibit a match to the competition model (Experiment 3, n=35, Figures 4AandB; Experiment 1, n=70, Figure 4-Supplements 2BandE; n=40, Maresch et al., 2021, Figure 4-Supplements 2AandD; n-25, Tsay et al., 2021, Figure 4-Supplements 2CandF).

In sum, we have added 7 new experiments to our revised manuscript, totaling 264 new participants. We use highly powered studies (e.g., n=25, n=35, n=40, n=70) to corroborate subject-to-subject correlations between implicit and explicit learning (Figure 3H in original manuscript). We also now consider implicit-total and explicit-total correlations across all these studies in the revised manuscript.

3.5. Justification of sample size is missing. What was the criteria to stop data collection? Why is the number of participants per group so variable (N=9 and N=13 for experiment 1, N=17 for experiment 2, N=10 for experiment 3, N=10 ? per group for experiment 4). Optional stopping is problematic when linked to data peeking (Armitage, McPherson and Rowe, Journal of the Royal Statistical Society. Series A (General), 1969). It is unclear how optional stopping influences the outcome of the statistical tests of this paper.

We did not engage in optional stopping. Unfortunately, our data collection was interrupted by the Covid19 pandemic. For this reason, we started using a laptop-based study design in Experiment 3. Furthermore, the primary data collector (Jihoon Jang) was not involved in data analysis (Scott Albert).

Nevertheless, our sample sizes are similar to other recent studies that also investigate error sources that drive implicit learning. These are studies the reviewer cites in their criticisms: (1) Kim et al., (2018): 10 participants in each group, (2) Day et al. (2016): 10 participants per group, (3) Morehead et al. (2017): 12 participants per group.

The primary exception is data in Experiment 2, where only 9 participants were recruited. For this reasons, in the revised paper we corroborate the subject-level correlations in Experiment 2, in Experiment 3, where n=35 participants. In addition, we analyze several datasets in our paper that were collected independently across many labs (Shadmehr Lab, Krakauer Lab, Henriques Lab, Ivry Lab, Donchin Lab, and the Della-Maggiore Lab).

Thus, we have done our best to verify our data sets and the competition model predictions across several studies, despite the challenges presented by the Covid-19 pandemic.

3.6. The authors should follow the best practices and add the individual data to all their bar graphs (Rousselet et al. EJN 2016).

We agree. Individual participant data are now provided for each experiment in the revised manuscript where possible:

1. Participants in Experiment 1 (Figures 1H, 1J, 2E, 3-Supp. 2, 5D, 4-Supps. 1B,E,H, 4-Supps. 3A,C,E, 4-Supp. 3B)

2. Participants in Experiment 2 (Figures 3G, 3N, 5A, 5B, Figure 3-Supplement 2)

3. Participants in Experiment 3 (Figures 3N, 3O, 3P, 3Q, 4G, 4H, 5A, 5C, 4-Supplements 2A,C,E, Figure 8-Supp. 1B)

4. Participants in Experiment 4 (Figure 8C)

5. Participants in Experiment 5 (Figure 9C)

6. Participants in Maresch et al., 2021 (Figure 4-Supplements 1A,D,G)

7. Participants in Tsay et al., 2021 (Figures 1M, 1O, 4-Supplements 1C,F,I, 4-Supps. 2A,C,E, 4-Supp. 3A)

8. Participants in Lerner et al., 2020 (Figure 9C)

9. Participants in Haith et al., 2015 (Figures 6B,C,D,E, 8C)

3.7. Missing interactions (Nieuwenhuis et al., Nature Neuro 2011), misinterpretation of non-significant p-values (Altman 1995 https://www.bmj.com/content/311/7003/485)

We appreciate this suggestion. We have now updated all relevant analyses in our revised manuscript. We used either a two-way ANOVA, two-way repeated measures ANOVA, or mixed-ANOVA depending on the experimental design. When interactions were statistically significant, we next measured simple main effects via one-way ANOVA. We outline this on Lines 1195-1207. All conclusions reached in the original manuscript remained the same using this updated statistical procedure. These include:

1. When measuring savings in Haith et al., we observed that the learning rate increased during the second exposure on high preparation time trials, but not on low preparation time trials (Figure 5B, right; two-way rm-ANOVA, prep. time by exposure number interaction, F(1,13)=5.29, p=0.039; significant interaction followed by one-way rm-ANOVA across Days 1 and 2: high preparation time with F(1,13)=6.53, p=0.024, η_p_^2^=0.335; low preparation time with F(1,13)=1.11, p=0.312, η_p_^2^=0.079). We corroborate this rate analysis by measuring early changes in reach angle (first 40 trials following rotation onset) across Days 1 and 2 (Figure 5C, left and middle). Only high preparation time trials exhibited a statistically significant increase in reach angles, consistent with savings (Figure 5C, right; two-way rm-ANOVA, preparation time by exposure interaction, F(1,13)=13.79, p=0.003; significant interaction followed by one-way rm-ANOVA across days: high preparation time with F(1,13)=11.84, p=0.004, η_p_^2^=0.477; low preparation time with F(1,13)=0.029, p=0.867, η_p_^2^=0.002).

2. When comparing implicit and explicit error sensitivities predicted by the competition model, the model predicted that both implicit system and explicit systems exhibited a statistically significant error sensitivity increase (Figure 5D, right; two-way rm-ANOVA, within-subject effect of exposure no., F(1,13)=10.14, p=0.007, η_p_^2^=0.438; within-subject effect of learning process, F(1,13)=0.051, p=0.824, η_p_^2^=0.004; exposure no. by learning process interaction, F(1,13)=1.24, p=0.285).

3. When comparing implicit and explicit error sensitivities predicted by the independence model, the model predicted that only the explicit system exhibited a statistically significant increase in error sensitivity (Figure 5E; two-way rm-ANOVA, learning process (implicit vs explicit) by exposure interaction, F(1,13)=7.016, p=0.02; significant interaction followed by one-way rm-ANOVA across exposures: explicit system F(1,13)=9.518, p=0.009, η_p_^2^=0.423; implicit system with F(1,13)=2.328, p=0.151, η_p_^2^=0.152).

4. When measuring savings under limited preparation time in Experiment 4, we still observed the opposite outcome to Haith et al., 2015. Notably, low preparation time learning rates increased by more than 80% in Experiment 4 (Figure 7C top; mixed-ANOVA exposure number by experiment type interaction, F(1,22)=5.993, p=0.023; significant interaction followed by one-way rm-ANOVA across exposures: Haith et al. with F(1,13)=1.109, p=0.312, η_p_^2^=0.079; Experiment 4 with F(1,9)=5.442, p=0.045, η_p_^2^=0.377). Statistically significant increases in reach angle were detected immediately following rotation onset in Experiment 4 (Figure 7B, bottom), but not our earlier data (Figure 7C, bottom; mixed-ANOVA exposure number by experiment interaction, F(1,22)=4.411, p=0.047; significant interaction followed by one-way rm-ANOVA across exposures: Haith et al. with F(1,13)=0.029, p=0.867, η_p_^2^=0.002; Experiment 4 with F(1,9)=11.275, p=0.008, η_p_^2^=0.556).

5. Lastly, we updated our anterograde interference analysis in Figure 8. While both low preparation time and high preparation time trials exhibited decreases in learning rate which improved with the passage of time (Figure 8C; two-way ANOVA, main effect of time delay, F(1,50)=5.643, p=0.021, η_p_^2^=0.101), these impairments were greatly exacerbated by limiting preparation time (Figure 8C; two-way ANOVA, main effect of preparation time, F(1,50)=11.747, p=0.001, η_p_^2^=0.19).

With regards to non-significant results. We agree that it is problematic when claims are made that two samples are the same, when no statistically significant difference is detected. We avoid such language in our revised manuscript.

3.8. Given these main limitations, I don't think that the authors have a convincing case in favor of their model and I don't think that any of these results actually support the error competition model.

We are very grateful to the reviewers for their incredibly detailed and insightful comments. We hope that the extensive changes we have made to the manuscript address your various concerns.

Detailed major comments per equation/figure/section:3.9. Equation 2: the authors note that the sensory prediction error is "anchored to the aim location " (line 104-105) which is exactly what Day et al. 2016 and McDougle et al. 2017 demonstrated. Yet, they did not fully take into account the implication of this statement. If it is so, it means that the optimal location to determine the extent of implicit motor adaptation is the aim location and not the target. Indeed, if the SPE is measured with respect to the aim location and is linked to the implicit system, it means that implicit adaptation will be maximum at the aim location and, because of local generalization, the amount of implicit adaptation will decay gradually when one wants to measure this system away of that optimal location (Figure 7 of Day et al). This means that, the further the aiming direction is from the target, the smaller the amount of implicit adaptation measured at the target location will be. This will result in an artificial negative correlation between the explicit and implicit system without having to relate to a common error source.

We appreciate this concern. We address these points in great detail in Points 1 and 3-1 above.

3.10. Equation 3: the authors do not take into account results from their lab (Marko et al., Herzfeld) and from others (Wei and Kording) that show that the sensitivity to error depends on the size of the rotation.

Yes, in three cases in our main figures, we assume that error sensitivity is the same across experimental groups. This occurs in our Neville and Cressman analysis, our Tsay et al. analysis, and our Experiment 1 analysis. These are now included in Figure 1 in the revised manuscript. We make this assumption to reduce model complexity. That is, with the assumption that error sensitivity is the same across groups, the model only has a single free parameter.

While we agree with the heart of the reviewer’s criticism, we should note an important correction. In our previous work we have shown that error sensitivity varies with error size, not the perturbation’s size. This is an important distinction. For example, in our Neville and Cressman analysis, rotation sizes vary between 20° and 60°. Yet, the terminal asymptotic target error across these groups varies less than 5° (while the rotation varies over 40°). For the largest variation in rotation conditions, rotation size varies by 75° in Tsay et al., but the terminal error varies only 13°. Thus, in the competition model, it is reasonable to assume that error sensitivities are similar across groups, because differences in their terminal errors are much smaller than differences in their rotation sizes.

With that being said, there is another potentially more important reason why assuming that implicit error sensitivity is constant across conditions has little effect on competition model predictions. The reason why error sensitivity plays a role in the competition equation, is due to the implicit system’s learning gain. This gain depends on implicit error sensitivity and retention according to: π = *b_i_*(1-*a_i_*+*b_i_*)^-1^. In other words, it is not error sensitivity that determines steady-state implicit learning, but the implicit learning gain.

Fortunately, this implicit learning gain responds weakly to changes in error sensitivity (which appears in both the numerator and denominator). For example, let us suppose that participants in the strategy group exhibited an implicit error sensitivity of 0.3, but in the no-strategy group, implicit error sensitivity was only 0.2. For an implicit retention factor of 0.9565 (see Methods in revised paper), the no-strategy learning gain would be 0.821 and the strategy learning gain would be 0.873. Thus, even though implicit error sensitivity was 50% larger in the strategy participants, the competitive implicit learning gain would change only 6.3%. For an even more extreme case where implicit error sensitivity was double (0.4) in the strategy group, this would still only lead to a 9.8% change in the competitive implicit learning gain.

Thus, while we appreciate the reviewer’s concern, our assumption that implicit error sensitivity is equal across learning conditions in Figure 1, is expected to have little effect on the model predictions. At the same time, it helps us to reduce model complexity, decreasing the probability of over-fitting. We explained here that the learning gain in the competition model is very insensitive to changes in implicit error sensitivity: even doubling error sensitivity increases the competition model’s learning gain by less than 10%. These new points have been added to Lines 1946-1973 in the revised paper.

3.11. Equation 5: In this equation, the authors suggest that the steady-state amount of implicit adaptation is directly proportional to the size of the rotation. Thanks to a paradigm similar to Mazzoni and Krakauer 2006, the team of Rich Ivry has demonstrated that the implicit response saturates very quickly with perturbation size (Kim et al., communications Biology, 2018, see their Figure 1).

We provide a thorough response to this concern in Point 6B (editor summary response). For convenience, we provide relevant excerpts from this response below.

Adapted excerpts from Point 6B in editor summary

These concerns are critical. We agree with the reviewers that Morehead et al., 2017 and Kim et al., 2018 suggest that implicit learning saturates quickly with rotation size. With that said, these studies both use invariant error-clamp perturbations, not standard visuomotor rotations. We are not sure that the implicit properties observed in an invariant error context apply to the conditions we consider in our manuscript. In our revised manuscript, we consider variations in steady-state implicit learning across two new data sets: (1) stepwise adaptation in Experiment 1 and (2) non-monotonic implicit responses in Tsay et al., 2021. As we will demonstrate below, our data and re-analysis strongly indicate that the implicit system does not always saturate with changes in rotation size. Rather, the implicit system exhibits a complicated response to both rotation size and explicit strategies, which together yield three implicit learning phenotypes:

1. Saturation in steady-state implicit learning despite increasing rotation size (analysis reported in original paper using Neville and Cressman, 2018).

2. Scaling in steady-state implicit learning with increasing rotation size (new analysis using stepwise learning in Experiment 1).

3. Non-monotonic (quadratic) steady-state implicit behavior due to increases in rotation magnitude (new analysis added which uses data from Tsay et al., 2021).

[…]

In sum, we now add two additional datasets to the revised manuscript: data collected using a stepwise rotation in Experiment 1, and data recently collected by Tsay et al., 2021. Collectively, these new data show that steady-state implicit learning is complex, exhibiting at least three contrasting phenotypes: saturation, scaling, and non-monotonicity. Remarkably, the competition theory provides a way to account for each of these patterns. Thus, the competition model does accurately describe how implicit learning varies due to both changes in rotation size and explicit strategy, at least in standard rotation paradigms. However, these rules do not explain implicit behavior in invariant error-clamp paradigms. In the revised manuscript, we delve further into a comparison between standard rotation learning and invariant error learning in our Discussion (Lines 911-941). The relevant passage is provided below:

“Although the implicit system is altered in many experimental conditions, one commonly observed phenomenon is its invariant response to changes in rotation size^3,31,35,36,40^. For example, in the Neville and Cressman^31^ dataset examined in Figure 1, total implicit learning remained constant despite tripling the rotation’s magnitude. While this saturation in implicit learning is sometimes interpreted as a restriction in implicit adaptability, this rotation-insensitivity may have another cause entirely: competition. That is, when rotations increase in magnitude, rapid scaling in the explicit response may prevent increases in total implicit adaptation. Critically, in the competition theory, implicit learning is driven not by the rotation, but by the residual error that remains between the rotation and explicit strategy. Thus, when we used gradual rotations to reduce explicit adaptation (Experiment 1), prior invariance in the implicit response was lifted: as the rotation increased, so too did implicit learning^37^ (Figure 1I). The competition theory could readily describe these two implicit learning phenotypes: saturation and scaling (Figures 1GandL). Furthermore, it also provided insight as to why implicit learning can even exhibit a non-monotonic response, as in Tsay et al. (2021)^36^. All in all, our data suggest that implicit insensitivity to rotation size is not due to a limitation mplycit learning, but rather a suppression created by competition with explicit strategy.

With that said, this competitive saturation in implicit adaptation should not be conflated with the upper limits in implicit adaptation that have been measured in response to invariant errors^3,11,40^. In this latter condition, implicit adaptation reaches a ceiling whose value varies somewhere between 15 degrees^3^ and 25 degrees^40^ across studies. In these experiments, participants adapt to an error that remains constant and is not coupled to the reach angle (thus, the competition theory cannot apply). While the state-space model naturally predicts that total adaptation can exceed the error size which drives learning in this errorclamp condition (as is observed in response to small error-clamps), it cannot explain why asymptotic learning is insensitive to the error’s magnitude. One idea is that proprioceptive signals^40,83^ may eventually outweigh the irrelevant visual errors in the clamped-error condition, thus prematurely halting adaptation. Another possibility is that implicit learning obeys the state-space competitive model described here, but only up until a ceiling that limits total possible implicit corrections. Indeed, in Experiment 1, we produced scaling in the implicit system in response to rotation size, but never evoked more than about 22° of implicit learning. However, when we used similar gradual conditions in the past to probe implicit learning^37^, we observed about 32° implicit learning in response to a 70° rotation. Further, in a recent study by Maresch and colleagues^47^, where strategies were probed only intermittently, implicit learning reached nearly 45°. Thus, there remains much work to be done to better understand variations in implicit learning across errorclamp conditions, and standard rotation conditions.”

3.12. Data from Cressman, Figure 1: Following Day et al., one should expect that, when the explicit strategy is larger (instruction group in Cressman et al. compared to the no-instruction group), the authors are measuring the amount of implicit adaptation further from aiming direction where it is maximum. As a result, the amount of implicit adaptation appears smaller in the instruction group simply because they were aiming more than the no-instruction group (Figure 1G).

Please see our response to Point 3-13 below.

3.13. Figure 1H: the absence of increase in implicit adaptation is not due to competition as claimed by the authors but is due to the saturation of the implicit response with increasing rotation size (Kim et al. 2018). It saturates at around 20{degree sign} for all rotations larger than 6{degree sign}.

In Points 3-12 and 3-13, the reviewer suggests that the Neville and Cressman dataset is explained by an SPE model that saturates with rotation size and exhibits plan-based generalization. That is, in Point 3-12 it is suggested that plan-based generalization explains why implicit learning decreases in the instruction group. In Point 3-13, it is suggested that implicit learning is similar across rotation sizes due to a saturation in implicit learning (Kim et al., 2018). However, these two ideas are not self-consistent.

To see this, consider as suggested in Point 3-12, that implicit learning is decreased in the instruction group due to generalization. This would mean that the 10° difference in strategy produces a 3° change in implicit learning (roughly a 30% decline). This hypothesis, however, is inconsistent with the implicit response to changes in rotation size. That is, implicit responses were similar across rotation sizes, but explicit strategies were not. For example, the 20° rotation group used about 6.4° explicit strategy and the 40° group used about 26.4° explicit strategy. Thus, given the reviewer’s hypothesis, a ‘saturated’ implicit learning system would exhibit a 6° decrease due to generalization (i.e., a 10° strategy change causes 3° decrease, so a 20° change would cause a 6° decrease). However, the true data only differ by approximately 0.3°: about 5% the amount predicted by the reviewer’s hypothesis. In other words, while plan-based generalization and ‘upper limits’ on implicit learning may be true in isolation, their combination makes incorrect predictions about the phenotypes exhibited in the Neville and Cressman dataset.

There are several other points in our manuscript that disprove the reviewers’ proposed model. These are all described in more detail in other places in this response. These points include:

1. In some experimental conditions, implicit learning scales with rotation size. This occurs in Experiment 1 (stepwise group) and also across the initial rotation sizes in Tsay et al., 2021. For more details, see Point 6B (Figure 1 in revised paper).

2. In some experimental conditions, implicit learning exhibits a non-monotonic response to changes in rotation size. These are neither consistent with Kim et al., 2018, nor do they match plan-based generalization. Please see Point 6B (Figure 1 in revised paper).

3. Also, note that Neville and Cressman checked whether plan-based generalization influenced their findings. They calculated implicit learning separately for their training targets, noting that some targets overlapped with aiming directions, and others did not. Plan-based generalization suggests that implicit learning would be greater at targets that overlapped with aiming directions. This was not the case.

4. Finally, in the revised manuscript, we thoroughly analyze a plan-based SPE generalization model. Its predictions do not match the data. See Point 1 above (Figure 5 in revised paper).

Thus, we hope that our data and new analysis in Figure 1, help to dispel the notion that the implicit system necessarily exhibits a saturated asymptotic response to rotation size. The implicit system exhibits many phenotypes, which are consistent with the competition theory.

3.14. Data from Saijo and Gomi: Here the authors interpret the after-effect as a measure of implicit adaptation but it is not as the participants were not told that the perturbation would be switched off.

Yes, we agree this is a limitation. It is possible and probable that participants re-aimed in the opposite direction during washout. The data and model may hint at this as shown in Author response image 2. For example, note the gradual group washout period (black arrow in Author response image 2). Here we can see that the washout proceeds at a slightly faster rate than the model fit. This could very well be due to explicit re-aiming in the opposite direction, which is not accounted for in the model.

Nevertheless, we do not believe this re-aiming alters our primary conclusions:

1. Firstly, aftereffects remain large across all washout cycles measured (see inset C above), which to us, indicates that there is a strong lingering difference in the implicit aftereffects between the abrupt and gradual groups.

2. Secondly, suppose strategies rapidly switched direction during the washout period. Because the gradual participants experienced a larger error during washout, they would adopt a larger strategy to counter the error, relative to the abrupt group. This would lessen the differences between the abrupt and gradual groups, which exceeded 15° on the first washout cycle. In other words, this would mean that the ‘true’ difference in implicit learning is likely *larger* between the two groups than the washout aftereffects would suggest (which would only further support the hypothesis that the gradual condition led to increased implicit learning).

This being said, we do agree with the overall sentiment that our Saijo and Gomi (2010) analysis is limited. Because participants were not instructed to stop aiming, we cannot know with certainty that the reach angles during the washout period were solely due to implicit learning. We have moved this analysis to an appendix (Appendix 2) in our revised manuscript.

Second, we have added new data to our revised paper to investigate abrupt vs. gradual learning (Experiment 1) where implicit learning is more measured via no-aiming exclusion trials. We discuss these points in greater detail in our response to Point 2. We reproduce parts of this response below, for convenience.

Adapted from Point 2 response to editorial comment

In our revised manuscript, we have collected new data to test the competition model’s prediction. These new data use similar conditions studied by Saijo and Gomi. In Experiment 1 (new data in the paper), participants were exposed to a 60° rotation, either abruptly (n=36), or in a stepwise manner (n=37) where the perturbation magnitude increased by 15° across 4 distinct learning blocks (Figure 2D). Unlike Saijo and Gomi, implicit learning was measured via exclusion during each learning period by instructing participants to aim directly towards the target. As we hypothesized, in Figure 1J, we now demonstrate that stepwise rotation onset (which yields smaller target errors) muted the explicit response to the rotation (compared to abrupt learning). The competition model predicts that decreases in explicit strategy should facilitate greater implicit adaptation. To test this prediction, we compared implicit learning across the gradual and abrupt groups during the fourth learning block, where both groups experienced the 60° rotation size (Figure 2E).

Consistent with our hypothesis, participants in the stepwise condition exhibited a 10° reduction in explicit re-aiming (Figure 2F, two-sample t-test, t(71)=4.97, p<0.001, d=1.16), but a concomitant 80% increase in their implicit recalibration (Figure 2G, data, two-sample t-test, t(71)=6.4, p<0.001, d=1.5). To test whether these changes in implicit learning matched the competition model, we fit the independence Equation (Equation 5) and the competition Equation (Equation 4) to the implicit and explicit reach angles measured in Blocks 14, across the stepwise and abrupt conditions, while holding implicit learning parameters constant. In other words, we asked whether the same parameters (*a_i_* and *b_i_*) could parsimoniously explain the implicit learning patterns measured across all 5 conditions (all 4 stepwise rotation sizes plus the abrupt condition). As expected, the competition model correctly predicted that implicit learning would increase in the stepwise group (Figure 2G, comp., two-sample t-test, t(71)=4.97, p<0.001, d=1.16), unlike the SPE-only learning model (Figure 2G, indep.).

Thus, in our revised manuscript we present new data that confirm the hypothesis we initially explored in the Saijo and Gomi (2010) dataset: decreasing explicit strategy enhances implicit learning. These new data do not present the same limitation noted by the reviewers. Lastly, it is critical to note that while stepwise participants showed greater implicit learning, their total adaptation was approximately 4° lower than the abrupt group (Figure 2E, right-most gray area (last 20 trials); two-sample t-test, t(71)=3.33, p=0.001, d=0.78). This surprising phenomenon is predicted by the competition equation. When strategies increase in the abrupt rotation group, this will tend to increase total adaptation. However, larger strategies leave smaller errors to drive implicit learning. Hence, greater adaptation will be associated with larger strategies, but less implicit learning. Indeed, the competition model predicted 53.47° total learning in the abrupt group but only 50.42° in the stepwise group. Recall that, we described this paradoxical phenomenon at the individual-participant level in Point 1. Note that this pattern was also observed by Saijo and Gomi. Surprisingly, when participants were exposed to a 60° rotation in a stepwise manner, total adaptation dropped over 10°, whereas the aftereffect exhibited during the washout period nearly doubled.

Summary

In our revised paper we have removed this Saijo and Gomi analysis from the main figures, and instead use this as supportive data in Figure 2-Supplement 2, and also in Appendix 2. We now state, “It is interesting to note that these implicit learning patterns are broadly consistent with the observation that gradual rotations improve procedural learning^34,43^, although these earlier studies did not properly tease apart implicit and explicit adaptation (see the Saijo and Gomi analysis described in Appendix 2).” In sum, our revised analyses no longer depend on the Saijo and Gomi dataset to test critical competition model predictions. For that matter, no critical hypothesis is tested with ill-estimated implicit learning in the revised manuscript. Importantly, our original conclusions are still supported by newly collected data in Experiment 1, now described in Figure 2.

3.15. Even if these after-effects did represent implicit adaptation, these results could be explained by Kim et al. 2018 and Day et al. 2016. For a 60{degree sign} rotation, the implicit component of adaptation will saturate around 15-20{degree sign} (like for any large perturbation). The explicit component has to compensate for that but it does a better job in the abrupt condition than in the gradual condition where some target error remains. Given that the aiming direction is larger in the abrupt case than in the gradual case, the amount of implicit adaptation is again measured further away from its optimal location in the abrupt case than in the gradual case (Day et al. 2016 and McDougle et al. 2017).

We agree that generalization is an important concern. Please see Point 1 (response to editor) where we describe new analyses on these matters. With that said, we disagree on this particular point that the Saijo and Gomi results can be due to plan-based generalization. Krakauer et al. (2000) measured generalization and varied the number of adaptation targets. With 8 training targets, they observed that adaptation completely generalized. Thus, 8 targets is sufficient to eliminate generalized decay in adaptation. In Saijo and Gomi, the authors used 12 training targets. Adaptation will not exhibit generalization-based decay under these conditions. Thus, this hypothesis is not sufficient to explain the 15° change in aftereffect, assuming as the reviewer premised, this is entirely due to implicit adaptation. We note the relationship between generalization and training targets on Lines 1002-1009 in the revised paper.

3.16. There is no proof that introducing a perturbation gradually suppresses explicit learning (line 191). The authors did not provide a citation for that and I don't think there is one. People confound awareness and explicit (also valid for line 212). Rather, given that the implicit system saturates at around 20{degree sign} for large rotation, I would expect that the re-aiming accounts for 20{degree sign} of the ~40{degree sign} of adaptation in the gradual condition.

A recent study by Yin and Wei (2020) (Figure 4) demonstrated that gradual perturbations suppress explicit strategy use. For example, in Figure 4A, the authors measured implicit and explicit learning using a report-paradigm. In C and D, they show implicit and explicit learning in the gradual group. 12 out of 28 participants never used a re-aiming strategy during gradual adaptation (Figure 4D). 16 out of 28 participants did report some reaiming during gradual adaptation, but this was suppressed by approximately 80% relative to the abrupt group (compare Figure C4, the gradual participants, to Figure 4A, the abrupt participants). We have added this citation at the appropriate location in the revised paper.

Finally, in the revised manuscript, we corroborate this recent study in Experiment 1. Participants in the stepwise adaptation group exhibited a 10° reduction in explicit strategy (see Figure 2F) relative to the abrupt group. Thus, our claim that gradual rotations reduce explicit strategy use is validated both by past literature and new data in our revised manuscript. Finally, please see Point 3-17 below where we corroborate the Saijo and Gomi analysis using a larger explicit strategy in the gradual rotation group.

3.17. Line 205-215: The chosen parameters appear very subjective. The authors should perform a sensitivity analyses to demonstrate that their conclusions do not depend on these specific parameters.

In our Saijo and Gomi (2010) analysis, we examined the implicit and explicit responses to abrupt and stepwise rotations. These simulations required two assumptions: (1) that explicit strategy reached 35.5° in the abrupt group, (2) but remained 0° in the gradual rotation group. The abrupt strategy level was not chosen arbitrarily: Neville and Cressman (2018) observed that the steady-state explicit response to a 60° rotation was equal to approximately 35.5°. However, we agree with the reviewer that our other assumption, that no strategy developed in the gradual group, is based on a subjective expectation. It could be that subjects in this group did develop a small strategy.

Fortunately, our simulations are not overly sensitive to the exact explicit strategies chosen in the abrupt and gradual conditions. The only important constraint is that abrupt strategy must be greater than gradual strategy. We have now corroborated this assumption in Experiment 1 in the revised manuscript. To demonstrate that altered strategy levels yield similar results in our Saijo and Gomi analysis, we have now repeated the simulations, but this time, chose a 10° strategy in the gradual condition (as opposed to 0°). We show the original analysis in Figure 2-Supplement 3 in the right column. We show the new control analysis in the left column.

Note that increasing explicit strategy use in the gradual condition to 10° (left column above) still produces the same qualitative results: (1) the abrupt group achieves greater total adaptation (see A), the implicit system reaches a larger saturation level in the gradual group (see C), which yields washout aftereffects in the competition model that closely resemble actual behavior (see D). We have added Figure 2-Supplement 3 to the revised manuscript and describe this analysis on Lines 1137-1138.

3.18. Figure 3: The after-effect in the study of Fernandez-Ruiz et al. does not solely represent the implicit adaptation component as these researchers did not tell their participants that they should stop aiming during the washout.

We appreciate this concern. We no longer include the Fernandez-Ruiz study in our revised Results section.

3.19. Correlations based on N=9 should not be considered as meaningful. Such correlation is subject to the statistical significance fallacy: it can only be true if it is significant with N=9 while this represents a fallacy (Button et al. 2013).

We agree with this concern. Unfortunately, our studies were halted in March 2020 due to the Covid-19 pandemic. However, we have now collected a laptop-based control study to corroborate this experiment with a much larger sample size. In Experiment 3 (previously Experiment 2) we tested two groups: a Limit PT and No PT Limit group. The latter group possessed n=35 participants to analyze correlations between implicit and explicit learning. We used the new Limit PT group to measure implicit learning properties that were used to make model predictions, which were missing in the original paper (see Point 1-11 above). We obtained similarly striking matches between the data and model, as shown in Figure 3Q.

In addition, our revised paper provided several ways to corroborate the implicit-explicit correlations noted by the reviewer in their concern. These new analyses are:

1. Experiment 1, added a stepwise learning group: n=37 (see Figure 5D).

2. Experiment 1, investigated abrupt and stepwise learning: n=70 (see Figure S4-1H).

3. Figure 1, added new analysis of Tsay et al., 2021 (n=25/group, see Figure S4-1I).

4. Figure 4-Supplement 2, added new analysis of Maresch et al., 2021 (n=40, See Figure S4-1G).

These new data corroborate and extend the conclusions we reached in our initial paper. Critically, we now reproduce our correlation analysis in Figure 3H (now Figure 3G) within our more highly powered experiments. For example, we show the same match between the competition model and data across participants in Experiment 3 (n=35, Figure 1Q). We also observe strong negative implicit-explicit correlations that agree with the competition model in the stepwise group in Experiment 1 (n=37, Figure 5D). Similar correlations were also detected in the 60° rotation conditions in Maresch et al., 2021 (n=40, Figure 4-Supplement 1G) and Tsay et al., 2021 (Figure 4-Supplement 1I, n=25) data sets.

Experiment 1 of the authors suffer from several limitations:3.20A. small sample size (N=9 for one group). Why is the sample size different between the two groups?

Yes, we agree that the sample sizes were limited in this experiment. Please see Point 3-19 above. Sample sizes did not match as these studies both used rolling recruitment that was interrupted by the Covid-19 pandemic. Note, however, we have corroborate these results several times in the revised paper across numerous experiments (again, see Point 3-19 above). Most notably, we have used a similar design in Experiment 3 (previously Experiment 2) and test n=35 participants in the No PT Limit group, and n=21 subjects in the Limit PT group. These new data are shown in our revisions to Figure 3. Note that more subjects were recruited in the No PT Limit group to bolster our participant-level implicit-explicit correlation analysis.

3.20B. Limiting RT does not abolish the explicit component of adaptation (Line 1027: where is the evidence that limit PT is effective in abolishing explicit re-aiming?). Haith and colleagues limited reaction time on a small subset of trials. If the authors want to use this manipulation to limit re-aiming, they should first demonstrate that this manipulation is effective in doing so (other authors that have used this manipulation but have failed to do validate it first). I wonder why the authors did not measure the implicit component for their limit PT group like they did in the No PT limit group. That is required to validate their manipulation.

We appreciate this concern. We have validated that limiting preparation time abolishes explicit strategy as claimed. We address this multiple times above, and also in our response to Point 3 (response to editor).

Excerpts adapted from Point 3 response

We agree that it is important to test how effectively this condition limits explicit strategy use. Therefore, we have performed two additional control studies to measure explicit strategy in the limited PT condition. First, we added a limited preparation time (Limit PT) group to our laptop-based study in Experiment 3 (Experiment 2 in original manuscript). In Experiment 3, participants in the Limit PT group (n=21) adapted to a 30° rotation but under a limited preparation time condition. As for the Limit PT group in Experiment 2 (Experiment 1 in original paper), we imposed a bound on reaction time to suppress movement preparation time. However, unlike Experiment 2, once the rotation ended, participants were told to stop re-aiming. This permitted us to examine whether limiting preparation time suppressed explicit strategy as intended. Our analysis of these new data is shown in Figures 3K-Q.

[…]

Summary

We collected additional experimental conditions in Experiment 3 in the revised manuscript (a Limit PT group and a decay-only group). Data in the Limit PT condition suggested that explicit strategies are suppressed to approximately 2° by our preparation time limit (compared to about 12° under normal conditions). Data in the decay-only group suggest that this 2° change in reach angle was not due to a cached explicit strategy but rather time-dependent implicit decay. Together, these conditions demonstrate that limiting reaction time in our experiments prevents the caching of explicit strategies; our limited preparation time measures do indeed reflect implicit learning.

3.20C. The negative correlation from Figure 3H can be explained by the fact that the SPE is anchored at the aiming direction and that, the larger the aiming direction is, the further the authors are measuring implicit adaptation away from its optimal location.

Yes, we appreciate that generalization may contribute to the negative correlations we observed between implicit and explicit learning. We have addressed this at several points above, most notably in Point 1 and also Point 3-1. Let us reiterate the most important pieces in our response below.

Adapted excerpts from Points 1 and 3-1

In the revised manuscript, we compare the relationship between implicit and explicit learning in ExperimentsExperiments 2 and 3 with past generalization curves. These curves were measured by Krakauer et al., 2000 and also Day et al., 2016. The latter paper is most relevant as it clearly delineated implicit and explicit learning. The new analysis is documented in Figure 4 in the revised manuscript. In Figures 5BandC, we show dimensioned (i.e., in degrees) relationships between implicit and explicit learning. In Figure 5A, we show a dimensionless implicit measure in which implicit learning is normalized to its “zero-strategy” level. Empirical analysis demonstrated that implicit learning in ExperimentsExperiments 2 and 3 (black and brown points) declined over 300% more rapidly with increases in explicit strategy, than predicted by the generalization curve measured by Day et al. (2016). Thus, plan-based generalization does not appear a viable alternative to the competition model, though it could make minor contributions to the implicit-explicit correlations we observed.

These analyses have relied on an empirical comparison between our data and past generalization studies. But we can also demonstrate mathematically that the implicit and explicit learning patterns we measured are inconsistent with the generalization hypothesis. We derived an SPE generalization model in response to Point 1 (response to editor). In this model, implicit learning is driven by SPEs, but exhibits plan-based generalization. The competition model and SPE generalization model both predict a negative correlation between implicit and explicit learning, but only the SPE model suggests that this relationship (i.e., gain) is altered by rotation magnitude. Thus, to compare these two models we would need to measure the gain relating implicit and explicit adaptation, across multiple rotation sizes. To do this, we have collected additional data. These data are reported in Experiment 1 in the revised manuscript. Experiment 1 includes a step-wise perturbation condition where participants (n=37) were initially exposed to a 15° rotation, then a 30° rotation, a 45° rotation, and lastly a 60° rotation. Implicit learning was measured via exclusion (i.e., no aiming) probes for each rotation size. To calculate explicit strategy, we subtracted these implicit measures from the total reach compensation measured on the last 10 trials in each rotation block. These data are reported in Figures 5D-F in the revised manuscript.

We calculated each model’s implicit learning gain (*p* in Point 1) that best matched the measured implicitexplicit relationship. We calculated this value in the 60° learning block alone, holding out all other rotation sizes. We then used this gain to predict the implicit-explicit relationship across the held-out rotation sizes. The competition model is shown in the solid black line in Figure 5D. The Day et al. (2016) generalization model is shown in the gray line. The total prediction error across the held-out 15°, 30°, and 45° rotation periods was two times larger in the SPE generalization model (Figure 5E, rm-ANOVA, F(2,35)=38.7, p<0.001, *η_p_^2^*=0.689; post-hoc comparisons all p<0.001). Issues with the SPE generalization model were not caused by misestimating the generalization gain, *m*. We fit the SPE generalization model again this time allowing both π and *m* to vary to best capture behavior in the 60° period (Figure 5D, SPE gen. best B4). This optimized model generalized very poorly to the held-out data, yielding a prediction error three times larger than the competition model (Figure 4G, SPE gen best B4).

To understand why the competition model yielded superior predictions, we fit separate linear regressions to the implicit-explicit relationship measured during each rotation period. The regression slopes and 95% CIs are shown in Figure 5F (data). Remarkably, the measured implicit-explicit slope appeared to be constant across all rotation magnitudes in agreement with the competition theory (Figure 5F, competition). These measurements sharply contrasted with the SPE generalization model, which predicted that the regression slope would decline as the rotation magnitude decreased.

In summary, our data in Experiment 1 were poorly described by an SPE learning model with aim-based generalization. While generalization may contribute to the measured relationship between implicit and explicit adaptation, its contribution is small relative to the competition theory. These new analyses are described in Section 2.2 in the revised paper.

3.20D. The difference between the PT limit and NO PT limit is not very convincing. First, the difference is barely significant (line 268). Why did the authors use the last 10 epochs for experiment 1 and the last 15 for experiment 2? This looks like a post-hoc decision to me and the authors should motivate their choice and should demonstrate that their results hold for different choice of epochs (last 5, 10, 15 and 20) to demonstrate the robustness of their results. Second, the degrees of freedom of the t-test (line 268) does not match the number of participants (9 vs. 13 but t(30)?)

We appreciate the reviewer’s concerns. We thank the reviewer for catching two typos. (1) First, we did use the last 10 epochs for both Experiment 1 (now Experiment 2) and Experiment 2 (now Experiment 3) but had incorrectly written 15 epochs in our original manuscript. This issue has been corrected. (2) Second, we misreported this statistic in our original paper. In the revised manuscript we have corrected this: “…learning proceeded more slowly and was less complete under the PT Limit (compare Figures 3BandE; two-sample t-test on last 10 adaptation epochs: t(20)=3.27, p=0.004, d=1.42).” In sum our results were significant at the level p=0.004. In addition, the large effect size (d=1.42) does not support the reviewer’s criticism that the PT Limit and No PT Limit had little difference. In addition, we have now corroborated this result in Experiment 3, where both the PT Limit and No PT Limit groups possessed higher sample sizes (n=35 for No PT Limit, n=21 for Limit PT). The new data supported the same conclusion (t(54)=5.58,p<0.001,d=1.54).

Nevertheless, despite this conclusive evidence that limiting preparation time suppressed total adaptation, we have completed the analysis the reviewer requested. We repeated our analysis in Experiment 1 (now Experiment 2) using the last 5, 15, and 20 cycles to measure whether limiting preparation time suppressed total learning. We detected statistically significant differences across all cases, each with similar effect sizes.

– 5 cycles: t(20)=3.21, p=0.0044, d=1.39

– 10 cycles: t(20)=3.27, p=0.004, d=1.42

– 15 cycles: t(20)=3.27, p=0.0038, d=1.42

– 20 cycles: t(20)=2.96, p=0.0077, d=1.28

Thus, our revised manuscript further corroborates that limiting preparation time leads to a reduction in total adaptation. We have also noted on Lines 1434-1437, that these conclusions are not dependent on the number of cycles used to calculate total adaptation.

3.20E. Why did the authors measure the explicit strategy via report (Figure S5E) while they don't use those values for the correlations? This looks like a post-hoc decision to me.

We ended Experiment 2 (Experiment 1 in the original manuscript) by asking participants to verbally report where they remembered aiming in order to hit the target. While many studies use similar reports to tease apart implicit and explicit strategy (e.g., McDougle et al., 2015), our probe had important limitations. Normally, participants are probed at various points throughout adaptation by asking them to report where they plan to aim on a given trial, and then allowing them to execute their reaching movement (thus they are able to continuously evaluate and familiarize themselves with their aiming strategy over time). However, we did not use this reporting paradigm given recent evidence (Maresch et al., 2021) that reporting explicit strategies increases explicit strategy use, which we expected would have the undesirable consequence of suppressing implicit learning (given the competition theory). Thus, in Experiment 2, reporting was measured only once; participants tried to recall where they were aiming throughout the experiment.

This methodology may have reduced the reliability in this measure of explicit adaptation. As stated in our Methods, we observed that participants were prone to incorrectly reporting their aiming direction, with several reported angles in the perturbation’s direction, rather than the compensatory direction. In fact, 25% of participant responses were reported in the incorrect direction. Thus, as outlined in our Methods, we chose to take the absolute value of these measures, given recent evidence that strategies are prone to sign errors (McDougle and Taylor, 2019). That is, we assumed that participants remembered their aiming magnitude, but misreported the orientation. In sum, we suspect that our report-based strategy measures are prone to inaccuracies and opted to interpret them cautiously and sparingly in our original manuscript.

Nevertheless, in the revised manuscript, we now also report the individual-level relationships between report-based implicit learning and report-based explicit strategy in Experiment 2 (previously Experiment 1). These are now illustrated in Figure 3-Supplement 2C. While reported explicit strategies were on average greater than our probe-based measure, and report-based implicit learning was smaller than our probe-based measure (Figures 3-Supplement 2AandB; paired t-test, t(8)=2.59, p=0.032, d=0.7), the report-based measures exhibited a strong correlation which aligned with competition model’s prediction (Figure 3-Supplement 2C; R^2^=0.95; slope is -0.93 with 95% CI [-1.11, -0.75] and intercept is 25.51° with 95% CI [22.69°, 28.34°]).

In summary, we have now added report-based implicit learning and explicit learning measurements to our revised manuscript. We show that these measures also exhibit a strong negative correlation with good agreement to the competition model. However, we still feel that it is important to question the reliability of these report-based measures (given that they were collected only once at the end of the experiment, when no longer in a ‘reaching context’). Our new analysis is described on Lines 319-327 in the paper.

Experiment 2:3.21A. This experiment is based on a small sample size to test for correlations (N=17). What was the objective criterion used to stop data collection at N=17 and not at another sample size? This should be reported.

We did not use a criterion when stopping data recruitment in our laptop-based studies (Experiment 3). To recruit participants, we used a rolling admission. At Johns Hopkins, there is service whereby the school advertises studies in an email newsletter (Today’s Announcements). We posted about this study in the newsletter, and then recruited all eligible and interested participants that responded to the advertisement. However, in response to your concern that n=17 is too small for our correlation analysis, we reached out to a new set of participants, and have roughly doubled our sample size in Experiment 3 (the No PT Limit group) to n=35. In our revised manuscript, we show strong correlations between explicit strategy and implicit learning in our revisions to Figure 3. In addition, we have added many new data sets to the revised manuscript to test the relationship between implicit and explicit learning. We observed strong negative correlations in each of these new datasets (Experiment 1 in Figures 5D and S4-1H, Maresch et al., 2021 in Figure S41G, and Tsay et al., in Figure S4-1I). Thus, our revised manuscript strongly indicates that a negative correlation exists between implicit and explicit learning. This conclusion is not based on a sole study with n=17, but now 4 studies with n=35, n=40, n=70, and n=25.

3.21B. This experiment does not decouple implicit and explicit in contrast to what the authors pretend. If the authors believe that the amount of implicit adaptation follows a state-space model, then the measure of early implicit is correlated to the amount of late implicit because Implicit_late = (ai)^N * Implicit_early where N is the number of trials between early and late. Therefore the two measures are not properly decoupled. To decouple them, the authors should use two separate ways of measuring implicit and explicit. To measure explicit, they could use aim report (Taylor, Krakauer and Ivry 2011) and to measure implicit independently, they could use the after-effect by asking the participants to stop aiming as done here. They actually have done so in experiment 1 but did not use these values.

We appreciate your concern. This is the same comment noted in Point 8B in the editorial summary. Please see our response to Point 8B.

Figure 4:3.22A. Data from the last panel of Figure 4B (line 321-333) should be analyzed with a 2x2 ANOVA and not by a t-test if the authors want to make the point that the learning differences were higher in high PT than low PT trials (Nieuwenhuis et al., Nature Neuroscience, 2011).

We appreciate this suggestion. We have now updated the relevant analysis in our revised manuscript. We used a two-way repeated measures ANOVA. We reached the same conclusion: when measuring savings in Haith et al., we observed that the learning rate increased during the second exposure on high preparation time trials, but not on low preparation time trials (Figure 6B, right; two-way rm-ANOVA, preparation time by exposure number interaction, F(1,13)=5.29, p=0.039; significant interaction followed by one-way rm-ANOVA across Days 1 and 2: high preparation time with F(1,13)=6.53, p=0.024, η_p_^2^=0.335; low preparation time with F(1,13)=1.11, p=0.312, η_p_^2^=0.079). We corroborate this rate analysis by measuring early changes in reach angle (first 40 trials following rotation onset) across Days 1 and 2 (Figure 6C, left and middle). Only high prep. time trials exhibited a statistically significant increase in reach angles, consistent with savings (Figure 6C, right; two-way rm-ANOVA, preparation time by exposure interaction, F(1,13)=13.79, p=0.003; significant interaction followed by 1-way rm-ANOVA over days: high prep. time with F(1,13)=11.84, p=0.004, η_p_^2^=0.477; low preparation time with F(1,13)=0.029, p=0.867, η_p_^2^=0.002).

3.22B. Lines 329-330: the authors should demonstrate that the explicit strategy is actually suppressed by measuring it via report. It is possible that limiting PT reduces the explicit strategy but it might not suppress it. Therefore, any remaining amount of explicit strategy could be subject to savings.

This is an important concern. In the revised manuscript, we demonstrate that our limited preparation time method eliminates explicit strategy use. We completely document our new data and analyses in Points 3 (response to editor summary) and 3-20B above.

3.22C. Coltman and Gribble (2019) demonstrated that fitting state-space models to individual subject's data was highly unreliable and that a bootstrap procedure should be preferred. Lines 344-353: the authors should replace their t-tests with permutation tests in order to avoid fitting the model to individual subject's data.

We appreciate the reviewer’s concern but would argue that this is not an issue in our analysis. We have rather extensive experience with state-space models. We have shown that individual participant fitting procedures can be unreliable when using a least-squares fitting technique (Albert and Shadmehr, 2018). This is a particular issue for two-state models. In the two-state model, it is hard to measure the retention factor and error sensitivities for the ‘slow state’ and ‘fast state’ because they are not measured. Rather, the data are fit to the sum of the two states, i.e., the overall measured reach angle. Many slow and fast state combinations can yield a similar overall behavior, which means there is little power to estimate the slow and fast state model parameters. This is the crux of the issue.

The competition and independence models would qualify as two-state models, because they possess an ‘implicit state’ and an ‘explicit state’. However, they do not suffer from the issue highlighted above. Our model assumes that low preparation trials are implicit learning, and high preparation trials are implicit learning plus explicit learning. Thus, the ground truth estimates for implicit learning (Low PT) and explicit learning (High PT minus Low PT), are inherently embedded in the fit. To ensure that our fitting procedure has access to both the implicit and explicit states, we fit both High PT trials and Low PT trials in our least-squares algorithm. We further describe our least-squares method in our response to Point 3-22D below.

3.22D. It is unclear how the error sensitivity parameters analysed in Figure 4D were obtained. I can follow in the Results section but there is basically nothing on this in the methods. This needs to be expanded.

We appreciate that our fitting procedure was difficult to interpret. In the revised paper, we have expanded our description about this analysis on Lines 1665-1692 in our Methods. This is reproduced below.

“Finally, we also used a state-space model of learning to measure properties of implicit and explicit learning during each exposure. We modeled implicit learning according to (Equation 3) and explicit learning according to (Equation 7). In our competition theory, we used target error as the error in both the implicit and explicit state-space equations. In our SPE model, we used target error as the explicit system’s error, and SPE as the implicit system’s error.

[…]

Finally, we also fit the target-error (Equation 1) model to the mean behavior across all participants in Exposure 1 and Exposure 2. We obtained the parameter set: a_i_=0.9829, a_e_=0.9278, b_i,1_=0.0629, b_i,2_=0.089, b_e,1_=0.0632, b_e,2_=0.1078. Note that the subscripts 1 and 2 denote error sensitivity during Exposure 1 and 2, respectively. These parameters were used for our simulations in Figure 7 (see Competition Map).”

3.22E. The authors make the assumption that the amount of implicit adaptation is the same for the high PT target and for the low PT target. What is the evidence that this assumption is reasonable? Those two targets are far apart while implicit adaptation only generalizes locally. Furthermore, low PT target is visited 4 times less frequently than the high PT target. The authors should redo the experiment and should measure the implicit component for the high PT trials to make sure that it is related to the implicit component for the low PT trials.

The reviewer appears to have misunderstood the task we used in Haith et al., 2015. The High PT and Low PT targets are one and the same. We should have explained this better in our Methods section. We have revised our Methods on 1638-1644.

“Briefly, participants (n=14) performed reaching movements to two targets, T1 and T2, under a controlled preparation time scenario. To control movement preparation time, four audio tones were played (at 500 ms intervals) and participants were instructed to reach coincident with the fourth tone. On high preparation time trials (High PT), target T1 was shown during the entire tone sequence. On low preparation time trials (Low PT), T2 was initially shown, but was then switched to target T1 approximately 300 ms prior to the fourth tone. High PT trials were more probable (80%) than Low PT trials (20%).”

3.22F. I don't understand why the authors illustrated the results from line 344-349 on Figure 4D and not the results of the following lines, which are also plausible. By doing so, the authors biased the results in favor of their preferred hypothesis. This figure is by no means a proof that the competition hypothesis is true. It shows that "if" the competition hypothesis is true, then there are surprising results ahead. The authors should do a better job at explaining that both models provide different interpretation of the data.

We agree with the reviewer. We have made the suggested changes. In the revised manuscript, we show the error sensitivities predicted by the competition model in Figure 6D and have now added independence model predictions to Figure 6E. In addition, we better explain that neither model is ‘true’. Rather, the models provide two different ways to interpret the data. Our revised text on Lines 555-560 is shown below:

“In summary, when we reanalyzed our earlier data, the competition and independence theories suggested that our data could be explained by two contrasting hypothetical outcomes. If we assumed that implicit and explicit systems were independent, then only explicit learning contributed to savings, as we concluded in our original report. However, if we assumed that the implicit and explicit systems learned from the same error (competition model), then both implicit and explicit systems contributed to savings. Which interpretation is more parsimonious with measured behavior?”

3.23. Figure 5: This figure also represent a biased view of the results (like Figure 4D). The competition hypothesis is presented in details while the alternative hypothesis is missing. How would the competition map look like with separate errors, especially when taking into account that the SPE (and implicit adaptation) is anchored at the aiming direction (generalization)?

We provided the competition map in Figure 7 (previously Figure 5) not to present a biased view, but because the competition model’s behavior is genuinely unintuitive. In the SPE model, implicit learning is driven by SPEs, which is unaltered by explicit strategy. Thus, when implicit learning increases, it must be due to an increase in implicit error sensitivity. Similarly, when implicit learning decreases, it must be due to a drop in implicit error sensitivity. We do not feel it is needed to explore this intuitive behavior, which is currently assumed in most motor learning studies.

The competition model, however, is not intuitive. An increase or decrease in implicit learning by itself, has no direct relation to an increase or decrease in implicit error sensitivity. Increases in implicit learning could be due to an increase in implicit error sensitivity, a drop in explicit strategy, or some combination thereof. Decreases in implicit learning could be due to a decrease in implicit error sensitivity, an increase in explicit strategy, or some combination thereof. Our goal with the competition map in Figure 7 is to illustrate each of these scenarios, in an attempt to clear up potential confusion that the reader may have.

With that said, in our revised manuscript, we now consider an SPE generalization model as the reviewer suggests. We agree that this model’s behavior is more complicated. We explore this model’s behavior in Section 2.2 in our revised manuscript. We show that this model is not consistent with our data in ExperimentsExperiments 13 in Figure 5 in the revised paper. For much greater detail on this matter, see our response to Point 1 (editor summary response in separate document).

Figure 6:3.24A. This figure is compatible with the fact that limiting PT on all trials is not efficient to suppress explicit adaptation, at least less so than doing it on 20% of the trials (see McDougle et al. 2019 for why this is the case) and that the remaining explicit adaptation leads to savings.

We appreciate this concern about how cached explicit strategy might contribute to savings. It still seems unlikely to us that the conditions in Experiment 4 (our savings experiment) are more likely to promote a cached explicit strategy than those used in Haith et al., 2015. For example, in the paper cited by the reviewer (McDougle et al., 2019), the authors show that caching is weakened by increasing the number of training targets. In Haith et al., 2015, while there are two targets, the rotation was only active at one target. Thus, participants would only need to a cache a single reaching plan. However, Experiment 4 used four training targets. On this point alone we would expect less explicit caching.

Nevertheless, we now clearly state in the revised text that explicit caching could have contributed to the savings measured in Experiment 4. The relevant passage in our Results now reads: “In sum, when explicit learning was inhibited on every trial, low preparation time behavior showed savings (Figure 8B). But when explicit learning was inhibited less frequently, low preparation time behavior did not exhibit a statistically significant increase in learning rate (Figure 8A). The competition theory provided a possible explanation; that an implicit system expressible at low preparation time exhibits savings, but these changes in implicit error sensitivity can be masked by competition with explicit strategy.

However, the savings we measured at limited preparation time may not solely be due changes in the implicit learning system, but also cached explicit strategies^22,45^.”

These points, however, are somewhat moot. In the revised manuscript, we now directly test whether our limit preparation time condition permits explicit caching. Please see our responses to Points 3 (response to editor summary) and 3-20B above for a more detailed response on this matter. Briefly, our new control experiments suggest that the limited preparation time conditions we use allow little to no explicit strategy. Thus, our revised manuscript more strongly supports our conclusion that implicit adaptation contributed to savings in Experiment 4.

3.24B. I don't understand why the authors did not measure implicit (and therefore explicit adaptation) directly in these experiment like they did in experiment 2. This would have given the authors a direct readout of the implicit component of adaptation and would have validated the fact that limiting PT might be a good way to suppress explicit adaptation. Again, this proof is missing in the paper.

We appreciate this concern. We also agree that these data would be valuable. In our task design, we did not want to interrupt the learning process by instructing participants to stop aiming. We were concerned that this interruption could lead to decay in implicit learning (due to time passage) that would impact our ability to measure implicit savings. Indeed, as described in Points 3 and 3-20B above, our newly collected decay-only group in Experiment 3 exhibits time-based decay in implicit learning. Nevertheless, we now show in the revised paper that limiting preparation time isolates the implicit learning system, by stopping explicit strategy use. Please see Points 3 (response to editor) and 3-20B above for more detail on this matter.

3.24C. Experiment 3 is based on a very limited number of participants (N=10!). No individual data are presented.

We have now added the individual participants to Figure 8 (Experiment 4, which was previously Experiment 3). As shown in Figure 8C, 9 out of 10 participants showed an increase in the initial response to the perturbation (Figure 8C) and an increase in the rate of learning (Figure 8Ct).

3.24D. Data from Figure 6C should be analyzed with an ANOVA and an interaction between the factor experiment (Haith vs. experiment 3) and block (1 vs. 2) should be demonstrated to support such conclusion (Nieuwenhuis et al. 2011).

We appreciate this suggestion. We have now updated the relevant analysis in our revised manuscript. We used a mixed-ANOVA. We reached the same conclusion as before: when measuring savings under limited preparation time in Experiment 4, we still observed the opposite outcome to Haith et al., 2015. Notably, low preparation time learning rates increased by more than 80% in Experiment 4 (Figure 8C top; mixed-ANOVA, exposure no. by experiment type interaction, F(1,22)=5.993, p=0.023; significant interaction followed by one-way rm-ANOVA across exposures: Haith et al. with F(1,13)=1.109, p=0.312, η_p_^2^=0.079; Experiment 4 with F(1,9)=5.442, p=0.045, η_p_^2^=0.377). Statistically significant increases in reach angle were detected immediately following rotation onset in Experiment 4 (Figure 8B, bottom), but not our earlier data (Figure 8C, bottom; mixed-ANOVA exposure number by experiment interaction, F(1,22)=4.411, p=0.047; significant interaction followed by one-way rm-ANOVA across exposures: Haith et al. with F(1,13)=0.029, p=0.867, η_p_^2^=0.002; Experiment 4 with F(1,9)=11.275, p=0.008, η_p_^2^=0.556).

Figure 7:3.25A. The data contained in this figure suffer from the same limitations as in the previous graphs: limiting PT does not exclude explicit strategy, sample size is small (N=10 per group), no direct measure of implicit or explicit is provided.

We appreciate this concern. As detailed above in several responses, we have now verified that our limited preparation time condition prevents explicit strategy use. Thus, by limiting preparation time in Experiment 5 (previously Experiment 4), we are measuring the implicit contributions to learning. Please see our responses to Points 3 (response to editor summary), 1-5, and 3-20B above for more details on this matter.

Secondly, we should note that we have added individual participant data to Figure 9C. We observed a very strong anterograde interference effect; all 20 participants (9 in 5-min group, 11 in 24-hr group) exhibited impaired learning rates upon exposure to the opposing rotation (as evidence by the normalized learning rates in Figure 9C all being less than 1).

3.25B. In addition, no statistical tests are provided beyond the two stars on the graph. The data should be analyzed with an ANOVA (Nieuwenhuis et al. 2011).

Thank you for this suggestion. In the revised manuscript we now use a two-way ANOVA to test whether the anterograde learning deficit was altered by limiting preparation time and increasing the time delay between exposures 1 and 2. We reached the same conclusion as before: while both low preparation time and high preparation time trials exhibited decreases in learning rate which improved with the passage of time (Figure 9C; two-way ANOVA, main effect of time delay, F(1,50)=5.643, p=0.021, η_p_^2^=0.101), these impairments were greatly exacerbated by limiting preparation time (Figure 9C; two-way ANOVA, main effect of preparation time, F(1,50)=11.747, p=0.001, η_p_^2^=0.19).

3.25C. The number of participants per group is never provided (N=20 for both groups together).

We apologize for the omission. In the revised manuscript we now show individual participants data in Figure 9C. In addition, we also state on that the 5-min. group and 24-hr group in Lerner et al., 2021 has n=16 and n=18 participants, respectively. Experiment 5 included n=20 participants (10 Male, 10 Female) with n=9 in the 5 min group and n=11 in the 24 group.

3.25D. It is unclear to me how this result contributes to the dissociation between the separate and shared error models.

Yes, the reviewer is correct. The anterograde interference experiment (Experiment 5) is not intended to test the dissociation between the separated and shared error models. Rather, it is intended to complement our savings paradigm in Experiment 4. The natural question arises: can the implicit system bi-directionally modulate its error sensitivity? We test for increases in Experiment 4. We test for decreases in Experiment 5.

Figure 8:3.26A.The whole explanation here seems post-hoc because none of the two models actually account for this data. The authors had to adapt the model to account for this data. Note that despite that, the model would fail to explain the data from Kim et al. 2018 that represents a very similar task manipulation.

The reason why we include Mazzoni and Krakauer, 2006, and Taylor and Ivry, 2011, data, is to show that both the competition model and independence model have limitations. The competition model appears to work very well in situations where participants reach to a target without aiming landmarks. However, it cannot explain the data in Figure 10. We think it is critical to show the reader that much is left to understand about the errors which drive adaptation, and how they can vary across experimental conditions (i.e., there is no universal model that applies equally well in all scenarios). To emphasize these points, we have now separated our Figure 10 (previously Figure 8) discussion into its own Results section entitled “*Part 4: Limitations of the competition theory”*.

And yes, we agree that we could do better to contrast our results, with data measured in invariant errorclamp experiments like Kim et al., 2019. Neither the competition model, nor independence model explains implicit behavior in these paradigms. For this reason, we have now added a passage to our Discussion:

“…competitive saturation in implicit adaptation should not be conflated with the upper limits in implicit adaptation that have been measured in response to invariant errors^3,11,40^. In this latter condition, implicit adaptation reaches a ceiling whose value varies somewhere between 15 degrees^3^ and 25 degrees^40^ across studies. In these experiments, participants adapt to an error that remains constant and is not coupled to the reach angle (thus, the competition theory cannot apply). While the state-space model naturally predicts that total adaptation can exceed the error size which drives learning in this error-clamp condition (as is observed in response to small error-clamps), it cannot explain why asymptotic learning is insensitive to the error’s magnitude. One idea is that proprioceptive signals^40,83^ may eventually outweigh the irrelevant visual errors in the clamped-error condition, thus prematurely halting adaptation. Another possibility is that implicit learning obeys the state-space competitive model described here, but only up until a ceiling that limits total possible implicit corrections. Indeed, in Experiment 1, we produced scaling in the implicit system in response to rotation size, but never evoked more than about 22° of implicit learning. However, when we used similar gradual conditions in the past to probe implicit learning^37^, we observed about 32° implicit learning in response to a 70° rotation. Further, in a recent study by Maresch and colleagues^47^, where strategies were probed only intermittently, implicit learning reached nearly 45°. Thus, there remains much work to be done to better understand variations in implicit learning across errorclamp conditions, and standard rotation conditions.”

3.26B. Line 463-467: the authors claim equivalence based on a non-significant p-value (Altman 1995). Given the small effect size, they don’t have any power to detect effects of small or medium size. They cannot conclude that there is no difference. They can only conclude that they don’t have enough power to detect a difference. As a result, it does NOT suggest that implicit adaptation was unaltered by the changes in explicit strategy.

We apologize for the misunderstanding here. The statistical test we report here was not conducted in the current paper. This is a test we ran and reported in Mazzoni and Krakauer, 2006. We agree, this p-value should not be taken to mean the implicit system is unaltered. Indeed, this is a major point in our revised analysis of these data: that there are differences in adaptation in the strategy and no-strategy groups that appeared to increase over exposure to the rotation.

We report this p-value again here, because of its dramatic impact on the field over the past 15 years. This statistical test was used as evidence that implicit learning is driven by SPEs. Thus, given its significance to the literature, we feel it is very useful to remind the reader of our analysis 15 years, and why it suggested to us at the time that the implicit system was only driven by SPEs.

We have revised the text to better explain where the origins of this statistical test on PxxLxx: “When we compared the rate of learning with and without strategy in Mazzoni and Krakauer, 2006, we found that it was not different during the initial exposure to the perturbation (Figure 10B, gray, mean adaptation over rotation trials 1-24, Wilcoxon rank sum, p=0.223). This statistical test led us to conclude in Mazzoni and Krakauer, 2006, that implicit adaptation was driven by a sensory prediction error that did not depend on the primary target and was not altered by explicit strategy.”

3.26C. In Mazzoni and Krakauer, the aiming direction was neither controlled nor measured. As a result, given the appearance of a target error with training, it is possible that the participants aimed in the direction opposite to the target error in order to reduce it. This would have reduced the apparent increase in implicit adaptation. The authors argue against this possibility based on the 47.8{degree sign} change in hand angle due to the instruction to stop aiming. I remained unconvinced by this argument as I would like to get more info about this change (Mean, SD, individual data). Furthermore, it is unclear what the actual instructions were. Asking to stop aiming or asking to bring one’s invisible hand on the primary target will have different effects on the change in hand angle.

We can appreciate the reviewer’s concern. Unfortunately, we no longer have access to these data, as this study was published 15 years ago, before it was the norm to make data publicly available on a repository. As detailed in our Methods, we could only extract the mean using GRABIT in MATLAB. With that said, we are not sure these data are necessary to answer the reviewer’s question. The notion that participants maintain the same strategy despite increasing target error was tested by Taylor and Ivry, 2011. Indeed, as the reviewer suggests, participants reverse their explicit strategy to reduce target error, but this starts to occur after somewhere between 80-100 trials: in Mazzoni and Krakauer, participants only experienced 70 rotation trials. We describe this in our revised paper (Lines 772-776): “Interestingly, while the reach angle exhibited the same implicit drift described by Mazzoni and Krakauer, with many more trials participants eventually counteracted this drift by modifying their explicit strategies, bringing their target error back to zero (Figure 10H, black). At the end of adaptation, participants exhibited large implicit aftereffects after being instructed to stop aiming (Figure 10H, right, aftereffect; t(9)=5.16, p<0.001, Cohen’s d=1.63).”

With regards to the reviewer’s second question, participants were told to stop using a strategy and to aim directly at the primary target. We have revised our text on Lines 713-716 to better indicate this: “When we asked participants to stop using their aiming strategy and to move instead toward the primary target (Figure 10A, do not aim rotation on) their movement angle changed by 47.8° (difference between 3 movements before and 3 movements after instruction), indicating that they had continued to maintain the instructed explicit re-aiming strategy near 45°.”

3.26D. The data during the washout period suffers from the fact that the participants from the no-strategy group were not told to stop aiming. The hypothesis that they did use an explicit strategy could explain why the difference between the two groups rapidly vanishes. In other words, if the authors want to use this experiment to demonstrate support any of their hypotheses, they should redo it properly by telling the participants to stop using any explicit strategy at the start of the washout period to make sure that the after-effect is devoid of any explicit strategy.

This is a good criticism, similar to that raised in Point 2. However, the Mazzoni and Krakauer, 2006, data are included not to support our hypotheses, but rather to show limitations in the competition model. For this reason, we still think it is important to show these data, even though 15 years later, we have improved the methods we use to measure implicit and explicit learning. Nevertheless, we have added a limitation to our Results section on PxxLxx, to recognize this concern: “It is important, however, to note a limitation in these analyses. Our earlier study did not employ the standard conditions used to measure implicit aftereffects: i.e., instructing participants to aim directly at the target, and also removing any visual feedback. Thus, the proposed dual-error model relies on the assumption that differences in washout were primarily related to the implicit system. These assumptions need to be tested more completely in future experiments.”

With that said, while we agree that the difference in Mazzoni and Krakauer (2006) diminishes rapidly, it did not vanish entirely. Below we show the mean aftereffect on the last 10 trials of washout (trials 71-80). Note that a 3° difference still persisted on these trials, consistent with a lingering difference in implicit learning. There are at least 3 reasons why this difference may have quickly diminished: (1) implicit learning decays in both groups on each trial, (2) there is implicit learning in the opposite direction in response to the washout errors, and (3) the explicit system comes online to mitigate the ‘negative’ washout errors.

3.26E. It is unclear whether the data from Taylor and Ivry (2011) are in favor of one of the models as the separate and shared error models are not compared.

These data are in favor of neither model. Neither a target error, or an SPE alone, could drive the complex responses observed in Taylor and Ivry (2011). This is exactly why we include these data, to demonstrate to the reader the limitations in our simple models, and to show that there is so much more to understand about the error sources which drive implicit learning. We have revised our Discussion to better highlight this (see 1063-1090).

“However, the nature of aim-cursor errors remains uncertain. For example, while this error source generates strong adaptation when the aim location coincides with a physical target (Figure 10H, instruction with target), implicit learning is observed even in the absence of a physical aiming landmark^9^ (Figure 10H, instruction without target), albeit to a smaller degree. This latter condition may implicate an SPE learning that does not require an aiming target. Thus, it may be that the aim-cursor error in Mazzoni and Krakauer is actually an SPE that is enhanced by the presence of a physical target. In this view, implicit learning is driven by a target error module and an SPE module that is enhanced by a visual target error^4,11,86^.

These various implicit learning modules are likely strongly dependent on experimental contexts, in ways we do not yet understand. For example, Taylor and Ivry (2011) would suggest that all experiments produce implicit some SPE learning, but less so in paradigms with no aiming targets. Yet, the competition equation accurately matched single-target behavior in Figures 1-9 without an SPE learning module. It is not clear why SPE learning would be absent in these experiments. One idea may be that the aftereffect observed by Taylor and Ivry (2011) in the absence of an aiming target, was actually a lingering associative motor memory that was reinforced by successfully hitting the target during the rotation period. Indeed, such a model-free learning mechanism^87^ should be included in a more complete implicit learning model. It is currently overlooked in error-based systems such as the competition and independence equations.

Another idea is that some SPE learning did occur in the no aiming target experiments we analyzed in Figures 1-9, but was overshadowed by the implicit system’s response to target error. A third possibility is that the SPE learning observed by Taylor and Ivry (2011) was contextually enhanced by participants implicitly recalling the aiming landmark locations (akin to memory-guided saccade adaptation) provided during the baseline period. This possibility would suggest SPEs vary along a complex spectrum: (1) never providing an aiming target causes little or no SPE learning (as in our experiments), (2) providing an aiming target during past training allows implicit recall that leads to small SPE learning, (3) providing an aiming target that disappears during the movement promotes better recall and leads to medium-sized SPE learning (i.e., the disappearing target condition in Taylor and Ivry), and (4) an aiming target that always remains visible leads to the largest SPE learning levels. This context-dependent SPE hypothesis may be related to recent work suggesting that both target errors and SPEs drive implicit learning, but implicit SPE learning is altered by distraction^54^.”

3.27. Discussion: it is important and positive that the authors discuss the limitation of their approach but I feel that they dismiss potential limitations rather quickly even though these are critical for their conclusions. They need to provide new data to prove those points rather than arguments.

We analyze new data in the revised manuscript to address reviewer concerns. These include, but are not limited to the following major R3 concerns:

1. For the reviewer’s concern about small sample size: we have increased the sample size in Experiment 3 (previously Experiment 2) to n=35 in our No PT Limit data which is used to measure correlations between implicit and explicit learning. We now corroborate these relationships in several additional data sets: Experiment 1 (n=73), Tsay et al. 2021 (n=25/group), and Maresch et al., 2021 (n=40).

2. For the reviewer’s concern about the saturation of implicit learning in the invariant error-clamp paradigm: we now show implicit learning is not limited to a saturation phenotype. In the stepwise group in Experiment 1 (new data, n=37), we show that the implicit system can increase with rotation size. In our new Tsay et al., (2021) analysis, we show that the implicit can also have a non-monotonic response to rotation size.

3. For the reviewer’s concern about increasing implicit learning with a gradual rotation: we compare the new stepwise and abrupt participants groups in Experiment 1 and show that gradual learning reduces explicit strategy and increases implicit learning.

4. For the reviewer’s concern about generalization. We now include an SPE generalization model in the paper. We show that its predictions are inconsistent with participant behavior in Experiment 1. We also show that our data diverge from plan-based generalization curves measured in past studies.

5. For the reviewer’s concern about limited preparation time experiments. We have collected two new groups in Experiment 3 (a Limit PT and a decay-only group) where we confirm our assumption that limiting preparation time prevents explicit strategy use.

Despite these new data, we note potential limitations in our work much more clearly in the revised paper. For example, see Lines 991-1033 (generalization) and Lines 862-878 (preparation time).

On limiting PT (lines 605-615):3.28A. The authors used three different arguments to support the fact that limiting PT suppresses explicit strategy.

We have included two new control experiments to address this point in the revised manuscript. Our new evidence that limiting preparation time suppresses explicit strategy is described at numerous points above (e.g., Points 3 (see response to editor summary) and 3-20B).

3.28B. Their first argument is that Haith did not observe savings in low PT trials. This is true but Haith only used low PT trials (with a change in target) on 20% of the trials. Restricting RT together with a switch in target location is probably instrumental in making the caching of the response harder. This is very different in the experiments done by the authors. In addition, one could argue that Haith et al. did not find evidence for savings but that these authors had limited power to detect a small or medium effect size (their N=12 per group). I agree that savings in low PT trials is smaller than in high PT trials but is savings completely absent in low PT trials? Figure 6H shows that learning is slightly better on block 2 compared to block 1. N=12 is clearly insufficient to detect such a small difference.

We agree that our limited preparation time method is not the same as that used in Haith et al., 2015. We do not necessarily agree with the reviewer that caching would be easier in our paradigm. Nevertheless, we have done two control experiments to test caching in our limited preparation time condition. We have found minimal levels of explicit caching (maximally about 2°), which is more likely explained by involuntary decay in implicit learning (due to 30 sec probe instruction period) than a voluntary change in aiming. We describe these new data many times above: most notably in Points 3 (response to editor) and 3-20B.

With regards to whether savings was occurred in Haith et al., 2015, but could not be detected: First, we would like to note that this study had n=14 participants (not n=12 as suggested by the reviewer). Second, it does not actually matter to our hypothesis whether a savings effect is entirely absent in Haith et al. or is simply ‘too small to detect’ as suggested by the reviewer. The main idea is that by limiting preparation time on more trials, this will give the implicit system a better chance to express savings. Consider Author response image 4. The gray point labelled Haith et al., 2015 matches our two-state model. This point lies within the ‘black zone’ which we arbitrarily set at a +/- 5% change in measured implicit learning. It could be as, the reviewer suggests, that this point should be shifted slightly to the right in the map (see the white ‘alternate possibility’). Nothing about this would violate the competition model. The main idea we explore in the paper is that by limiting preparation time (black point, limit prep. time) the savings effect should increase in magnitude (in this hypothetical point in the map, this would be a >20% increase in implicit learning).

**Author response image 4. sa2fig4:** Shows the hypothetical scenario described by Reviewer 3.

Our data in Figure 8C above clearly demonstrate this phenomenon. While both learning rate (top row) and early implicit difference (bottom row) show no change between the two exposures (left and right bars) in the Haith et al., 2015, data set (purple, comp.), this difference is dramatically enhanced under our limited preparation time conditions in Experiment 4 (yellow lines, no comp.). Lastly, we used the statistical approach that R3 suggested in Point 3-24D to ensure that Experiment 4 yields a change in reaching behavior exceeding Haith et al.: we observed that low prep. time learning rates increased by more than 80% in Experiment 4 (Figure 8C top; mixed-ANOVA, exposure no. by experiment type interaction, F(1,22)=5.993, p=0.023; significant interaction followed by one-way rm-ANOVA across exposures: Haith et al. with F(1,13)=1.109, p=0.312, η_p_^2^=0.079; Experiment 4 with F(1,9)=5.442, p=0.045, η_p_^2^=0.377). Statistically significant increases in reach angle were detected immediately following rotation onset in Experiment 4 (Figure 8B, bottom), but not our earlier data (Figure 8C, bottom; mixed-ANOVA exposure number by experiment interaction, F(1,22)=4.411, p=0.047; significant interaction followed by one-way rm-ANOVA across exposures: Haith et al. with F(1,13)=0.029, p=0.867, η_p_^2^=0.002; Experiment 4 with F(1,9)=11.275, p=0.008, η_p_^2^=0.556).

In sum, we have addressed the reviewer’s concern by demonstrating our limited preparation time method suppresses explicit strategy. In addition, the possibility that some small and undetected implicit savings occurred in Haith et al., has no impact on our paper. The important point in the competition model, is that limiting preparation time should enhance one’s ability to observe implicit savings. We confirmed this idea using a mixed-ANOVA as suggested by R3 in Point 3-24D.

3.28C. Their second argument is that they used four different targets. McDougle et al. demonstrated caching of explicit strategy without RT costs for two targets and impossibility to do so for 12 targets. The authors could use the experimental design of McDougle if they wanted to prove that caching explicit strategies is impossible for four targets. I don't see why you could cache strategies without RT cost for 2 targets but not for 4 targets. This argument in not convincing.

As described in Points 3-28A, 3-28B (evidence in Points 3 and 3-20B), this concern is now moot. We show in the revised paper that our limited preparation time condition suppresses explicit strategy.

While no longer pertinent, we do wish to note that the McDougle et al. experiments were not similar to Experiment 4 in our paper or to the Haith et al. 2015 experiment. Thus, it is challenging to use these data to make exact predictions. For example, if caching occurs with two targets as in McDougle et al., why was caching absent in Haith et al., 2015, where two targets were also used, with only one target experiencing a cursor rotation? In any case, we have removed the statement in question in the revised manuscript; we no longer compare Experiment 4 and Haith et al. based on target numbers.

3.28D. The last argument of the authors is that they imposed even shorter latencies (200ms) than Haith (300ms). Yet, if one can cache explicit strategies without reaction time cost, it does not matter whether a limit of 200 or 300ms is imposed as there is no RT cost.

As described in Points 3-28A, B, and C (evidence in Points 3 and 3-20B), this concern is now moot. We show in the revised paper that our limited preparation time condition suppresses explicit strategy.

While no longer pertinent, we do agree with the reviewer that the reaction time limit may not alter one’s ability to retrieve the cached memory. However, the overarching question, is whether explicit re-aiming could have occurred on limited preparation time trials. While caching may be insensitive to reaction time, McDougle et al. show that general strategy use employs a mental rotation. Cutting this rotation short in time, results in intermediate strategies (at least with 12 targets in their paper). Thus, while a stricter time limit (200 ms in our data, as opposed to 300 ms in Haith et al.) may still lead to caching in conditions where caching is permitted, it still would be likely to interrupt mental rotations, thus producing smaller explicit strategies. Nevertheless, given our new experimental data (see Points 3 and 3-20B), we no longer compare Experiment 4 with Haith et al. based on reaction time limits in the revised manuscript.

On Generalization (line 678-702).3.29A. How much does the amount of implicit adaptation decays with increasing aiming direction? Above, I argued that the data from Day et al. would predict a negative correlation between the explicit and the implicit components of adaptation and a decrease in implicit adaptation with increasing rotation size. The authors clearly disagree on the basis of four arguments. None of them convinced me.

We do not disagree with the reviewer that plan-based generalization would cause a negative correlation between implicit and explicit learning. We do however disagree that this phenomenon matches our data, given its small effect size which has been documented in past studies. Regardless, we now compare the competition model to an SPE generalization model in the revised manuscript. We show in Section 2.2 in the paper that plan-based generalization does not match our data in Experiments 1-3. This analysis is highlighted in Figure 5 in the revised manuscript. Lastly, we have also added an entire section to our Discussion where we describe similarities and differences between these models (see “The relationship between competition and implicit generalization”). Please see our response to Point 1 above for a detailed description of these new data and analyses.

3.29B. First, they estimate this decrease to be only 5{degree sign} based on these two papers (FigS5A and B but 5C shows ~10{degree sign}). This seems to be a very conservative estimate as Day et al. reported a 10{degree sign} reduction in after-effects for 40{degree sign} of aiming direction (see their Figure 7). An explicit component of 40{degree sign} was measured by Neville and Cressman for a 60{degree sign} rotation. The 10{degree sign} reduction based on a 40{degree sign} explicit strategy fits perfectly with data from Figure 1G (black bars) and Figure 2F. Off course, the experimental conditions will influence this generalization effect but this should push the authors to investigate this possibility rather than to dismiss it because the values do not precisely match. How close should it match to be accepted?

The reason why we suggested 5° in our original paper was due to our data in Experiments 2 and 3 (previously 1 and 2) where we measured the correlation between implicit learning and explicit strategy across subjects. As shown in Figures 5A-C, explicit strategies varied across a 20-25° range. The Day et al. experiment would predict maximally a 5° change in implicit learning. We have made this point clearer in the revised paper, by examining our new data and Day et al. predictions in Figure 5 above. We elaborate on this in our response to Point 1 (response to editor). To summarize, Day et al. drastically underpredicts the implicit decline observed in Experiments 2 and 3 (Figures 5A-C). It also substantially underpredicts the gain we measured in Experiment 1 (see gray lines in Figure 5D above). Moreover, an SPE generalization model violates the invariance in the implicit-explicit correlation gain we observed in Experiment 1 (see Figures 5D-F). On a related note, we do agree with the reviewers, that studies are much needed to investigate how generalization properties change with experimental conditions. We note this on Lines 1029-1033 in the revised paper:

“With that said, while the generalization hypothesis did not match important patterns in our data, it remains a very important phenomena that may alter implicit learning measurements. It is imperative that implicit generalization is more thoroughly examined to determine how it varies across experimental methodologies. These data will be needed to accurately evaluate the competitive relationship between implicit and explicit learning.”

3.29C. Second, it is unclear how this generalization changes with the number of targets (argument on lines 687-689). This has never been studied and cannot be used as an argument based on further assumptions. Furthermore, I am not sure that the generalization would be so different for 2 or 4 targets.

We are a bit confused on this comment. Krakauer et al., 2000, how the generalization curve varies with target number. This paper was cited as evidence in the original manuscript. We now include all the curves measured for each target number (1, 2, 4, and 8) in Figures 5A-C in our revised manuscript.

3.29D. Third, the authors measured the explicit strategy in experiment 1 via report in a very different way than what is usually used by the authors as the participants do not have to make use of them. It seems to be suboptimal as the authors did not use them for their correlation on Figure 3H and the difference reported in Figure S5E is tiny (no stats are provided) but is based on a very limited number of participants with no individual data to be seen. If it is suboptimal for Figure 3H why is it sufficient as an argument?

To put this in context, in Figure 5-Supplement 2E (previously Figure S5E), shows how implicit learning differed across our probe measurement and our explicit report measurement. Plan-based generalization predicts that implicit learning measured via aftereffect (i.e., no aiming) should be smaller than one’s true implicit learning, which could be measured via report. Indeed, Day et al. showed this in their study. Our data did not show this. In Figure 3-Supplement 2B and Figure 5-Supplement 2E, we show that the implicit aftereffect in Experiment 1 was larger than the report-based estimate by about 4°. This is supported by a paired t-test which we have added to the revised paper (paired t-test, t(8)=2.59, p=0.032, d=0.7).

While the reviewer may argue this is small, this should be considered relative to the Day et al. prediction. The generalization curve in Day et al., would predict an opposite relationship by about 5°. Thus, in total our measures are about 9° (over 50% the total implicit response measured) in the opposite direction predicted by Day et al. Furthermore, the reviewer’s comment overlooks our additional point in the original manuscript, that this same discrepancy was seen in Maresch et al., 2021, thus bolstering our point. Thus, it seems imperative that future work examine how experimental conditions alter implicit generalization, which we state on Lines 1029-1033 in the revised paper. We suspect that the generalization curve may substantially vary across the conditions used in Day et al. (1 target, aim report on each trial) and those used in Experiment 1 (4 targets, no aim reports during adaptation).

We agree with the reviewer, however, that our implicit report-based measures are suboptimal. As in our response to Point 4 above, it should be noted that we ended Experiment 2 (Experiment 1 in the original manuscript) by asking participants to verbally report where they remembered aiming in order to hit the target. While many studies use similar reports to tease apart implicit and explicit strategy (e.g., McDougle et al., 2015), our probe had important limitations. Normally, participants are probed at various points throughout adaptation by asking them to report where they plan to aim on a given trial, and then allowing them to execute their reaching movement (thus they are able to continuously evaluate and familiarize themselves with their aiming strategy over time).

Our methodology may have reduced the reliability in this measure of explicit adaptation. As stated in our Methods, we observed that participants were prone to incorrectly reporting their aiming direction, with several reported angles in the perturbation’s direction, rather than the compensatory direction. In fact, 25% of participant responses were reported in the incorrect direction. Thus, as outlined in our Methods, we chose to take the absolute value of these measures, given recent evidence that strategies are prone to sign errors (McDougle and Taylor, 2019). That is, we assumed that participants remembered their aiming magnitude, but misreported the orientation.

Nevertheless, as detailed in Point 4 above, we now also report the correlations in Figure 3G (previously 3H) using report-based implicit learning in Figure 3-Supplement 2C. Despite the issue with our measures, we still obtained a close match to the competition model prediction.

3.29E. Fourth, when interpreting the data of Neville and Cressman (line 690-692), the authors mention that there were no differences between the three targets even though two of them corresponded to aim directions for other targets. As far as I can tell, the absence of difference in implicit adaptation across the three targets is not mentioned in the paper by Neville and Cressman as they collapsed the data across the three targets for their statistical analyses throughout the paper. In addition, I don't understand why we should expect a difference between the three targets. If the SPE and the implicit process are anchored to the aiming direction and given that the aiming direction is different for the three targets, I would not expect that the aiming direction of a visible target would be influenced by the fact that, for some participants, this aiming direction corresponds to the location of an invisible target.

We are not sure that we understand the reviewer’s suggestion. To rephrase our point (which was made initially by Neville and Cressman), there are 3 adaptation targets. Suppose we label them Targets 1, 2, and 3. Given their arrangement, to hit Target 1 with a rotation cursor, participants needed to aim near Target 2. To hit Target 2 with a rotation cursor, participants needed to aim near Target 3. However, to hit Target 3, participants had to aim to a location that did not correspond to a target. What this means given the SPE generalization hypothesis, is that there is an implicit memory that should be centered in space on Targets 2 and 3, but not Target 1. Thus, when participants are told to reach straight towards Targets 2 and 3, this would correspond to local peaks in the generalization function. Target 1, however, is not associated with any such peak. Overall, given plan-based generalization, one would expect a larger aftereffect to occur at Targets 2 and 3, than Target 1. Neville and Cressman (2018), however, did not observe this. The reviewer may have missed this in the original paper, because the analysis is described only in their Discussion and Supplementary Materials (Figure S.2 in their manuscript). Overall, this represents additional evidence that in addition to Experiment 1 and Maresch et al., 2021 (as detailed in Point 3-29D above), that generalization may differ between studies that do not use aim reports and those that do.

3.29F. Finally, the authors argue here about the size of the influence of the generalization effect on the amount of implicit adaptation. They never challenge the fact that the anchoring of implicit adaptation on the aiming direction and the presence of a generalization effect (independently of its size) leads to a negative correlation between the implicit and explicit component of adaptations (their Figure 3) without any need for the competition model.

We agree that an SPE generalization model could also produce a negative relationship between implicit and explicit learning. However, there are numerous phenomena described in both our original and revised manuscripts, that are inconsistent with plan-based generalization. Most importantly, please see Point 1 (response to editor summary), where we document our new analyses in Figure 5, which demonstrate how our data in Experiments 1-3 support the competition model over this generalization alternative.

However, while we do not go into these details in the revised manuscript, an SPE generalization model is simply ill-equipped to describe several other results we explore. Most notably, this model does not have the flexibility needed to account for the 3 implicit learning phenotypes described in Figure 1 in the revised paper (see Point 6B in editor response): (1) invariance, (2) scaling, and (3) non-monotonicity. The SPE generalization model is not able to account for the invariance phenotype (see Point 3-13 above), nor the non-monotonic phenotype. The competition model predicts all 3 phenotypes.

In sum, while we still maintain that generalization may have made a small contribution to our data, there is overwhelming evidence that it plays little role in the implicit-explicit relationships we measure.

1. An SPE generalization model cannot describe the 3 fundamental implicit learning phenotypes we examine in Figure 1 in the revised manuscript.

2. The model’s predictions about report vs. aftereffect-based implicit learning are opposite the data measured in Experiment 2 (previously Experiment 1), Maresch et al., 2021, and Neville and Cressman (2018).

3. The model incorrectly predicts how the implicit and explicit relationship will vary across changes in rotation size, which we now examine in Experiment 1 (see Point 1 above and Figure 5).

4. Previous generalization curves (e.g., Day et al., 2016) drastically underpredict the implicit learning changes we measured in Experiments 2 and 3 (see Figures 5A-C).

We have devoted two entire sections to the SPE generalization model in our revised paper (Section 2.2 in our Results, and “The relationship between competition and implicit generalization” in our Discussion).

Final recommendations3.30. The authors should perform an unbiased analysis of a model that include separate error sources, the generalization effect and a saturation of implicit adaptation with increasing rotation size. In my opinion, such model would account for almost all of the presented results.

We have included an unbiased analysis of an SPE generalization model in our revised paper. We detail this completely in Point 1 above. To summarize our findings, this model does not match our data in Experiments 1, 2, and 3. This is detailed in Figure 5 in the revised manuscript, and Section 2.2 in our Results.

The notion that implicit learning has an invariant saturation across increasing rotation size (as in Kim et al., 2018), is entirely inconsistent with our new data. To test this, we collected a control experiment: the stepwise group in Experiment 1. Here we show that implicit learning can respond proportionally to rotation size under the right experimental conditions. Our complete analysis on this point is described in Point 6B (response to the editorial summary) and in Figures 1H-L in the revised paper. In addition, generalization also disagrees with non-monotonic saturation points noted in Tsay et al., 2021, which are now examined in Figures 1M-Q in our revised manuscript.

3.31. They should redo all the experiments based on limited preparation time and should include direct measures of implicit or/and explicit strategies (for validation purposes). This would require larger group size.

In the revised manuscript we have added two control experiments to validate our limited preparation time condition. These include the Limit PT group and decay-only group in Experiment 3. We describe these new data in Point 3 above. To summarize our findings, our limited preparation time condition does suppress explicit strategy use as claimed in our original manuscript. These new data are shown in Figures 3K-M and Figure 8-Supplement 1 in the revised paper.

3.32. They should replicate the experiments where they need a measure of after-effect devoid of any explicit strategies as this has only become standard recently (experiment for Figure 2 and Figure 8). Not that for Figure 8, they might want to measure the explicit aim during the adaptation period as well.

In our revised manuscript, we replicate our Saijo and Gomi (2011) analysis (Figure 2 in original paper) using the abrupt and stepwise groups in Experiment 1. These new data corroborate our initial conclusion that gradual rotations suppress explicit strategy, thus diminishing competition and enhancing explicit learning. These new data ‘correctly’ measure implicit and explicit learning. The relevant analyses are shown in Figures 1H-L and Figures 2D-G in the revised paper. Please see our response to Point 2 (editor response) for more details.

Lastly, we do agree that our data in Mazzoni and Krakauer would be improved by directly measuring the implicit system. We list this as a limitation on Lines 752-756. Repeating such experiments is beyond our paper’s scope but is desperately needed to understand how implicit error sources may vary across paradigms that use aiming landmarks. In any case, our primary reason for analyzing these data in our paper, is to show limitations in the competition model. The existing data accomplish this goal, as we show in Section 4.1 in our revised paper. Finally, note that we include several passages in our revised Discussion (see “*Error sources that drive implicit adaptation*”) that expand on these issues.

[Editors’ note: what follows is the authors’ response to the second round of review.]

The manuscript has been improved but there are some remaining issues that need to be addressed, as outlined below:One review raised the concern about Experiment 1, which presented perturbations with increasing rotation size and elicited larger implicit learning than the abrupt perturbation condition. However, this design confounded the condition/trial order and the perturbation size. The other concern is the newly added non-monotonicity data set from Tsay's study. On the one hand, the current paper states that the proposed model might not apply to error clamp learning; on the other hand, this part of the results was "predicted" by the model squarely. Thus, can it count as evidence that the proposed model is parsimonious for all visuomotor rotation paradigms? This message must be clearly stated with a special reference to error-clamp learning.Please find the reviewers' comments and recommendations below, and I hope this will be helpful for revising the paper.

We are grateful as always, for the reviewers’ constructive and insightful criticisms. Below, we address each point in great detail. To summarize briefly:

Steady-states in Experiment 1

We appreciate the reviewer’s concern about steady-state implicit learning in Experiment 1. Please see our response to Point 4-1 below. There we argue that the experimental conditions we used were likely to produce steady-state adaptation, based on analyzing the multiple learning blocks in the abrupt group, as well as past evidence in Neville and Cressman, as well as Salomonczyk et al. (2011). For the abrupt condition in Experiment 1, we did not detect any statistically significant increase in implicit learning beyond the initial learning period. This suggests the initial block was sufficient to achieve steady-state learning in a 60° rotation. Similarly, we considered past data we collected in Salomonczyk et al. (2011) where participants had multiple exposures to a 30° rotation. Here as well, we did not detect a statistically significant change in implicit learning after the first learning block. Lastly, while we cannot statistically assess trends in Neville and Cressman (without subject data), the average change in implicit learning following the initial rotation block was incredibly limited in their 20° and 40° rotation groups: about 0.7° and 1.4°, respectively. These changes are grossly mismatched with the 14.5° variation in implicit learning we observed in Experiment 1. Thus, it is rotation size, not time, that causes the large changes in implicit learning in Experiment 1.

Furthermore, as we explain in Point 4-1, the competition theory is not specific to steady-state learning. That is, the scaling, saturation, and nonmonotonic phenotypes can be observed and validated at any point during the learning period; the effect size is largest during asymptotic performance. Thus, while we maintain our steady-state measures in Experiment 1 are robust, reaching steady-state is not needed to validate the competition model as we do in Figures 1 and 2.

In the revised paper, these important matters are discussed on P4L103, P4L120, P5L169, Appendices 1and2, and Figure 1-Supplements 3 and 4.

Nonmonotonic learning in Tsay et al., 2021

We appreciate the reviewers’ concern about potential similarities between implicit attenuation in Tsay et al., 2021 and Morehead et al., 2017. With that said, we do not agree that these learning patterns match one another. Firstly, the reduction in implicit learning in the 15° group in Tsay et al. cannot be explained via the error-clamp properties suggested by Morehead et al. The ‘error cancelation’ they observed in their 7.5° rotation group occurred because total learning reached 7.5°, completely eliminating the error source. In Tsay et al., implicit learning only reached half the rotation’s magnitude in the 15° group. Thus, error was not cancelled here.

Second, while the reduction in implicit learning in the 95°+ rotations in Morehead et al. is intriguingly similar to the decrease in implicit learning in the 90° Tsay et al. group, these paradigms possess critical differences that complicate their direct comparison. That is, errors drive learning, not rotations per se. In Tsay et al. because participants were allowed to aim, their residual steady-state errors were only 5°. In Morehead et al., participants were not allowed to aim, and residual errors were >80°. Given that implicit error sensitivity is exceedingly small in response to such large residual errors (Wei and Kording, 2009; Marko et al., 2012), this seems to us the most likely reason why implicit learning was attenuated in Morehead et al.

Thus, the competition model remains the most likely candidate in Tsay et al., where extreme residual errors such as Morehead et al. were never encountered. Its applicability is bolstered by the fact that the competition model alone can both qualitatively and quantitatively explain implicit phenotypes exhibited in Experiment 1, Salomonczyk et al., 2021, Neville and Cressman, 2018, the 15-60° rotations in Tsay et al., as well as the individual-level correlations between implicit, explicit, and total learning in Figures 3-5 (Experiments 1-3, Tsay et al., 2021, Maresch et al., 2021).

Finally, in our previous manuscript, we did not provide enough mathematical context for why the competition and independence models cannot be applied to error-clamp paradigms. The steady-state equations we provide in (Equations 4) and (5) are only appropriate for standard rotation learning, where errors decrease over time. These equations differ in the error-clamp condition, where errors remain constant. As we describe in Points 1-7 and 4-3, below, the ‘corrected’ steady-state equations show that for error-clamp learning, implicit learning must reach unattainable levels in order to achieve the dynamic steady-states described by the competition or independence theories. For error-clamp adaptation, the only possible way to reach “steady-state” is when the implicit system saturates at a physiologic upper bound.

We have greatly expanded on comparisons between error-clamp learning and standard rotation learning on P23L918 in our revised Discussion. We also better compare Tsay et al. and Morehead et al. on P21L849 in our Discussion, as well as Appendix 6.6.

Additional major changes

There are many other critical changes we have made to our revised paper in response to reviewer concerns. The 2 most important additions are detailed in Points 1-15 and 4-4 below. In 1-15, we show that the variations in implicit learning we examine in Part 1, cannot be explained via changes in implicit error sensitivity. This is discussed on P8L276, in Appendix 4, and Figure 2-Supplement 2. In 4-4, we describe updates to our generalization analyses. Namely, we use Gaussian generalization properties measured by McDougle et al., 2017, to validate the claims we made in our past manuscript. Using data where CW and CCW rotations are counterbalanced did not rescue the SPEgeneralization model. Using Gaussian generalization properties over linear properties did not improve the model either. We have greatly updated our Results on P10L374, Figure 4, Figure 4-Supplement 1, and Appendix 6.

Reviewer #4:In this paper, Albert et al. test a novel model where explicit and implicit motor adaptation processes share an error signal. Under this error sharing scheme, the two systems compete – that is, the more that explicit learning contributes to behavior, the less that implicit adaptation does. The authors attempt to demonstrate this effect over a variety of new experiments and a comprehensive re-analysis of older experiments. They contend that the popular model of SPEs exclusively driving implicit adaptation (and implicit/explicit independence) does not account for these results. Once target error sensitivity is included into the model, the resulting competition process allows the model to fit a variety of seemingly disparate results. Overall, the competition model is argued to be the correct model of explicit/implicit interactions during visuomotor learning.I'm of two minds on this paper. On the positive side, this paper has compelling ideas, a laudable breadth and amount of data/analyses, and several strong results (mainly the reduced-PT results). It is important for the motor learning field to start developing a synthesis of the 'Library of Babel' of the adaptation literature, as is attempted here and elsewhere (e.g., D. Wolpert lab's 'COIN' model). On the negative side, the empirical support feels a bit like a patch-work – some experiments have clear flaws (e.g., Exp 1, see below), others are considered in a vacuum that dismisses previous work (e.g., nonmonotonicity effect), and many leftover mysteries are treated in the Discussion section rather than dealt with in targeted experiments. While some of the responses to the previous reviewers are effective (e.g., showing that reduced PT can block strategies), others are not (e.g., all of Exp 1, squaring certain key findings with other published conflicting results, the treatment of generalization). The overall effect is somewhat muddy – a genuinely interesting idea worth pursuing, but unclear if the burden of proof is met.(4.1) The stepwise condition in Exp 1 is critically flawed. Rotation size is confounded with time. It is unclear – and unlikely – that implicit learning has reached an asymptote so quickly. Thus, the scaling effect is at best significantly confounded, and at worst nearly completely confounded. I think that this flaw also injects uncertainty into further analyses that use this experiment (e.g., Figure 5).

We appreciate the reviewer’s concern. However, we are not sure that this issue is quite as large as suggested, nor is likely to alter our primary conclusion. The stepwise condition in Experiment 1 is intended to show that total implicit learning is modulated by rotation size when explicit strategies are muted (i.e., when competition is partially alleviated). We observed that implicit adaptation increased across the 15°, 30°, 45°, and 60° rotations (see Figure 1—figure supplement 4A, rm-ANOVA, F(3,108)=99.9, p<0.001, *η_p_^2^*=0.735). The reviewer suggests a potential issue in this analysis is that implicit learning requires time to saturate, and thus, may not reach its total asymptotic level in the 15°, 30°, and 45° rotations.

Fortunately, the abrupt rotation group provides a way to verify the timecourse of implicit learning in this task. Total implicit learning was assayed 4 times in the abrupt group. Each probe overlapped with a stepwise rotation period. For example, the initial probe in the abrupt condition occurred during Block 1, when implicit learning was measured in the 15° stepwise rotation. Similarly, the second, third, and fourth implicit measurements in the abrupt group overlapped with the 30°, 45°, and 60° rotation periods is the stepwise groups. Implicit learning measures across all 4 abrupt periods are shown in Figure 1—figure supplement 4.

We tested whether total implicit learning varied across the 4 blocks in the abrupt condition. We did not detect any statistically significant effect of Block No. on total implicit learning (rm-ANOVA, F(3,105)=2.21, p=0.091, *η_p_^2^*=0.059). The same was true when we compared solely the first and last blocks (paired t-test, t(35)=1.53, p=0.134). Thus, even in the 60° rotation condition, which arguably provides the largest dynamic range to measure implicit variation, there was little or no change in implicit learning following its initial measurement during Block 1. This proves that it is the gradual change in rotation size that induces greater implicit learning, not the length of exposure to the 60° rotation. It also suggests that one exposure block was sufficient to achieve steady-state adaptation.

How can we be sure this generalizes to smaller rotation sizes? While we did not measure implicit learning across each block in the 15°, 30°, and 45° stepwise conditions, we used a very similar experimental protocol in Salomonczyk et al. (2011). These data provide another example where the implicit response scales with rotation size during a stepwise rotation sequence (see Figure 1—figure supplement 4C). Here each block used 39 rotation cycles (117 total trials in each block).

When we increased the rotation in a stepwise manner across 3 blocks: 30° then 50° and lastly 70°, we observed strong increases in asymptotic implicit learning (Figure 1—figure supplement 4C, p<0.001). In a second group, participants were exposed to a 30° rotation and implicit learning was measured across 3 consecutive blocks (Figure 1—figure supplement 4D). Critically, we did not detect any change in the implicit aftereffect across the 3 learning periods (all contrasts, p>0.05). Thus, the result was the same as in our 60° abrupt data in Figure 1—figure supplement 4B. The implicit aftereffect in Block 1, served as an appropriate measure for asymptotic implicit learning. These data strongly argue against the reviewer’s concern.

Lastly, while we did not measure extended exposures to a 15° or 45° rotation, Neville and Cressman (2018) tested prolonged exposure to a 20° and 40° rotations in a similar paradigm (3 targets). These data are reproduced in Figure 1—figure supplement 4E,F. While we do not have access to the raw data to run a repeated-measures ANOVA, we can say that the total change in implicit aftereffect was no larger than 0.7-1.4° across the first and third rotation blocks. Such changes are an order of magnitude smaller than the changes in implicit learning we observed across the 15-60° rotation sizes in the Experiment 1 stepwise group (see Figure 1—figure supplement 4A).

In sum, the 60° implicit learning timecourse we measured in the abrupt group, the 30° implicit learning timecourse measures in Salomonczyk et al. (2011), and the 20° and 40° groups in Neville and Cressman (2018), suggest that the implicit learning system exhibits little to no increase beyond the initial learning block in all experiments. Each study is rather comparable, testing participants with 3 targets, in an upper wedge spanning 45-135° in the workspace. Even in the case where the 15°, 30°, and 45° responses could exhibit an increase with additional rotation exposure, the data suggest these additional gains would be limited to 1.4°. This change would be an order of magnitude smaller than the overall implicit difference (14.5°) exhibited across stepwise rotation periods (Figure 1—figure supplement 4A), and thus would not meaningfully impact our conclusions. Thus, we are confident in our analysis. We agree that the reviewer’s concern is important to consider, and now describe it on P5L170, Appendix 2, and in Figure 1-Supplement 4 in the revised paper.

Lastly, while we maintain our implicit measures in Experiment 1 do appear to provide adequate asymptotic estimates, we wish to emphasize that the competition theory would still apply prior to reaching the asymptotic state. In our paper, we focus on asymptotic learning, because it provides a relationship between implicit and explicit learning (i.e., the competition equation) that is easy to validate mathematically. However, the scaling, saturation, and nonmonotonic phenotypes we discuss in Figure 1 can be observed well before the implicit-explicit system reaches its steady-state. Consider the state-space model where the implicit system adapts to target error driven by the rotation *r*, as in the competition model (the model used in Figures 6D and 7; the term multiplied by *b_i_* is target error (see Equation 1)).xi(n)≈bi1−(ai−bi)n−11−(ai−bi)(r−xeavg)

This equation can be rewritten recursively to represent *x_i_*^(n)^ with respect to all prior trials:In the case where implicit learning starts at zero (naïve learner), this equation simplifies to:This equation shows how the implicit system on trial *n* is driven by the explicit system on all prior trials, the rotation, and the implicit error sensitivity and retention factor. An excellent approximation to this equation can be obtained by replacing *x_e_*^(k)^ as the average *x_e_* across all prior trials (to observe this approximation’s accuracy, compare the red and blue lines in Figure —figure supplement 3A). This approximation yields:Remarkably, this equation is analogous to the competition model in Equation 4 in our manuscript. It states that implicit learning on a given trial *n* is proportional to , the difference between the rotation and the average explicit strategy used by the participant. As *n* goes to infinity, we obtain the competition model in (Equation 4). The implication of this equation is that a linear competition between implicit learning and explicit learning can be observed throughout the adaptation process, not only in the asymptotic learning phase. Thus, the scaling, saturation, and non-monotonic phenotypes we describe in Figure 1, can be observed throughout the adaptation process.

To illustrate this, consider the simulations below. In Figure —figure supplement 3A, we simulate the implicit and explicit response to a 30° rotation as in Figure 6D in the paper. In this simulation both the implicit and explicit systems are driven by target error. The red line shows the implicit approximation above, the blue line shows exact implicit learning, and the magenta line shows explicit strategy. In this simulation, we used the *a_i_*, *b_i_*, and *a_e_* parameters identified in our model fit to Haith et al., (2015), as in our paper’s competition map. Next, recall that the scaling, saturation, and nonmonotonic phenotypes are due to how the explicit system responds to changes in rotation size. When the explicit system response increases more slowly than the rotation, implicit learning will increase in the scaling phenotype. When it increases at the same rate as the rotation, implicit learning will stay the same, leading to the saturation phenotype. When it increases more rapidly than the rotation, implicit learning will decrease as the rotation increases: the nonmonotonic phenotype.

To show these phenotypes, we simulated the implicit and explicit response to a 30° and 45° rotation and tuned the explicit error sensitivity: *b_e_*. For all 30° simulations, *b_e_* remained 0.2. To obtain the scaling implicit phenotype, explicit error sensitivity remained 0.2 in the 45° simulation. To obtain the saturation implicit phenotype, we increased *b_e_* to 0.435 in the 45° condition (i.e., the explicit system became more reactive to the higher rotation). Lastly, to obtain the nonmonotonic phenotype, we increased explicit error sensitivity dramatically, to 0.93. The plots above show the implicit and explicit states obtained at various points throughout learning: from left to right, 5, 10, 20, 40, and 150 rotation cycles. The explicit responses are shown in the bottom row. Implicit responses are shown in the top row.

There are many critical things to note. Firstly, at all time points, changes in explicit error sensitivity produce 3 distinct levels: less explicit learning when *b_e_*=0.2, medium explicit learning when *b_e_*=0.435, and high explicit learning when *b_e_*=0.93. These changes have dramatic effects on the implicit system. For the low explicit strategy level, the implicit system scales when the rotation increases from 30° to 45°. For the medium explicit strategy level, the implicit system remains the same when the rotation increases from 30° to 45°. For the high explicit strategy level, implicit learning decreases when the rotation increases from 30° to 45°. Note, most critically, all 3 phenotypes occur at all phases in the implicit learning time course: as early as cycle 5, and as late as cycle 150 (top row, compare each bar set). The only thing that changes is the total difference between 30° and 45° implicit learning, which is smallest (and hardest to detect) early in learning (cycle 5), and largest (easiest to detect) after many cycles of exposure to the rotation.

What this means is that the competition model does not solely pertain to asymptotic learning. The same phenomena that occur due to implicit-explicit competition, also appear throughout all phases in the learning process. To simplify matters we have chosen in our manuscript to focus on steady-state learning, where the mathematical relationship between steady-state implicit and explicit learning converges and is easy to test (i.e., the competition equation).

Summary

We do appreciate the reviewer’s criticism. However, we maintain our implicit learning probes in Experiment 1 do provide accurate estimates of asymptotic learning. In Experiment 1, Salomonczyk et al., and Neville and Cressman, 3 targets situated in an upper triangular wedge (45° spacing) were used.

1. We did not detect any statistically significant change in implicit learning after the first learning block in the 60° abrupt group in Experiment 1.

2. We did not detect any statistically significant change in implicit learning after the first learning block in the 30° group in Salomonczyk et al. (2011).

3. Neville and Cressman (2018) measured implicit learning at 3 different timepoints for 20° and 40° rotations. While we cannot assess statistical significance, changes in implicit learning after the first learning block were exceedingly small (less than 1.4°).

Thus, we are confident that the implicit system approaches its asymptotic level within the first block’s duration in Experiment 1. Any additional growth in implicit learning is predicted to be quite small (less than 1.5°) – a level that has no effect on our primary conclusion (given that we measured a 14.5° change in implicit learning across the 4 stepwise rotation periods). Finally, our conclusions in Experiment 1, are not dependent on the implicit system reaching its asymptotic state. As shown above, the scaling, saturation, and nonmonotonic implicit learning phenotypes will occur throughout all adaptation phases. Thus, the critical point we are attempting to make, is that only a competition model (not an SPE model, or the implicit properties demonstrated in error-clamp studies) can exhibit the versatility to capture these distinct implicit modes (let alone the many other phenomena we explore throughout the paper), whether this be at asymptote, or earlier during adaptation.

We have decided that a critical point to describe to the reader is that the competition model does not solely apply to asymptotic learning. Thus, we have added derivations to Appendix 1 to start with describing *x_i_* on any trial *n*, prior to describing the limit as *n* gets large, which yields the asymptotic states in the competition Equation (Equation 4). Next, we also highlight the potential issue noted by the reviewer, that implicit learning has not saturated in Block 1 in Experiment 1. We describe the 60° abrupt response, the Salomonczyk et al. (2011) data, and Neville and Cressman data in Figure 1-Supplement 3, and also in Appendix 1.2 in the revised paper. These data show that our implicit learning measures provide close approximations for steady-state implicit learning which changes very slowly after the initial rotation block, if at all. We also note that the arbitrary point at which we decide implicit learning has reached the asymptotic state does not alter the implicit learning phenotypes predicted by the competition model (it still exhibits the scaling, saturation, and nonmonotonic phenotypes prior to steady-state).

(4.2) It could be argued that the direct between-condition comparison of the 60° blocks in Exp 1 rescues the flaw mentioned above, in that the number of completed trials is matched. However, plan- or movement-based generalization (Gonzalez Castro et al., 2011) artifacts, which would boost adaptation at the target for the stepwise condition relative to the abrupt one, are one way (perhaps among others) to close some of that gap. With reasonable assumptions about the range of implicit learning rates that the authors themselves make, and an implicit generalization function similar to previous papers that isolate implicit adaptation (e.g., σ around ~30°), a similar gap could probably be produced by a movement- or plan-based generalization model. [I note here that Day et al., 2016 is not, in my view, a usable data set for extracting a generalization function, see point 4 below.]

We appreciate the reviewer’s concern. But the variation between implicit learning and explicit strategy across abrupt and stepwise learning cannot not be captured by any plausible generalization curve.

The reviewer suggests that one explanation is that both groups have equal implicit learning, but the abrupt condition leads to greater re-aiming. More aiming in the abrupt group causes a decrease in the implicit learning measured at the target due to generalization. This hypothesis could be summarized with an SPE generalization model in which measured implicit learning at the target is equal to total implicit learning via: ximeasured=xissexp⁡(−0.5(xessσ)2), where σ is the generalization curve’s width and xiss is total implicit learning (measured at aim direction). Could this model lead to the observed data?

Initially, let us assume that σ = 37.76°. This is the generalization curve that was measured by McDougle et al. (2017), as proposed in Point 4-4 below. To estimate the change in implicit learning between the abrupt and gradual groups given generalization, we can calculate the reduction in implicit aftereffect expected given a normal distribution with σ=37.76°, for the 39.5° and 29.9° explicit strategies measured in the abrupt and stepwise conditions. Aiming straight to the target in the abrupt group would yield 57.86% remaining aftereffect, and 73.09% remaining implicit aftereffect in the stepwise group. Altogether, this would mean that the stepwise rotation increased the implicit aftereffect by 100(73.09/57.86 – 1) = 26.3% over the abrupt group. On the contrary, the abrupt and stepwise implicit aftereffects were 11.72° and 21.36°, respectively. This is an 82.3% increase in implicit learning.

In sum, similar to our analysis of Experiments 2 and 3 in Figure 5, while generalization will produce a negative implicit-explicit relationship, the implicit learning variations we observed were much larger in Experiment 1 than predicted by generalization alone. In this case, the 82.3% increase in implicit learning is more than 3-fold larger than the 26.3% increase predicted by the implicit generalization properties measured by McDougle et al. (2017).

Suppose σ = 37.76° is not true in our data. To match the measured data, σ would need to be smaller, to narrow the generalization curve. This is unlikely to be the case given that in Experiment 1 we used 3 targets, whereas McDougle et al. used only 1. Additional targets would not narrow the generalization curve, it would widen it (Krakauer et al., 2000). Still, let us proceed. Rather than assume that σ = 37.76°, let us fit a normal distribution to the measured data. In the abrupt group, implicit and explicit learning were 11.72° and 39.5°, respectively. In the stepwise group, implicit and explicit learning were 21.36° and 29.9°, respectively. Fitting a normal generalization curve to these data, would yield the curve shown in Figure 4—figure supplement 1A. The optimal σ is 23.6°. Surprisingly, the generalization curve would require that implicit learning (measured at the aiming location) needs to be 47.8°. This value substantially exceeds that deemed possible by Morehead et al. (2017) and Kim et al. (2018). Much more importantly, these values are unphysical. For abrupt learning an *x_i_^ss^* = 47.8° and *x_e_^ss^* = 39.5° would indicate that total adaptation should equal 47.8° + 39.5° = 87.3° (see Figure 4—figure supplement 1B). Not only does this exceed measured abrupt adaptation by about 60%, but it is also larger than the rotation magnitude – thus, entirely unphysical. In the stepwise group as well, total predicted learning would be about 77.7°, larger than the rotation size (hence unphysical).

Clearly, there is a deep issue here. The problem is that as the generalization curve narrows (e.g., σ = 23.6° vs. 37.76°), not only does implicit learning measured at the target drastically underapproximate total implicit learning at the aim location, but the explicit strategy we estimated via xess=xTss−xiss will substantially overestimate true explicit learning (because explicit strategy is estimated using implicit learning). As σ gets smaller, the issue will grow and the analyses above become inappropriate, leading to unphysical systems. The key idea is that both implicit and explicit learning need to be corrected by the generalization curve in our data. Note that:

Equation 1:ximeasured=xissexp⁡(−0.5(xessσ)2)

The discrepancy between total and measured implicit learning is: (xiss−ximeasured). The amount that implicit learning is underapproximated is the same amount that explicit learning is overapproximated. Thus, we have:

Equation 2:xemeasured=xess+xiss−ximearured

We can re-arrange Equation 2:

Equation 3:xess=xemeasured−xiss+ximeasured

Combining Equations 1 and 3 yields the expression:

Equation 4:xess=xemeasured−xiss+xissexp(−0.5(xess/σ)2)

Equations 1 and 4 correctly express and constrain the relationship between (1) total implicit learning, (2) total explicit learning, (3) measured implicit learning and (4) measured explicit learning. We identified the σ and *x_i_^ss^* that minimized the squared error between ximeasured predicted by Equation 1 above and the respective stepwise and abrupt values, subject to the constraint that Equation 4 must be satisfied exactly. The model revealed that the optimal σ and xiss were 3.87° and 45.69°, which produced the curve shown in Figure 4—figure supplement 1C (corrected model). This shows how measured implicit learning and explicit learning will interact. Figure 4—figure supplement 1D, however, shows the associated relationship between *measured* implicit learning and total strategy (see the corrected model). Figure 4—figure supplement 1C demonstrates one oddity: the model requires that the total implicit learning equal 45.69° implicit learning, or roughly 90% the total adapted response. More importantly, Figure 4—figure supplement 1D reveals that the model requires an extreme generalization curve, with σ = 3.87°, as compared to 37.76° as measured by McDougle et al. This value is not physiological, being an order of magnitude smaller than any generalization curve measured to date. Thus, we can conclude that generalization is extremely unlikely to yield the measured data and is not a viable alternative to the competition model.

Help interpreting the relationship in Figure 4-supplement 1: how are C and D related? Let us begin with the stepwise point in C. This lies roughly at 20° implicit learning and 30° explicit strategy. This explicit strategy mirrors that measured in the paper: total adaptation – measured implicit learning. Note that total implicit learning is about 45°. Thus, measured implicit learning is about 45-20 = 25° smaller than total implicit learning. This means that our estimated explicit strategy at 30°, is about 25° too large. Thus, the actual strategy is much smaller: 30-25 = 5°. In Part D, the x-axis explicit strategy will be about 5° and implicit learning will be about 20/45 x 100 = 44.4%.

Summary

Our abrupt vs. stepwise analysis in Figures 2D-G does not match implicit generalization. While incorrect, we began with the assumption that the implicit learning measures in Experiment 1 represented generalized learning, but explicit strategies represented total re-aiming. The change in implicit learning we measured, however, was three times larger than that predicted by generalization as measured in McDougle et al. Moreover, this analysis is flawed, in that only correcting implicit measures with generalization, but not explicit measures, produced a situation where total adaptation would have exceeded the rotation’s magnitude. This is because our explicit strategies are estimated using total adaptation and implicit learning. When we corrected the SPE generalization model so that both the implicit and explicit learning we measured were corrected by a generalization curve, the model required that plan-based generalization resemble a Gaussian with σ = 3.87°, an unphysiological scenario.

These analyses show that a generalization model cannot explain the measured data. We have updated our text on P10L374, as well as Figure 4-Supplement 1, and Appendix 6. These issues also occur in Figure 4, when we compare Experiments 2 and 3 to past generalization curves (see Point 4-4 below). Also note that we have conducted similar analyses for the Neville and Cressman (2018) dataset, analyzing the response to rotation size, and instructions. These are shown in Appendix 6.5 in the revised paper and demonstrate the same limitations in generalization as shown above.

(4.3) The nonmonotonicity result requires disregarding the results of Morehead at al., 2017, and essentially, according to the rebuttal, an entire method (invariant clamp). The authors do mention and discuss that paper and that method, which is welcome. However, the authors claim that "We are not sure that the implicit properties observed in an invariant error context apply to the conditions we consider in our manuscript." This is curious and raises several crucial parsimony issues, for instance: the results from the Tsay study are fully consistent with Morehead 2017. First, attenuated implicit adaptation to small rotations (not clamps; Morehead Figure 5) could be attributed to error cancellation (assuming aiming has become negligible to nonexistent, or never happened in the first place). This (and/or something like the independence model) thus may explain the attenuated 15° result in Figure 1N. Second, and more importantly, the drop-off of several degrees of adaptation from 30°/60° to 90° in Figure 1N is eerily similar to that seen in Morehead '17. Here's what we're left with: An odd coincidence whereby another method predicts essentially these exact results but is simultaneously (vaguely) not applicable. If the authors agree that invariant clamps do limit explicit learning contributions (see Tsay et al. 2020), it would seem that similar nonmonotonicity being present in both rotations and invariant error-clamps works directly against the competition model. Moreover, it could be argued that the weakened 90° adaptation in clamps (Morehead) explains why people aim more during 90° rotations. A clear reason, preferably with empirical support, for why various inconvenient results from the invariant clamp literature (see point 6 for another example) are either erroneous, or different enough in kind to essentially be dismissed, is needed.

We appreciate the reviewer’s criticisms here, but think it is important not to overlook the overall picture. While we are not sure we agree that Tsay et al. is consistent with Morehead et al. (elaborated on below), even in such a case, the implicit learning properties suggested by Morehead et al., are at odds with most data considered in our paper. Here are some examples:

– The scaling phenotype in Experiment 1, and also Salomonczyk et al. 2011, directly contradicts the idea that implicit learning reaches a rotation-invariant saturation point – the hallmark phenotype in Morehead et al., 2017. These studies also show that the implicit response can vary greatly across 15-60° rotations, vastly exceeding the 4.4° upper-limit on the proportional zone estimated in Kim et al., 2018.

– The variations in implicit learning across the abrupt and stepwise conditions in Experiment 1, are at a minimum 3fold larger than one can obtain with standard generalization curves (see Point 4-2 above), and minimally 2fold larger in response to instruction in Neville and Cressman (2018). Moreover, as described in Point 4-2, a model where implicit learning is equal between instruction/no-instruction conditions or abrupt/gradual conditions requires generalization properties that are unphysiologically narrow. In sum, it is not possible that implicit learning was equal across these various conditions but altered by generalization.

– A similar argument is true when considering the variations in implicit learning in Neville and Cressman (2018). There is no way to reconcile the variation in implicit learning in response to instruction, with the invariance in learning across rotation sizes. The latter requires a Morehead-like response, and complete generalization in implicit learning. However, with complete generalization, there can be no decrement in implicit learning with instruction, unless competition with explicit strategy is permitted.

Thus, these studies alone show it is not possible that the implicit system shows the same response to rotation size as in Morehead et al. Now, let us consider the assertion that the rotation response in Tsay et al. is consistent with Morehead et al. To be clear, error-clamp learning in Morehead et al., does not show a nonmonotonic response, one that goes up and then down (or vice versa). Rather, it shows a saturated response to rotation size initially, and then a decrease in implicit learning when rotations get very large (i.e., >95°). As the reviewer notes, when Morehead et al. tested a 7.5° standard rotation (or at least somewhat standard, minus the “ignore the cursor” instruction), smaller implicit learning was observed (than the saturated zone). This reduction in implicit learning was attributed to “error cancellation”. Namely, the participants adapted to 7.5°, which eliminated error and halted implicit learning (see Morehead et al). We do not agree with the reviewer that this phenomenon may have occurred in Tsay et al. The critical point is that Tsay et al. tested a 15° rotation. As shown in Figure 1N, implicit learning reached only 7.6°. Thus, it is not possible that the error was cancelled by implicit learning, which only reached half the rotation magnitude. Morehead et al. would have predicted that implicit learning should continue until 15° to cancel the error. Thus, the implicit responses in Tsay et al. are not consistent with those in Morehead et al.

This now brings us to the decrement in implicit learning we observed in Figure 1N with the 90° rotation. Yes, we agree that Morehead et al. observed an attenuation in implicit learning with very large rotations. It is possible that such an attenuation may have contributed to the substantial implicit reduction in the 90° group in Tsay et al. We reference this possibility in our revised manuscript on P21L849. With that said, we wish to emphasize a very important detail. It is errors that drive learning, not rotations. While the large rotations may have been similar across these two studies, errors subjects experienced in both experiments were totally mismatched. In Morehead et al., participants in both the error-clamp and standard rotation groups were told not to aim and to ignore the cursor. The implicit learning curve reached approximately 10°, leaving an 85° target error. Given past studies (e.g., Wei and Kording, 2009; Marko et al., 2012), error sensitivity will be exceedingly tiny for such errors. In our view, this insensitivity to extremely large errors likely led to the attenuation in implicit learning observed in Morehead et al. Instructions to “ignore the cursor” may have further exacerbated the reduction in sensitivity to these large errors (e.g., if participants did not foveate the cursor during adaptation).

However, in Tsay et al., participants were allowed to aim. Total learning reached about 85°, leaving a 5° target error: an error much more inclined to drive implicit learning. Comparing steady-state adaptation to this 5° residual error with the 85° residual error in Morehead et al., is not reasonable in our view. The only way to compare learning across these experiments, would be in the case that SPE is the sole driver of implicit learning (which is not altered by explicit strategy). This, however, is directly contradictory to the analyses in the current paper, and other studies which show the critical role visual target error has in implicit learning: Miyamoto et al. (*Nat Neuro*, 2020), Ranjan and Smith (MLMC 2020), Tsay et al. (*bioRxiv*, 2021), and Taylor and Ivry (2011).

Thus, we would argue that direct comparison between Tsay et al. and Morehead et al., is not advisable. Attenuation in implicit learning in these two studies may have entirely separate causes: a drastic reduction in target errors (the competition hypothesis) in Tsay et al., and an unresponsiveness to extreme target error in Morehead et al. (which could have been exacerbated by telling participants to ignore the cursor), which appears largely specific to this one study. For the learning patterns in Tsay et al., the competition model seems the most parsimonious choice, not only given its quantitative match to the data (Figures 1Q and Figure 1-Supplement 2), but also because it alone (not the errorclamp learning properties in Morehead et al.) can explain implicit responses across the many other cases we consider in Figures 1 and 2: abrupt and stepwise responses in Experiment 1 (as well as Salomonczyk et al.), rotation responses between 15-60° in Tsay et al., as well as implicit behavior in Neville and Cressman. This is not to mention that the implicit learning properties in Morehead et al. provide no clear way to interpret the pairwise and counterintuitive (e.g., a negative implicit-total learning) relationships between implicit learning, explicit strategy, and total learning detailed at length in Figures 3-5 in our paper at the individual-participant level. That is, we tested the reviewer’s suggestion that negative correlations between implicit-explicit learning were due to strategies responding to implicit variation (see P11L417 in the revised paper). The data were inconsistent with this possibility (Figure 5). However, we have now revised our Discussion section to note other possible reasons why implicit learning declined with large rotation sizes in Tsay et al. (see P21L849).

With that said, there is a much larger and more central point to be made here, which we have not clearly described in our previous paper and reviewer responses. Our previous statement, that ‘the implicit properties observed in an invariant error context [may not] apply to [standard rotation] conditions’, misses critical mathematical context, and creates an unnecessary contention between these two experimental conditions.

At a core level, the equation that governs steady-state behavior in an error-clamp condition, is not the same as in a standard rotation condition. Excluding any role strategic learning may play in competing with implicit learning, an SPE model predicts steady-state implicit learning via the independence equation:xiss=bi(1−ai+bi)−1r . In a constant error-clamp study, where errors remain invariant, the correct steady-state prediction is:xiss=bi(1−ai)−1r . These two steady-states possess different implicit learning gains: bi(1−ai)−1 for constant error-clamp and bi(1−ai+bi)−1 for standard rotation learning. This has critical implications. For example, in an error-clamp condition, for an *a_i_* = 0.98 and *b_i_* = 0.3, the state-space model predicts an implicit steady-state of 0.3(1-0.98)^-1^*r* = 15*r*. What that means is that in an errorclamp condition, steady-state implicit adaptation would need to exceed the rotation size by at least an order of magnitude to reach a dynamic equilibrium between learning and forgetting. For *p_i_*=15 would mean that a 5° errorclamp would require 75° implicit learning to reach a dynamic steady-state. A 30° rotation would require 450° implicit learning. Thus, very rapidly, even small rotations would require implicit learning that is either physiologically unlikely, or physically impossible, in order to reach a steady-state between learning and forgetting.

Thus, not only do the competition and independence Equations Equations (4) and (5) not apply to error-clamp learning, their ‘corrected’ variants show us that the steady-states we describe in our paper cannot be reached in an errorclamp paradigm. For these reasons, the steady-states reached in error-clamp studies are likely caused by another mechanism: the ceiling effect as described in Morehead et al. (2017) and Kim et al. (2018). That is, the large implicit learning gain (e.g., 15) in error-clamp studies will drive implicit systems to a ceiling in the total amount it is able to adapt (because it cannot reach its dynamic equilibrium). In the error-clamp scenario, implicit forgetting is simply not strong enough to compete with learning from the constant, unchanging error, yielding large theoretical steady-state levels, that across many rotation sizes, greatly exceed the implicit system’s capacity to correct. On the other hand, in standard rotation conditions, the implicit learning gain is much smaller (e.g., 0.6-0.8 in the data sets we consider here), meaning that the implicit system will reach a dynamic steady-state prior to a rotation-invariant ceiling.

Now, a separate but related question, is what causes the implicit system’s “upper capacity limit”, and does the limit vary across experimental settings. We suspect that it does. For example, in Morehead et al. (2017) the implicit system was limited to about 10-15° learning, but in Kim et al. (2018) this limit increased to 20-25°. The limit increased greatly, despite using similar experimental conditions across these tasks. We speculate that an important component in this limit is proprioception which may play a role in halting implicit learning when the hand strays too far from the target. We speculate that this halting is much more aggressive in error-clamp paradigms, when the participant’s goal is to move their hand straight to the target and ignore the cursor (here proprioception is the only input that participants have about whether or not they are achieving their “reach straight” goal). In the standard rotation, however, changes in reach angle are implicitly encouraged, because the participant’s goal is to hit the target with the cursor, not their hand. Thus, we suspect that the brain is less inclined to halt implicit learning due to “proprioceptive discrepancies” in standard rotation conditions; visual errors dominate here. This may explain why some authors have observed implicit learning levels (e.g., about 35° in Salomonczyk et al., 2011, and even 45° in Maresch et al., 2021) which greatly exceed the error-clamp limits observed in Morehead et al. and Kim et al.

We have amended our Discussion to better compare and contrast standard rotation learning and invariant errorclamp learning on P23L918 in the revised paper. We also better compare Tsay et al. and Morehead et al. on P21L849 in our Discussion, as well as Appendix 6.6. We outline the critical discrepancies in the state-space model, i.e., the theoretical implicit learning gain. We hope our discussion can help the reader understand why the implicit system may reach a ceiling in error-clamp conditions, but not standard rotation conditions. This ceiling effect makes it appear as if implicit steady-states are not sensitive to rotation size. Overall, we do think that a competition or independence model can apply in error-clamp learning conditions, when we allow the idea that implicit learning can reach a “capacity limit”, and also that it may respond differentially to visual and proprioceptive errors across errorclamp studies and standard rotation conditions (that is, we should not expect the same ceiling across task variants).

Summary

We appreciate the reviewer’s concern about potential similarities between implicit attenuation in Tsay et al., 2021 and Morehead et al., 2017. With that said, there are critical pieces that do not match: e.g., reduced implicit learning in the 15° rotation group, which cannot be chalked up to ‘error cancelation’ as in Morehead et al. Further, there are fundamental differences in the errors experienced across these two experimental paradigms: large residual errors (>80°) in Morehead et al., versus 5° residual error in the 90° rotation group in Tsay et al. These substantial variations in error make it challenging to compare these two studies. In the Morehead et al., study, we suspect drastic reduction in implicit error sensitivity is likely behind attenuations in implicit learning: the large residual errors encountered in this study do not generalize well to the experiments we consider here. Given that the competition model alone can explain the implicit learning phenotypes in Experiment 1, Salomonczyk et al., Neville and Cressman, and the 15-60° rotation range in Tsay et al., this model provides the most parsimonious explanation to the implicit attenuation in Tsay et al.

Lastly, in our past paper and reviewer responses, we did not provide enough mathematical context for why implicit steady-states may differ across error-clamp and standard rotation conditions. The issue lies in that the steady-state equations which describe a dynamic equilibrium between learning and forgetting, differ across these experimental paradigms. For error-clamp learning, implicit learning must reach unattainable levels in order to reach the dynamic steady-states described by the competition or independence theories. For error-clamp adaptation, the only possible way to reach “steady-state” is when the implicit system saturates at a physiologic upper bound. Thus, we think that the competition model describes implicit learning up until a physiologic upper bound – the one highlighted in errorclamp studies. This upper bound’s nature remains unclear, as it appears to vary across similar studies: e.g., Morehead et al., and Kim et al., and exceeded in some standard rotation conditions (e.g., Salomonczyk et al., and Maresch et al., 2021). Thus, we think it is likely that several experimental factors may play a role in determining the “stopping point” for implicit learning, when errors cannot be reduced through adaptation: e.g., differential responses to visual and proprioceptive signals.

We have now greatly expanded on these issues and speculations in our revised Discussion. It is our hope that these mathematical and empirical considerations inspire the community to better compare and contrast implicit responses across standard rotations and error-clamp conditions: especially in scenarios where explicit strategy is also present to modulate the implicit response.

(4.4) The treatment given to non-target based generalization (i.e., plan/aim) is useful and a nice revision w/r/t previous reviewer comments. However, there are remaining issues that muddy the waters. First, it should be noted that Day et al. did not counterbalance the rotation sign. This might seem like a nitpick, but it is well known that intrinsic biases will significantly contaminate VMR data, especially in e.g. single target studies. I would thus not rely on the Day generalization function as a reasonable point of comparison, especially using linear regression on what is a Gaussian/cosine-like function. It appears that the generalization explanation is somewhat less handicapped if more reasonable generalization parameterizations are considered. Admittedly, they would still likely produce, quantitatively, too slow of a drop-off relative to the competition model for explaining Experiments 2 and 3. This is a quantitative difference, not a qualitative one. W/r/t point 1 above, the use of Exp 1 is an additional problematic aspect of the generalization comparison (Figure 5, lower panels). All in all, I think the authors do make a solid case that the generalization explanation is not a clear winner; but, if it is acknowledged that it can contribute to the negative correlation, and is parameterized without using Day et al. and linear assumptions, I’d expect the amount of effect left over to be smaller than depicted in the current paper. If the answer comes down to a few degrees of difference when it is known that different methods of measuring implicit learning produce differences well beyond that range (Maresch), this key result becomes less convincing. Indeed, the authors acknowledge in the Discussion a range of 22°-45° of implicit learning seen across studies.

We thank the reviewer for these excellent suggestions. We have made several changes to the manuscript to address these concerns. First, we have now added the generalization curve measured by McDougle et al. (2017) to the paper. These authors measured implicit generalization via aftereffect, and also balanced CW and CCW rotations. To do this, we used the generalization curve measured in Figure 3A in their paper. We used the aftereffects measured at the target locations labeled: 22.5°, 0°, -22.5°, -45°, and -67.5°. We used the “left-hand-side” of the curve as these data represent how hand angle would change when participants abandon their strategy and are told to reach straight to the target. We added these reach angles to Figure 4A in the revised manuscript (see 4A below). To do this, we normalized data to the maximal aftereffect at 22.5° (result is shown in McDougle 1T in 4A below).

There are some important things to note about these data. As the reviewer suggested, the McDougle et al. and Day et al. generalization curves do diverge. However, this departure seems to occur around when explicit strategies are greater than about 25-30° or so. For angles less than 25°, they are overall quite similar. This is important given our data in Experiments 2 and 3. Here explicit strategies were maximally 20-25°. Thus, in this region, the Day et al. and McDougle et al. curves are very similar. As such, when the generalization curves are used to predict how implicit learning should decline in Experiments 2 and 3 (see Figure 5B and 5C), McDougle et al. is no better at matching the measured data than the Day et al. curve highlighted in the previous manuscript. Thus, while we agree that other studies such as McDougle et al. provide better generalization estimates, our conclusions that the implicit decline far exceeds that predicted by generalization in Experiments 2 and 3, remains unchanged.

Recall, however, that the analysis in Figures 4AandB above, requires an important correction (see Point 4-2). Namely, the explicit strategies along the x-axis are not *true* explicit learning. This is explicit strategy estimated via total adaptation minus the measured implicit learning (on reach-to-target probes). Thus, it is actually not appropriate to compare the data in these insets directly to past generalization curves. Under the generalization hypothesis, because measured implicit learning will underestimate total implicit, the estimated explicit strategy will overestimate the true explicit strategy. While we considered this idea in a supplementary figure in our previous manuscript, we think this is a much more central point that should be made. Thus, in Figure 4C we now show the implicit-explicit generalization curve that would be required to produce the implicit-explicit measures we obtained in Experiments 2 and 3. To obtain these data, we fit a normal SPE generalization model to the data in Experiment 2 and 3, separately, and identified the total implicit learning and generalization width that best matched the data. To do this, we used the method described in Point 4-2 above. The optimal generalization curves had σ = 5.16° and 5.76° in Experiments 2 and 3, respectively. Note that this generalization, is exceedingly narrow, and inconsistent with McDougle et al. (σ = 37.76°). Thus, as in Point 4-2 above, plan-based generalization is not consistent with the measured data.

Next, to comply with the reviewer’s suggestion, we have also updated our Experiment 3 analysis in Figures 4D-F. We wish to reiterate again that our data (and previous data sets by Salomonczyk et al. and Neville and Cressman) suggest that our data in Experiment 1 do provide good asymptotic estimates (see our Point 4-1 response above) for the implicit system. Also, please recall (see Point 4-1) that the data in Experiment 1 can still be analyzed whether or not implicit and explicit learning have reached asymptote; the competition model (and similarly the independence model) apply to the entire learning timecourse (only the effect sizes scale as total adaptation is approached). Thus, with these clarifications, we hope the reviewer agrees there is no issue with our Experiment 1 analysis in Figure 4.

Instead, the reason why the Experiment 1 analysis is so critical is it speaks to the reviewer’s point about ‘qualitative versus quantitative differences.’ Experiment 1 tests multiple rotation sizes, and as such, allows us to test qualitative differences between generalization and the competition model. As described in the paper, the competition model predicts that the implicit gain relating implicit learning and explicit strategy should be similar across rotation sizes. A generalization model predicts that this gain will increase as the rotation increases. This change in gain is due to increases in implicit learning as rotation size increases. A Gaussian implicit-explicit relationship would exacerbate these variations in gain; because the normal distribution is nonlinear, linear approximations to the curve will vary in slope as explicit strategy gets larger (which occurs as rotation size gets larger).

In our revised Experiment 1 analysis, we made two major improvements. We now split our analysis into two pieces, one in which the relationship between implicit learning and explicit learning is assumed to be well-approximated by a line (SPE gen. linear in Figures 4D-F). Second, we repeat these analyses using a normal distribution to represent the implicit-explicit relationship in the SPE generalization model (SPE gen. normal in Figures 4D-F). In our linear analysis, we do not use Day et al. generalization curve properties. We replaced these with the McDougle et al. generalization curve properties. We use σ = 37.76° in our Gaussian model, which is the width measured by McDougle et al.

As in the previous manuscript, we fit the model to the B4 period only (the 60° period). Figure 4D shows the updated comparison to the data. The black line shows the competition model. The cyan line (SPE gen. linear) shows the linear generalization model. The gold dashed line (SPE gen. normal) shows the Gaussian model. Figure 4E calculates the prediction error in each model. This is the RMSE between the data and the model predictions across the held-out B1 (15°), B2 (30°), and B3 (45°) periods. Prediction error was about 60% larger in the SPE model (Figure 4E, rm-ANOVA, F(2,72)=13.7, p<0.001, *η_p_^2^*=0.276). The linear model performed slightly better than the Gaussian model (post-hoc test, p=0.006). Thus, as previously reported, the competition model more accurately matched the data measured in Experiment 1.

Issues with both the linear and normal SPE generalization functions were due in part to an intrinsic property in each model; the relationship between implicit and explicit learning should vary as implicit learning increases. For example, suppose that there is a 50% decrease in implicit learning due to generalization. A condition with 15° implicit learning produces a 7.5° change, but a condition with 30° implicit learning produces a 15° change – thus, the total change in learning will vary with total implicit adaptation, which increased as rotation size increased. Another issue with the Gaussian model is that the implicit-explicit curve varies with explicit strategy, which ‘moves the data’ to different regions along the nonlinear normal distribution. As rotation gets larger, explicit strategies increase, which results in systematic changes in where the normal distribution is sampled, yielding variable implicit-explicit relationships when assessed in the linear sense. These variations in implicit-explicit learning did not match the measured data (Figure 4F), which exhibited little to no change in the slope relating implicit and explicit learning across rotation sizes. On the other hand, as discussed in the paper, the invariance in slope matches the competition model, where the implicit explicit correlation has a constant slope -*p_i_* which does not depend on the rotation (see black lines in Figure 4F).

Last, we conducted a model comparison between all 3 models (competition, linear generalization with McDougle et al. properties, and normal generalization with McDougle et al. properties) using AIC. For these analyses, we used the stepwise participants in Experiment 1. These are the only data where such a comparison can be done, because implicit and explicit learning was assayed across 4 rotation sizes. Thus, we fit all 3 models in Figures 5D-F to individual participants. As expected, the competition model was more likely to explain the data (Figure 4G); 31 participants were best described by the competition model, compared to 2 participants in the linear SPE generalization case, and 4 in the normal SPE generalization case (Figure 4G).

Perhaps the poor SPE model performance was linked to an inaccurate estimate for the width of the generalization curve (the slope in the linear case, the standard deviation in the normal case). In other words, maybe generalization in McDougle et al. differed from that in Experiment 1. To test this, we conducted a sensitivity analysis. We repeated our prediction error (Figure 4H, left) and AIC (Figure 4H, right) analyses, but varied the generalization width assumed by the model. As a lower bound, we used one-half the width measured by McDougle et al. As an upper bound, we used twice the width measured by McDougle et al. As shown in Figure 4H, the competition model remained superior across this range in generalization parameters.

Summary

In the revised paper, we now highlight the McDougle et al. generalization curve over that measured in the Day et al. study. In addition, we now consider whether a normal distribution can better match the measured implicit-explicit relationships in Experiment 1. Our results remain unchanged. The SPE generalization model exhibits two key limitations, a qualitative one, and a quantitative one:

1. Past generalization curves (even that measured by McDougle et al.) drastically underpredict the decline in implicit learning we measured in Experiments 2 and 3. This quantitative deviation would only be worsened had the McDougle et al. study used 4 targets (as in our data) which would have widened their generalization curve as observed in Krakauer et al., 2000. The existing deviations are not small – instead, implicit learning drops nearly 3 times more rapidly across the explicit strategy range probed in Experiments 2 and 3. The discrepancy is substantially worsened when we correct the “explicit strategies” we estimated, using generalization.

2. SPE generalization models predict that the decline in implicit learning will vary with changes in rotation size and explicit strategy. The competition model, however, predicts that the gain relating implicit and explicit learning is invariant across rotation sizes and changes in explicit strategy. These predictions were tested in Experiment 1. These data agreed with the competition model: there was a constant slope relating implicit-explicit learning across all rotation sizes. The competition model’s dominance was also shown in an AIC comparison that we have added to the paper. Finally, the SPE model’s poor performance was not caused by incorrectly estimating generalization parameters: in a sensitivity analysis we varied the generalization curve’s width and observed no appreciable effect on our results.

In the revised manuscript we have made these critical updates, and again thank the reviewer for strengthening our analysis. These changes can be observed in Section 1.6 in the paper, Figure 4, and Appendix 6.

(4.5) I may be missing something here, but is the error sensitivity finding reported in Figure 6 not circular? No savings is seen in the raw data itself, but if the proposed model is used, a sensitivity effect is recovered. This wasn't clear to me as presented.

This is indeed a puzzling phenomenon, hence why we unpack this in our competition map simulations in Figure 7. The apparent contradiction lies in how one defines savings. Generally, savings is intuitively defined as the upregulation in a learning process upon re-exposure to a perturbation. So, in this sense, in Haith et al. (2015), the implicit system does not exhibit savings. The issue arises, however, when the assumption is made that an invariance in the implicit learning *timecourse* implies an invariance in implicit learning *properties* (e.g., implicit error sensitivity).

As shown in Figure 6 (and the competition map in Figure 7E), the competition model predicts that the implicit system’s error sensitivity increased in Exposure 2 relative to Exposure 1. At the very same time, simulating the implicit system with these model parameters (see blue lines in Figure 5D and black lines in Figure 7E) yields similar implicit learning curves across Exposures 1 and 2. How is this possible? The answer is that the implicit learning *timecourse* depends not only on implicit error sensitivity (which has increased according to the model), but also competition with explicit strategy. Thus, an increase in implicit error sensitivity which would tend to increase implicit learning during Exposure 2, is counteracted by an increase in strategic learning, which would tend to decrease implicit learning via competition. These two competing forces yield an invariance in the implicit learning *timecourse*, but a variation in implicit learning *properties* (e.g., error sensitivity).

Another way to understand this (which we have now added to the paper on P13L510) is as follows. The competition model suggests that while the increased implicit error sensitivity does not lead to greater implicit learning during Exposure 2, it nonetheless contributes to overall savings. That is, had implicit error sensitivity not increased, implicit learning would decrease during Exposure 2 (because strategies are larger), and less overall savings in learning rate would occur.

These paradoxical situations have extremely important implications. From Haith et al. (2015), the invariance in the implicit learning curve might intuitively suggest that the implicit system does not experience any changes due to the initial rotation exposure. However, this expectation is not matched in Experiment 4 (Figure 8). Here, suppressing reaction times reveals a change in implicit learning that was not observed in Haith et al. These data sets are entirely consistent with each other when explicit strategy is considered (Figures 7EandF). Suppressing explicit strategy eliminates competition, thus allowing the implicit system to express its increased error sensitivity within its learning timecourse, which were ‘hiding’ in Haith et al.

What our model suggests is an unappreciated possibility in how we interpret savings experiments. The notion that increases in error sensitivity will enhance the implicit system’s learning timecourse, is not true. There is a distinction between learning properties and learning timecourses. Properties are specific to the learning system. Timecourses depend in part on interactions with parallel learning systems. The state-space model attempts to reveal properties. Empirical analysis in datasets such at Haith et al., measures timecourses. Thus, the competition model suggests that we must distinguish between two questions in savings experiments: (1) “did implicit learning increase?” and (2) “did implicit error sensitivity change?” As a corollary to this, the competition model suggests that invariance in an implicit learning timecourse should not be taken to mean that the implicit system’s response to error has not changed. We think these ideas are critical in the ongoing debate as to whether the implicit system changes its response to errors with re-exposure to a rotation.

We appreciate the confusion surrounding these points, as it is a genuinely unintuitive phenomenon. We hope that the passage added on P13L510 provides some additional context.

(4.6) The attenuation (anti-savings) findings of Avraham et al. 2021 are not sufficiently explained by this model.

Absolutely. We had noted this puzzling deviation between standard rotation and error-clamp learning in our last Discussion section: “… it is important to note that the increased implicit error sensitivity observed in Experiment 4 (Figure 8) contrasts with the implicit system’s response to invariant errors^25^; in cases where participants can neither control nor reduce their target error, the implicit system appears attenuated upon re-exposure to a perturbation…”

Indeed, this discrepancy provides reason to question whether standard rotation learning, and invariant error-clamp paradigms engage the implicit learning system in the same way (see Point 4-3 response). Avraham et al. (2021) argues that the error-clamp attenuation they observed, can also be seen in standard rotation learning (their Figure 3). Our work however, casts doubt on this: in our view, decreases in implicit learning is much more likely due to a competition with explicit strategy, rather than a downregulation in implicit learning properties. Thus, we think it is much too early to tell whether the attenuation induced by re-exposure in Avraham et al., occurs in standard rotation learning paradigms. Given the scope of our work, we feel the best course of action is to note these discrepancies in order to inspire future research into these central matters. We can at present, only speculate as to the cause of the discrepancy in implicit behavior across the standard rotation and error-clamp paradigm.

One working hypothesis may relate to the reward system. In standard rotation learning, the goal is to move a cursor to the target. In invariant error-clamp paradigms, participants cannot control the cursor; their goal is to try to move their arm as straight to the target as possible. Thus, in the standard rotation paradigm, it is optimal to potentiate the implicit system in response to past errors. In the error-clamp paradigm, it is optimal to suppress the implicit system to prevent deviations between the hand and target. The proprioceptive system may play a large role in suppressing implicit learning in the error-clamp paradigm, and less role in standard rotation learning. In sum, we speculate that the participant’s internal goal may play a strong role in savings, and that the stark contrast in goals across the errorclamp paradigm and standard rotation paradigm, may at least partly lead to their diverging implicit phenotypes.

Along these lines, Sedaghat-nejad et al. (2021) has shown that saccade adaptation (an implicit learning process) is accelerated when learning improves task success. In other words, when being in an adapted state improves reward acquisition probability, saccadic learning accelerates. On the other hand, when learning does not translate to improvements in task success, adaptation rates are not improved, in response to the same perturbation. These ideas are eerily similar to the current conundrum: implicit error sensitivity may be upregulated in standard rotations, because in these tasks learning improves one’s ability to acquire a reward (i.e., hit the target with the cursor). In an invariant error-clamp context, learning does not alter one’s ability to acquire the target, but rather (as noted above) is detrimental to the participant’s goal: move the arm straight to the target.

These suppositions somewhat align with known error-clamp learning properties established by Kim et al. (2019). In this paradigm, learning rate and total implicit adaptation are accelerated when the cursor is programmed to hit the target, and attenuated when the cursor completely or partially hits the target. One interpretation here, is that when the cursor does not hit the target, the brain predicts that adaptation can restore one’s ability to hit the target, and thus learning is improved. When the brain experiences the same error, but the cursor is already hitting the target, there is no need to adapt, and the brain may attenuate its implicit learning rate. Thus, while the mechanism is very unclear, one idea is with re-exposure to the perturbation as in Avraham et al., 2021, the brain has already determined that adaptation does not improve one’s ability to acquire the target, and then invokes the same attenuation process in Kim et al., 2019, to slow learning rates/extents. These processes would be reversed in standard rotation learning, where past adaptation has improved one’s ability to hit the target, and thus will promote enhanced learning rates.

In sum, we can do little more than speculate as to why the implicit response to rotation re-exposure varies so greatly between standard rotation learning and error-clamp learning. In our revised paper (P24L954), we have expanded on our previous comparison between savings-related phenomena across both tasks. These matters will need to be studied in much greater depth in future work devoted solely to this topic. At a minimum, Experiment 4, shows the implicit savings phenotype may vary across tasks/contexts. This is a fascinating possibility which we hope will spark future research directions.

(4.7) Have the authors considered context and inference effects (Heald et al. 2020) as possible drivers of several of the presented results?

This is an interesting point. While Heald et al., 2021 have limited treatment of implicit-explicit learning components, they do provide some extensions to their model which will exhibit properties common with the competition model. Heald et al. considers implicit and explicit learning in a spontaneous recovery paradigm (McDougle et al., 2015). They speculate that explicit memory components correspond to the ‘most responsible context on the previous trial’. They introduce a ‘bias’ term to the model, representing a mismatch between vision and proprioception. Implicit learning is the weighted average of this bias over each context. These two ‘states’ map well onto implicit and explicit learning in McDougle et al.

While we do not examine McDougle et al. (2015) in our manuscript, the implicit and explicit COIN-model extensions will compete as in the competition model. Incidentally, the authors state in their Supplementary Material that the ‘state [explicit] and bias [implicit] interact competitively within a context to account for the total state feedback.’ In their work, this competitive interaction gives rise to the non-monotonic explicit response, which at first increases, and then decreases as in McDougle et al. We noted that the competition model also produces this same phenotype, albeit in a different experiment (e.g., see P23L909, and Figure 6D).

Thus, it is interesting that both the COIN model realization, and our target-error learning model do indeed share a common property: namely, competition. For the COIN model, these states compete because they attempt to sum to the total state feedback. For the competition model, they attempt to sum to the total error. The models differ of course, in how they define implicit-explicit learning, and how they contextualize the learning process. For us, these are two parallel ‘states’ which respond to a common error. For the COIN model, these represent two components in a credit assignment: explicit being one’s estimate of an ‘external’ perturbation to the environment, implicit being one’s ‘internal’ estimate for a miscalibration between vision and proprioception. It seems possible that the COIN model could produce the competitive behaviors we consider, when ‘credit’ is more readily assigned to the external perturbation in some rotation conditions (e.g., large rotations) than others (e.g., small rotations).

We thank the reviewer for raising this interesting possibility and have added a discussion on P22L862.